# SCALING LAWS OF SIGNSGD IN LINEAR REGRESSION: WHEN DOES IT OUTPERFORM SGD?

**Jihwan Kim**
Seoul National University & KAIST InnoCORE LLM
aqua4689@snu.ac.kr

**Dogyoon Song**
University of California, Davis
dgsong@ucdavis.edu

**Chulhee Yun**
KAIST
chulhee.yun@kaist.ac.kr

## ABSTRACT

We study scaling laws of signSGD under a power-law random features (PLRF) model that accounts for both feature and target decay. We analyze the population risk of a linear model trained with one-pass signSGD on Gaussian-sketched features. We express the risk as a function of model size, training steps, learning rate, and the feature and target decay parameters. Comparing against the SGD risk analyzed by Paquette et al. (2024), we identify a *drift-normalization effect* and a *noise-reshaping effect* unique to signSGD. We then obtain compute-optimal scaling laws under the optimal choice of learning rate. Our analysis shows that the noise-reshaping effect can make the compute-optimal slope of signSGD steeper than that of SGD in regimes where noise is dominant. Finally, we observe that the widely used warmup-stable-decay (WSD) schedule further reduces the noise term and sharpens the compute-optimal slope, when feature decay is fast but target decay is slow.

## 1 INTRODUCTION

In large-scale language model training, neural scaling laws are a well-documented empirical regularity: performance tends to improve predictably as data, parameters, and compute increase. Kaplan et al. (2020) observe that the language model cross-entropy loss scales as a power-law of model size $M$ and number of steps $N$ in terms of the risk formula $R(M, N) \asymp M^{-\tau_1} + N^{-\tau_2}$ for some $\tau_1, \tau_2 > 0$.[1] Also, they observe that loss scales as the power of training compute, under optimal allocation of compute between model size and number of steps.

A growing body of theory has sought to explain this phenomenon, most prominently by analyzing the stochastic gradient descent (SGD) optimizer under the power-law random features (PLRF) model (Paquette et al., 2024; Lin et al., 2024; 2025). Yet, in practice, SGD is not the optimizer that powers today's state-of-the-art LLMs. Instead, their training is dominated by Adam (Kingma & Ba, 2015) and its variants. While Adam is considerably more difficult to analyze theoretically, it is often approximated in theory by the simpler signSGD (Bernstein et al., 2018), which captures Adam's coordinate-wise adaptivity. This gap between practice and theory motivates a natural question: *how do scaling laws change when we replace SGD with signSGD?* Addressing this question can help align theory with optimizer choices used in practice, and clarify how adaptive updates could reshape compute-optimal scaling regimes in the PLRF setting.

### 1.1 OUR CONTRIBUTION

We study the scaling law of signSGD in the power-law random features (PLRF) model, and our contributions are as follows.

---

[1] We use $\asymp$ to denote equality up to a multiplicative constant, i.e., $f(x) \asymp g(x)$ means $c_1 g(x) \le f(x) \le c_2 g(x)$ for some constants $c_1, c_2 > 0$.

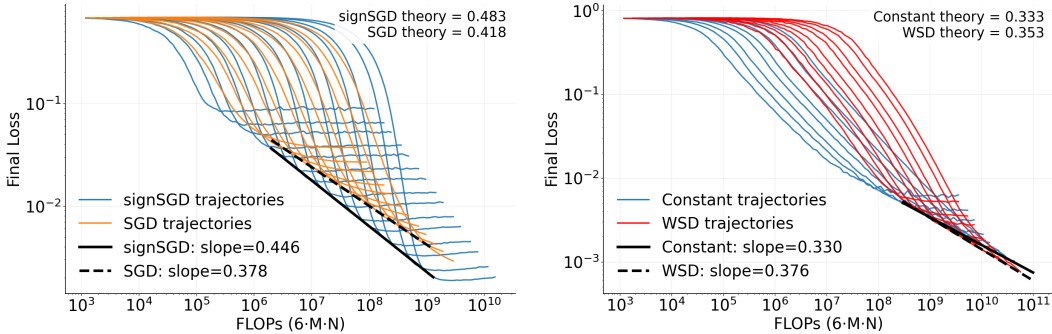

Figure 1: **Left: SGD vs. signSGD; Right: signSGD with constant vs. warmup-stable-decay schedules.** Colored lines represent the training trajectories of each algorithm, and black lines denote the compute-optimal curves. The upper right legend in each panel shows the theoretical value of the compute-optimal slope. SignSGD achieves a steeper compute-optimal slope than SGD (left panel), and warmup-stable-decay scheduling sharpens the compute-optimal slope relative to a constant schedule (right panel), for some parameter configurations. See Appendix C for parameters used in the experiment.

1. We derive a scaling law of signSGD with constant learning rates involving three variables (model size $M$, training steps $N$, learning rate $\gamma_0$) and two PLRF model parameters (feature decay $\alpha$, target decay $\beta$); see (12). By comparing with the SGD scaling laws of Paquette et al. (2024) and Lin et al. (2024), we observe two effects of signSGD: a *drift-normalization effect* and a *noise-reshaping effect*, inside the scaling law (see Section 4.1).

2. Under the fixed compute budget, we balance model size $M$ and training steps $N$, and optimize over learning rate $\gamma_0$. This allows us to characterize the compute-optimal loss decay rate and optimal model size with respect to the compute budget (see Table 1). Comparing against the compute-optimal scaling laws of SGD from Paquette et al. (2024) across regimes of the $(\alpha, \beta)$-parameter plane, we find that signSGD can achieve better exponents in the SGD noise bottleneck regimes, due to the noise-reshaping effect (see Figure 1).

3. We show that learning rate scheduling can further reduce the stochastic noise of signSGD. We analyze a warmup-stable-decay (WSD) schedule (Hu et al., 2024) widely used in large language model training. By maintaining drift velocity during the stable interval and reducing stochastic noise by the polynomially decaying interval, this schedule increases the compute-optimal slope in the PLRF setting for large $\alpha$ and small $\beta$ (see Section 4.3 and Figure 1).

4. We empirically validate our theory; see Figure 1 and Appendix C for details.

## 1.2 RELATED WORK

Here we discuss directly relevant results; additional related work is deferred to Appendix B.

**Empirical Scaling Laws.** Modern empirical work shows that performance improves with scale across data, parameters, and compute, following power laws across many domains (Hestness et al., 2017). In language modeling, Kaplan et al. (2020) document power-law loss trends over multiple orders of magnitude and simple budgeting rules linking model size, data, and compute. Henighan et al. (2020) extend these curves to images, video, and multimodal settings. Building on this, Hoffmann et al. (2022) argue that many LMs were under-trained on tokens and propose data-optimal scaling that substantially improves accuracy at fixed compute. Tissue et al. (2025) investigate the empirical scaling law with learning rate annealing.

**Scaling Law Theory.** Our work starts from the SGD scaling law in the PLRF model in Paquette et al. (2024) and Lin et al. (2024). In particular, Paquette et al. (2024) derive a scaling-law formula for one-pass SGD, where $M$, $N$, and $\gamma_0$ denote the model size, number of training steps, and learning

rate, respectively, and $\alpha$ and $\beta$ are the feature- and target-decay parameters.

$$R(M, N, \gamma_0) \approx \underbrace{M^{-2\alpha + \max(0, 1-2\beta)}}_{=:\mathcal{A}(M)} + \underbrace{(N\gamma_0)^{-\frac{2\alpha+2\beta-1}{2\alpha}}}_{=:\mathcal{D}_{\text{al}}^{\text{SGD}}(N,\gamma_0)} + \underbrace{M^{-1}(N\gamma_0)^{-\frac{2\alpha-1}{2\alpha}}}_{=:\mathcal{D}_{\text{dis}}^{\text{SGD}}(M,N,\gamma_0)} + \underbrace{\gamma_0(N\gamma_0)^{-\frac{4\alpha-1}{2\alpha}}}_{=:\mathcal{N}^{\text{SGD}}(N,\gamma_0)}.$$

(1)

The $\mathcal{A}(M)$ term corresponds to the *approximation error*, i.e., the loss as $N \to \infty$. Paquette et al. (2024) explain that $\mathcal{D}_{\text{al}}^{\text{SGD}}(N, \gamma_0)$ represents the *aligned feature loss*, as it coincides with the loss for a diagonal sketch matrix $\boldsymbol{S}$ (see Section 2.2 for a formal description). They also explain that $\mathcal{D}_{\text{dis}}^{\text{SGD}}(M, N, \gamma_0)$ corresponds to the *distorted feature loss*, arising from projection with a random matrix $\boldsymbol{S}$, and it decays more slowly than the aligned feature loss. Finally, $\mathcal{N}^{\text{SGD}}(N, \gamma_0)$ captures the *SGD noise*, stemming from the quadratic term in the Taylor expansion of the SGD update.

Several subsequent papers extend this baseline along two axes: (i) optimizer changes and (ii) model/training-protocol variations. On the optimizer side, Ferbach et al. (2025) investigate dimension-adapted Nesterov acceleration in the PLRF model and argue that it gives a better scaling law for $2\alpha > 1$ regime. Kunstner & Bach (2025) compare the gradient descent and sign descent scaling law in the linear bigram model. Comparison with their work is in Appendix B.1. Lin et al. (2025) cover the multi-pass SGD scaling law and identify the effect of data reuse for the scaling law. Discussion of the model/training-protocol changes is deferred to Appendix B.

**Scaling Behavior of Linear Models in the Context of Kernel Methods.** The power-law settings for data and targets adopted in our work are deeply rooted in the literature on kernel methods and their finite-width approximations. In this context, the power-law decays of the covariance spectrum and target coefficients are analogous to the classical capacity and source conditions, respectively. These spectral assumptions have been extensively investigated in kernel ridge regression (Caponnetto & De Vito, 2007; Cui et al., 2021) and random-features ridge regression (Rudi & Rosasco, 2017; Bach, 2017; Defilippis et al., 2024). Furthermore, similar conditions are fundamental to prior theoretical works on SGD that are closely related to our setting, including studies on one-pass SGD (Yao et al., 2007; Ying & Pontil, 2008; Carratino et al., 2018; Berthier et al., 2020) and multi-pass SGD (Pillaud-Vivien et al., 2018). Detailed comparison with these works is in Appendix B.3.

**SignSGD Dynamics.** Bernstein et al. (2018) give the non-convex convergence rate of signSGD. Xiao et al. (2025) derive the SDE and ODE for one-pass signSGD in the linear regression setting with squared loss. The ODE we derive matches theirs in final form; however, we obtain it in an alternative route that does not require a spectral lower bound on the covariance matrix that they imposed. Detailed comparison with Xiao et al. (2025) is in Appendix B.2. Compagnoni et al. (2025) derive SDEs for adaptive methods, including signSGD.

## 2 PROBLEM SETUP

### 2.1 NOTATION

We use bold lowercase letters (e.g., $\boldsymbol{u}$) to denote vectors and bold uppercase letters (e.g., $\boldsymbol{A}$) to denote matrices. For vectors $\boldsymbol{u}$ and $\boldsymbol{v}$, we denote the outer product by $\boldsymbol{u} \otimes \boldsymbol{v} := \boldsymbol{u}\boldsymbol{v}^{\mathsf{T}}$. $\lambda_i(\boldsymbol{A})$ denotes the i-th eigenvalue of the matrix $\boldsymbol{A}$. For positive-valued functions $f(x)$ and $g(x)$, we use $f(x) \lesssim g(x)$ if there exists $C > 0$ such that $f(x) \leq Cg(x)$ for sufficiently large $x$, and we use $f(x) \approx g(x)$ if there exist $c, C > 0$ such that $cg(x) \leq f(x) \leq Cg(x)$ for sufficiently large $x$.

### 2.2 MODEL

We consider the power-law random features (PLRF) model, parameterized by $\boldsymbol{\theta} \in \mathbb{R}^M$. Given a feature-label pair $(\boldsymbol{x}, y) \in \mathbb{R}^d \times \mathbb{R}$, the parameter $\boldsymbol{\theta}$ plays the role of a linear regression coefficient vector on the sketched features $\boldsymbol{S}\boldsymbol{x}$ (for some $\boldsymbol{S} \in \mathbb{R}^{M \times d}$), and the population risk function is

$$L(\boldsymbol{\theta}) = \mathbb{E}_{\boldsymbol{x}}\big[(\langle \boldsymbol{S}\boldsymbol{x}, \boldsymbol{\theta}\rangle - y)^2\big].$$

The data are generated as follows: the feature vector $\boldsymbol{x} \in \mathbb{R}^d$ is drawn from $\mathcal{N}(0, \boldsymbol{H})$ where $\boldsymbol{H}$ has eigenvalues $1^{-2\alpha}, \dots, d^{-2\alpha}$, and the label is $y = \langle \boldsymbol{x}, \boldsymbol{w}^* \rangle$ with $\langle \boldsymbol{v}_i, \boldsymbol{w}^* \rangle = i^{-\beta}$, where $\boldsymbol{v}_i$ is an

eigenvector of $\boldsymbol{H}$ corresponding to eigenvalue $i^{-2\alpha}$ for $i = 1, \ldots, d$; we call $\alpha$ and $\beta$ feature-decay and target-decay parameters, respectively. The sketch matrix $\boldsymbol{S} \in \mathbb{R}^{M \times d}$ is a random matrix that has i.i.d. entries $\mathcal{N}(0, 1/M)$, is drawn once and then held fixed throughout training; we refer to $M$ (with $M \leq d$) as the model size. Under these model assumptions,

$$L(\boldsymbol{\theta}) = \|\boldsymbol{H}^{1/2}(\boldsymbol{S}^\mathsf{T}\boldsymbol{\theta} - \boldsymbol{w}^*)\|^2.$$

We assume $d \geq r_0 M$ for some $r_0 > 1$, and let $d/M \to r \in [r_0, \infty]$ as $d, M \to \infty$ when $2\alpha > 1$, and $d/M \to r \in [r_0, \infty)$ when $2\alpha < 1$. The projected optimal parameter is

$$\boldsymbol{\theta}^* = (\boldsymbol{S}\boldsymbol{H}\boldsymbol{S}^\mathsf{T})^{-1}\boldsymbol{S}\boldsymbol{H}\boldsymbol{w}^*. \tag{2}$$

Define $\boldsymbol{w}_\perp = \boldsymbol{w}^* - \boldsymbol{S}^\mathsf{T}\boldsymbol{\theta}^*$ so that $\boldsymbol{w}^* = \boldsymbol{S}^\mathsf{T}\boldsymbol{\theta}^* + \boldsymbol{w}_\perp$ and $\boldsymbol{S}\boldsymbol{H}\boldsymbol{w}_\perp = 0$. The loss decomposes as

$$L(\boldsymbol{\theta}) = \|\boldsymbol{H}^{1/2}\boldsymbol{S}^\mathsf{T}(\boldsymbol{\theta} - \boldsymbol{\theta}^*)\|^2 + \|\boldsymbol{H}^{1/2}\boldsymbol{w}_\perp\|^2,$$

where the second term represents the approximation error.

**SignSGD.** We estimate the minimizer of the population risk via empirical risk minimization using signSGD. At step $k$, we draw a fresh sample $(\boldsymbol{x}_k, y_k)$ from our data model and form the stochastic gradient

$$\boldsymbol{g}_k = 2\big(\langle \boldsymbol{S}\boldsymbol{x}_k, \boldsymbol{\theta}_k \rangle - y_k\big)\boldsymbol{S}\boldsymbol{x}_k. \tag{3}$$

The signSGD update rule is

$$\begin{aligned}
\boldsymbol{\theta}_{k+1} &= \boldsymbol{\theta}_k - \gamma_k \operatorname{sign}(\boldsymbol{g}_k) \\
&= \boldsymbol{\theta}_k - \gamma_k \operatorname{sign}\big(\langle \boldsymbol{S}\boldsymbol{x}_k, \boldsymbol{\theta}_k \rangle - y_k\big) \operatorname{sign}(\boldsymbol{S}\boldsymbol{x}_k).
\end{aligned}$$

### 2.3 Representation of the Result

Let $R(M, N, \gamma_0)$ denote the loss $L(\boldsymbol{\theta}_N)$ under learning rate $\gamma_0$ and fixed model size $M$. We define the computational budget in terms of FLOPS[2] as $\mathfrak{f} = MN$, and consider the optimal model size $M^\star$ under fixed $\mathfrak{f}$, and optimal scaling of learning rate in the form $\gamma_0^\star = (M^\star)^{-e^*}$. For SGD, Paquette et al. (2024) derive compute-optimal scaling laws of the following form:

$$M^\star \asymp \mathfrak{f}^\xi, \qquad R\left(M^\star, \tfrac{\mathfrak{f}}{M^\star}, \gamma_0^\star\right) \asymp \mathfrak{f}^{-\eta}.$$

Our objective is to derive analogous formulas for signSGD, namely, $R(M, N, \gamma_0)$ and $R\big(M^\star, \tfrac{\mathfrak{f}}{M^\star}, \gamma_0^\star\big)$, and to compare them with the corresponding results for SGD.

## 3 Analyzing the SignSGD

In this section, we formulate the implicit integral equation for signSGD. We define

$$\boldsymbol{K} = \boldsymbol{S}\boldsymbol{H}\boldsymbol{S}^\mathsf{T}, \quad \overline{\boldsymbol{K}} = \operatorname{diag}(\boldsymbol{K})^{-1/2}\boldsymbol{K}, \quad \boldsymbol{K}_\sigma = \arcsin\big(\operatorname{diag}(\boldsymbol{K})^{-1/2}\boldsymbol{K}\operatorname{diag}(\boldsymbol{K})^{-1/2}\big), \quad (4)$$

where $\arcsin$ is applied entry-wise; we use these matrices and notation throughout the paper. We decompose the loss via

$$r_i(N) := (\boldsymbol{\theta}_N - \boldsymbol{\theta}^*)^\mathsf{T}(\boldsymbol{K}\boldsymbol{u}_i \otimes \boldsymbol{w}_i)(\boldsymbol{\theta}_N - \boldsymbol{\theta}^*),$$

where $\boldsymbol{u}_i, \boldsymbol{w}_i$ are the right/left eigenvectors of $\overline{\boldsymbol{K}}$ corresponding to the $i$th eigenvalue $\lambda_i(\overline{\boldsymbol{K}})$. This modal decomposition matches that of Xiao et al. (2025). For brevity we write $L(N) := L(\boldsymbol{\theta}_N)$.

$$L(N) = \sum_{i=1}^{M} r_i(N) + \|\boldsymbol{H}^{1/2}\boldsymbol{w}_\perp\|^2. \tag{5}$$

In Appendix E.1, we derive the one-step update formula for signSGD on a quadratic objective, using a second-order Taylor expansion and sign–Gaussian identities. Applying this to $r_i$ yields

$$\mathbb{E}[r_i(k+1) - r_i(k) \,|\, \mathcal{F}_k] = -\underbrace{\frac{4\gamma_k}{\pi\sqrt{L(k)}}\,\lambda_i(\overline{\boldsymbol{K}})\,r_i(k)}_{\text{drift}} + \underbrace{\frac{2\gamma_k^2}{\pi}\,\boldsymbol{w}_i^\mathsf{T}\boldsymbol{K}_\sigma\boldsymbol{K}\boldsymbol{u}_i}_{\text{quadratic noise}}. \tag{6}$$

---

[2]floating point operations per second

1. **Drift.** The first term in (6) yields a systematic decrease of mode $i$: it is proportional to the curvature $\lambda_i(\overline{\boldsymbol{K}})$ and the learning rate $\gamma_k$, while the factor $1/\sqrt{L(k)}$ self–normalizes the step. Note that the directions corresponding to larger eigenvalues contract faster.

2. **Quadratic noise.** The second term in (6) is an $O(\gamma_k^2)$ variance injection shaped by curvature and the sign–noise covariance. It is independent of $r_i(k)$ and may set a noise floor of $r_i$, unless $\gamma_k$ decays.

Overall, one-step progress reflects a balance between drift and quadratic noise: when $r_i(k)$ is large, the drift decreases $r_i(k)$; near the optimum, quadratic noise can dominate and cause $r_i(k)$ to plateau.

Converting the one-step update formula to the continuous-time ODE, we obtain [3]

$$\frac{dr_i}{dt} = -\underbrace{\frac{4\gamma_{t/\gamma_0}}{\pi\gamma_0\sqrt{L(t)}}\lambda_i(\overline{\boldsymbol{K}})\, r_i(t)}_{=:\Phi_i^{\mathbf{drift}}(t)} + \underbrace{\frac{2\gamma_{t/\gamma_0}^2}{\pi\gamma_0}\, \boldsymbol{w}_i^\mathsf{T}\boldsymbol{K}_\sigma\boldsymbol{K}\boldsymbol{u}_i}_{=:\Phi_i^{\mathbf{noise}}(t)}. \tag{7}$$

Compared to SGD, the drift is self-normalized by $1/\sqrt{L(t)}$ and the quadratic noise term does *not* carry the extra $L(t)$ factor present in SGD. So, for the constant learning rate, the quadratic noise does not decrease over time. The variation-of-constants formula gives the implicit integral representation

$$r_i(N) = r_i(0)\,\exp\left\{-\int_0^N \Phi_i^{\mathbf{drift}}(u)\,du\right\} + \int_0^N \exp\left\{-\int_z^N \Phi_i^{\mathbf{drift}}(u)\,du\right\}\times\Phi_i^{\mathbf{noise}}(z)\,dz. \tag{8}$$

Summing over modes, we define

$$L^{\mathrm{drift}}(N) = \sum_{i=1}^M r_i(0)\,\exp\left\{-\int_0^N \Phi_i^{\mathbf{drift}}(u)\,du\right\}, \tag{9}$$

$$L^{\mathrm{noise}}(N) = \sum_{i=1}^M \int_0^N \exp\left\{-\int_z^N \Phi_i^{\mathbf{drift}}(u)\,du\right\}\times\Phi_i^{\mathbf{noise}}(z)\,dz. \tag{10}$$

Exact formulation of $L^{\mathrm{drift}}(N)$ and $L^{\mathrm{noise}}(N)$ can be found in (27) of Appendix E.2. Then by (5) our risk is decomposed as

$$L(N) = L^{\mathrm{drift}}(N) + L^{\mathrm{noise}}(N) + \underbrace{\|\boldsymbol{H}^{1/2}\boldsymbol{w}_\perp\|^2}_{\mathrm{approx}}. \tag{11}$$

## 4 MAIN RESULTS

### 4.1 LOSS FORMULA FOR CONSTANT LEARNING RATE

We now analyze (11) to express it in the form of $R(M, N, \gamma_0)$, the loss after $N$ steps with constant learning rate $\gamma_0$ and model size $M$; time-varying schedules are discussed later.

- For $L^{\mathrm{drift}}(N)$, we use a deterministic approximation (Appendix E.2.2) similar to Paquette et al. (2024), and obtain the asymptotic self-consistent equation: with $\Gamma_M = M^{\min(\alpha,0.5)}\gamma_0$,

$$L^{\mathrm{drift}}(N) \approx \left(\Gamma_M\int_0^N L^{\mathrm{drift}}(u)^{-1/2}\,du\right)^{-\frac{2\alpha+2\beta-1}{2\alpha}} + M^{-1}\left(\Gamma_M\int_0^N L^{\mathrm{drift}}(u)^{-1/2}\,du\right)^{-\frac{2\alpha-1}{2\alpha}}.$$

Solving this yields signSGD counterparts of the aligned- and distorted- feature loss terms in (1), denoted by $\mathcal{D}_{\mathrm{al}}^{\mathrm{sign}}(M, N, \gamma_0)$ and $\mathcal{D}_{\mathrm{dis}}^{\mathrm{sign}}(M, N, \gamma_0)$; see (12) below for their precise forms.

- For $L^{\mathrm{noise}}(N)$ and approximation term, we calculate the limit loss $L_\infty$ and get

$$L_\infty \approx \max\left\{\gamma_0^2 M^{2-\min(1,2\alpha)},\ \|\boldsymbol{H}^{1/2}\boldsymbol{w}_\perp\|^2\right\}$$

Lastly we use approximation error result from Paquette et al. (2024); Lin et al. (2024),

$$\|\boldsymbol{H}^{1/2}\boldsymbol{w}_\perp\|^2 \approx M^{-2\alpha+\max(0,1-2\beta)}.$$

---

[3]We treat $L$, $r_i$, and the learning-rate $\gamma_k$ as continuous extensions, so $\gamma_{t/\gamma_0}$ is well-defined for any $t > 0$.

Combining two parts yields a proxy for the loss formula, and we prove that it satisfies the implicit integral equation (11) in Appendix E.3.4 and E.4.4. Finally, we get the following four-term scaling law formula for one-pass signSGD in the regime $-\alpha + 0.5 < \beta < \alpha + 0.5$:[4]

$$R(M, N, \gamma_0) \asymp \underbrace{M^{-2\alpha + \max(0,\, 1-2\beta)}}_{=:\mathcal{A}(M)} + \underbrace{\left(M^{\min(\alpha, 0.5)} N \gamma_0\right)^{-\frac{2(2\alpha + 2\beta - 1)}{2\alpha - 2\beta + 1}}}_{=:\mathcal{D}_{\mathrm{al}}^{\mathrm{sign}}(M, N, \gamma_0)} \tag{12}$$
$$+ \underbrace{M^{-\frac{6\alpha - 1}{2\alpha + 1}} (N \gamma_0)^{-\frac{2(2\alpha - 1)}{2\alpha + 1}}}_{=:\mathcal{D}_{\mathrm{dis}}^{\mathrm{sign}}(M, N, \gamma_0)} + \underbrace{\gamma_0^2\, M^{2 - \min(1, 2\alpha)}}_{=:\mathcal{N}^{\mathrm{sign}}(M, \gamma_0)}.$$

**Interpretation.** The term $\mathcal{A}(M)$ is the approximation error. The terms $\mathcal{D}_{\mathrm{al}}^{\mathrm{sign}}(M, N, \gamma_0)$ and $\mathcal{D}_{\mathrm{dis}}^{\mathrm{sign}}(M, N, \gamma_0)$ arise from the drift's exponential damping $r_i(0) \exp\left\{ - \int_0^N \Phi_i^{\mathbf{drift}}(u)\, du \right\}$ and correspond to the aligned and distorted feature losses of SGD scaling law in Paquette et al. (2024). The term $\mathcal{N}^{\mathrm{sign}}(M, \gamma_0)$ captures the quadratic noise from the one-step Taylor expansion, specific to one-pass signSGD.

**Comparison.** We compare our signSGD scaling law formula with the SGD formula (1) of Paquette et al. (2024). Since the approximation error is optimizer-independent, the term $\mathcal{A}(M)$ remains unchanged. For the $N$-exponent in $\mathcal{D}_{\mathrm{al}}$ and $\mathcal{D}_{\mathrm{dis}}$, when the absolute value of the exponent is $x$ for SGD, then it changes to $\frac{2}{2-x} x$ in signSGD, which is strictly larger than $x$. Therefore, $\mathcal{D}_{\mathrm{al}}^{\mathrm{sign}}(M, N, \gamma_0)$ and $\mathcal{D}_{\mathrm{dis}}^{\mathrm{sign}}(M, N, \gamma_0)$ decrease faster in the number of steps $N$ under signSGD. By contrast, the signSGD noise term $\mathcal{N}^{\mathrm{sign}}(M, \gamma_0)$ does not decay with $N$, whereas the SGD noise $\mathcal{N}^{\mathrm{SGD}}(N, \gamma_0)$ does.[5]

We discuss the underlying mechanism that modifies the drift terms $\mathcal{D}_{\mathrm{al}}$, $\mathcal{D}_{\mathrm{dis}}$, and the noise term $\mathcal{N}$.

- **Drift terms (Drift-normalization effect):** In signSGD, the drift in (6) is $\frac{4\gamma_k}{\pi \sqrt{L(k)}} \lambda_i(\overline{\boldsymbol{K}})$, whereas for SGD it is $2\gamma_k \lambda_i(\boldsymbol{K})$; see (4) for the definition of $\boldsymbol{K}$ and $\overline{\boldsymbol{K}}$. The diagonal preconditioning embedded in $\overline{\boldsymbol{K}}$ contributes an extra factor $M^{\min(\alpha, 1/2)}$, since the scale of the matrix $\mathrm{diag}(\boldsymbol{K})^{-1/2}$, which is multiplied in $\overline{\boldsymbol{K}}$, is $M^{\min(\alpha, 1/2)}$. The normalization by $\sqrt{L(k)}$ replaces the effective flow time $N\gamma_0$ with $\gamma_0 \int_0^N L(u)^{-1/2}\, du$, which accelerates progress in training whenever $L(u) \lesssim 1$. Thus, in the aligned/distorted drift terms, $N\gamma_0$ is replaced by $M^{\min(\alpha, 1/2)} \gamma_0 \int_0^N L(u)^{-1/2}\, du$. It leads to the self-consistent equation, which does not occur in SGD, and the solution of the self-consistent equation includes powers of $M^{\min(\alpha, 1/2)} N \gamma_0$. The absolute value of the exponent increases compared to SGD due to the acceleration in the regime $L(u) \lesssim 1$.

- **Noise term (Noise-reshaping effect):** The signSGD noise in (6) is $\frac{2\gamma_k^2}{\pi} \boldsymbol{w}_i^\top \boldsymbol{K}_\sigma \boldsymbol{K} \boldsymbol{u}_i$, while for SGD it is $\gamma_k^2 (\boldsymbol{v}_i^\top \boldsymbol{K} \boldsymbol{v}_i) L(k)$ with $\boldsymbol{v}_i$ an eigenvector of $\boldsymbol{K}$. The normalization removes the multiplicative $L(k)$ in signSGD, eliminating the Volterra structure present in Paquette et al. (2024). This difference is crucial: the lack of $L(k)$ in the quadratic term ultimately yields a noise term that does not decay in $N$. In the final formula, it removes the $(N\gamma_0)^{-\frac{4\alpha - 1}{2\alpha}}$ factor which appears in the SGD noise term, and therefore the noise term of signSGD increases as the learning rate $\gamma_0$ grows for all $(\alpha, \beta)$. In contrast, when the learning rate $\gamma_0$ grows, the noise term of SGD decreases for $\alpha > 0.5$ and increases for $\alpha < 0.5$. Meanwhile, an additional $M$-dependence arises from working in the $\overline{\boldsymbol{K}}$-eigenbasis (rather than $\boldsymbol{K}$-eigenbasis) due to the sign operation.

## 4.2 COMPUTE-OPTIMAL RESULT UNDER OPTIMAL CONSTANT LEARNING RATE

In the constant learning-rate schedule, we allow $\gamma_0$ to scale with the model size via $\gamma_0 = M^{-e}$. The hyperparameter $e$ directly influences the compute-optimal scaling law.[6]

---

[4]For the case $\beta > \alpha + 0.5$, $\mathcal{D}_{\mathrm{al}}^{\mathrm{sign}}(M, N, \gamma_0)$ takes form of $\left(1 - \kappa M^{\min(\alpha, 0.5)} N \gamma_0\right)^{-\frac{2(2\alpha + 2\beta - 1)}{2\alpha - 2\beta + 1}}$. See Appendix E.5 for more details.

[5]Decay with respect to $M$ depends on the choice of $\gamma_0$; in the subsequent sections, we set $\gamma_0$ as $M^{-e}$.

[6]One may wonder why we do not parameterize by $N$. Setting $\gamma_0 = M^{-e}$ is without loss of generality, since in the compute-optimal case both $M$ and $N$ are expressed as powers of the total compute $\mathfrak{f}$.

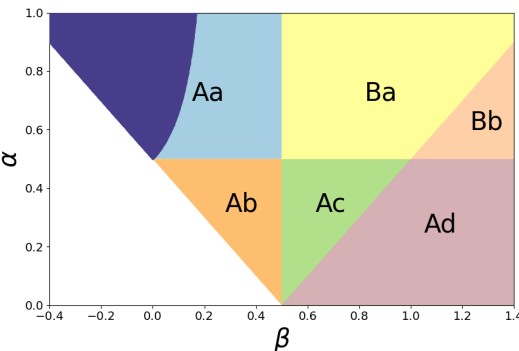 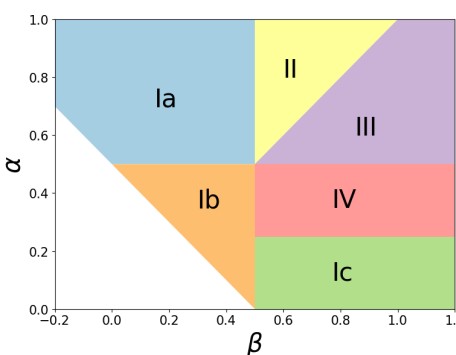

Figure 2: **Left: Phase plane for signSGD; Right: Phase plane for SGD.** The white region indicates parameter values with no power-law scaling. The dark blue area represents the region where warmup-stable-decay scheduling (Section 4.3) yields a better compute-optimal exponent.

Following Paquette et al. (2024), we distinguish the *maximal* and *optimal* learning rates for SGD. The maximal rate is the largest step that yields a stable (non-exploding) recursion; for signSGD, it leads to a zero compute-optimal slope (see Appendix F.1). We therefore focus on the optimal learning rate $\gamma_0^\star$, which maximizes the decay exponent $\eta$ in

$$R(M^\star, \mathfrak{f}/M^\star, \gamma_0^\star) \asymp \mathfrak{f}^{-\eta},$$

where $M^\star$ denotes the model size minimizing $R(\cdot)$ at fixed compute budget $\mathfrak{f}$.

To characterize the compute-optimal scaling, set $\gamma_0 = M^{-e}$, $M = \mathfrak{f}^x$, and $N = \mathfrak{f}^{1-x}$ (with $x \in [0, 1]$), and solve

$$(e^*, x^*) \in \arg\min_{e,x} R(M, N, \gamma_0) = \arg\min_{e,x} R(\mathfrak{f}^x, \mathfrak{f}^{1-x}, \mathfrak{f}^{-ex}). \tag{13}$$

Then $M^\star = \mathfrak{f}^{x^*}$, $N^\star = \mathfrak{f}^{1-x^*}$, and $\gamma_0^\star = (M^\star)^{-e^*}$, and at the optimum

$$R(M^\star, \mathfrak{f}/M^\star, \gamma_0^\star) \asymp \mathfrak{f}^{-\eta(\alpha,\beta)},$$

for some $\eta(\alpha, \beta) > 0$, which we refer to as the compute-optimal slope.

In problem (13), each of the four terms in (12) scales as $\mathfrak{f}^{-\ell_i(e,x)}$, so minimizing $R$ is equivalent to maximizing $\min\{\ell_1, \ell_2, \ell_3, \ell_4\}$. The optimal value $(e^*, x^*)$ is obtained by balancing three active exponents. The resulting formulas and dominant and balancing terms are summarized in Table 1; see Appendix F.2 for details.

We follow Paquette et al. (2024) in defining phases by dominant terms; to avoid confusion with their SGD phases, we label our signSGD phases by uppercase letters. Accordingly, any reference to Phase I–IV hereafter refers exclusively to the SGD phases of Paquette et al. (2024). For signSGD, the phase plane is simpler: when $\alpha > 0.5$ and $\beta > 0.5$ (Phase B) all four terms are dominant; otherwise (Phase A) the dominant terms are $\mathcal{A}(M)$, $\mathcal{D}_{\mathrm{al}}^{\mathrm{sign}}(M, N, \gamma_0)$, and $\mathcal{N}^{\mathrm{sign}}(M, \gamma_0)$. We declare *subphases* whenever the formula of at least one of $\gamma_0 = M^{-e^*}$, $M^\star$, or $R(M^\star, \mathfrak{f}/M^\star, \gamma_0^\star)$ changes. These changes occur across the boundaries $\alpha = 0.5$, $\beta = 0.5$, and $\beta = \alpha + 0.5$, yielding six subphases in total (Phase A split into four, Phase B into two). We provide a formula of approximation, drift, and noise term for each subphase in Table 2. For context, Paquette et al. (2024) partition the $(\alpha, \beta)$-plane into four phases with six subphases for optimal learning rate (and seven for the maximal learning rate).

*Remark* 1 (Dominant vs. balancing terms). Dominant terms refer to those that dominate the risk for some $(\gamma_0, M, N)$. *Balancing terms* are the ones that tie (hence "balancing") at the compute-optimal choice $(\gamma_0^\star, M^\star, N^\star)$ and therefore determine the slope; they form a subset of the dominant terms.

**Comparison of Compute-optimal Results.** For the intersection of Phase Aa, Ab, Ac, Ba and Phase I, II, the compute-optimal slope $\eta(\alpha, \beta)$ and optimal model size $M^\star$ are the same for signSGD

Table 1: Dominant and balancing terms, optimal learning rate, compute-optimal model size, and risk across different $(\alpha, \beta)$ phases. Refer to (12) for the definitions of the terms $\mathcal{A}, \mathcal{D}_{\mathrm{al}}, \mathcal{D}_{\mathrm{dis}}, \mathcal{N}$. See Figures 8 to 12 in the Appendix for empirical validation of the theoretical exponents.

| Phase | Dominant terms | Balancing terms | | $\gamma_0^\star$ | $M^\star$ | $R\left(M^\star, \frac{\mathfrak{f}}{M^\star}, \gamma_0^\star\right)$ |
|---|---|---|---|---|---|---|
| | | **Term structure** | | | **Compute–optimal** | |
| **Phase A** | $\mathcal{A}, \mathcal{D}_{\mathrm{al}}, \mathcal{N}$ | $\mathcal{A}, \mathcal{D}_{\mathrm{al}}, \mathcal{N}$ | Aa | $M^{-(\alpha+\beta)}$ | $\mathfrak{f}^{\frac{1}{2\alpha+1}}$ | $\mathfrak{f}^{-\frac{2\alpha+2\beta-1}{2\alpha+1}}$ |
| | | | Ab | $M^{-\frac{2\beta+1}{2}}$ | $\mathfrak{f}^{\frac{1}{2}}$ | $\mathfrak{f}^{-\frac{2\alpha+2\beta-1}{2}}$ |
| | | | Ac | $M^{-1}$ | $\mathfrak{f}^{\frac{2\alpha+2\beta-1}{2(2\beta-\alpha(2\beta-3)-1)}}$ | $\mathfrak{f}^{-\frac{\alpha(2\alpha+2\beta-1)}{2\beta-\alpha(2\beta-3)-1}}$ |
| | | | Ad | $M^{-1}$ | $\mathfrak{f}^{\frac{1}{2-\alpha}}$ | $\mathfrak{f}^{-\frac{2\alpha}{2-\alpha}}$ |
| **Phase B** | $\mathcal{A}, \mathcal{D}_{\mathrm{al}}, \mathcal{D}_{\mathrm{dis}}, \mathcal{N}$ | $\mathcal{D}_{\mathrm{al}}, \mathcal{D}_{\mathrm{dis}}, \mathcal{N}$ | Ba | $M^{-\frac{2\alpha+4\beta-1}{4\beta}}$ | $\mathfrak{f}^{\frac{\beta}{\alpha+\beta}}$ | $\mathfrak{f}^{-\frac{2\alpha+2\beta-1}{2\alpha+2\beta}}$ |
| | | | Bb | $M^{-\frac{6\alpha+1}{4\alpha+2}}$ | $\mathfrak{f}^{\frac{2\alpha+1}{4\alpha+1}}$ | $\mathfrak{f}^{-\frac{4\alpha}{4\alpha+1}}$ |

and SGD. In contrast, for the area of Phase III, IV excluding the case $0.25 < \alpha < 1/3$, $\beta > (1-\alpha)(1-2\alpha)/(2(1-3\alpha))$ (See Figure 4 in the Appendix for the visualization of this area), the compute-optimal slope $\eta(\alpha, \beta)$ for signSGD is *steeper* than that for SGD, and the optimal model size is bigger in signSGD. We refer to this region as the Area III-IV$_{\mathrm{sub}}$. Finally, for the optimal learning rate $\gamma_0 = M^{-e^*}$, the exponent $e^*$ is always bigger than SGD in signSGD, which means signSGD always has a smaller optimal learning rate.

## 4.3 EFFECT OF WARMUP-STABLE-DECAY SCHEDULING

We next study the widely used warmup-stable-decay schedule (Hu et al., 2024), which reduces late-stage noise via the decay interval while maintaining the drift rate over the stable interval.

For the warmup-stable-decay schedule, we set the learning rate to $\gamma_k = \gamma_0 f(k)$ with

$$
f(k) = \begin{cases} k/(wN), & k \le wN, \\ 1, & wN \le k \le pN, \\ \left(1 + \tau(k - pN)\right)^{-c}, & k > pN, \end{cases} \tag{14}
$$

where $w, p, c \in (0, 1)$ and $\tau > 0$. In other words, the learning rate increases linearly for the first $wN$ steps, stays constant for the next $(p - w)N$ steps, and finally decays as a polynomial of exponent $c$ for the remaining $(1 - p)N$ steps. Throughout, we additionally assume $w < p/2$.

In Phase Aa, the $f$-scheduled noise bound can improve over constant LR:

$$
L^{\mathrm{noise}}(N) \lesssim \gamma_0 M^{\frac{1}{2}} N^{-c} \sqrt{L(N)} + \gamma_0^{\frac{1}{2\alpha}} M^{\frac{1}{4\alpha}} N^{-(1-c)(1-\frac{1}{2\alpha})}.
$$

Combining this with the drift and approximation terms, and then optimizing over $e$ of $\gamma_0 = M^{-e}$, the decay parameter $c$, and the model size $M$, yields the $f$-scheduled risk bound [7]

$$
R_f(M^\star, \mathfrak{f}/M^\star, (M^\star)^{-e^*}) \lesssim \mathfrak{f}^{-\frac{2(4\alpha-1)(2\alpha+2\beta-1)}{16\alpha^2+8\alpha\beta+2\alpha-2\beta-1}}. \tag{15}
$$

The absolute value of the exponent in (15) exceeds the compute-optimal slope under constant learning rate when $\alpha > 0.5$ and $0.5 - \alpha < \beta < \frac{2\alpha-1}{2(4\alpha-1)}$. Thus, warmup-stable-decay scheduling yields a strictly larger compute-optimal slope in the upper left region of Phase Aa (marked with dark blue in Figure 2). We will refer to this region as Area Aa$^\star$ throughout the paper.

Scheduling does not improve the SGD compute-optimal exponent in Phases I–II (see Appendix G.5). Thus, with scheduling, signSGD achieves a larger compute-optimal exponent compared to SGD in Area Aa$^\star$. [8]

---

[7]The loss bound (15) also holds for stable-decay scheduling without a warmup stage, as well as for cosine and linear scheduling. Refer to Appendices G and H.

[8]Whether scheduling benefits other regions of signSGD or other phases of SGD remains open, since for both methods the scheduled noise upper and lower bounds do not match tightly, even up to constant factors.

## 5 Discussion: Where and Why SignSGD Provides Benefits?

With a constant learning rate $\gamma_0 = M^{-e}$, signSGD yields improvements over SGD in Area III-IV$_{\text{sub}}$. Under warmup-stable-decay scheduling, we find signSGD also provides benefits in Area Aa$^\star$.

**Mechanisms.** These gains can be explained by *noise-reshaping*, together with *drift-normalization*. In Paquette et al. (2024), Phases III–IV are the SGD noise-bottleneck regimes. By contrast, noise-reshaping in signSGD can alleviate this bottleneck with a suitable learning-rate choice, yielding improved compute-optimal slopes.

**Role of Learning-rate Scaling.** The signSGD noise term with constant LR is $\mathcal{N}^{\text{sign}}(M, \gamma_0) = \gamma_0^2 M^{2-\min(1,2\alpha)}$, whereas for SGD it is $\mathcal{N}^{\text{SGD}}(N, \gamma_0) = \gamma_0(N\gamma_0)^{-(4\alpha-1)/(2\alpha)}$. If $\gamma_0 \asymp 1$, $\mathcal{N}^{\text{sign}}(M, \gamma_0)$ is much larger than $\mathcal{N}^{\text{SGD}}(N, \gamma_0)$, making the compute-optimal slope asymptotically zero. Hence, we set $\gamma_0 = M^{-e}$ and optimize $e$ to balance terms and obtain a steep compute-optimal curve: decreasing $\gamma_0$ lowers $\mathcal{N}^{\text{sign}}(M, \gamma_0)$ while increasing the drift terms $\mathcal{D}^{\text{sign}}_{\text{al}}(M, N, \gamma_0)$ and $\mathcal{D}^{\text{sign}}_{\text{dis}}(M, N, \gamma_0)$, and the optimal $e$ strikes the balance.

**Why Gains Arise in Area III-IV$_{\text{sub}}$.** For SGD, the shape of $\mathcal{N}^{\text{SGD}}(N, \gamma_0)$ makes it dominate $\mathcal{D}^{\text{SGD}}_{\text{al}}(N, \gamma_0)$ at the compute-optimal point in Phases III–IV. It is because the absolute value of exponent in $\mathcal{N}^{\text{SGD}}(N, \gamma_0) = \gamma_0(N\gamma_0)^{-\frac{4\alpha-1}{2\alpha}}$ is smaller than that of $\mathcal{D}^{\text{SGD}}_{\text{al}}(N, \gamma_0) = (N\gamma_0)^{-\frac{2\alpha+2\beta-1}{2\alpha}}$ in Area III-IV$_{\text{sub}}$. For signSGD, noise-reshaping alters $\mathcal{N}^{\text{sign}}(M, \gamma_0)$ so it can *balance* against $\mathcal{D}^{\text{sign}}_{\text{al}}(M, N, \gamma_0)$. Note that the noise term takes a completely different form: $\mathcal{N}^{\text{sign}}(M, \gamma_0) = \gamma_0^2 M^{2-\min(1,2\alpha)}$, therefore dominance against the aligned drift term disappears. On the other hand, drift-normalization steepens the decay of $\mathcal{D}^{\text{sign}}_{\text{al}}(M, N, \gamma_0)$ by increasing the absolute value of the exponent with respect to $N$. This creates room for a balance in which both terms are smaller than the SGD noise $\mathcal{N}^{\text{SGD}}(N, \gamma_0)$ at optimum, explaining the improvements in Area III-IV$_{\text{sub}}$. For example, in the intersection between Phase Ba and Phase III, balancing $\mathcal{N}^{\text{sign}}(M, \gamma_0)$ and $\mathcal{D}^{\text{sign}}_{\text{al}}(M, N, \gamma_0)$ leads to $\mathfrak{f}^{-\frac{2\alpha+2\beta-1}{2\alpha+2\beta}}$, whereas $\mathcal{N}^{\text{SGD}}(N, \gamma_0)$ takes bigger value $\mathfrak{f}^{-\frac{4\alpha-1}{4\alpha}}$.

**Why Warmup-stable-decay Scheduling Helps.** For a learning-rate schedule $\gamma_k = \gamma_0 f(k)$ with general $f$, the drift-only self-consistent solution in Phase Aa takes the form

$$\left( M^{1/2} \gamma_0 F(N) \right)^{-\frac{2(2\alpha+2\beta-1)}{2\alpha-2\beta+1}}, \qquad \text{where} \qquad F(N) := \int_0^N f(u)\, du.$$

This can be viewed as $\mathcal{D}^{\text{sign}}_{\text{al}}(M, N, \gamma_0)$ with $N$ replaced by $F(N)$. This aligns with empirical observations that a loss term can decay polynomially with the area under the learning-rate curve (Tissue et al., 2025).

In contrast, the noise term depends most heavily on the learning rate *near the end* of training, since earlier noise can be damped by later drift; see (8). Warmup-stable-decay preserves the total area $F(N)$ asymptotically while shrinking the late-stage learning rate, thereby reducing noise without sacrificing drift. As a result, warmup-stable-decay scheduling yields a larger compute-optimal slope in Area Aa$^\star$ (upper-left Phase Aa; see Section 5.1 for intuition). More broadly, we conjecture that appropriate scheduling can further reduce the signSGD noise term, enabling improvements beyond Area III-IV$_{\text{sub}}$ and Area Aa$^\star$.

### 5.1 Hypothesis for the Position of the Beneficial Area

Here, we hypothesize why the areas with improved scaling law lie near the left edge (small $\beta$) and the right side ($\beta > \alpha$) of the phase plane.

**Heuristic Criterion.** Let "target decay" denote the coordinate-wise decay of $\boldsymbol{\theta}^*$ in (2), and let "stochastic-gradient decay" denote that of the stochastic gradient in (3). SignSGD is advantageous when the *target decays more slowly* than the stochastic gradient. Under SGD, coordinates with smaller gradients take smaller updates; if the target does not decay much, those coordinates still require learning targets of comparable magnitude, so more iterations are needed—an inefficiency that signSGD mitigates by normalizing per-coordinate updates via the sign operation.

**When Does This Occur? Observations and Conjecture**  Let $SHS^\top = U\Lambda U^\top$ be the eigendecomposition. Then the expected stochastic-gradient expressed in the $U$-eigenbasis has $i$th coordinate magnitude that decays as $i^{-2\alpha}$. See Appendix I for details of analysis.

Next, we examine how the target decays in the basis of the columns of $U$. For that, we have to consider $U^\top\theta^*$. Since $\mathbb{E}[S^\top S] = I$, we decompose

$$S^\top S = I + E, \qquad E := S^\top S - I,$$

so that $E$ represents the zero-mean fluctuation around the identity. Then we have

$$\begin{aligned}
U^\top\theta^* &= U^\top(SHS^\top)^{-1}SHw^* \\
&= U^\top(SHS^\top)^{-1}SH(S^\top S - E)w^* \\
&= U^\top Sw^* - U^\top(SHS^\top)^{-1}SHEw^*.
\end{aligned}$$

Since $SHS^\top = U\Lambda U^\top$ and the columns of $U$ and $S$ are well aligned, we expect that $U^\top Sw^*$ would exhibit a decay pattern similar to $w^*$. The second term $U^\top(SHS^\top)^{-1}SHEw^*$ could be thought of as a stochastic error which hinders the decay. For small $\beta$, as the decay of $w^*$ is slow, the decay of $U^\top Sw^*$ is expected to be slow, and therefore the overall decay of $U^\top\theta^*$ will be slow as well. If we increase the $\beta$, the decay of $U^\top Sw^*$ will become faster, which also drives a faster decay of $U^\top\theta^*$. However, when $\beta$ becomes too big, as the first term $U^\top Sw^*$ decays rapidly, the second term $U^\top(SHS^\top)^{-1}SHEw^*$ dominates quickly, and therefore $U^\top\theta^*$ will plateau quickly after some steep decay.

Figure 3 empirically validates our intuition for the decay of $U^\top\theta^*$. For $(\alpha,\beta) = (0.7, 1.1)$, $U^\top\theta^*$ plateaus quickly; for $(0.7, 0.6)$ it decays longer; and for $(0.7, 0.1)$, since $w^*$ hardly decays, the target also shows little decay.

These observations suggest that in the left region (small $\beta$) and the right region ($\beta > \alpha$), the targets decay more slowly than the stochastic gradient, whereas in the middle band ($0.5 < \beta < \alpha$) they do not. This could potentially explain why the signSGD-beneficial area appears near the left edge and the right side of the phase plane.

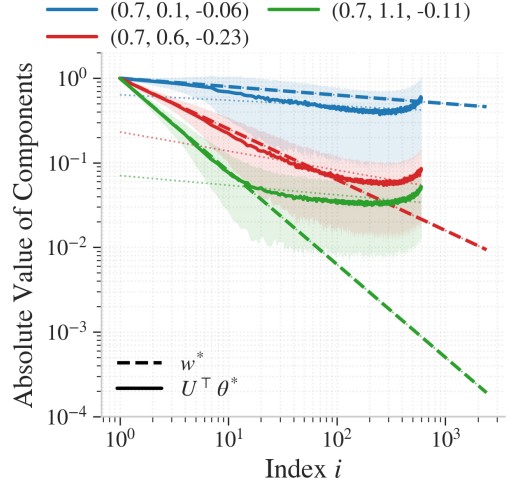

Figure 3: **Decay of $\theta^*$ in the basis of columns of $U$ compared to $w^*$.** The legend on the top shows ($\alpha$, $\beta$, fitted slope of $U^\top\theta^*$).

### 5.2 Conjecture for Adam

We conjecture that Adam with $\beta_2$ parameter sufficiently close to 1 follows the same scaling law with signSGD, based on the heuristic analysis in Appendix J. In detail, we expect Adam to follow the same asymptotic loss formula (12) with signSGD, and therefore to follow the same compute-optimal scaling law with respect to FLOPS $\mathfrak{f}$ in the Table 1. We also conducted an experiment on Adam and checked that the exponents in the Table 1 and the measured compute-optimal loss exponents and optimal model size exponents for Adam match well (see Figure 25).

## 6 Conclusion

We derived the scaling law of signSGD under the PLRF model and identified two distinctive effects—drift-normalization and noise-reshaping—relative to SGD. Analyzing compute-optimal tradeoffs, we showed that signSGD achieves steeper slopes than SGD in the noise-bottleneck regimes, and that the warmup-stable-decay schedule further improves performance in the Area Aa$^\star$. Additionally, in Appendix J, we analyze Adam using the heuristic of Xiao et al. (2025) and observe the same scaling law as signSGD. Deriving Adam's scaling law without heuristic assumptions is a compelling direction. We defer the discussion of limitations and additional future work to Appendix A.

## ACKNOWLEDGMENTS

Jihwan Kim thanks Junghyun Lee for helpful discussions and insightful feedback. This work was supported in part by an Institute of Information & communications Technology Planning & Evaluation (IITP) grant (No. RS-2024-00457882, National AI Research Lab Project) funded by the Korean government (MSIT) and the InnoCORE program of the Ministry of Science and ICT (No. N10250156).

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

# Supplementary Materials for "Scaling Laws of SignSGD in Linear Regression: When Does It Outperform SGD?"

## Usage of LLM

We primarily used LLMs to polish the English writing throughout the paper. They were also employed to help us identify additional related work beyond those we were already familiar with. When preparing well-formatted tables, we relied on LLMs for assistance. We also used LLMs to refine LaTeX code so that complicated formulas appeared clean and readable in the manuscript. Finally, we sought LLM support for debugging code used in our experiments.

## Overview of Appendix

(1) In Appendix A we discuss limitations and future work.

(2) In Appendix B we discuss more related works beyond those discussed in Section 1.2, and provide a detailed comparison with closely related works.

(3) In Appendix C we present experimental results which support our theory.

(4) In Appendix D, we prove that the general setting with a feature covariance $H$ can be reduced to the diagonal covariance case without loss of generality.

(5) In Appendix E we derive the scaling law formula (12) of $R(M, N, \gamma_0)$ under constant learning rate. We first derive a one-step update formula and convert it to an ODE to get an integral equation. We use a deterministic approximation for the integral equation with experimental results. Then we set a proxy of the loss function and verify that it satisfies the integral equation.

(6) In Appendix F we discuss the maximal learning rate deferred from the main text, and derive the optimal learning rate, compute-optimal loss, and optimal model size in Table 1.

(7) In Appendix G we derive the result for the warmup-stable-decay learning rate in Section 4.3.

(8) In Appendix H, we provide an analysis for the linear decaying scheduling and the cosine scheduling.

(9) In Appendix I we provide analysis for stochastic gradient decay deferred from Section 5.1.

(10) In Appendix J we derive scaling law of Adam under heuristic proposed by Xiao et al. (2025), and verify our results with experiment.

(11) In Appendix K we provide omitted analysis from Appendix E.

(12) In Appendix L, we provide an analysis for the case with noisy labels.

## A    Limitation and Future Work

**Limitation.**    Our analysis assumes batch size 1 and focuses on the PLRF setting; we leave theoretical extensions to minibatches for future work and provide only empirical evidence in Section C.4. We also use a deterministic approximation whose accuracy we verify empirically; tightening constants and extending the formal guarantees are left for future work.

**Future Work.**    Combining signSGD with dimension-adapted acceleration (Ferbach et al., 2025) and extending the framework to more complex architectures (e.g., two-layer linear networks or self-attention) are promising avenues.

## B    Additional Related Work

**More Related Works on Empirical Scaling Laws.**    Porian et al. (2024) resolve discrepancy between Kaplan et al. (2020) and Hoffmann et al. (2022). Kumar et al. (2025) investigate precision-aware scaling law.

**More Related Works on Scaling Law Theory.** There are lines of work analyzing more complex models compared to the power-law random features (PLRF) model. Bordelon et al. (2025) investigate the scaling law of a two-layer linear neural network with projected gradient descent, and argue the benefit compared to the PLRF model, which is one-layer. Ding et al. (2025) cover the scaling law of quadratically parameterized linear regression with SGD. Lyu et al. (2025) cover the scaling law of linear self-attention under gradient flow.

Sharma & Kaplan (2020) show that test loss scales as a power-law of model size in regression problems. Hutter (2021) investigates binary classification using a tabulation learning algorithm, deriving a power-law scaling with respect to dataset size. Bahri et al. (2024) analyze a linear random features model with SGD, showing a power-law decay in test loss with respect to sample size (or model size, when the other is infinite). Bordelon et al. (2024) derive a power law over model size, dataset size, and time for the linear random features model under gradient flow dynamics.

**More Related Works about signSGD and sign descent.** Balles et al. (2020) investigate the geometry of sign gradient descent. Kunstner et al. (2023) discover that sign descent could be the key factor making the gap between SGD and Adam on Transformers. Bernstein et al. (2019) propose signSGD with majority vote, which is communication efficient and fault-tolerant. Karimireddy et al. (2019) prove that error-feedback can make the rate of convergence of signSGD better.

### B.1 COMPARISON WITH KUNSTNER & BACH (2025)

First, their work compares the scaling laws of sign descent and gradient descent, whereas our work compares the scaling laws of signSGD and SGD. Second, they analyze a Linear Bigram Model, while we analyze for the power-law random features (PLRF) model. The advantage of the PLRF model is that it models two parameters each for feature vector decay and target decay, while the Linear Bigram Model has one parameter for data frequency decay. Lastly, they derived a scaling law where the model size goes to infinity; in contrast, our scaling law covers both finite model size and infinite limit by representing the loss as a function of model size, number of steps, and learning rate. This makes it possible for us to analyze the compute-optimal scaling law.

### B.2 COMPARISON WITH XIAO ET AL. (2025)

ODE for signSGD in Xiao et al. (2025) is equivalent to the ODE that occurs during our analysis. The reason that we were not able to directly use their ODE is that they derived it under the spectrum lower bound assumption for the covariance matrix. In our case, the spectrum of the covariance matrix $SHS^\mathsf{T}$ decays asymptotically as $i^{-2\alpha}$, so their assumption does not hold for our setup. So we re-derived the ODE without the spectrum lower bound assumption. Due to the spectrum lower bound assumption, they led to an exponential decay to limit risk, which is completely different from the polynomial neural scaling law derived from our paper. They discussed the noise-reshaping effect on the level of SDE. In contrast, we observed noise reshaping on the level of scaling law and investigated its effect on compute-optimal scaling.

### B.3 COMPARISON WITH THE WORKS IN THE CONTEXT OF KERNEL METHODS

Yao et al. (2007) study deterministic Gradient Descent and SGD under the reproducing kernel Hilbert space (RKHS) model. Their setup captures the infinite-dimensional case, while our paper handles model size $M$ as a tunable parameter to achieve optimal risk. They analyze the Early Stopping and that concept is closely related to the number of optimal steps $N = \mathfrak{f}/M^\star$ under fixed compute in our paper. Both imply that stopping the algorithm before the convergence can be helpful. The strength of our paper compared to theirs is that we provide an asymptotic loss function with model size and number of steps (which is the same as sample count in one-pass setting), while they provide an upper bound of loss by a polynomial of the sample count. They use the source parameter $r$ and relation $r = (2\alpha + 2\beta - 1)/(4\alpha)$ was indicated in Paquette et al. (2024). The authors derive $m^{-(\alpha+\beta-0.5)/(6\alpha+2\beta-1)}$ rate under condition $\alpha + \beta > 0.5$, where $m$ is sample count. Our signSGD rate with respect to $N$ for noisy labels in Section L is better than their rate. Their strength compared to our paper is that they also cover the classification setting, not only the regression setting. We leave the classification setting as future work.

Ying & Pontil (2008) study online gradient descent without regularization under the reproducing kernel Hilbert space (RKHS) model. They represent the expected loss as a function of the number of online steps $T$. They derive loss formula $T^{-(2\alpha+2\beta-1)/(4\alpha+2\beta-1)}\ln T$. Similar to Yao et al. (2007), our signSGD rate with respect to $N$ for noisy labels in Section L is better than their rate. Their source parameter $\beta$ is related to the target decay parameter $\beta$ in our paper. Note that they use the same Greek letter but have different meanings. They focus on the number of online steps $T$, while we handle two variables: model size $M$ and number of steps $N$. Their paper investigates the universal polynomially decaying step size and constant step size depending on the number of online steps $T$. The first one is similar to the polynomially decaying part of the warmup-stable-decay scheduling. One major difference is that we tune the learning rate based on model size $M$.

Carratino et al. (2018) study both multiple and single pass SGD under a random feature model with a connection to the RKHS setting. In their random feature model, non-linearity is included by the continuous map $\psi$, we leave the analysis of signSGD under the nonlinear model for future work. They provide a bound of risk with high probability, while we focus on the average asymptotic behavior of signSGD. They handle both model size $M$ and number of iterations $t$, and it is the same as our setting. Their strength compared to our paper is that they cover minibatching, while we focus on batch size 1. For the signSGD batch size bigger than 1 makes the problem significantly complicated to solve compared to the case of SGD, so we leave minibatching for future work. Their rate with sample count $n$ is $n^{-(2\alpha+2\beta-1)/(2(\alpha+\beta))}$. Our signSGD rate with respect to $N$ for noisy labels in Section L is better than their rate for the case $\beta > 0$, and theirs is better for the case $\beta < 0$.

Berthier et al. (2020) has a closer setting to our paper. They study linear regression with SGD and assume a noiseless label. Their upper bound of loss is $n^{-\min((2\alpha+2\beta-1)/(2\alpha),1-1/(2\alpha))}$ where $n$ is number of samples. Later work Paquette et al. (2024) has the same exponents for drift terms, as they also use SGD and assume a noiseless label. The difference between exponents in Berthier et al. (2020) and the exponents of the drift term in our work stems from the drift-normalization effect of signSGD. Also note that our work is different in several other aspects: (i) we consider a model size parameter $M$; (ii) we cover the regime $2\alpha < 1$; (iii) we derive the asymptotic loss formula rather than an upper bound; (iv) we consider the compute-optimal aspect.

Pillaud-Vivien et al. (2018) investigate multi-pass SGD in least-squares regression with bounded label noise. They got a rate $n^{-(2\alpha+2\beta-1)/(2\alpha+2\beta)}$ where $n$ is the number of samples, and it is better than single-pass SGD in the regime $\beta < 0$. Compared to the signSGD rate with respect to $N$ for noisy labels in Section L, our signSGD rate is better when $\beta > 0$ and worse for regime $\beta < 0$ than the single-pass SGD. Investigating multi-pass signSGD for $\beta < 0$ will be an interesting future direction.

Much earlier work Caponnetto & De Vito (2007) study kernel ridge regression in the RKHS model. Their rate is $l^{-\frac{2\alpha+2\beta-1}{2\alpha+2\beta}}$ where $l$ is number of samples. Their rate is better than our signSGD rate with respect to $N$ for noisy labels in Section L for the case $\beta < 0$, and worse for the case $\beta > 0$.

Later work Cui et al. (2021) also investigate kernel ridge regression in the RKHS model. Different from Caponnetto & De Vito (2007), they also consider a noiseless target and get a rate of $n^{-(2\alpha+2\beta-1)}$ for that case, where $n$ is the number of samples. Our noiseless drift exponent $-\frac{2(2\alpha+2\beta-1)}{2\alpha-2\beta+1}$ is better when $\alpha > \beta + 0.5$, and worse otherwise.

Rudi & Rosasco (2017) consider random-features ridge regression under the RKHS model. They give a rate of $n^{-(2\alpha+2\beta)/(2\alpha+2\beta+1)}$ where $n$ is the number of samples. Compared to our signSGD rate with respect to $N$ for noisy labels in Section L, ours is better when $\beta > 0$, $\alpha > 1/(4\beta) - \beta$ holds, and worse otherwise.

Bach (2017) also considers random-features ridge regression under the RKHS model, and gives a different upper bound rate $n^{-\alpha}$ where $n$ is the number of samples. Compared to our signSGD rate with respect to $N$ for noisy labels in Section L, ours is better when $\beta > \alpha^2 - \alpha + 0.5$, and worse otherwise.

Defilippis et al. (2024) derive a deterministic equivalent for random-features ridge regression under the RKHS model. Their rate is $n^{-(2\beta-1)/(2\beta)}$ for $\beta \leq 0.5 + 2\alpha$ and $n^{-(4\alpha)/(4\alpha+1)}$ for $\beta \geq 0.5 + 2\alpha$. Compared to our signSGD rate with respect to $N$ for noisy labels in Section L, ours is better when $\alpha > -2\beta^2 + \beta$, $\beta \leq 0.5 + 2\alpha$ or $\beta > (2\alpha+1)/(8\alpha+2)$, $\beta \geq 0.5 + 2\alpha$ holds, and worse otherwise.

### B.4 TABLE OF ASYMPTOTIC FORMS OF APPROXIMATION, DRIFT, AND NOISE TERM FOR SIGNSGD AND SGD

We added Table 2 and Table 3, which show asymptotic forms of approximation, drift, and noise term for signSGD and SGD, for comparison.

Table 2: Asymptotic forms of approximation, drift, and noise term for signSGD in different $(\alpha, \beta)$ phases. In this table, we provide a formula of approximation, drift, and noise term for 6 subphases.

| Phase | Approx | Drift | Noise |
|---|---|---|---|
| Phase Aa | $M^{-(2\alpha+2\beta-1)}$ | $(M^{1/2}N\gamma_0)^{-\frac{2(2\alpha+2\beta-1)}{2\alpha-2\beta+1}}$ | $\gamma_0^2 M$ |
| Phase Ab | $M^{-(2\alpha+2\beta-1)}$ | $(M^{\alpha}N\gamma_0)^{-\frac{2(2\alpha+2\beta-1)}{2\alpha-2\beta+1}}$ | $\gamma_0^2 M^{2-2\alpha}$ |
| Phase Ac | $M^{-2\alpha}$ | $(M^{\alpha}N\gamma_0)^{-\frac{2(2\alpha+2\beta-1)}{2\alpha-2\beta+1}}$ | $\gamma_0^2 M^{2-2\alpha}$ |
| Phase Ad | $M^{-2\alpha}$ | $(\max(1-M^{\alpha}N\gamma_0, 0))^{\frac{2(2\alpha+2\beta-1)}{-2\alpha+2\beta-1}}$ | $\gamma_0^2 M^{2-2\alpha}$ |
| Phase Ba | $M^{-2\alpha}$ | $(M^{1/2}N\gamma_0)^{-\frac{2(2\alpha+2\beta-1)}{2\alpha-2\beta+1}} + M^{-\frac{6\alpha-1}{2\alpha+1}}(N\gamma_0)^{-\frac{2(2\alpha-1)}{2\alpha+1}}$ | $\gamma_0^2 M$ |
| Phase Bb | $M^{-2\alpha}$ | $(\max(1-M^{1/2}N\gamma_0, 0))^{\frac{2(2\alpha+2\beta-1)}{-2\alpha+2\beta-1}} + M^{-\frac{6\alpha-1}{2\alpha+1}}(N\gamma_0)^{-\frac{2(2\alpha-1)}{2\alpha+1}}$ | $\gamma_0^2 M$ |

Table 3: Asymptotic forms of approximation, drift, and noise term for SGD in different $(\alpha, \beta)$ phases. In this table, we provide a formula of approximation, drift, and noise term for 6 subphases.

| Phase | Approx | Drift | Noise |
|---|---|---|---|
| Phase Ia | $M^{-(2\alpha+2\beta-1)}$ | $(N\gamma_0)^{-\frac{2\alpha+2\beta-1}{2\alpha}}$ | $\gamma_0(N\gamma_0)^{-\frac{4\alpha-1}{2\alpha}}$ |
| Phase Ib | $M^{-(2\alpha+2\beta-1)}$ | $(N\gamma_0)^{-\frac{2\alpha+2\beta-1}{2\alpha}}$ | $\gamma_0(N\gamma_0)^{-\frac{4\alpha-1}{2\alpha}}$ |
| Phase Ic | $M^{-2\alpha}$ | $(N\gamma_0)^{-\frac{2\alpha+2\beta-1}{2\alpha}}$ | $\gamma_0(N\gamma_0)^{-\frac{4\alpha-1}{2\alpha}}$ |
| Phase II | $M^{-2\alpha}$ | $(N\gamma_0)^{-\frac{2\alpha+2\beta-1}{2\alpha}} + M^{-1}(N\gamma_0)^{-\frac{2\alpha-1}{2\alpha}}$ | $\gamma_0(N\gamma_0)^{-\frac{4\alpha-1}{2\alpha}}$ |
| Phase III | $M^{-2\alpha}$ | $(N\gamma_0)^{-\frac{2\alpha+2\beta-1}{2\alpha}} + M^{-1}(N\gamma_0)^{-\frac{2\alpha-1}{2\alpha}}$ | $\gamma_0(N\gamma_0)^{-\frac{4\alpha-1}{2\alpha}}$ |
| Phase IV | $M^{-2\alpha}$ | $(N\gamma_0)^{-\frac{2\alpha+2\beta-1}{2\alpha}}$ | $\gamma_0(N\gamma_0)^{-\frac{4\alpha-1}{2\alpha}}$ |

### B.5 ADDITIONAL PHASE PLANE PLOTS TO COMPARE WITH PRIOR WORK

Figure 4 indicates the area where signSGD has a steeper compute-optimal slope compared to SGD, by coloring it with Mint green. It lies in Phase Ac, Ad, Ba, Bb, and covers all areas of Phase Bb. In terms of the SGD Phase, it covers all areas of Phase III and most of the areas of Phase IV.

Figure 5 indicates the area where signSGD has a steeper compute-optimal slope compared to DANA-decaying in Ferbach et al. (2025), by coloring it with Lime green. It lies in Phase Ac, Ad, Ba, Bb. It is smaller than the Mint green area, and this is natural, since DANA-decaying in Ferbach et al. (2025) has a steeper slope compared to SGD.

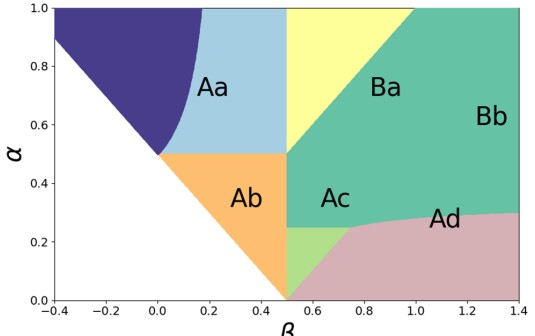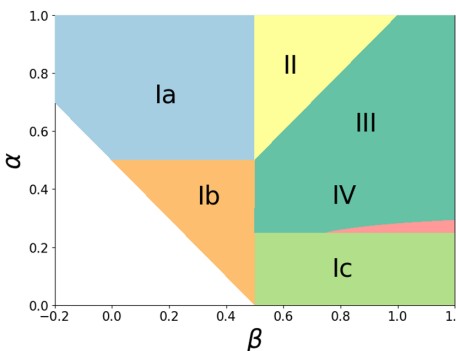

Figure 4: **Phase planes to compare signSGD and SGD.** Mint green area covering all of Phase Bb and III, and some part of Phase Ac, Ad, Ba, IV is the area where signSGD has a steeper compute-optimal slope compared to SGD. The left side is the signSGD phase plane, and the right side is the SGD phase plane. We placed the Mint green area for both of them for clarity. We will call this Mint green area as Area III-IV$_{\text{sub}}$.

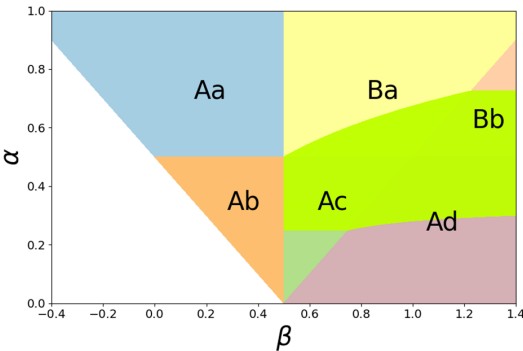

Figure 5: **Phase plane to compare signSGD and DANA-decaying in Ferbach et al. (2025).** Lime green area covering some part of Phase Ac, Ad, Ba, Bb is the area where signSGD has a steeper compute-optimal slope compared to DANA-decaying in Ferbach et al. (2025).

## C  EXPERIMENTS

### C.1  EXPLANATION FOR FIGURE 1.

**Parameters.** Left parameters: $(\alpha, \beta) = (0.4, 0.8)$, $\gamma_0 = 0.006$, $e^* = 1.0$ for signSGD, $e^* = 0.4571$ for SGD, 20 runs. Right parameters: $(\alpha, \beta) = (1.0, 0)$, $\gamma_0 = 0.002$ for both, $e^* = 1.0$ for constant, $e^* = 0.833$, $c = 0.091$, $w = 0.05$, $p = 0.9$, $\tau = 1$ for warmup-stable-decay, 10 runs.

**Takeaways.** In Figure 1, the left panel demonstrates the steeper compute-optimal slope of signSGD for $(\alpha, \beta) = (0.4, 0.8)$ in the area of Phase Ac. The right panel shows the increase in compute-optimal slope achieved by warmup-stable-decay scheduling for $(\alpha, \beta) = (1.0, 0)$. The theoretical and experimental compute-optimal slopes agree within errors of $0.04$ (left) and $0.01$ (right), which are well within the error margins reported in prior works.

Additionally, Figure 6 demonstrates the steeper compute-optimal slope of signSGD for $(\alpha, \beta) = (0.4, 1.0)$ in the Phase Ad and $(\alpha, \beta) = (0.7, 1.1)$ in Phase Ba.

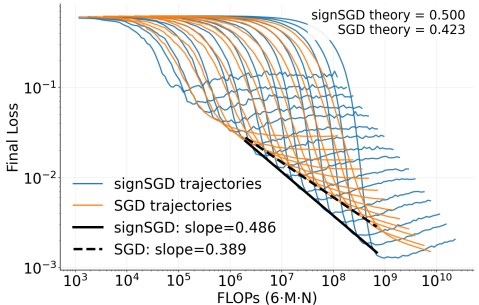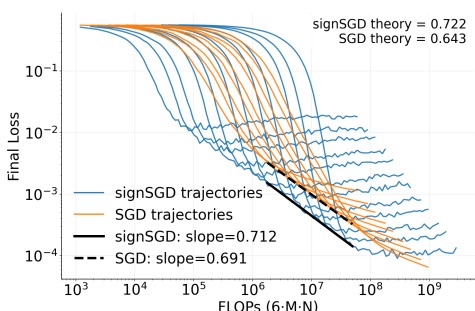

Figure 6: **comparison of SGD and signSGD on Compute-Optimal Scaling.** Colored lines represent the training trajectories of each algorithm, while black lines denote the compute-optimal curves. In both panels, the theoretical compute-optimal predictions closely follow the observed scaling. Both plots show that signSGD has a steeper compute-optimal slope than SGD. Left parameters: $(\alpha, \beta) = (0.4, 1.0)$, $\gamma_0 = 0.01$, $e^* = 1.0$ for signSGD, $e^* = 0.533$ for SGD, 5 runs. Right parameters: $(\alpha, \beta) = (0.7, 1.1)$, $\gamma_0 = 0.01$, $e^* = 1.09$ for signSGD, $e^* = 0$ for SGD, 20 runs.

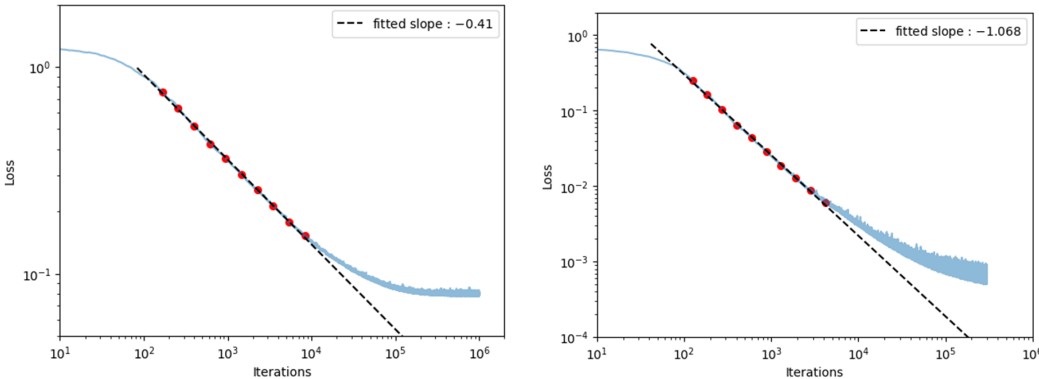

Figure 7: $\mathcal{D}_{\mathrm{al}}^{\mathrm{sign}}(M, N, \gamma_0)$ **term exponent.** Blue curves: true signSGD trajectories. Black dotted curves: linear fits over the early-iteration interval in log-log scale. Left: parameters $(\alpha, \beta) = (0.75, 0)$, $\gamma_0 = 0.0012$, $f(z) = 1$, $M = 200$, $d = 400$. The theoretical exponent is $-2(2\alpha + 2\beta - 1)/(2\alpha - 2\beta + 1) = -0.4$, which matches the experiment. Right: parameters $(\alpha, \beta) = (1.0, 0.2)$, $\gamma_0 = 0.0006$, $f(z) = 1$, $M = 400$, $d = 1600$. The theoretical exponent is $-2(2\alpha + 2\beta - 1)/(2\alpha - 2\beta + 1) = -1.077$, again consistent with the experiment.

## C.2   EXPERIMENT FOR ALIGNED DRIFT

In Figure 7, we examine the exponent of the $\mathcal{D}_{\mathrm{al}}^{\mathrm{sign}}(M, N, \gamma_0)$ term,

$$\left(M^{\min(\alpha, 0.5)} \gamma_0 N\right)^{-\frac{2(2\alpha+2\beta-1)}{2\alpha-2\beta+1}},$$

of signSGD. For the Phase Aa, the $\mathcal{D}_{\mathrm{al}}^{\mathrm{sign}}(M, N, \gamma_0)$ term dominates in the early iterations over a sufficient interval, allowing us to evaluate the exponent by line fitting on a log-log plot. The experimental results align well with the theoretical formula $-\frac{2(2\alpha+2\beta-1)}{2\alpha-2\beta+1}$.

## C.3   VALIDATION OF THE TABLE 1

Figures 8 through 12 validates the exponent in Table 1 for various $(\alpha, \beta)$. On the left plots, we draw multiple curves with different model size $M$ while setting the learning rate as $\gamma_0 = M^{-e^*}$. Then the lower envelope becomes the compute-optimal curve, and by measuring the slope in a log-log plot, we can validate the compute-optimal loss exponent in the Table 1. On the right plots, we draw the

optimal model size at each FLOPS. Here, the optimal model size is the model size of the curve that meets the lower envelope at that FLOPS. By measuring the slope in a log-log plot, we can validate the optimal model size exponent in the Table 1. Note that we use a similar experimental setting to Paquette et al. (2024). In most cases, the error between the measured exponent and the theoretical exponent was less than $0.04$, and the error was less than $0.06$ even for the worst case. This error lies within the error margins reported in prior works (Paquette et al., 2024; Ferbach et al., 2025).

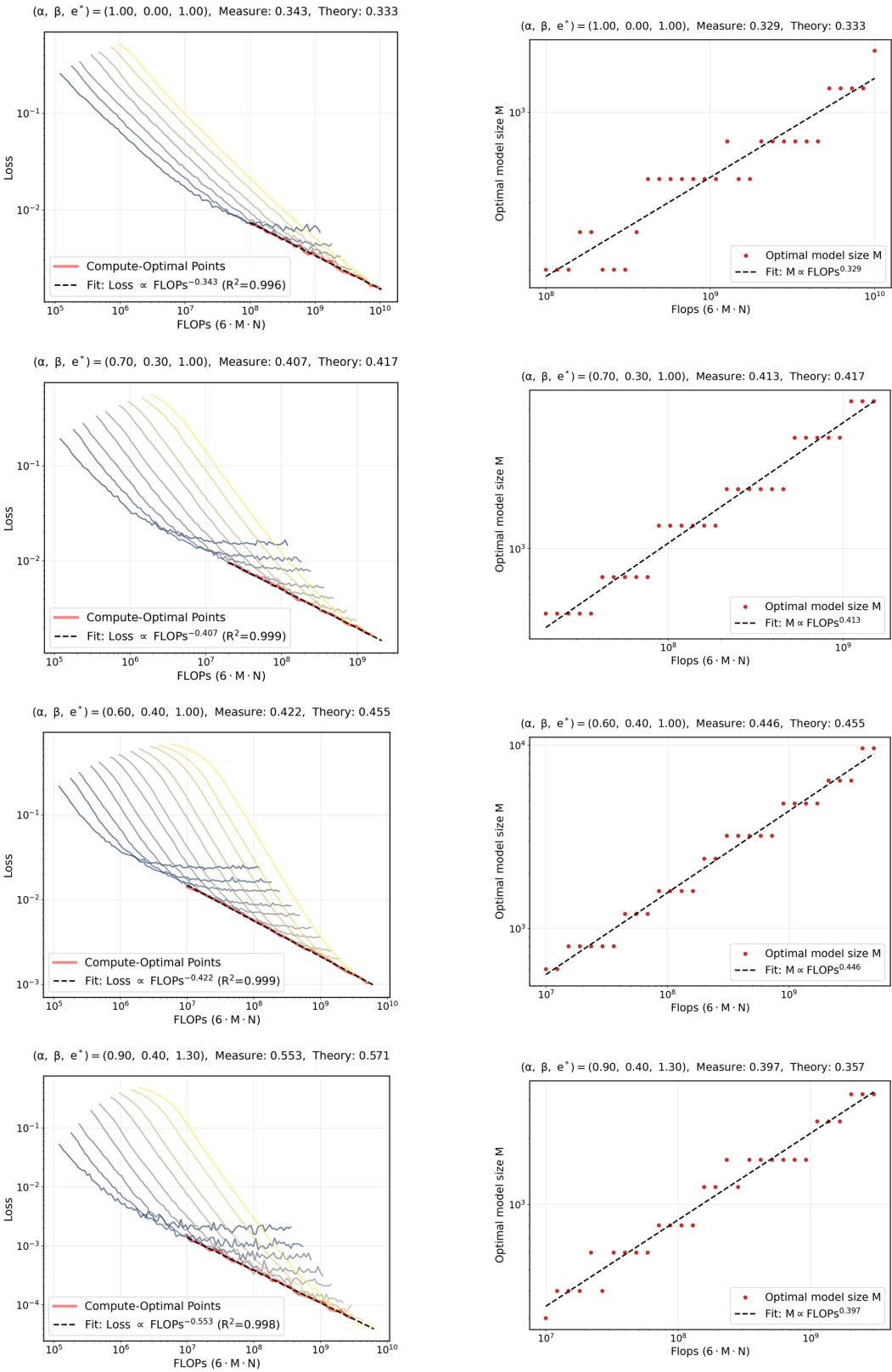

Figure 8: **Measure of compute-optimal loss slope and optimal model size slope.** We validate the exponent of $R\left(M^\star, \frac{\mathfrak{f}}{M^\star}, \gamma_0^\star\right)$ and $M^\star$ with respect to $\mathfrak{f}$ in the Table 1. The left plot shows the compute-optimal loss with respect to FLOPS $6MN$. The right plot shows the optimal model size with respect to FLOPS $6MN$. Each plot includes the measured slope and the theoretical slope from the Table 1.

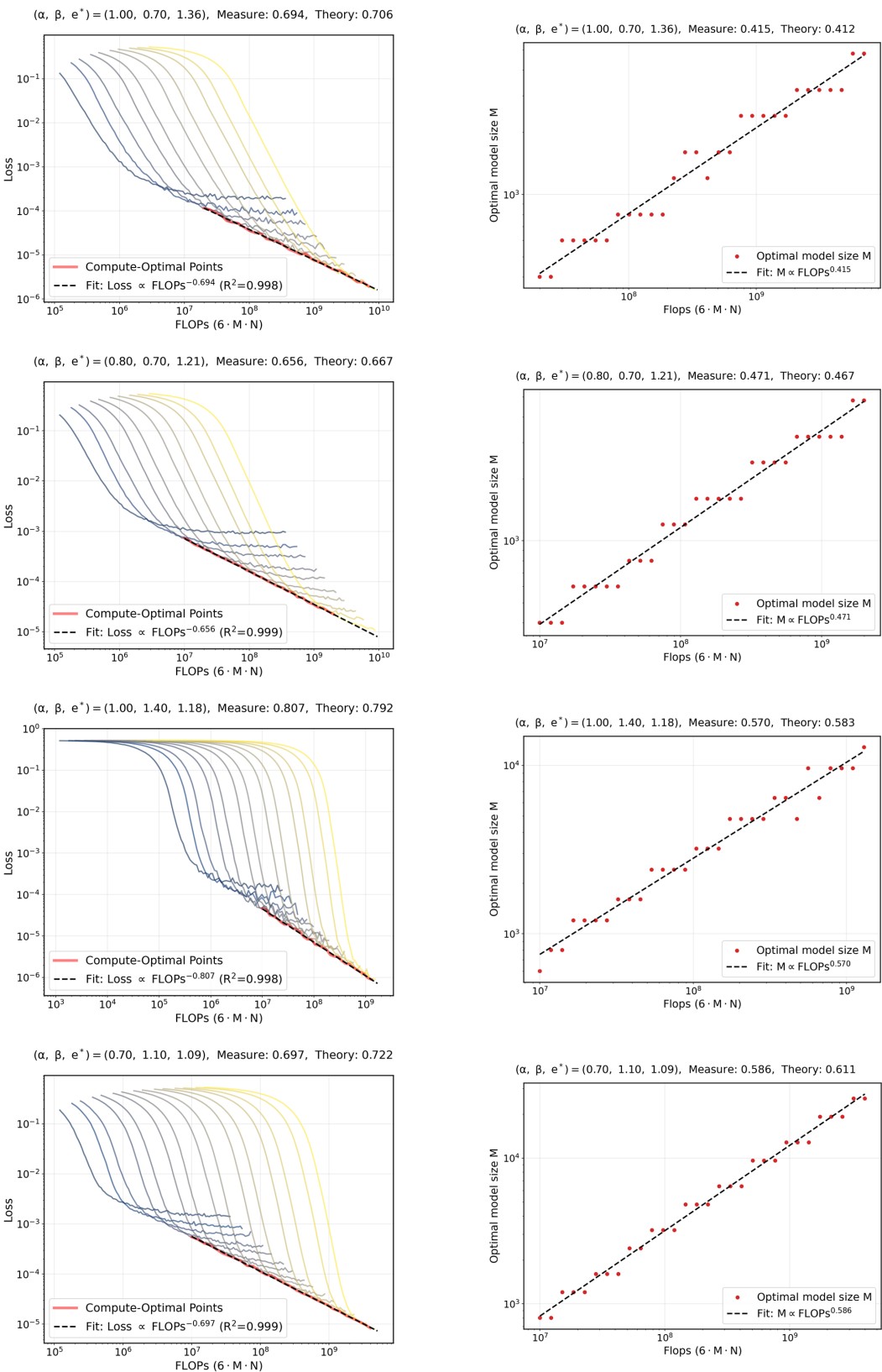

Figure 9: **Measure of compute-optimal loss slope and optimal model size slope.** We validate the exponent of $R\left(M^{\star}, \frac{\mathfrak{f}}{M^{\star}}, \gamma_0^{\star}\right)$ and $M^{\star}$ with respect to $\mathfrak{f}$ in the Table 1. The left plot shows the compute-optimal loss with respect to FLOPS $6MN$. The right plot shows the optimal model size with respect to FLOPS $6MN$.

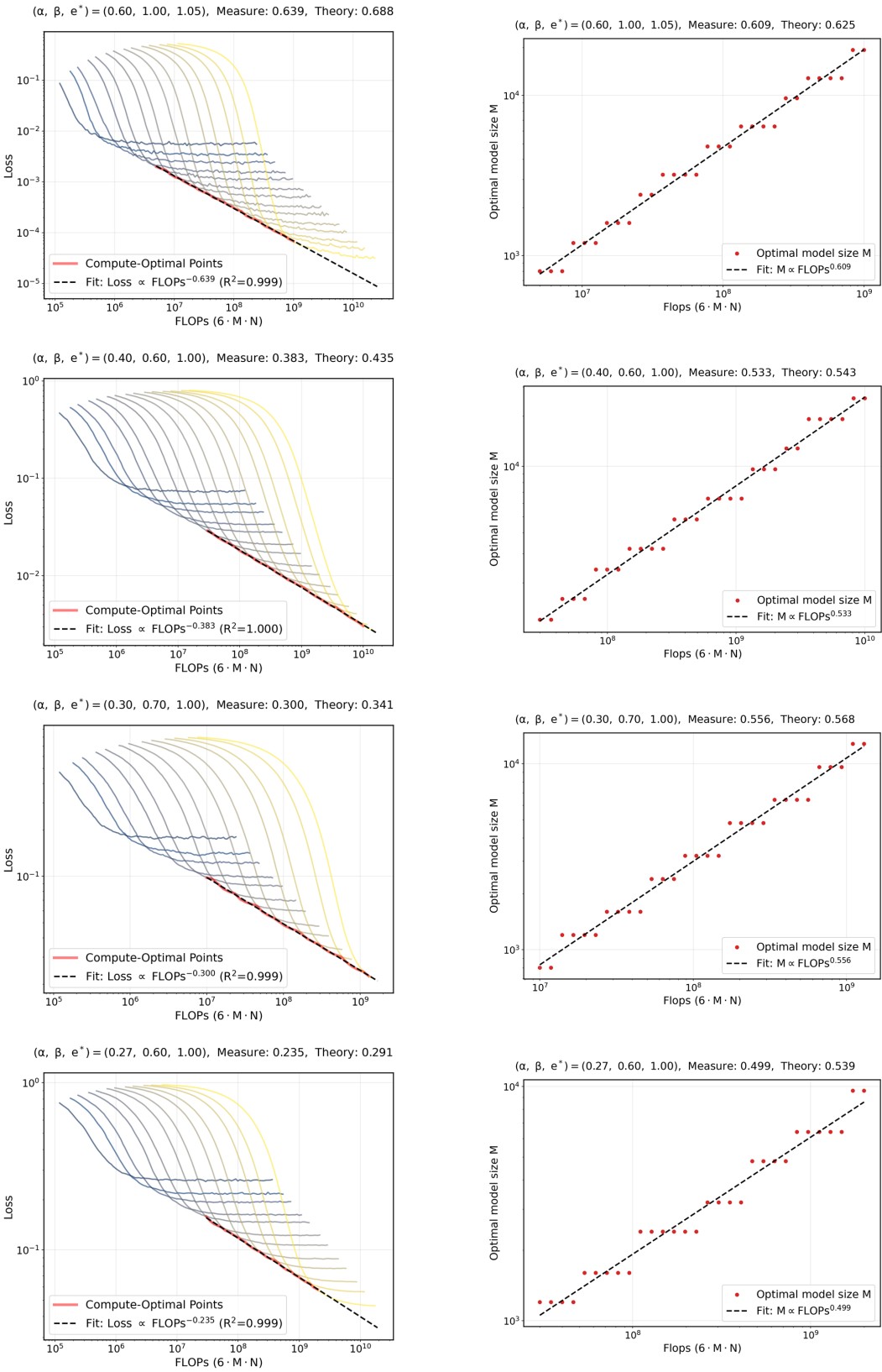

Figure 10: **Measure of compute-optimal loss slope and optimal model size slope.** We validate the exponent of $R\left(M^\star, \frac{\mathfrak{f}}{M^\star}, \gamma_0^\star\right)$ and $M^\star$ with respect to $\mathfrak{f}$ in the Table 1. The left plot shows the compute-optimal loss with respect to FLOPS $6MN$. The right plot shows the optimal model size with respect to FLOPS $6MN$.

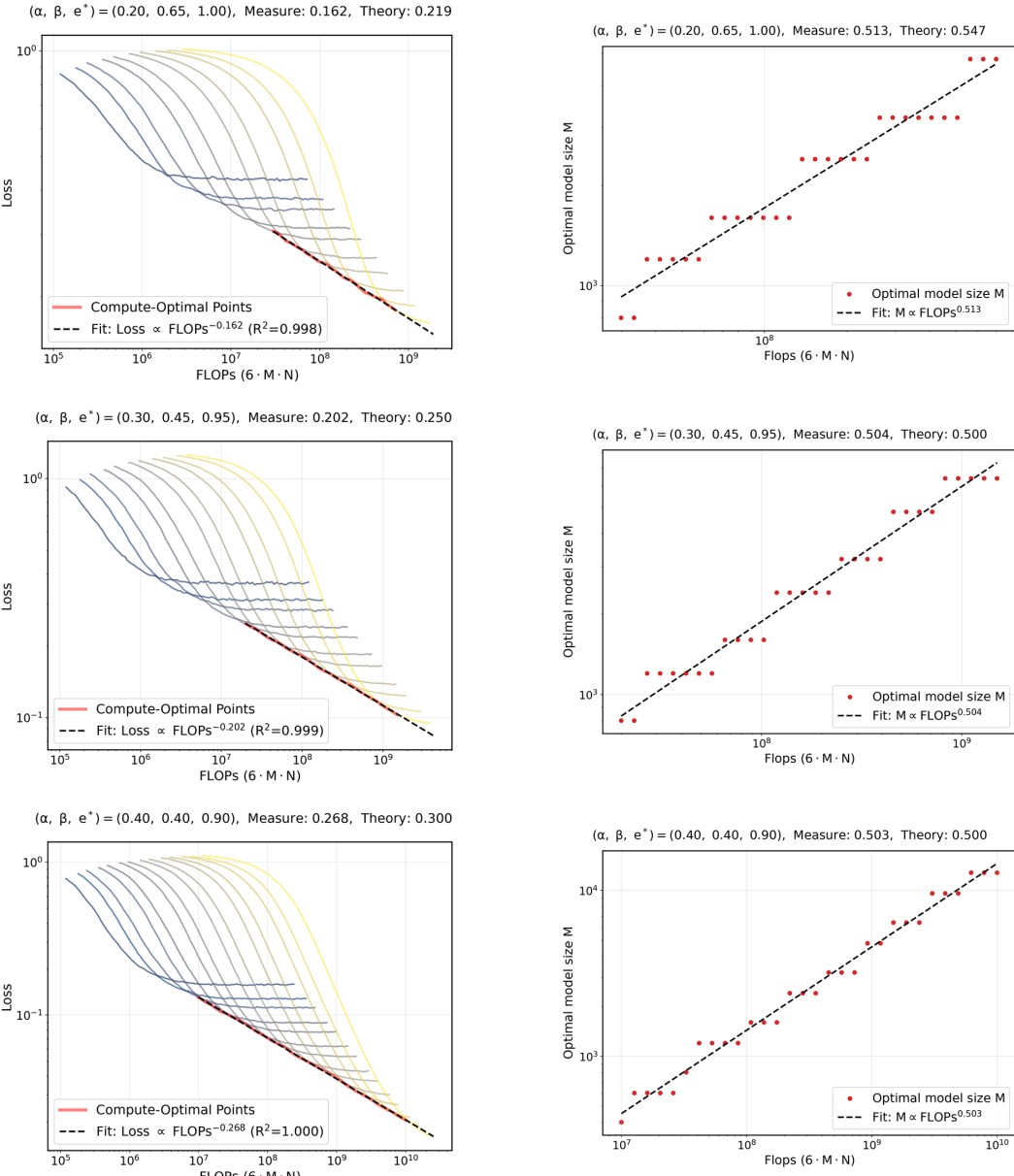

Figure 11: **Measure of compute-optimal loss slope and optimal model size slope.** We validate the exponent of $R\left(M^\star, \frac{\mathfrak{f}}{M^\star}, \gamma_0^\star\right)$ and $M^\star$ with respect to $\mathfrak{f}$ in the Table 1. The left plot shows the compute-optimal loss with respect to FLOPS $6MN$. The right plot shows the optimal model size with respect to FLOPS $6MN$. Each plot includes the measured slope and the theoretical slope from the Table 1.

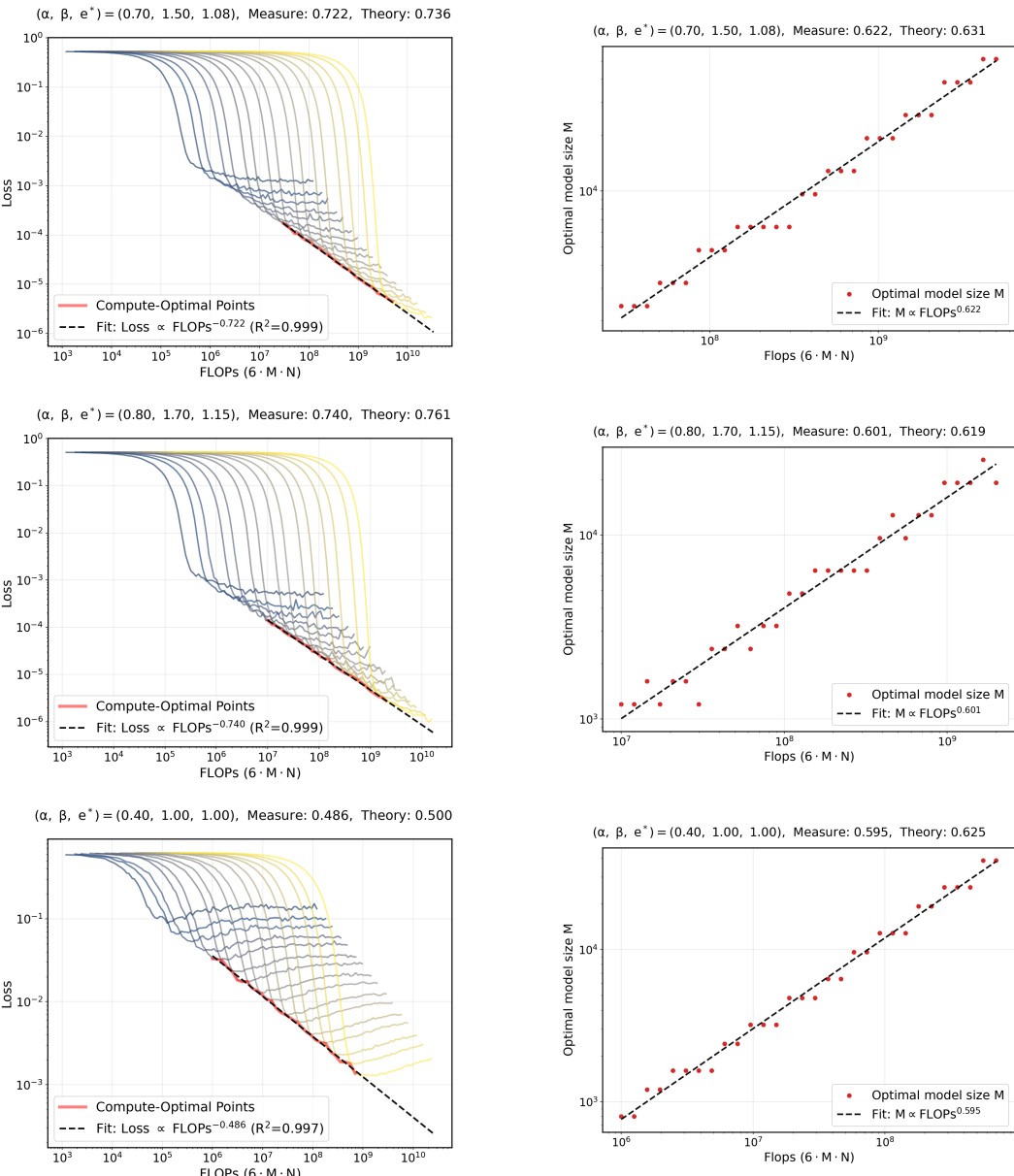

Figure 12: **Measure of compute-optimal loss slope and optimal model size slope.** We validate the exponent of $R\left(M^\star, \frac{\mathfrak{f}}{M^\star}, \gamma_0^\star\right)$ and $M^\star$ with respect to $\mathfrak{f}$ in the Table 1. The left plot shows the compute-optimal loss with respect to FLOPS $6MN$. The right plot shows the optimal model size with respect to FLOPS $6MN$. Each plot includes the measured slope and the theoretical slope from the Table 1.

## C.4 EXPERIMENT FOR MINIBATCHING

In this subsection, we provide an experiment with batch sizes 10 and 128. Figures 13 and 14 show the measured compute-optimal loss slope and optimal model size slope for batch sizes 10 and 128, respectively. The theory slope in the figure is the theory value for batch size 1. We can see that the difference between the measured value for batch sizes 10 and 128 and the theoretical value for batch size 1 is less than or equal to 0.042. Therefore, we conjecture that mini-batching with a constant-order batch size has the same compute-optimal exponents as the batch size 1 case; this is plausible because constant factors in the loss formula are ignored in the exponent analysis. Mathematically analyzing mini-batch signSGD is an important direction for research, which we leave for future work.

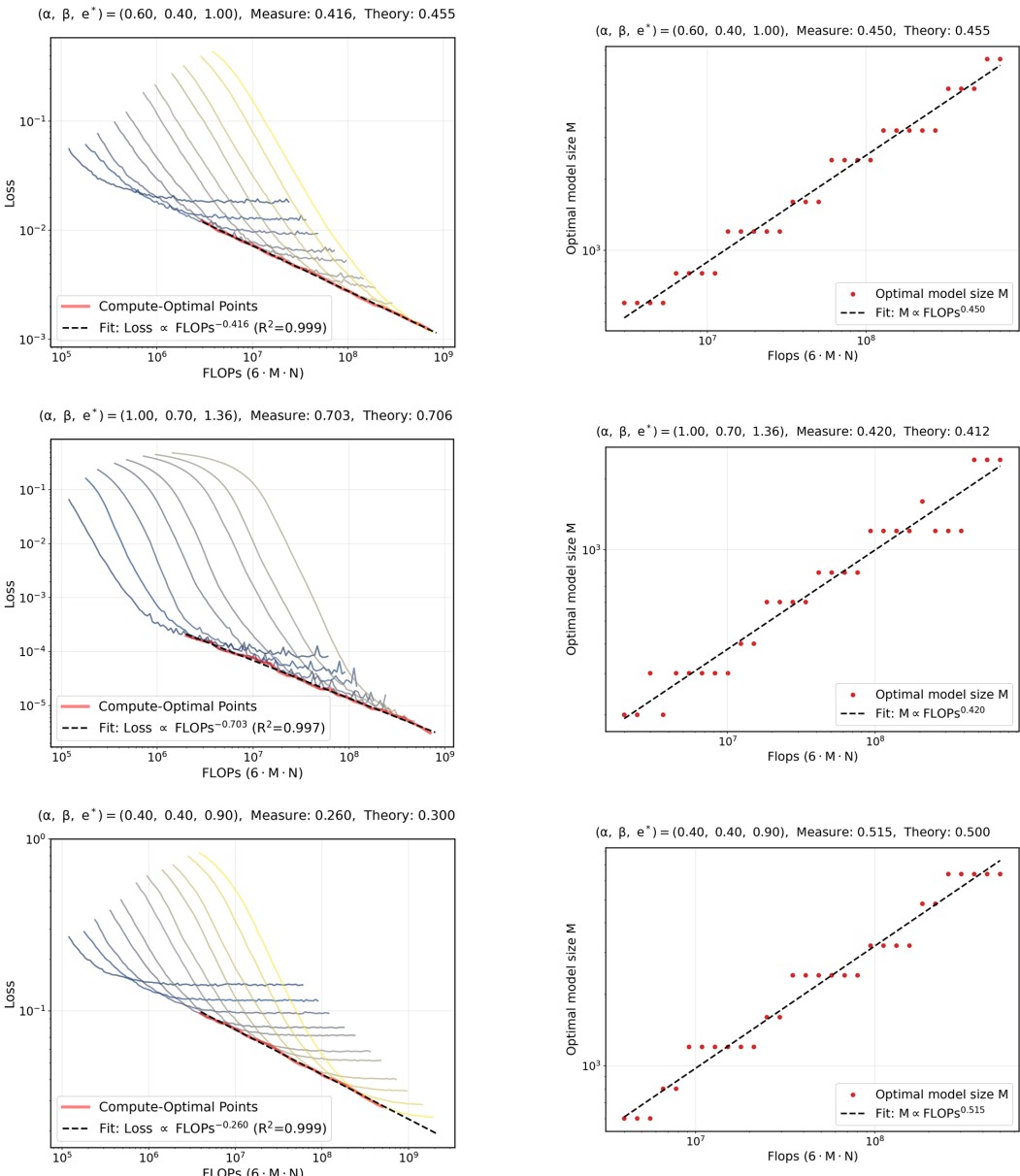

Figure 13: **Measure of compute-optimal loss slope and optimal model size slope for batch size 10.** We calculate the exponent of $R\left(M^{\star}, \frac{\mathfrak{f}}{M^{\star}}, \gamma_0^{\star}\right)$ and $M^{\star}$ with respect to $\mathfrak{f}$. The left plot shows the compute-optimal loss with respect to FLOPS $6MN$. The right plot shows the optimal model size with respect to FLOPS $6MN$. Each plot includes the measured slope and the theoretical slope for the batch size 1 case.

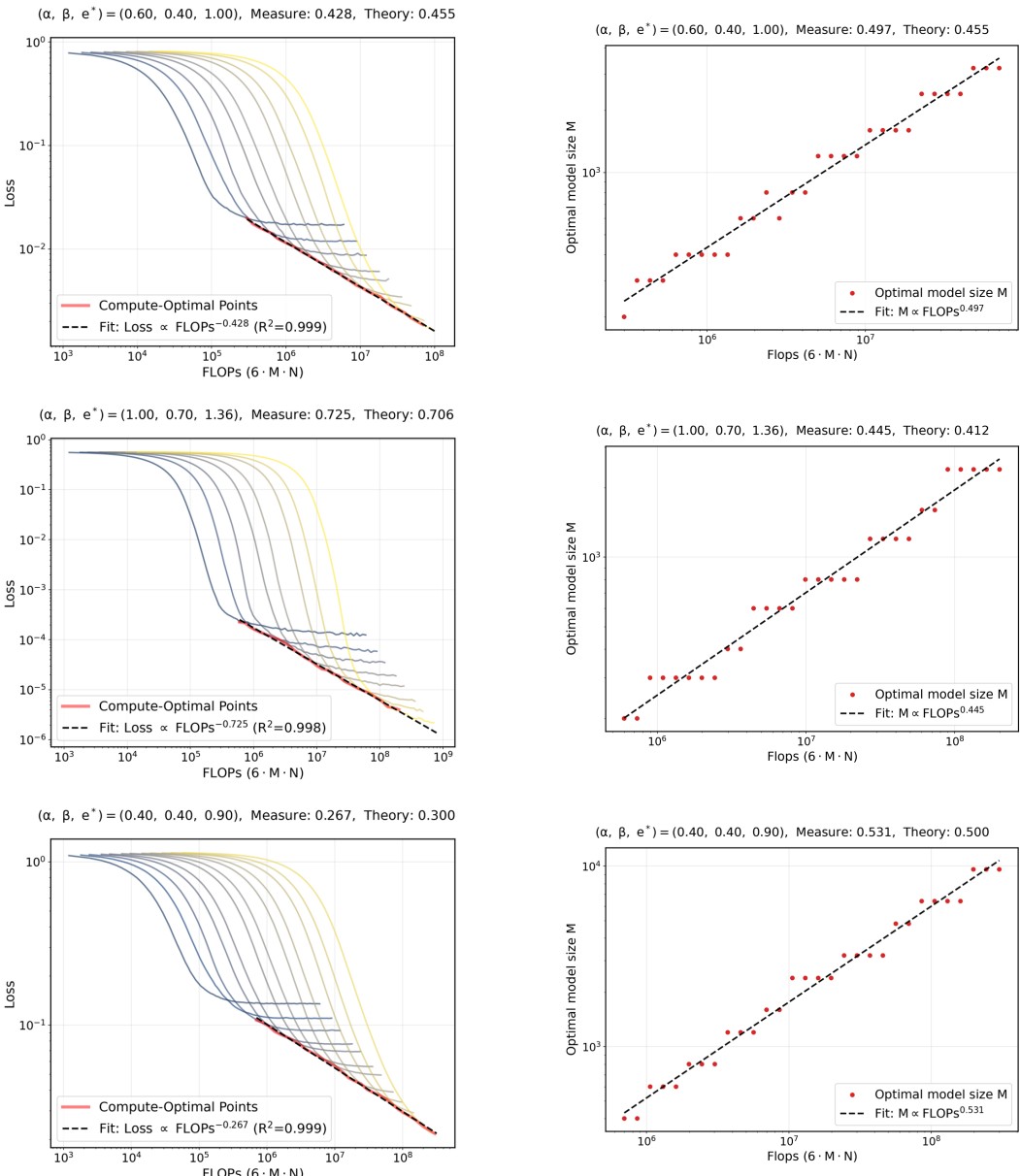

Figure 14: **Measure of compute-optimal loss slope and optimal model size slope for batch size 128.** We calculate the exponent of $R\left(M^\star, \frac{\mathfrak{f}}{M^\star}, \gamma_0^\star\right)$ and $M^\star$ with respect to $\mathfrak{f}$. The left plot shows the compute-optimal loss with respect to FLOPS $6MN$. The right plot shows the optimal model size with respect to FLOPS $6MN$. Each plot includes the measured slope and the theoretical slope for the batch size 1 case.

## C.5 Experiment of AdamW and SGD with Transformer

### C.5.1 Compute-optimal Exponent

We calculated the loss decaying exponent with respect to the compute for AdamW (Loshchilov & Hutter, 2019) and SGD optimizer on the Transformer architecture (Vaswani et al., 2017). We conducted an experiment based on the GitHub code of Shehper (2025). In our experiment, we evaluated five different model sizes: $(\text{number of layers}, \text{embedding dimension}) = (4, 64), (8, 64), (8, 96), (8, 128), (8, 160)$. We used a constant learning rate and gradient clipping with 1.0 for both AdamW and SGD. We set $\beta_1 = 0.9$, $\beta_2 = 0.95$ for AdamW. We trained for $10^5$ steps for each run. We set both batch size and gradient accumulation steps as 1. We set dropout as 0.1, and set weight decay as 0.1. We used 1024 tokens per iteration. Amount of compute is calculated by $6 \times (\text{number of model parameters}) \times (\text{iterations}) \times (\text{tokens per iteration})$. The validation loss is a cross-entropy loss with 200 sets of 1024 tokens. We used the OpenWebText dataset (EleutherAI, 2024) for training.

Figure 15 shows that the exponent of AdamW is -0.021 and the exponent of SGD is -0.005. It means AdamW has better compute-optimal scaling compared to SGD in this experiment. Our experiment implies that a practical optimizer, AdamW, on a practical deep network, Transformer, can have a better compute-optimal exponent compared to SGD. Although our analysis is about signSGD—studied as an approximate surrogate of Adam and its variants—and a simple linear model, our experiment implies that an advantage in the compute-optimal scaling aspect may also occur in a practical optimizer, AdamW, with a deep neural network Transformer.

### C.5.2 Drift-normalization Effect and Noise-reshaping Effect

To observe the drift-normalization effect, we experimented with a batch size of 16 and gradient accumulation steps of 32 to decrease the noise term. As the loss curve is the sum of the drift term, noise term, and approximation term, decreasing the noise term allows us to observe the drift-normalization effect more clearly. We experimented for $(\text{number of layers}, \text{embedding dimension}) = (8, 96)$ for each AdamW and SGD. Other experimental settings are the same as the section C.5.1. In Figure 16, we measure the slope of the loss curve in a log-log plot for the linear decaying interval, where the drift term is dominant. We can observe that the slope for AdamW is larger than SGD, and this is consistent with the drift-normalization effect in PLRF, which increased the exponent of the drift term in signSGD compared to SGD.

To observe the noise-reshaping effect, we focus on the plateau regime of the batch size 1 experiment. To see how the loss value of the plateau regime is influenced by the size of the learning rate, we experiment with two learning rate values: 0.00266 and 0.00133 for both AdamW and SGD. We experimented for $(\text{number of layers}, \text{embedding dimension}) = (8, 96)$ for each AdamW and SGD. Other experimental settings are the same as the section C.5.1, including batch size 1 and gradient accumulation steps 1. In Figure 17, we can see that the loss value at the plateau regime, which is dominated by the noise term, increases for AdamW when we take a bigger learning rate, but does not increase for SGD. This is consistent with the noise-reshaping effect in PLRF, which made the size of the noise term in signSGD increase as we take a larger learning rate, in contrast to SGD.

### C.6 Other Synthetic Task Experiment

We experimented with feature learning based on the setting of Bordelon et al. (2025). In the feature learning, the sketch matrix $\boldsymbol{S}$ becomes learnable, in contrast to the fixed Gaussian sketch setting of the PLRF model. We let $\boldsymbol{S} = \boldsymbol{B}(t)\boldsymbol{S}_0$, where $\boldsymbol{B}(t)$ is $M \times M$ square matrix and $\boldsymbol{B}(0) = I$. During the training, we update the square matrix $\boldsymbol{B}(t)$ at each time step with the optimizer. Other settings, except for this learnable sketch matrix, are the same as the settings for PLRF.

Figure 18 shows our evaluation of the compute-optimal slope for Adam, signGD (full-batch sign descent; deterministic signSGD), and GD in the feature learning setting. We experimented with a full batch due to the training instability of small batch cases. We experimented for the parameter $(\alpha, \beta) = (1.0, 1.25)$ which is included in the Area III-IV$_{\text{sub}}$. In this feature learning experiment, Adam and signGD had similar slopes, and those two had a steeper slope compared to GD. The result is consistent with the phenomena in PLRF that signSGD has a steeper compute-optimal slope

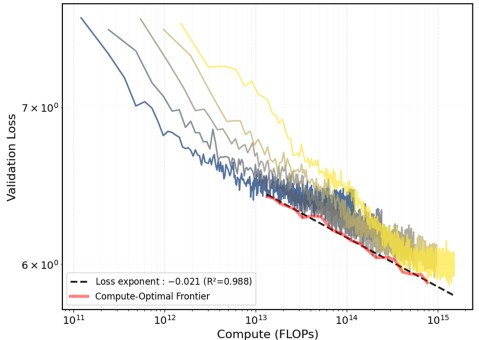 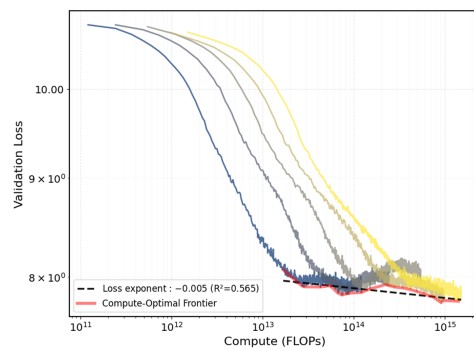

Figure 15: **Measure of compute-optimal loss slope for AdamW and SGD on Transformer architecture. Left: AdamW, Right: SGD** The x-axis shows the amount of compute calculated by $6 \times$ (number of model parameters) $\times$ (iterations) $\times$ (tokens per iteration). The y-axis shows the validation loss, which is a cross-entropy loss with 200 sets of 1024 tokens.

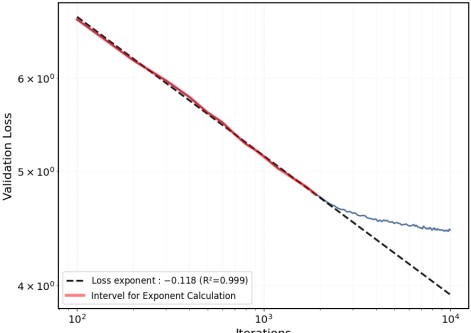 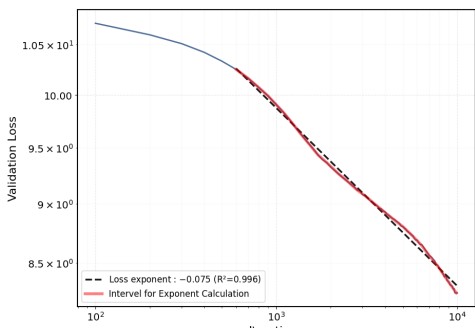

Figure 16: **Measure of drift term slope for AdamW and SGD on Transformer architecture. Left: AdamW, Right: SGD** The x-axis shows the iterations. The y-axis shows the validation loss, which is a cross-entropy loss with 200 sets of 1024 tokens.

compared to SGD in the Area III-IV$_{sub}$, and also consistent with the conjecture in PLRF that Adam has the same compute-optimal slope as signSGD.

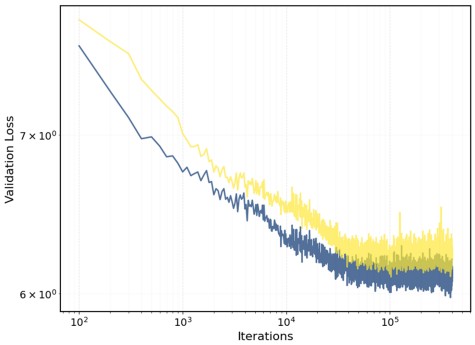 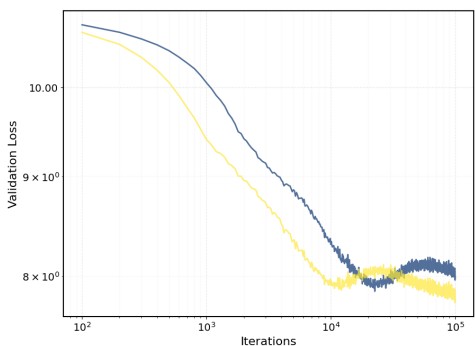

Figure 17: **Plateau loss value for two different learning rate. Left: AdamW, Right: SGD** The blue curve is the trajectory with a learning rate of 0.00133, and the yellow curve is the trajectory with a learning rate of 0.00266. The x-axis shows the iterations. The y-axis shows the validation loss, which is a cross-entropy loss with 200 sets of 1024 tokens.

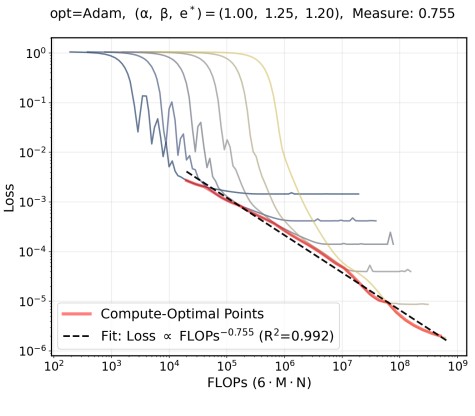 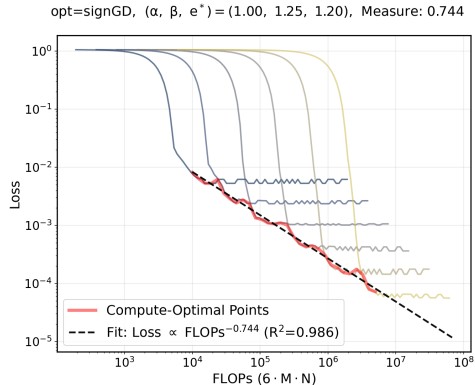

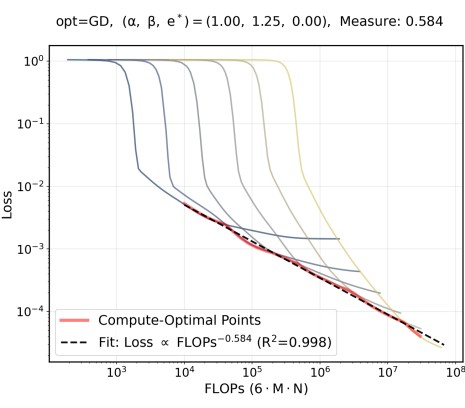

Figure 18: **Compute-optimal exponent for feature learning. Left upper: Adam, Right upper: signGD, Left lower: GD.** The x-axis shows the FLOPS. The y-axis shows the loss value. We experimented for $(\alpha, \beta) = (1.0, 1.25)$. We set $e$ of $\gamma_0 = M^{-e}$ as the optimal value derived in PLRF. We use a constant learning rate. Dimension before projection is 2000, and each loss curve is experimented with projected dimensions 32, 64, 128, 256, 512, 1024.

## D    EQUIVALENCE TO DIAGONAL COVARIANCE $H$

In this section, we will prove that general covariance $\boldsymbol{H}$ with eigenvalues $1^{-2\alpha}, 2^{-2\alpha}, \ldots, d^{-2\alpha}$ can be reduced to diagonal covariance $\mathrm{diag}(1^{-2\alpha}, 2^{-2\alpha}, \ldots, d^{-2\alpha})$. Thereby, for the following sections, we will assume $\boldsymbol{H} = \mathrm{diag}(1^{-2\alpha}, 2^{-2\alpha}, \ldots, d^{-2\alpha})$ without loss of generality.

Recall that we assume $\langle \boldsymbol{v}_i, \boldsymbol{w}^* \rangle = i^{-\beta}$ where $\boldsymbol{v}_i$ is a eigenvector of $\boldsymbol{H}$ corresponding to eigenvalue $i^{-2\alpha}$ for $i = 1, \ldots, d$.

Let $\boldsymbol{D} = \mathrm{diag}(1^{-2\alpha}, 2^{-2\alpha}, \ldots, d^{-2\alpha})$. Then $\boldsymbol{H} = \boldsymbol{U}\boldsymbol{D}\boldsymbol{U}^\mathsf{T}$ holds for some orthogonal matrix $\boldsymbol{U}$ by the eigenvalue decomposition. The $i$-th column of $\boldsymbol{U}$ can be thought as $\boldsymbol{v}_i$. Then the following holds for $\boldsymbol{w}_0^* = [1^{-\beta}, 2^{-\beta}, \ldots, d^{-\beta}]^\mathsf{T}$.

$$\boldsymbol{w}^* = \sum_{i=1}^{d} i^{-\beta} \cdot \boldsymbol{v}_i = \boldsymbol{U}\boldsymbol{w}_0^*.$$

The signSGD update rule is

$$\boldsymbol{\theta}_{k+1} = \boldsymbol{\theta}_k - \gamma_k \, \mathrm{sign}\left(\langle \boldsymbol{S}\boldsymbol{x}_k, \boldsymbol{\theta}_k \rangle - y_k\right) \mathrm{sign}(\boldsymbol{S}\boldsymbol{x}_k).$$

With label assumption $y_k = \langle \boldsymbol{x}_k, w^* \rangle$, the signSGD update rule converts to

$$\boldsymbol{\theta}_{k+1} = \boldsymbol{\theta}_k - \gamma_k \, \mathrm{sign}\left(\langle \boldsymbol{S}\boldsymbol{x}_k, \boldsymbol{\theta}_k \rangle - \langle \boldsymbol{x}_k, w^* \rangle\right) \mathrm{sign}(\boldsymbol{S}\boldsymbol{x}_k).$$

We let $\boldsymbol{x}_k' = \boldsymbol{U}^\mathsf{T}\boldsymbol{x}_t$. By substituting $\boldsymbol{x}_k = \boldsymbol{U}\boldsymbol{x}_t'$ and $\boldsymbol{w}^* = \boldsymbol{U}\boldsymbol{w}_0^*$, we get

$$\boldsymbol{\theta}_{k+1} = \boldsymbol{\theta}_k - \gamma_k \, \mathrm{sign}\left(\langle \boldsymbol{S}\boldsymbol{U}\boldsymbol{x}_k', \boldsymbol{\theta}_k \rangle - \langle \boldsymbol{U}\boldsymbol{x}_k', \boldsymbol{U}\boldsymbol{w}_0^* \rangle\right) \mathrm{sign}(\boldsymbol{S}\boldsymbol{U}\boldsymbol{x}_k').$$

As $\boldsymbol{U}$ is orthogonal, it leads to

$$\boldsymbol{\theta}_{k+1} = \boldsymbol{\theta}_k - \gamma_k \, \mathrm{sign}\left(\langle \boldsymbol{S}\boldsymbol{U}\boldsymbol{x}_k', \boldsymbol{\theta}_k \rangle - \langle \boldsymbol{x}_k', \boldsymbol{w}_0^* \rangle\right) \mathrm{sign}(\boldsymbol{S}\boldsymbol{U}\boldsymbol{x}_k'). \tag{16}$$

Also, the loss formula

$$L(\boldsymbol{\theta}) = \|\boldsymbol{H}^{1/2}(\boldsymbol{S}^\mathsf{T}\boldsymbol{\theta} - \boldsymbol{w}^*)\|^2 = (\boldsymbol{S}^\mathsf{T}\boldsymbol{\theta} - \boldsymbol{w}^*)^\mathsf{T}\boldsymbol{H}(\boldsymbol{S}^\mathsf{T}\boldsymbol{\theta} - \boldsymbol{w}^*)$$

converts to

$$L(\boldsymbol{\theta}) = (\boldsymbol{S}^\mathsf{T}\boldsymbol{\theta} - \boldsymbol{U}\boldsymbol{w}_0^*)^\mathsf{T}\boldsymbol{U}\boldsymbol{D}\boldsymbol{U}^\mathsf{T}(\boldsymbol{S}^\mathsf{T}\boldsymbol{\theta} - \boldsymbol{U}\boldsymbol{w}_0^*) = ((\boldsymbol{S}\boldsymbol{U})^\mathsf{T}\boldsymbol{\theta} - \boldsymbol{w}_0^*)^\mathsf{T}\boldsymbol{D}((\boldsymbol{S}\boldsymbol{U})^\mathsf{T}\boldsymbol{\theta} - \boldsymbol{w}_0^*).$$

Now the covariance of $\boldsymbol{x}_k'$ is $\boldsymbol{D} = \mathrm{diag}(1^{-2\alpha}, 2^{-2\alpha}, \ldots, d^{-2\alpha})$ and target $\boldsymbol{w}_0^* = [1^{-\beta}, 2^{-\beta}, \ldots, d^{-\beta}]^\mathsf{T}$ is same with the diagonal covariance case. Lastly, the distribution of $\boldsymbol{S}\boldsymbol{U}$ is identical to the distribution of $\boldsymbol{S}$. This is because each row $\boldsymbol{s}_i$ of $\boldsymbol{S}$ follows the distribution $\mathcal{N}(0, \boldsymbol{I}_d/M)$, and $\boldsymbol{s}_i\boldsymbol{U}$, which is each row of $\boldsymbol{S}\boldsymbol{U}$, follows the distribution $\mathcal{N}(0, \boldsymbol{U}^\mathsf{T}\boldsymbol{I}_d\boldsymbol{U}/M) = \mathcal{N}(0, \boldsymbol{I}_d/M)$. Also note that $\boldsymbol{s}_i$s are independent and $\boldsymbol{s}_i\boldsymbol{U}$s are independent.

So the converted update rule (16) is equivalent to the case with diagonal covariance $\mathrm{diag}(1^{-2\alpha}, 2^{-2\alpha}, \ldots, d^{-2\alpha})$.

# E   DERIVATION OF THE SCALING LAW FORMULA $R(M, N, \gamma_0)$

**Goal.**   In this section, our goal is to derive the scaling law formula (12) of $R(M, N, \gamma_0)$. On the area $\alpha < 0.5$ or $\beta < 0.5$ with $-\alpha + 0.5 < \beta < \alpha + 0.5$, $\mathcal{D}_{\mathrm{dis}}^{\mathrm{sign}}(M, N, \gamma_0)$ term is smaller than at least one of the other three terms. So it is enough to show

$$R(M, N, \gamma_0) \asymp \underbrace{M^{-2\alpha+\max(0, 1-2\beta)}}_{=:\mathcal{A}(M)} + \underbrace{\left(M^{\min(\alpha, 0.5)} N\gamma_0\right)^{-\frac{2(2\alpha+2\beta-1)}{2\alpha-2\beta+1}}}_{=:\mathcal{D}_{\mathrm{al}}^{\mathrm{sign}}(M,N,\gamma_0)} + \underbrace{\gamma_0^2 M^{2-\min(1,2\alpha)}}_{=:\mathcal{N}^{\mathrm{sign}}(M,\gamma_0)}.$$

for that area.

For the area $\alpha > 0.5$ and $\beta > 0.5$ with $-\alpha + 0.5 < \beta < \alpha + 0.5$, as all four terms are dominant, we will prove

$$R(M, N, \gamma_0) \asymp \underbrace{M^{-2\alpha+\max(0, 1-2\beta)}}_{=:\mathcal{A}(M)} + \underbrace{\left(M^{\min(\alpha, 0.5)} N\gamma_0\right)^{-\frac{2(2\alpha+2\beta-1)}{2\alpha-2\beta+1}}}_{=:\mathcal{D}_{\mathrm{al}}^{\mathrm{sign}}(M,N,\gamma_0)}$$

$$+ \underbrace{M^{-\frac{6\alpha-1}{2\alpha+1}}\left(N\gamma_0\right)^{-\frac{2(2\alpha-1)}{2\alpha+1}}}_{=:\mathcal{D}_{\mathrm{dis}}^{\mathrm{sign}}(M,N,\gamma_0)} + \underbrace{\gamma_0^2 M^{2-\min(1,2\alpha)}}_{=:\mathcal{N}^{\mathrm{sign}}(M,\gamma_0)}.$$

**Proof Overview.**   As a first step, we obtain the ODE

$$\frac{dp_i}{dt} = -\frac{4}{\pi\sqrt{P(t)}} \lambda_i(\overline{\boldsymbol{K}}) f(t/\gamma_0) p_i(t) + \frac{2f(t/\gamma_0)^2\gamma_0}{\pi} V_i. \tag{17}$$

where $P(t) = L(t/\gamma_0)$ and $p_i(t) = r_i(t/\gamma_0)$.

Then we derive the following integral equation from the ODE.

$$L(N) = \|\boldsymbol{H}^{1/2}\boldsymbol{w}_\perp\|^2 + \sum_{i=1}^{M} r_i(0)\, e^{-\frac{4\lambda_i\gamma_0}{\pi}\int_0^N \frac{f(u)}{\sqrt{L(u)}}\,du} + \frac{2\gamma_0^2}{\pi}\sum_{i=1}^{M} V_i \int_0^N e^{-\frac{4\lambda_i\gamma_0}{\pi}\int_z^N \frac{f(u)}{\sqrt{L(u)}}\,du} f(z)^2\,dz. \tag{18}$$

Going through the arguments, including the contour integral, our integral equation converts to the following equation, where $Q(z) = \frac{4\gamma_0}{\pi}\int_0^z \frac{f(u)}{\sqrt{L(u)}}\,du$.

$$L(N) \asymp \underbrace{M^{-2\alpha+\max(0,1-2\beta)}}_{\textbf{approx}} + \underbrace{\left(M^{\min(\alpha,\,0.5)} Q(N)\right)^{-\frac{2\alpha+2\beta-1}{2\alpha}}}_{\textbf{drift}} \tag{19}$$

$$+ \underbrace{\frac{2\gamma_0^2}{\pi}\sum_{i=1}^{M} V_i \int_0^N \exp\!\left(-\frac{4\gamma_0}{\pi}\lambda_i(\overline{K})\int_z^N \frac{du}{\sqrt{L(u)}}\right) dz}_{\textbf{noise}}. \tag{20}$$

for $\alpha < 0.5$ or $\beta < 0.5$, and

$$L(N) \asymp \underbrace{M^{-2\alpha}}_{\textbf{approx}} + \underbrace{\left(M^{1/2}Q(N)\right)^{-\frac{2\alpha+2\beta-1}{2\alpha}}}_{\text{drift}_1} + \underbrace{M^{-1}\left(M^{1/2}Q(N)\right)^{-1+\frac{1}{2\alpha}}}_{\text{drift}_2} \tag{21}$$

$$+ \underbrace{\frac{2\gamma_0^2}{\pi}\sum_{i=1}^{M} V_i \int_0^N \exp\!\left(-\frac{4\gamma_0}{\pi}\lambda_i(\overline{K})\int_z^N \frac{du}{\sqrt{L(u)}}\right) dz}_{\textbf{noise}}, \tag{22}$$

for $\alpha > 0.5$ and $\beta > 0.5$.

Solving the early stage and the limit stage separately, we get the following proxy for $\alpha < 0.5$ or $\beta < 0.5$.

$$L_{\mathrm{px}}(N) := \left(\gamma_0 M^{\min(\alpha,\,0.5)} N\right)^{-p} + \underbrace{\gamma_0^2 M^{2-2\min(\alpha,\,0.5)} + M^{-2\alpha+\max(0,1-2\beta)}}_{=:C}, \qquad p = \frac{2(2\alpha+2\beta-1)}{2\alpha+1-2\beta}. \tag{23}$$

For $\alpha > 0.5$ and $\beta > 0.5$, we get the proxy

$$L_{\mathrm{px}}(N) = (\gamma_0\, M^{0.5} N)^{-p_1} + (\gamma_0\, M^{\frac{6\alpha-1}{4\alpha-2}} N)^{-p_2} + C, \tag{24}$$

where

$$p_1 = \frac{2(2\alpha + 2\beta - 1)}{2\alpha + 1 - 2\beta}, \qquad p_2 = \frac{4\alpha - 2}{2\alpha + 1}.$$

As a last step, we verify the proxies by proving that they satisfy the converted integral equations.

### E.1 ONE-STEP UPDATE FORMULA OF SIGNSGD

Xiao et al. (2025) approximate the signSGD trajectory using SDE and ODE techniques. Their proof relies on a spectral lower bound assumption of the covariance matrix, so their results are not directly applicable to our setting.

For a quadratic function $q$, by Taylor's theorem, we have

$$\mathbb{E}[q(\boldsymbol{\theta}_{k+1}) - q(\boldsymbol{\theta}_k)\,|\,\mathcal{F}_k] = \mathbb{E}[\langle \nabla q(\boldsymbol{\theta}_k), \boldsymbol{\theta}_{k+1} - \boldsymbol{\theta}_k \rangle\,|\,\mathcal{F}_k] + \tfrac{1}{2}\,\mathbb{E}\big[\langle \nabla^2 q, (\boldsymbol{\theta}_{k+1} - \boldsymbol{\theta}_k)^{\otimes 2}\rangle\,\big|\,\mathcal{F}_k\big],$$

where $\mathcal{F}_k = \sigma(\boldsymbol{S}, \boldsymbol{\theta}_0, \ldots, \boldsymbol{\theta}_k)$. Since

$$\boldsymbol{\theta}_{k+1} - \boldsymbol{\theta}_k = -\gamma_k\,\mathrm{sign}(\langle \boldsymbol{S}\boldsymbol{x}_k, \boldsymbol{\theta}_k\rangle - y_k)\,\mathrm{sign}(\boldsymbol{S}\boldsymbol{x}_k),$$

We can expand the two terms using sign-Gaussian identities.

**Gradient term.**

$$\mathbb{E}[\langle \nabla q(\boldsymbol{\theta}_k), \boldsymbol{\theta}_{k+1} - \boldsymbol{\theta}_k \rangle\,|\,\mathcal{F}_k]$$
$$= -\gamma_k\,\Big\langle \nabla q(\boldsymbol{\theta}_k),\, \mathbb{E}\big[\mathrm{sign}(\boldsymbol{S}\boldsymbol{x}_k)\,\mathrm{sign}(\langle \boldsymbol{x}_k, \boldsymbol{S}^{\mathsf{T}}\boldsymbol{\theta}_k - \boldsymbol{w}^*\rangle)\,\big|\,\mathcal{F}_k\big]\Big\rangle$$
$$= -\gamma_k\,\left\langle \nabla q(\boldsymbol{\theta}_k),\, \frac{2}{\pi}\arcsin\left(\frac{\mathrm{diag}(\boldsymbol{S}\boldsymbol{H}\boldsymbol{S}^{\mathsf{T}})^{-1/2}\,\boldsymbol{S}\boldsymbol{H}\,(\boldsymbol{S}^{\mathsf{T}}\boldsymbol{\theta}_k - \boldsymbol{w}^*)}{\sqrt{(\boldsymbol{S}^{\mathsf{T}}\boldsymbol{\theta}_k - \boldsymbol{w}^*)^{\mathsf{T}}\,\boldsymbol{H}\,(\boldsymbol{S}^{\mathsf{T}}\boldsymbol{\theta}_k - \boldsymbol{w}^*)}}\right)\right\rangle$$
$$= -\gamma_k\,\left\langle \nabla q(\boldsymbol{\theta}_k),\, \frac{2}{\pi}\arcsin\left(\frac{\mathrm{diag}(\boldsymbol{K})^{-1/2}\,\boldsymbol{K}\,(\boldsymbol{\theta}_k - \boldsymbol{\theta}^*)}{\big\|\boldsymbol{H}^{1/2}(\boldsymbol{S}^{\mathsf{T}}\boldsymbol{\theta}_k - \boldsymbol{w}^*)\big\|}\right)\right\rangle,$$

where $\boldsymbol{K} = \boldsymbol{S}\boldsymbol{H}\boldsymbol{S}^{\mathsf{T}}$.

**Quadratic term.**

$$\mathbb{E}\Big[\big\langle \nabla^2 q, (\boldsymbol{\theta}_{k+1} - \boldsymbol{\theta}_k)^{\otimes 2}\big\rangle\,\Big|\,\mathcal{F}_k\Big]$$
$$= \gamma_k^2\,\Big\langle \nabla^2 q,\, \mathbb{E}\big[(\mathrm{sign}(\boldsymbol{S}\boldsymbol{x}_k)\,\mathrm{sign}(\langle \boldsymbol{x}_k, \boldsymbol{S}^{\mathsf{T}}\boldsymbol{\theta}_k - \boldsymbol{w}^*\rangle))^{\otimes 2}\,\big|\,\mathcal{F}_k\big]\Big\rangle$$
$$= \gamma_k^2\,\Big\langle \nabla^2 q,\, \mathbb{E}\big[(\mathrm{sign}(\boldsymbol{S}\boldsymbol{x}_k))^{\otimes 2}\,\big|\,\mathcal{F}_k\big]\Big\rangle$$
$$= \gamma_k^2\,\Big\langle \nabla^2 q,\, \frac{2}{\pi}\arcsin\big(\mathrm{diag}(\boldsymbol{S}\boldsymbol{H}\boldsymbol{S}^{\mathsf{T}})^{-1/2}\,\boldsymbol{S}\boldsymbol{H}\boldsymbol{S}^{\mathsf{T}}\,\mathrm{diag}(\boldsymbol{S}\boldsymbol{H}\boldsymbol{S}^{\mathsf{T}})^{-1/2}\big)\Big\rangle$$
$$= \gamma_k^2\,\Big\langle \nabla^2 q,\, \frac{2}{\pi}\arcsin\big(\mathrm{diag}(\boldsymbol{K})^{-1/2}\,\boldsymbol{K}\,\mathrm{diag}(\boldsymbol{K})^{-1/2}\big)\Big\rangle.$$

**One-step update formula.** Substituting the gradient and quadratic terms yields the desired one-step update formula for signSGD.

$$\mathbb{E}[q(\boldsymbol{\theta}_{k+1}) - q(\boldsymbol{\theta}_k)\,|\,\mathcal{F}_k] = -\frac{2\gamma_k}{\pi}\,\left\langle \nabla q(\boldsymbol{\theta}_k),\, \arcsin\left(\frac{\overline{\boldsymbol{K}}\,(\boldsymbol{\theta}_k - \boldsymbol{\theta}^*)}{\sqrt{L(k)}}\right)\right\rangle + \frac{\gamma_k^2}{\pi}\,\langle \nabla^2 q, \boldsymbol{K}_\sigma\rangle.$$

Let $\lambda_i(\overline{\boldsymbol{K}})$, $\boldsymbol{u}_i$, and $\boldsymbol{w}_i$ denote the eigenvalue, right eigenvector, and left eigenvector of $\overline{\boldsymbol{K}}$, respectively. Then $\overline{\boldsymbol{K}} = \sum_{i=1}^{M} \lambda_i(\overline{\boldsymbol{K}})\,\boldsymbol{u}_i \otimes \boldsymbol{w}_i$ and $I = \sum_{i=1}^{M} \boldsymbol{u}_i \otimes \boldsymbol{w}_i$.

Define
$$r_i(k) = (\boldsymbol{\theta}_k - \boldsymbol{\theta}^*)^\mathsf{T}(\boldsymbol{K}\boldsymbol{u}_i \otimes \boldsymbol{w}_i)(\boldsymbol{\theta}_k - \boldsymbol{\theta}^*).$$

The loss decomposes as

$$L(k) = \left\|\boldsymbol{H}^{1/2}\boldsymbol{S}^\mathsf{T}(\boldsymbol{\theta}_k - \boldsymbol{\theta}^*)\right\|^2 + \left\|\boldsymbol{H}^{1/2}\boldsymbol{w}_\perp\right\|^2 = (\boldsymbol{\theta}_k - \boldsymbol{\theta}^*)^\mathsf{T}\boldsymbol{K}(\boldsymbol{\theta}_k - \boldsymbol{\theta}^*) + \|\boldsymbol{H}^{1/2}\boldsymbol{w}_\perp\|^2 = \sum_{i=1}^d r_i(k) + \|\boldsymbol{H}^{1/2}\boldsymbol{w}_\perp\|^2.$$

We now apply the one-step update formula to $r_i(k)$. Note that

$$\nabla r_i(k) = \boldsymbol{K}\boldsymbol{u}_i \langle \boldsymbol{w}_i, \boldsymbol{\theta}_k - \boldsymbol{\theta}^* \rangle + \boldsymbol{w}_i \langle \boldsymbol{K}\boldsymbol{u}_i, \boldsymbol{\theta}_k - \boldsymbol{\theta}^* \rangle, \qquad \nabla^2 r_i = \boldsymbol{K}\boldsymbol{u}_i\boldsymbol{w}_i^\mathsf{T} + \boldsymbol{w}_i\boldsymbol{u}_i^\mathsf{T}\boldsymbol{K}^\mathsf{T}.$$

Approximating $\arcsin(x) \approx x$ and using $\boldsymbol{K}^\mathsf{T} = \boldsymbol{K}$ together with $\boldsymbol{K}^\mathsf{T}\overline{\boldsymbol{K}} = \overline{\boldsymbol{K}}^\mathsf{T}\boldsymbol{K}^\mathsf{T}$, we obtain

$$\mathbb{E}[r_i(k+1) - r_i(k)\,|\,\mathcal{F}_k] \approx -\frac{2\gamma_k}{\pi}\left(\langle \boldsymbol{w}_i, \boldsymbol{\theta}_k - \boldsymbol{\theta}^* \rangle \left\langle \boldsymbol{K}\boldsymbol{u}_i, \frac{\overline{\boldsymbol{K}}(\boldsymbol{\theta}_k - \boldsymbol{\theta}^*)}{\sqrt{L(k)}}\right\rangle + \langle \boldsymbol{K}\boldsymbol{u}_i, \boldsymbol{\theta}_k - \boldsymbol{\theta}^* \rangle \left\langle \boldsymbol{w}_i, \frac{\overline{\boldsymbol{K}}(\boldsymbol{\theta}_k - \boldsymbol{\theta}^*)}{\sqrt{L(k)}}\right\rangle\right)$$
$$+ \frac{2\gamma_k^2}{\pi}\boldsymbol{w}_i^\mathsf{T}\boldsymbol{K}_\sigma\boldsymbol{K}\boldsymbol{u}_i$$
$$= -\frac{4\gamma_k}{\pi\sqrt{L(k)}}\lambda_i(\overline{\boldsymbol{K}})\,r_i(k) + \frac{2\gamma_k^2}{\pi}\boldsymbol{w}_i^\mathsf{T}\boldsymbol{K}_\sigma\boldsymbol{K}\boldsymbol{u}_i.$$

It is possible to replace the linear approximation $\arcsin(x) \approx x$ by an inequality, and the main results of our paper remain unchanged. We explain it in Appendix K.2. Hence,

$$\mathbb{E}[r_i(k+1) - r_i(k)\,|\,\mathcal{F}_k] \approx -\frac{4\gamma_k}{\pi\sqrt{L(k)}}\lambda_i(\overline{\boldsymbol{K}})\,r_i(k) + \frac{2\gamma_k^2}{\pi}\boldsymbol{w}_i^\mathsf{T}\boldsymbol{K}_\sigma\boldsymbol{K}\boldsymbol{u}_i.$$

### E.2 ODE Approximation and Implicit Integral Equation of signSGD

Let the learning rate be $\gamma_k = \gamma_0 f(k)$. Define $V_i = \boldsymbol{w}_i^\mathsf{T}\boldsymbol{K}_\sigma\boldsymbol{K}\boldsymbol{u}_i$, then our one-step update formula becomes

$$\mathbb{E}[r_i(k+1) - r_i(k)\,|\,\mathcal{F}_k] = -\frac{4\gamma_k}{\pi\sqrt{L(k)}}\lambda_i(\overline{\boldsymbol{K}})\,r_i(k) + \frac{2\gamma_k^2}{\pi}V_i.$$

Dividing by $\gamma_0$ gives

$$\mathbb{E}\left[\frac{r_i(k+1) - r_i(k)}{\gamma_0}\,\middle|\,\mathcal{F}_k\right] = -\frac{4}{\pi\sqrt{L(k)}}\lambda_i(\overline{\boldsymbol{K}})\,f(k)\,r_i(k) + \frac{2f(k)^2\gamma_0}{\pi}V_i.$$

Interpreting $\gamma_0$ as the time step, the discrete index $k$ corresponds to continuous time $t = k\gamma_0$. Let $P(t) = L(t/\gamma_0)$ and $p_i(t) = r_i(t/\gamma_0)$. We then obtain the ODE

$$\frac{dp_i}{dt} = -\frac{4}{\pi\sqrt{P(t)}}\lambda_i(\overline{\boldsymbol{K}})\,f(t/\gamma_0)\,p_i(t) + \frac{2f(t/\gamma_0)^2\gamma_0}{\pi}V_i. \tag{25}$$

From this point onward in the analysis, we treat $P$, $p_i$, $L$, and $r_i$ as their continuous extensions, allowing arbitrary positive real inputs.

**Integral formulation.** Solving the ODE yields

$$p_i(t) = p_i(0)\,e^{-\frac{4\lambda_i}{\pi}\int_0^t \frac{f(u/\gamma_0)}{\sqrt{P(u)}}\,du} + \frac{2\gamma_0}{\pi}V_i\int_0^t e^{-\frac{4\lambda_i}{\pi}\int_s^t \frac{f(u/\gamma_0)}{\sqrt{P(u)}}\,du}\,f(s/\gamma_0)^2\,ds.$$

Since $P(t) = \sum_{i=1}^M p_i(t) + \|\boldsymbol{H}^{1/2}\boldsymbol{w}_\perp\|^2$, we obtain

$$P(t) = \|\boldsymbol{H}^{1/2}\boldsymbol{w}_\perp\|^2 + \sum_{i=1}^M p_i(0)\,e^{-\frac{4\lambda_i}{\pi}\int_0^t \frac{f(u/\gamma_0)}{\sqrt{P(u)}}\,du} + \frac{2\gamma_0}{\pi}\sum_{i=1}^M V_i\int_0^t e^{-\frac{4\lambda_i}{\pi}\int_s^t \frac{f(u/\gamma_0)}{\sqrt{P(u)}}\,du}\,f(s/\gamma_0)^2\,ds.$$

**Integral equation in discrete form.** Note that $L(N) = P(N\gamma_0)$. With a change of variables, we obtain

$$L(N) = \|\boldsymbol{H}^{1/2}\boldsymbol{w}_\perp\|^2 + \sum_{i=1}^{M} r_i(0)\, e^{-\frac{4\lambda_i\gamma_0}{\pi}\int_0^N \frac{f(u)}{\sqrt{L(u)}}\,du} + \frac{2\gamma_0^2}{\pi}\sum_{i=1}^{M} V_i \int_0^N e^{-\frac{4\lambda_i\gamma_0}{\pi}\int_z^N \frac{f(u)}{\sqrt{L(u)}}\,du}\, f(z)^2\, dz. \tag{26}$$

**Drift and noise decomposition.** Define

$$L^{\text{drift}}(N) = \sum_{i=1}^{M} r_i(0)\, e^{-\frac{4\lambda_i\gamma_0}{\pi}\int_0^N \frac{f(u)}{\sqrt{L(u)}}\,du}, \quad L^{\text{noise}}(N) = \frac{2\gamma_0^2}{\pi}\sum_{i=1}^{M} V_i \int_0^N e^{-\frac{4\lambda_i\gamma_0}{\pi}\int_z^N \frac{f(u)}{\sqrt{L(u)}}\,du}\, f(z)^2\, dz. \tag{27}$$

Then

$$L(N) = \|\boldsymbol{H}^{1/2}\boldsymbol{w}_\perp\|^2 + L^{\text{drift}}(N) + L^{\text{noise}}(N), \tag{28}$$

and we will analyze $\|\boldsymbol{H}^{1/2}\boldsymbol{w}_\perp\|^2 + L^{\text{drift}}(N)$ and $L^{\text{noise}}(N)$ separately.

Figure 19 shows dynamics of three terms $\|\boldsymbol{H}^{1/2}\boldsymbol{w}_\perp\|^2$, $L^{\text{drift}}(N)$, $L^{\text{noise}}(N)$ referring each as Approx, Drift, Noise. The right plot in Figure 19 validates the equality in (28).

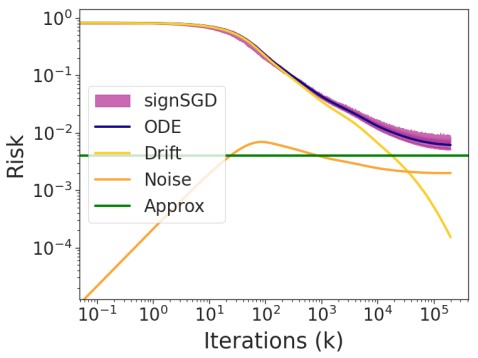 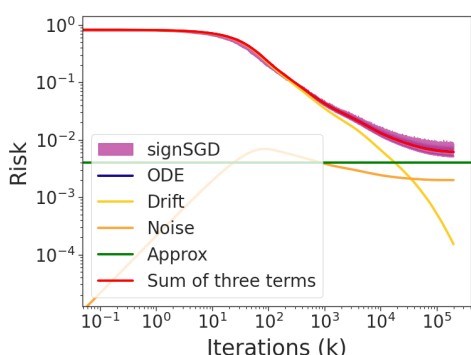

Figure 19: **Dynamics of Drift and Noise.** Left: the purple curve is the 80% confidence interval of the true signSGD trajectory, while the blue curve is the numerical ODE solution. The yellow, orange, and green curves correspond to the approximation, drift, and noise terms in (28). Right: the red curve shows the sum of these three terms, matching both the true trajectory and the ODE solution. Parameters: $\alpha = 1.0$, $\beta = 0$, $\gamma_0 = 0.003$, $f(z) = 1$, $M = 200$, $d = 800$.

### E.2.1 TRANSFORMATION OF THE DRIFT TERM AND APPROXIMATION ERROR

Let

$$Q(z) = \frac{4\gamma_0}{\pi}\int_0^z \frac{f(u)}{\sqrt{L(u)}}\,du, \qquad \overline{\boldsymbol{K}}_1 = \boldsymbol{H}^{1/2}\boldsymbol{S}^\mathsf{T}\,\text{diag}(\boldsymbol{S}\boldsymbol{H}\boldsymbol{S}^\mathsf{T})^{-1/2}\boldsymbol{S}\boldsymbol{H}^{1/2}.$$

Then

$$\boldsymbol{K}\,\overline{\boldsymbol{K}}^p = \boldsymbol{S}\boldsymbol{H}^{1/2}\overline{\boldsymbol{K}}_1^p\boldsymbol{H}^{1/2}\boldsymbol{S}^\mathsf{T}$$

holds.

Define

$$\boldsymbol{A} = \boldsymbol{H}^{1/2}e^{-\overline{\boldsymbol{K}}_1 Q(N)}\boldsymbol{H}^{1/2}, \qquad \boldsymbol{u} = \boldsymbol{S}^\mathsf{T}\boldsymbol{\theta}_0 - \boldsymbol{S}^\mathsf{T}\boldsymbol{\theta}^* - \boldsymbol{w}_\perp = \boldsymbol{S}^\mathsf{T}\boldsymbol{\theta}_0 - \boldsymbol{w}^*.$$

From $\boldsymbol{S}\boldsymbol{H}\boldsymbol{w}_\perp = 0$ we get

$$\overline{\boldsymbol{K}}_1(\boldsymbol{H}^{1/2}\boldsymbol{w}_\perp) = 0,$$

and this implies

$$e^{-\overline{\boldsymbol{K}}_1 Q(N)}(\boldsymbol{H}^{1/2}\boldsymbol{w}_\perp) = e^0(\boldsymbol{H}^{1/2}\boldsymbol{w}_\perp) = \boldsymbol{H}^{1/2}\boldsymbol{w}_\perp.$$

Thus,

$$\boldsymbol{A}\boldsymbol{w}_\perp = \boldsymbol{H}\boldsymbol{w}_\perp, \qquad \boldsymbol{w}_\perp^\mathsf{T} \boldsymbol{A}\boldsymbol{w}_\perp = \boldsymbol{w}_\perp^\mathsf{T} \boldsymbol{H}\boldsymbol{w}_\perp,$$

and

$$\boldsymbol{u}^\mathsf{T} \boldsymbol{A}\boldsymbol{w}_\perp = \boldsymbol{u}^\mathsf{T} \boldsymbol{H}\boldsymbol{w}_\perp = (\boldsymbol{\theta}_0 - \boldsymbol{\theta}^*)^\mathsf{T} \boldsymbol{S}\boldsymbol{H}\boldsymbol{w}_\perp - \boldsymbol{w}_\perp^\mathsf{T} \boldsymbol{H}\boldsymbol{w}_\perp = -\boldsymbol{w}_\perp^\mathsf{T} \boldsymbol{H}\boldsymbol{w}_\perp.$$

Using these identities, we can convert the drift term as follows:

$$
\begin{aligned}
L^{\mathrm{drift}}(N) &= \sum_{i=1}^{M} r_i(0) \cdot e^{-\lambda_i(\overline{\boldsymbol{K}})Q(N)} \\
&= \sum_{i=1}^{M} (\boldsymbol{\theta}_0 - \boldsymbol{\theta}^*)^\mathsf{T} (\boldsymbol{K}\boldsymbol{u}_i \otimes \boldsymbol{w}_i)(\boldsymbol{\theta}_0 - \boldsymbol{\theta}^*) \cdot e^{-\lambda_i(\overline{\boldsymbol{K}})Q(N)} \\
&= \sum_{i=1}^{M} (\boldsymbol{\theta}_0 - \boldsymbol{\theta}^*)^\mathsf{T} \Big( (\boldsymbol{K}\boldsymbol{u}_i \otimes \boldsymbol{w}_i) \cdot e^{-\lambda_i(\overline{\boldsymbol{K}})Q(N)} \Big)(\boldsymbol{\theta}_0 - \boldsymbol{\theta}^*) \\
&= (\boldsymbol{\theta}_0 - \boldsymbol{\theta}^*)^\mathsf{T} \boldsymbol{K} e^{-\overline{\boldsymbol{K}}Q(N)} (\boldsymbol{\theta}_0 - \boldsymbol{\theta}^*) \\
&= (\boldsymbol{\theta}_0 - \boldsymbol{\theta}^*)^\mathsf{T} \boldsymbol{S}\boldsymbol{H}^{1/2} \left( \boldsymbol{H}^{1/2} \boldsymbol{S} e^{-\overline{\boldsymbol{K}}Q(N)} \right)(\boldsymbol{\theta}_0 - \boldsymbol{\theta}^*) \\
&= (\boldsymbol{\theta}_0 - \boldsymbol{\theta}^*)^\mathsf{T} \boldsymbol{S}\boldsymbol{H}^{1/2} \left( e^{-\overline{\boldsymbol{K}}_1 Q(N)} \boldsymbol{H}^{1/2} \boldsymbol{S} \right)(\boldsymbol{\theta}_0 - \boldsymbol{\theta}^*) \\
&= (\boldsymbol{u} + \boldsymbol{w}_\perp)^\mathsf{T} \boldsymbol{A}(\boldsymbol{u} + \boldsymbol{w}_\perp) \\
&= \boldsymbol{u}^\mathsf{T} \boldsymbol{A}\boldsymbol{u} + \boldsymbol{u}^\mathsf{T} \boldsymbol{A}\boldsymbol{w}_\perp + \boldsymbol{w}_\perp^\mathsf{T} \boldsymbol{A}\boldsymbol{u} + \boldsymbol{w}_\perp^\mathsf{T} \boldsymbol{A}\boldsymbol{w}_\perp \\
&= \boldsymbol{u}^\mathsf{T} \boldsymbol{H}^{1/2} e^{-\overline{\boldsymbol{K}}_1 Q(N)} \boldsymbol{H}^{1/2} \boldsymbol{u} - \boldsymbol{w}_\perp^\mathsf{T} \boldsymbol{H}\boldsymbol{w}_\perp - \boldsymbol{w}_\perp^\mathsf{T} \boldsymbol{H}\boldsymbol{w}_\perp + \boldsymbol{w}_\perp^\mathsf{T} \boldsymbol{H}\boldsymbol{w}_\perp \\
&= \boldsymbol{u}^\mathsf{T} \boldsymbol{H}^{1/2} e^{-\overline{\boldsymbol{K}}_1 Q(N)} \boldsymbol{H}^{1/2} \boldsymbol{u} - \|\boldsymbol{H}^{1/2} \boldsymbol{w}_\perp\|^2.
\end{aligned}
$$

**Drift term plus approximation error.** Adding the approximation error gives

$$
\begin{aligned}
L^{\mathrm{drift}}(N) + \|\boldsymbol{H}^{1/2}\boldsymbol{w}_\perp\|^2 &= \boldsymbol{u}^\mathsf{T} \boldsymbol{H}^{1/2} e^{-\overline{\boldsymbol{K}}_1 Q(N)} \boldsymbol{H}^{1/2} \boldsymbol{u} \\
&= \Big\langle e^{-\overline{\boldsymbol{K}}_1 Q(N)}, \ \big(\boldsymbol{H}^{1/2}(\boldsymbol{S}^\mathsf{T}\boldsymbol{\theta}_0 - \boldsymbol{w}^*)\big)^{\otimes 2} \Big\rangle.
\end{aligned}
$$

Also we assume $\boldsymbol{\theta}_0 = 0$, then

$$\big\langle e^{-\overline{\boldsymbol{K}}_1 Q(N)}, (\boldsymbol{H}^{1/2}(\boldsymbol{S}^\mathsf{T}\boldsymbol{\theta}_0 - \boldsymbol{w}^*))^{\otimes 2} \big\rangle = \big\langle e^{-\overline{\boldsymbol{K}}_1 Q(N)}, (\boldsymbol{H}^{1/2}\boldsymbol{w}^*)^{\otimes 2} \big\rangle.$$

In the next subsection, we will describe how to apply a deterministic approximation, similar to Paquette et al. (2024), to the following term:

$$\mathcal{H} := \big\langle e^{-\overline{\boldsymbol{K}}_1 Q(N)}, \boldsymbol{v}^{\otimes 2} \big\rangle,$$

where $\boldsymbol{v} := \boldsymbol{H}^{1/2}\boldsymbol{w}^* \in \mathbb{R}^d$.

### E.2.2 DETERMINISTIC APPROXIMATION

Note that we assume $d \geq rM$ for some $r > 1$, and let $d/M \to (1, \infty]$ as $d, M \to \infty$ when $2\alpha > 1$, and $d/M \to (1, \infty)$ when $2\alpha < 1$. In our setup, $S \in \mathbb{R}^{M \times d}$ have i.i.d. $\mathcal{N}(0, 1/M)$ entries, and we will write the $k$th column of $S^\mathsf{T}$ as $\frac{1}{\sqrt{M}}\boldsymbol{s}_k \in \mathbb{R}^d$; columns are independent.

Define

$$\boldsymbol{y}_k := \boldsymbol{H}^{1/2}\boldsymbol{s}_k \in \mathbb{R}^d, \qquad a_k := \frac{1}{\sqrt{\frac{1}{M}\boldsymbol{y}_k^\mathsf{T}\boldsymbol{y}_k}} = \frac{\sqrt{M}}{\sqrt{\boldsymbol{s}_k^\mathsf{T}\boldsymbol{H}\boldsymbol{s}_k}} > 0.$$

The unnormalized baseline and the column–normalized matrices are

$$\widehat{\boldsymbol{K}} := \boldsymbol{H}^{1/2}\boldsymbol{S}^{\mathsf{T}}\boldsymbol{S}\boldsymbol{H}^{1/2} = \frac{1}{M}\sum_{k=1}^{M}\boldsymbol{y}_k\boldsymbol{y}_k^{\mathsf{T}}, \qquad \overline{\boldsymbol{K}}_1 := \boldsymbol{H}^{1/2}\boldsymbol{S}^{\mathsf{T}}\operatorname{diag}(\boldsymbol{S}\boldsymbol{H}\boldsymbol{S}^{\mathsf{T}})^{-1/2}\boldsymbol{S}\boldsymbol{H}^{1/2} = \frac{1}{M}\sum_{k=1}^{M}a_k\,\boldsymbol{y}_k\boldsymbol{y}_k^{\mathsf{T}}.$$

For $z \in \mathbb{C}^{+} := \{z : \Im z > 0\}$, define the resolvents

$$\boldsymbol{L}(z) := (\overline{\boldsymbol{K}}_1 - z\boldsymbol{I})^{-1}, \qquad \boldsymbol{R}^{(k)}(z) := \left(\frac{1}{M}\sum_{\ell \neq k}a_\ell\boldsymbol{y}_\ell\boldsymbol{y}_\ell^{\mathsf{T}} - z\boldsymbol{I}\right)^{-1}.$$

Note that

$$\boldsymbol{y}_k\boldsymbol{B}\,\boldsymbol{y}_k \approx \operatorname{Tr}(\boldsymbol{H}\boldsymbol{B})$$

for matrix $\boldsymbol{B}$. In particular,

$$\boldsymbol{y}_k^{\mathsf{T}}\boldsymbol{y}_k \approx \operatorname{Tr}\boldsymbol{H}, \qquad a_k \approx \frac{\sqrt{M}}{\sqrt{\operatorname{Tr}\boldsymbol{H}}}.$$

Also note that

$$a_k\,\boldsymbol{y}_k^{\mathsf{T}}\boldsymbol{R}\,\boldsymbol{R}^{(k)}\,\boldsymbol{y}_k \approx \frac{\sqrt{M}}{\sqrt{\operatorname{Tr}\boldsymbol{H}}}\cdot\operatorname{Tr}(\boldsymbol{H}\,\boldsymbol{R}\,\boldsymbol{R}^{(k)}),$$

and

$$a_k\boldsymbol{y}_k^{\mathsf{T}}\boldsymbol{R}^{(k)}\boldsymbol{y}_k \approx \frac{\sqrt{M}}{\sqrt{\operatorname{Tr}\boldsymbol{H}}}\operatorname{Tr}(\boldsymbol{H}\,\boldsymbol{R}^{(k)}).$$

By the Sherman–Morrison expansion,

$$\boldsymbol{R} = \boldsymbol{R}^{(k)} - \frac{M^{-1}a_k\,\boldsymbol{R}^{(k)}\boldsymbol{y}_k\boldsymbol{y}_k^{\mathsf{T}}\boldsymbol{R}^{(k)}}{1 + M^{-1}a_k\,\boldsymbol{y}_k^{\mathsf{T}}\boldsymbol{R}^{(k)}\boldsymbol{y}_k}.$$

Multiplying on the left by $\boldsymbol{R}$ and sandwiching with $\boldsymbol{y}_k^{\mathsf{T}}(\cdot)\boldsymbol{y}_k$, we get

$$a_k\boldsymbol{y}_k^{\mathsf{T}}\boldsymbol{R}\,\boldsymbol{R}\,\boldsymbol{y}_k = a_k\boldsymbol{y}_k^{\mathsf{T}}\boldsymbol{R}\,\boldsymbol{R}^{(k)}\boldsymbol{y}_k - \frac{M^{-1}a_k\boldsymbol{y}_k^{\mathsf{T}}\boldsymbol{R}\,\boldsymbol{R}^{(k)}\boldsymbol{y}_k\cdot a_k\boldsymbol{y}_k^{\mathsf{T}}\boldsymbol{R}^{(k)}\boldsymbol{y}_k}{1 + M^{-1}a_k\,\boldsymbol{y}_k^{\mathsf{T}}\boldsymbol{R}^{(k)}\boldsymbol{y}_k}.$$

Now we will replace terms on the right side by

$$a_k\boldsymbol{y}_k^{\mathsf{T}}\boldsymbol{R}\,\boldsymbol{R}^{(k)}\boldsymbol{y}_k \approx \frac{\sqrt{M}}{\sqrt{\operatorname{Tr}\boldsymbol{H}}}\operatorname{Tr}(\boldsymbol{H}\,\boldsymbol{R}\,\boldsymbol{R}^{(k)}),$$

and

$$a_k\boldsymbol{y}_k^{\mathsf{T}}\boldsymbol{R}^{(k)}\boldsymbol{y}_k \approx \frac{\sqrt{M}}{\sqrt{\operatorname{Tr}\boldsymbol{H}}}\operatorname{Tr}(\boldsymbol{H}\,\boldsymbol{R}^{(k)}).$$

Thus

$$a_k\boldsymbol{y}_k^{\mathsf{T}}\boldsymbol{R}\,\boldsymbol{R}\,\boldsymbol{y}_k \approx \frac{\frac{\sqrt{M}}{\sqrt{\operatorname{Tr}\boldsymbol{H}}}\operatorname{Tr}(\boldsymbol{H}\,\boldsymbol{R}\,\boldsymbol{R}^{(k)})}{1 + M^{-1}\frac{\sqrt{M}}{\sqrt{\operatorname{Tr}\boldsymbol{H}}}\operatorname{Tr}(\boldsymbol{H}\,\boldsymbol{R}^{(k)})}.$$

Replacing $\boldsymbol{R}^{(k)}$ by $\boldsymbol{R}$ and averaging over $k$, we obtain

$$\frac{1}{M}\sum_{k=1}^{M}a_k\boldsymbol{y}_k^{\mathsf{T}}\boldsymbol{R}\,\boldsymbol{R}\,\boldsymbol{y}_k \approx \frac{p_d\operatorname{Tr}(\boldsymbol{H}\,\boldsymbol{R}\,\boldsymbol{R})}{1 + M^{-1}p_d\operatorname{Tr}(\boldsymbol{H}\,\boldsymbol{R})}, \qquad p_d := \frac{\sqrt{M}}{\sqrt{\operatorname{Tr}\boldsymbol{H}}}.$$

It implies

$$\operatorname{Tr}(\boldsymbol{R}(\boldsymbol{R}^{-1} + z\boldsymbol{I})\boldsymbol{R}) \approx \frac{p_d\operatorname{Tr}(\boldsymbol{R}\,\boldsymbol{H}\,\boldsymbol{R})}{1 + M^{-1}p_d\operatorname{Tr}(\boldsymbol{H}\,\boldsymbol{R})}.$$

This implies

$$\boldsymbol{L}(z)^{-1} + z\boldsymbol{I} \approx \frac{p_d}{1 + M^{-1}p_d\operatorname{Tr}(\boldsymbol{H}\,\boldsymbol{L}(z))}\boldsymbol{H}.$$

Let

$$m(z/p_d) = \frac{1}{1 + M^{-1}p_d \operatorname{Tr}(\boldsymbol{H}\,\boldsymbol{L}(z))}.$$

Then

$$\boldsymbol{L}(z) \approx (-z\boldsymbol{I} + p_d m(z/p_d)\boldsymbol{H})^{-1}.$$

Thus

$$(\overline{\boldsymbol{K}}_1 - z\boldsymbol{I})^{-1} \approx (-z\boldsymbol{I} + p_d m(z/p_d)\boldsymbol{H})^{-1}.$$

Therefore,

$$m(z) = \frac{1}{1 + M^{-1}p_d \operatorname{Tr}(\boldsymbol{H}\,\boldsymbol{R}(p_d z))} \approx \frac{1}{1 + M^{-1}\operatorname{Tr}(\boldsymbol{H}(-z\boldsymbol{I} + m(z)\boldsymbol{H})^{-1})}$$

holds. This fixed–point equation is identical to the one in Paquette et al. (2024).

**Contour representation.** Let $\boldsymbol{v} := \boldsymbol{H}^{1/2}\boldsymbol{w}^* \in \mathbb{R}^d$ and consider

$$\mathcal{H} := \left\langle e^{-\overline{\boldsymbol{K}}_1 Q(N)}, \boldsymbol{v}^{\otimes 2} \right\rangle.$$

For any analytic $g$ on a contour $\Gamma_2$ enclosing $\operatorname{Spec}(\overline{\boldsymbol{K}}_1)$,

$$g(\overline{\boldsymbol{K}}_1) = -\frac{1}{2\pi i} \oint_{\Gamma_2} g(z)(\overline{\boldsymbol{K}}_1 - z\boldsymbol{I})^{-1} \, dz.$$

We prove

$$c_1 \, M^{\min(0.5,\alpha)} \, I \;\preceq\; \operatorname{diag}(\boldsymbol{S}\boldsymbol{H}\boldsymbol{S}^{\mathsf{T}})^{-1/2} \;\preceq\; c_2 \, M^{\min(0.5,\alpha)} \, I$$

in Section K.4. It leads to

$$c_1 \, M^{\min(0.5,\alpha)} \widehat{\boldsymbol{K}} \preceq \overline{\boldsymbol{K}} \preceq c_2 \, M^{\min(0.5,\alpha)} \widehat{\boldsymbol{K}}.$$

$\overline{\boldsymbol{K}}_1$ has eigenvalues scaled by $M^{\min(0.5,\alpha)}$ compared to $\widehat{\boldsymbol{K}}$ excluding constant. Note that $p_d \asymp M^{\min(0.5,\alpha)}$. So, there exists a contour $\Gamma_2$ enclosing the spectrum of $\overline{\boldsymbol{K}}_1$, and its $1/p_d$–scaled version $\Gamma$ encloses the spectrum of $\widehat{\boldsymbol{K}}$.

Taking $g(z) = e^{-Q(N)z}$,

$$\begin{aligned}
\mathcal{H} &= -\frac{1}{2\pi i} \oint_{\Gamma_2} e^{-Q(N)z} \left\langle (\overline{\boldsymbol{K}}_1 - z\boldsymbol{I})^{-1}, \boldsymbol{v}^{\otimes 2} \right\rangle dz \\
&\approx -\frac{1}{2\pi i} \oint_{\Gamma_2} e^{-Q(N)z} \left\langle (-z\boldsymbol{I} + p_d m(z/p_d)\boldsymbol{H})^{-1}, \boldsymbol{v}^{\otimes 2} \right\rangle dz \\
&= -\frac{1}{2\pi i} \oint_{\Gamma} e^{-p_d Q(N)z} \left\langle (-z\boldsymbol{I} + m(z)\boldsymbol{H})^{-1}, \boldsymbol{v}^{\otimes 2} \right\rangle dz.
\end{aligned}$$

Let $\mathcal{R}(z) = (-z\boldsymbol{I} + m(z)\boldsymbol{H})^{-1}$, then our objective converts to

$$\mathcal{H} \approx -\frac{1}{2\pi i} \oint_{\Gamma} e^{-p_d Q(N)z} \left\langle \mathcal{R}(z), \boldsymbol{v}^{\otimes 2} \right\rangle dz.$$

### E.2.3 FINAL TRANSFORMATION RESULT

Paquette et al. (2024) evaluate the contour integrals with $\mathcal{R}(z)$. When $\alpha < 0.5$ or $\beta < 0.5$, they show

$$-\frac{1}{2\pi i} \oint_{\Gamma} \left(1 - 2\gamma Bz + \gamma^2 B(B+1)z^2\right)^r \left\langle \mathcal{L}(z), v^{\otimes 2} \right\rangle dz \;\asymp\; M^{-2\alpha + \max(0,\,1-2\beta)}$$

$$+ \; (2\gamma Br)^{-\frac{2\alpha + 2\beta - 1}{2\alpha}}. \qquad (29)$$

When $\alpha > 0.5$ and $\beta > 0.5$, they obtained

$$-\frac{1}{2\pi \mathrm{i}} \oint_\Gamma \left(1 - 2\gamma Bz + \gamma^2 B(B+1)z^2\right)^r \left\langle \mathcal{L}(z), v^{\otimes 2} \right\rangle dz \asymp M^{-2\alpha + \max(0, 1-2\beta)}$$
$$+ \ (2\gamma Br)^{-\frac{2\alpha + 2\beta - 1}{2\alpha}}$$
$$+ \ M^{-1} (2\gamma Br)^{-2 + \frac{1}{2\alpha}}. \qquad (30)$$

For the case $\alpha < 0.5$ or $\beta < 0.5$, applying a similar method to our objective yields

$$-\frac{1}{2\pi \mathrm{i}} \oint_\Gamma e^{-p_d Q(N)z} \left\langle \mathcal{L}(z), v^{\otimes 2} \right\rangle dz \asymp M^{-2\alpha + \max(0, 1-2\beta)}$$
$$+ \ \left(M^{\min(\alpha, 0.5)} Q(N)\right)^{-\frac{2\alpha + 2\beta - 1}{2\alpha}}, \qquad (31)$$

with details provided in Appendix K.1. Hence,

$$\left\langle e^{-\overline{K}_1 Q(N)}, (\boldsymbol{H}^{1/2}\boldsymbol{w}^*)^{\otimes 2} \right\rangle \asymp M^{-2\alpha + \max(0, 1-2\beta)}$$
$$+ \ \left(M^{\min(\alpha, 0.5)} Q(N)\right)^{-\frac{2\alpha + 2\beta - 1}{2\alpha}}. \qquad (32)$$

For the case $\alpha > 0.5$ and $\beta > 0.5$, a similar argument gives

$$-\frac{1}{2\pi \mathrm{i}} \oint_\Gamma e^{-p_d Q(N)z} \left\langle \mathcal{L}(z), v^{\otimes 2} \right\rangle dz \asymp M^{-2\alpha + \max(0, 1-2\beta)}$$
$$+ \ \left(M^{\min(\alpha, 0.5)} Q(N)\right)^{-\frac{2\alpha + 2\beta - 1}{2\alpha}}$$
$$+ \ M^{-1} \left(M^{\min(\alpha, 0.5)} Q(N)\right)^{-1 + \frac{1}{2\alpha}}, \qquad (33)$$

with details in Appendix K.1. Consequently,

$$\left\langle e^{-\overline{K}_1 Q(N)}, (\boldsymbol{H}^{1/2}\boldsymbol{w}^*)^{\otimes 2} \right\rangle \asymp M^{-2\alpha + \max(0, 1-2\beta)}$$
$$+ \ \left(M^{\min(\alpha, 0.5)} Q(N)\right)^{-\frac{2\alpha + 2\beta - 1}{2\alpha}}$$
$$+ \ M^{-1} \left(M^{\min(\alpha, 0.5)} Q(N)\right)^{-1 + \frac{1}{2\alpha}}. \qquad (34)$$

In summary, we obtain

$$L^{\mathrm{drift}}(N) + \|\boldsymbol{H}^{1/2}\boldsymbol{w}_\perp\|^2 \asymp M^{-2\alpha + \max(0, 1-2\beta)} + \left(M^{\min(\alpha, 0.5)} Q(N)\right)^{-\frac{2\alpha + 2\beta - 1}{2\alpha}}, \qquad (35)$$

for $\alpha < 0.5$ or $\beta < 0.5$, and

$$L^{\mathrm{drift}}(N) + \|\boldsymbol{H}^{1/2}\boldsymbol{w}_\perp\|^2 \asymp M^{-2\alpha + \max(0, 1-2\beta)}$$
$$+ \ \left(M^{\min(\alpha, 0.5)} Q(N)\right)^{-\frac{2\alpha + 2\beta - 1}{2\alpha}}$$
$$+ \ M^{-1} \left(M^{\min(\alpha, 0.5)} Q(N)\right)^{-1 + \frac{1}{2\alpha}}, \qquad (36)$$

for $\alpha > 0.5$ and $\beta > 0.5$.

Figure 20 shows that our transformed result in (35) and (36) based on deterministic approximation match the true signSGD trajectory up to a constant factor. When interpreting the figure, note that our analysis is asymptotic; hence, discrepancies may appear in the very early iterations.

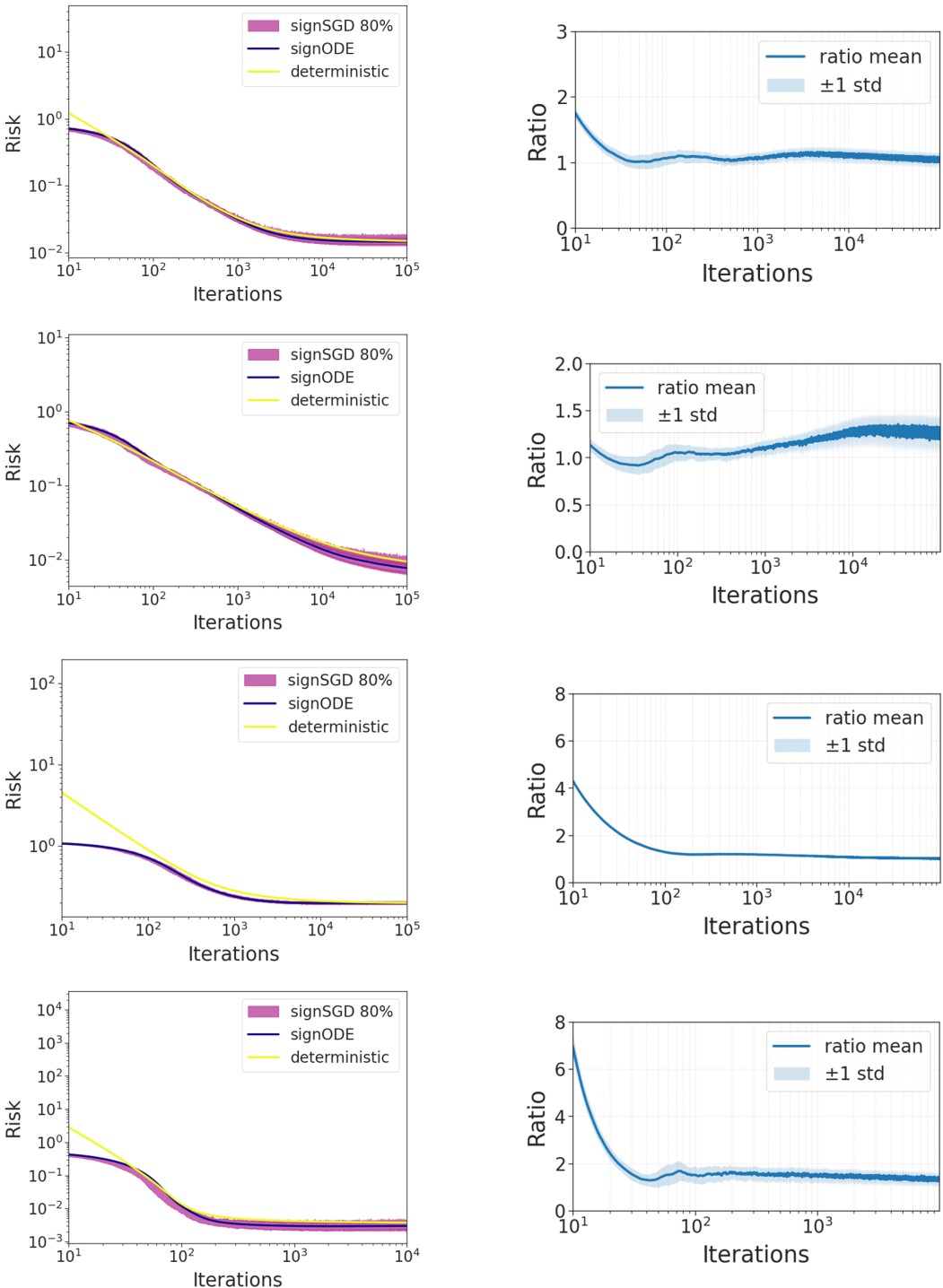

Figure 20: **Verification of the deterministic approximation and drift-term transformation.** Left: the purple curve denotes the 80% confidence interval of the true signSGD trajectory, the blue curve represents the numerical ODE solution, and the yellow curve corresponds to the deterministic approximation after drift-term transformation in (35) and (36). Deterministic approximation matches the true trajectory up to a constant factor. It should be noted that our analysis is asymptotic, and thus, discrepancies may occur in the very early iterations. Right: the ratio between the approximation and the true trajectory remains bounded by a constant factor, confirming the validity of our approach. Parameters: $(\alpha, \beta) = (0.7, 0.3), (1.0, 0), (0.4, 0.4), (0.7, 1.1)$ from top to bottom, $\gamma_0 = 0.003$, $f(z) = 1$, $M = 200$, $d = 800$, 100 runs.

### E.3 CONSTANT LEARNING RATE: PROXY AND VERIFICATION FOR THE CASE $\alpha < 0.5$ OR $\beta < 0.5$ (PHASE A)

Throughout this section, we set $f(z) \equiv 1$; hence

$$Q(N) = \frac{4\gamma_0}{\pi} \int_0^N \frac{du}{\sqrt{L(u)}}.$$

Applying the drift/approximation-term transformation to the ODE solution yields the implicit relation

$$L(N) \;\approx\; \underbrace{M^{-2\alpha+\max(0,1-2\beta)}}_{\textbf{approx}} + \underbrace{\left(M^{\min(\alpha,\,0.5)}\,Q(N)\right)^{-\frac{2\alpha+2\beta-1}{2\alpha}}}_{\textbf{drift}} \tag{37}$$

$$+ \underbrace{\frac{2\gamma_0^2}{\pi} \sum_{i=1}^M V_i \int_0^N \exp\!\left(-\frac{4\gamma_0}{\pi}\lambda_i(\overline{K})\!\int_z^N \frac{du}{\sqrt{L(u)}}\right) dz}_{\textbf{noise}}. \tag{38}$$

#### E.3.1 EARLY STAGE (DOMINANCE OF THE DRIFT TERM)

At $N = 0$, the noise integral is $0$, the approximation term is independent of $N$, and the drift term is large and decreases with $N$. Thus, in the early phase,

$$L(N) \;\approx\; \left(M^{\min(\alpha,\,0.5)}\,Q(N)\right)^{-\frac{2\alpha+2\beta-1}{2\alpha}}. \tag{39}$$

Since $Q(N) = \frac{4\gamma_0}{\pi}\int_0^N L(u)^{-1/2}\,du$, (39) is equivalent (up to absolute constants) to

$$L(N)^{-\frac{2\alpha}{2\alpha+2\beta-1}} \;\approx\; M^{\min(\alpha,\,0.5)}\,\gamma_0 \int_0^N \frac{du}{\sqrt{L(u)}}. \tag{40}$$

To obtain a proxy profile, we replace $\approx$ by equality in (40) and differentiate both sides:

$$-\frac{2\alpha}{2\alpha+2\beta-1}\,L(t)^{-\frac{2\alpha}{2\alpha+2\beta-1}-1}\,L'(t) \;=\; M^{\min(\alpha,\,0.5)}\,\gamma_0\,\frac{1}{\sqrt{L(t)}}. \tag{41}$$

Solving (41) for $L'(t)$ and separating variables gives the separable ODE

$$\frac{dL}{dt} = -\kappa\,L^{\zeta}, \qquad \zeta = \frac{2\alpha}{2\alpha+2\beta-1} + \frac{1}{2}, \qquad \kappa = \frac{2\alpha+2\beta-1}{2\alpha}\,M^{\min(\alpha,\,0.5)}\,\gamma_0.$$

Assuming $\zeta > 1$ (i.e. $2\alpha + 2\beta < 4\alpha + 1$), we integrate to obtain

$$-\frac{L(t)^{-(\zeta-1)}}{\zeta-1} = -\kappa t + \text{constant} \quad\Longrightarrow\quad L(t) = \left[(\zeta-1)\,\kappa\,t\right]^{-1/(\zeta-1)}. \tag{42}$$

Substituting $\zeta = \frac{2\alpha}{2\alpha+2\beta-1} + \frac{1}{2}$ and $\kappa = \frac{2\alpha+2\beta-1}{2\alpha}\,M^{\min(\alpha,\,0.5)}\,\gamma_0$ yields the early-phase proxy

$$L(N) \;\approx\; \left(\gamma_0\,M^{\min(\alpha,\,0.5)}\,N\right)^{-p}, \qquad p := \frac{2(2\alpha+2\beta-1)}{2\alpha+1-2\beta}. \tag{43}$$

By construction, (43) satisfies (40) (hence (39)) up to absolute constants.

#### E.3.2 LIMIT STAGE (STATIONARY ANALYSIS AND FLOOR)

With $f \equiv 1$, the mode-wise ODE is

$$\frac{dp_i}{dt} = -\frac{4}{\pi\sqrt{P(t)}}\,\lambda_i(\overline{K})\,p_i(t) + \frac{2\gamma_0}{\pi}\,V_i.$$

At stationarity, $p_i(t) \to s_i$ and $P(t) \to L_\infty$, we must have

$$-\frac{4}{\pi\sqrt{L_\infty}}\,\lambda_i(\overline{K})\,s_i + \frac{2\gamma_0}{\pi}\,V_i = 0 \quad\Longrightarrow\quad s_i = \frac{\gamma_0\sqrt{L_\infty}}{2\,\lambda_i(\overline{K})}\,V_i = \frac{\gamma_0\sqrt{L_\infty}}{2\,\lambda_i(\overline{K})}\,(\boldsymbol{w}_i^\mathsf{T}\boldsymbol{K}_\sigma\boldsymbol{K}\boldsymbol{u}_i).$$

Using the loss decomposition $P(t) = \sum_{i=1}^{M} p_i(t) + \|\boldsymbol{H}^{1/2}\boldsymbol{w}_\perp\|^2$, we obtain

$$L_\infty = \sum_{i=1}^{M} s_i + \|\boldsymbol{H}^{1/2}\boldsymbol{w}_\perp\|^2 = \frac{\gamma_0}{2}\Big(\sum_{i=1}^{M} \frac{\boldsymbol{w}_i^\mathsf{T}\boldsymbol{K}_\sigma\boldsymbol{K}\boldsymbol{u}_i}{\lambda_i(\overline{K})}\Big)\sqrt{L_\infty} + \|\boldsymbol{H}^{1/2}\boldsymbol{w}_\perp\|^2$$

$$= \frac{\gamma_0}{2}\operatorname{Tr}\big(\operatorname{diag}(\boldsymbol{K})^{1/2}\boldsymbol{K}_\sigma\big)\sqrt{L_\infty} + \|\boldsymbol{H}^{1/2}\boldsymbol{w}_\perp\|^2 = \frac{\gamma_0\pi}{4}\operatorname{Tr}\big(\operatorname{diag}(\boldsymbol{K})^{1/2}\big)\sqrt{L_\infty} + \|\boldsymbol{H}^{1/2}\boldsymbol{w}_\perp\|^2.$$

Solving the quadratic in $\sqrt{L_\infty}$ gives

$$L_\infty = \left(\frac{\frac{\gamma_0\pi}{4}\operatorname{Tr}\big(\operatorname{diag}(\boldsymbol{K})^{1/2}\big) + \sqrt{\big(\frac{\gamma_0\pi}{4}\operatorname{Tr}\big(\operatorname{diag}(\boldsymbol{K})^{1/2}\big)\big)^2 + 4\|\boldsymbol{H}^{1/2}\boldsymbol{w}_\perp\|^2}}{2}\right)^2 \tag{44}$$

$$\eqsim \max\Big\{\big(\gamma_0\operatorname{Tr}(\operatorname{diag}(\boldsymbol{K})^{1/2})\big)^2,\ \|\boldsymbol{H}^{1/2}\boldsymbol{w}_\perp\|^2\Big\}. \tag{45}$$

Under our setup,

$$\operatorname{Tr}(\operatorname{diag}(\boldsymbol{K})^{1/2}) = \sum_{i=1}^{M}\sqrt{(\boldsymbol{S}\boldsymbol{H}\boldsymbol{S}^\mathsf{T})_{ii}} \eqsim M \cdot \sqrt{\frac{1}{M}M^{\max(1-2\alpha,0)}} \eqsim M^{1-\min(\alpha,\,0.5)}.$$

By the results from Paquette et al. (2024); Lin et al. (2024), and note in Appendix K.3,

$$\|\boldsymbol{H}^{1/2}\boldsymbol{w}_\perp\|^2 \eqsim M^{-2\alpha+\max(0,\,1-2\beta)}.$$

Hence

$$L_\infty \eqsim \max\Big\{\gamma_0^2 M^{2-2\min(\alpha,\,0.5)},\ M^{-2\alpha+\max(0,\,1-2\beta)}\Big\}. \tag{46}$$

### E.3.3   PROXY

Combining the early-phase decay (43) with the floor (46), we adopt

$$L_{\mathrm{px}}(N) := \big(\gamma_0 M^{\min(\alpha,\,0.5)} N\big)^{-p} + \underbrace{\gamma_0^2 M^{2-2\min(\alpha,\,0.5)} + M^{-2\alpha+\max(0,\,1-2\beta)}}_{=:C},\qquad p = \frac{2(2\alpha+2\beta-1)}{2\alpha+1-2\beta}. \tag{47}$$

### E.3.4   VERIFICATION OF THE PROXY

We show that $L_{\mathrm{px}}$ satisfies (37) up to absolute constants. Equivalently, writing $Q_{L_{\mathrm{px}}}(N) := \frac{4\gamma_0}{\pi}\int_0^N \frac{du}{\sqrt{L_{\mathrm{px}}(u)}}$, we establish

$$\underbrace{\big(M^{\min(\alpha,\,0.5)} Q_{L_{\mathrm{px}}}(N)\big)^{-\frac{2\alpha+2\beta-1}{2\alpha}}}_{\textbf{drift}} + \underbrace{M^{-2\alpha+\max(0,\,1-2\beta)}}_{\textbf{approx}} + \underbrace{\frac{2\gamma_0^2}{\pi}\sum_{i=1}^{M} V_i\int_0^N \exp\Big(-\frac{4\gamma_0}{\pi}\lambda_i(\overline{K})\int_z^N \frac{du}{\sqrt{L_{\mathrm{px}}(u)}}\Big)dz}_{\textbf{noise}} \tag{48}$$

$$\eqsim \underbrace{\big(\gamma_0 M^{\min(\alpha,\,0.5)} N\big)^{-p} + C}_{L_{\mathrm{px}}(N)}. \tag{49}$$

**Lower Bound**   We prove

$$\textbf{drift} + \textbf{approx} + \textbf{noise} \gtrsim \big(\gamma_0 M^{\min(\alpha,\,0.5)} N\big)^{-p} + C. \tag{50}$$

Since $L_{\mathrm{px}}(u) \geq (\gamma_0 M^{\min(\alpha,\,0.5)} u)^{-p}$,

$$\textbf{drift} = \big(M^{\min(\alpha,\,0.5)} Q_{L_{\mathrm{px}}}(N)\big)^{-\frac{2\alpha+2\beta-1}{2\alpha}} \gtrsim \Big(M^{\min(\alpha,\,0.5)} \cdot \gamma_0 \int_0^N \big(\gamma_0 M^{\min(\alpha,\,0.5)} u\big)^{p/2}\,du\Big)^{-\frac{2\alpha+2\beta-1}{2\alpha}}$$

$$\eqsim \big(\gamma_0 M^{\min(\alpha,\,0.5)} N\big)^{-p}.$$

Since $L_{\mathrm{px}}(u) \geq C$ for all $u$,

$$\int_z^N \frac{du}{\sqrt{L_{\mathrm{px}}(u)}} \leq \frac{N-z}{\sqrt{C}}.$$

Hence

$$\mathbf{noise} \geq \frac{2\gamma_0^2}{\pi} \sum_{i=1}^M V_i \int_0^N \exp\left(-\frac{4\gamma_0}{\pi} \lambda_i(\overline{K}) \frac{N-z}{\sqrt{C}}\right) dz \tag{51}$$

$$= \frac{2\gamma_0^2}{\pi} \sum_{i=1}^M V_i \frac{\sqrt{C}}{\frac{4\gamma_0}{\pi} \lambda_i(\overline{K})} \left(1 - e^{-\frac{4\gamma_0}{\pi} \lambda_i(\overline{K}) \frac{N}{\sqrt{C}}}\right) \tag{52}$$

$$\gtrsim \gamma_0 \sqrt{C} \sum_{i=1}^M \frac{V_i}{\lambda_i(\overline{K})} = \frac{\gamma_0}{2} \operatorname{Tr}(\operatorname{diag}(K)^{1/2}) \sqrt{C} \asymp \gamma_0 M^{1-\min(\alpha,0.5)} \sqrt{C} \gtrsim \gamma_0^2 M^{2-2\min(\alpha,0.5)}. \tag{53}$$

Adding the approximation term $M^{-2\alpha+\max(0,1-2\beta)}$ gives $\mathbf{noise} + \mathbf{approx} \gtrsim C$. Combining with the drift contribution yields (50).

**Upper Bound**   We establish

$$\mathbf{drift} + \mathbf{approx} + \mathbf{noise} \lesssim \left(\gamma_0 M^{\min(\alpha,0.5)} N\right)^{-p} + C. \tag{54}$$

Let

$$A(N) := \max\left\{\left(\gamma_0 M^{\min(\alpha,0.5)} N\right)^{-p}, C\right\}, \qquad p = \frac{2(2\alpha+2\beta-1)}{2\alpha+1-2\beta}.$$

Then $L_{\mathrm{px}}(N) \asymp A(N)$. Define $N_0$ by $(\gamma_0 M^{\min(\alpha,0.5)} N_0)^{-p} = C$, i.e.

$$A(N) = \begin{cases} \left(\gamma_0 M^{\min(\alpha,0.5)} N\right)^{-p}, & N \leq N_0, \\ C, & N > N_0. \end{cases}$$

There exists a constant $B \geq 1$ such that

$$L_{\mathrm{px}}(N) \leq B\,A(N) \qquad (\forall N \geq 0). \tag{55}$$

**Upper bound for the drift term.** Since $L \leq BA$ by (73) and $Q$ is decreasing in its denominator,

$$\mathbf{drift} = \left(M^{\min(\alpha,0.5)} Q_L(N)\right)^{-\frac{2\alpha+2\beta-1}{2\alpha}} \lesssim \left(M^{\min(\alpha,0.5)} Q_{BA}(N)\right)^{-\frac{2\alpha+2\beta-1}{2\alpha}}.$$

We evaluate the right-hand side by cases.

*Case $N \leq N_0$.* Then $A(u) = (\gamma_0 M^{\min(\alpha,0.5)} u)^{-p}$ for $u \leq N$, so

$$Q_{BA}(N) = \frac{4\gamma_0}{\pi} \int_0^N \frac{du}{\sqrt{BA(u)}} = \frac{c}{\sqrt{B}} \gamma_0 \int_0^N \left(\gamma_0 M^{\min(\alpha,0.5)} u\right)^{p/2} du$$

for an absolute constant $c > 0$, which implies

$$\mathbf{drift} \lesssim \left(\gamma_0 M^{\min(\alpha,0.5)} N\right)^{-\frac{2\alpha+2\beta-1}{2\alpha}(1+p/2)} = \left(\gamma_0 M^{\min(\alpha,0.5)} N\right)^{-p}.$$

*Case $N > N_0$.* Split the integral at $N_0$:

$$M^{\min(\alpha,0.5)} Q_{BA}(N) = \frac{c}{\sqrt{B}} \gamma_0 M^{\min(\alpha,0.5)} \left[\int_0^{N_0} \left(\gamma_0 M^{\min(\alpha,0.5)} u\right)^{p/2} du + \int_{N_0}^N \frac{du}{\sqrt{BC}}\right]$$

$$= \frac{c}{\sqrt{B}} \left[\left(\gamma_0 M^{\min(\alpha,0.5)} N_0\right)^{1+p/2} + \gamma_0 M^{\min(\alpha,0.5)} \frac{N-N_0}{\sqrt{BC}}\right].$$

Raising to the power $-\frac{2\alpha+2\beta-1}{2\alpha}$ and using $(\gamma_0\, M^{\min(\alpha,\,0.5)}\, N_0)^{-p} = C$,

$$\textbf{drift} \lesssim \left[C^{-\frac{1+p/2}{p}} + \gamma_0\, \frac{N-N_0}{\sqrt{BC}}\right]^{-\frac{2\alpha+2\beta-1}{2\alpha}} \leq \left(C^{-\frac{1}{(2\alpha+2\beta-1)/(2\alpha)}}\right)^{-\frac{2\alpha+2\beta-1}{2\alpha}} = C.$$

Combining the two cases,

$$\textbf{drift} \lesssim \left(\gamma_0\, M^{\min(\alpha,\,0.5)}\, N\right)^{-p} + C. \tag{56}$$

**Upper bound for the noise integral.** By the monotonicity of $r \mapsto r^{-1/2}$,

$$\int_z^N \frac{du}{\sqrt{L(u)}} \geq \frac{1}{\sqrt{B}} \int_z^N \frac{du}{\sqrt{A(u)}}.$$

Therefore,

$$\textbf{noise} \leq \frac{2\gamma_0^2}{\pi} \sum_{i=1}^M V_i \int_0^N \exp\left(-\frac{4\gamma_0}{\pi\sqrt{B}}\, \lambda_i(\overline{K}) \int_z^N \frac{du}{\sqrt{A(u)}}\right) dz. \tag{57}$$

We again split into two cases.

*Case $N \leq N_0$.* Then $A(u) = (\gamma_0\, M^{\min(\alpha,\,0.5)}\, u)^{-p}$ on $[0, N]$, hence

$$\int_z^N \frac{du}{\sqrt{A(u)}} = \left(\gamma_0\, M^{\min(\alpha,\,0.5)}\right)^{p/2} \int_z^N u^{p/2}\, du = \left(\gamma_0\, M^{\min(\alpha,\,0.5)}\right)^{p/2} \frac{N^{1+p/2} - z^{1+p/2}}{1+p/2}.$$

Plugging this into (57) and factoring,

$$\begin{aligned}
\textbf{noise} &= \frac{2\gamma_0^2}{\pi} \sum_{i=1}^M V_i \int_0^N \exp\left(-\frac{4\gamma_0}{\pi\sqrt{B}}\, \lambda_i(\overline{K}) \left(\gamma_0\, M^{\min(\alpha,\,0.5)}\right)^{p/2} \frac{N^{1+p/2} - z^{1+p/2}}{1+p/2}\right) dz \\
&= \frac{2\gamma_0^2}{\pi} \sum_{i=1}^M V_i\, \exp\left(-\frac{4\gamma_0}{\pi\sqrt{B}}\, \lambda_i(\overline{K}) \left(\gamma_0\, M^{\min(\alpha,\,0.5)}\right)^{p/2} \frac{N^{1+p/2}}{1+p/2}\right) \\
&\qquad\qquad \times \int_0^N \exp\left(\frac{4\gamma_0}{\pi\sqrt{B}}\, \lambda_i(\overline{K}) \left(\gamma_0\, M^{\min(\alpha,\,0.5)}\right)^{p/2} \frac{z^{1+p/2}}{1+p/2}\right) dz.
\end{aligned}$$

Make the change of variables $y = z^{1+p/2}$ so that $dz = \frac{1}{1+p/2}\, y^{\frac{1}{1+p/2}-1}\, dy$ and the upper limit becomes $N^{1+p/2}$:

$$\textbf{noise} = \frac{2\gamma_0^2}{\pi} \sum_{i=1}^M V_i\, e^{-\alpha_i N^{1+p/2}} \int_0^{N^{1+p/2}} e^{\alpha_i y}\, \frac{1}{1+p/2}\, y^{\frac{1}{1+p/2}-1}\, dy,$$

$$\alpha_i := \frac{4\gamma_0}{\pi\sqrt{B}}\, \lambda_i(\overline{K})\, \frac{\left(\gamma_0\, M^{\min(\alpha,\,0.5)}\right)^{p/2}}{1+p/2}.$$

Let $X := N^{1+p/2}$ and

$$g(y) := \frac{1}{1+p/2}\, y^{\frac{1}{1+p/2}-1} = \frac{1}{1+p/2}\, y^{-\frac{p}{2+p}}.$$

Since $e^{\alpha_i y}$ is increasing and $g(y)$ is decreasing on $(0, X]$, Chebyshev's integral inequality (oppositely monotone) yields

$$\frac{1}{X} \int_0^X e^{\alpha_i y} g(y)\, dy \leq \left(\frac{1}{X} \int_0^X e^{\alpha_i y}\, dy\right)\left(\frac{1}{X} \int_0^X g(y)\, dy\right).$$

Hence

$$e^{-\alpha_i X}\int_0^X e^{\alpha_i y} g(y)\,dy \le e^{-\alpha_i X}\,\frac{e^{\alpha_i X}-1}{\alpha_i}\,\frac{1}{X}\int_0^X g(y)\,dy$$

$$= \frac{1-e^{-\alpha_i X}}{\alpha_i}\,\frac{1}{1+p/2}\cdot\frac{1}{1-\frac{p}{2+p}}\,X^{-\frac{p}{2+p}}$$

$$= \frac{1-e^{-\alpha_i X}}{\alpha_i}\,X^{-\frac{p}{2+p}}\qquad\left(\text{since }(1-\tfrac{p}{2+p})(1+\tfrac{p}{2})=1\right)$$

$$\le \frac{1}{\alpha_i}\,X^{-\frac{p}{2+p}} = \frac{1}{\alpha_i}\,N^{-p/2}.$$

Therefore

$$\textbf{noise}\ \le\ \frac{2\gamma_0^2}{\pi}\sum_{i=1}^M V_i\,\frac{1}{\alpha_i}\,N^{-p/2},$$

and with $\alpha_i=\frac{4\gamma_0}{\pi\sqrt{B}}\,\lambda_i(\overline{K})\,\frac{\left(\gamma_0 M^{\min(\alpha,\,0.5)}\right)^{p/2}}{1+p/2}$ this becomes

$$\textbf{noise}\ \le\ \frac{\gamma_0\sqrt{B}}{2}\,(1+p/2)\sum_{i=1}^M\frac{V_i}{\lambda_i(\overline{K})}\left(\gamma_0 M^{\min(\alpha,\,0.5)}\,N\right)^{-p/2}.$$

Using $\sum_i\frac{V_i}{\lambda_i(\overline{K})}=\mathrm{Tr}(\mathrm{diag}(\boldsymbol{K})^{1/2})\approx M^{1-\min(\alpha,\,0.5)}$, we get

$$\textbf{noise}\ \lesssim\ \gamma_0\,M^{1-\min(\alpha,\,0.5)}\left(\gamma_0 M^{\min(\alpha,\,0.5)}\,N\right)^{-p/2}$$

$$= \gamma_0\,M^{1-\min(\alpha,\,0.5)}\left(\gamma_0 M^{\min(\alpha,\,0.5)}\,N\right)^{p/2}\left(\gamma_0 M^{\min(\alpha,\,0.5)}\,N\right)^{-p}$$

$$\le \gamma_0\,M^{1-\min(\alpha,\,0.5)}\,\frac{1}{\sqrt{C}}\left(\gamma_0 M^{\min(\alpha,\,0.5)}\,N\right)^{-p}\ \lesssim\ \left(\gamma_0 M^{\min(\alpha,\,0.5)}\,N\right)^{-p},$$

where we used $(\gamma_0 M^{\min(\alpha,\,0.5)}\,N)^{p/2}\le(\gamma_0 M^{\min(\alpha,\,0.5)}\,N_0)^{p/2}=C^{-1/2}$.

*Case $N>N_0$.* Split the $z$–integral at $N_0$:

$$\textbf{noise}\le\frac{2\gamma_0^2}{\pi}\sum_{i=1}^M V_i\left[\int_0^{N_0}\exp\!\left(-\frac{4\gamma_0}{\pi\sqrt{B}}\,\lambda_i(\overline{K})\int_z^{N_0}\frac{du}{\sqrt{A(u)}}\right)dz+\int_{N_0}^N\exp\!\left(-\frac{4\gamma_0}{\pi\sqrt{B}}\,\lambda_i(\overline{K})\int_z^N\frac{du}{\sqrt{A(u)}}\right)dz\right].$$

The first integral is the $N=N_0$ case just handled, hence

$$\int_0^{N_0}\cdots dz\ \lesssim\ (\gamma_0 M^{\min(\alpha,\,0.5)}\,N_0)^{-p}=C.$$

For the second integral, we use that $A\equiv C$ on $[N_0,N]$:

$$\int_{N_0}^N\exp\!\left(-\frac{4\gamma_0}{\pi\sqrt{B}}\,\lambda_i(\overline{K})\int_z^N\frac{du}{\sqrt{A(u)}}\right)dz=\int_{N_0}^N\exp\!\left(-\frac{4\gamma_0}{\pi\sqrt{B}}\,\lambda_i(\overline{K})\,\frac{N-z}{\sqrt{C}}\right)dz$$

$$= \frac{\sqrt{C}}{\frac{4\gamma_0}{\pi\sqrt{B}}\,\lambda_i(\overline{K})}\left(1-e^{-\frac{4\gamma_0}{\pi\sqrt{B}}\,\lambda_i(\overline{K})\,\frac{N-N_0}{\sqrt{C}}}\right)$$

$$\le \frac{\pi\sqrt{B}}{4\gamma_0}\,\frac{\sqrt{C}}{\lambda_i(\overline{K})}.$$

Therefore,

$$\textbf{noise}\ \lesssim\ (\gamma_0 M^{\min(\alpha,\,0.5)}\,N_0)^{-p}+\frac{2\gamma_0^2}{\pi}\sum_{i=1}^M V_i\cdot\frac{\pi\sqrt{B}}{4\gamma_0}\,\frac{\sqrt{C}}{\lambda_i(\overline{K})}$$

$$= C+\frac{\gamma_0\sqrt{B}}{2}\,\sqrt{C}\sum_{i=1}^M\frac{V_i}{\lambda_i(\overline{K})}=C+\frac{\gamma_0\sqrt{B}}{2}\,\sqrt{C}\,\mathrm{Tr}(\mathrm{diag}(\boldsymbol{K})^{1/2})$$

$$\lesssim C+\gamma_0 M^{\min(\alpha,\,0.5)}\sqrt{C}\ \lesssim\ C+\sqrt{C}\cdot\sqrt{C}\ \lesssim\ C.$$

Combining both cases,

$$\mathbf{noise} \lesssim \left(\gamma_0 \, M^{\min(\alpha,\,0.5)} \, N\right)^{-p} + C. \tag{58}$$

**Conclusion of the upper bound.** From (56), (58), and $\mathbf{approx} = M^{-2\alpha+\max(0,\,1-2\beta)} \le C$, we obtain (54).

Finally, combining the lower bound (50) and the upper bound (54) proves (49). Therefore, the proxy (67) satisfies the implicit relation (37) up to absolute constants, with the three contributions labeled as **approx**, **drift**, and **noise**.

### E.4  CONSTANT LEARNING RATE: PROXY AND VERIFICATION FOR THE CASE $\alpha > 0.5$ AND $\beta > 0.5$ (PHASE B)

We now handle the case $\alpha > 0.5$ and $\beta > 0.5$. Since $\alpha > 0.5$, we have $\min(\alpha, 0.5) = 0.5$, and because $\beta > 0.5$, we have $\min(2\alpha, 2\alpha + 2\beta - 1) = 2\alpha$. Applying the drift/approximation-term transformation to the ODE solution yields

$$L(N) \asymp \underbrace{M^{-2\alpha}}_{\text{approx}} + \underbrace{\left(M^{1/2}Q(N)\right)^{-\frac{2\alpha+2\beta-1}{2\alpha}}}_{\text{drift}_1} + \underbrace{M^{-1}\left(M^{1/2}Q(N)\right)^{-1+\frac{1}{2\alpha}}}_{\text{drift}_2} \tag{59}$$

$$+ \underbrace{\frac{2\gamma_0^2}{\pi} \sum_{i=1}^{M} V_i \int_0^N \exp\left(-\frac{4\gamma_0}{\pi}\lambda_i(\overline{K})\int_z^N \frac{du}{\sqrt{L(u)}}\right) dz}_{\text{noise}}, \tag{60}$$

where

$$Q(N) = \frac{4\gamma_0}{\pi}\int_0^N \frac{du}{\sqrt{L(u)}}.$$

#### E.4.1  EARLY STAGE PROXIES (DRIFT$_1$ AND DRIFT$_2$)

We extract proxies from the two drift terms in (59) by the same differentiate-and-separate trick as before.

**drift$_1$:** $\left(M^{1/2}Q(N)\right)^{-(2\alpha+2\beta-1)/(2\alpha)}$.  Assuming this term dominates and replacing $\asymp$ by equality,

$$L(N)^{-\frac{2\alpha}{2\alpha+2\beta-1}} = M^{1/2}\gamma_0 \int_0^N \frac{du}{\sqrt{L(u)}}.$$

Differentiation gives the separable ODE $L'(t) = -\kappa_1 L(t)^{\beta_1}$ with

$$\beta_1 = \frac{2\alpha}{2\alpha+2\beta-1} + \frac{1}{2}, \qquad \kappa_1 = \frac{2\alpha+2\beta-1}{2\alpha}M^{1/2}\gamma_0.$$

For $\beta_1 > 1$ (equivalently $2\alpha + 2\beta < 4\alpha + 1$) we obtain

$$L_1(N) \asymp \left(\gamma_0 M^{1/2}N\right)^{-p_1}, \qquad p_1 = \frac{2(2\alpha+2\beta-1)}{2\alpha+1-2\beta}. \tag{61}$$

**drift$_2$:** $M^{-1}\left(M^{1/2}Q(N)\right)^{-1+\frac{1}{2\alpha}}$.  Assume $\alpha > \frac{1}{2}$ and, in the early phase, the second drift term dominates:

$$L(N) \asymp M^{-1}\left(M^{1/2}Q(N)\right)^{-\frac{2\alpha-1}{2\alpha}}, \qquad Q(N) \asymp \gamma_0 \int_0^N \frac{du}{\sqrt{L(u)}}.$$

Expanding the $M$–exponent,

$$\left(M^{1/2}Q\right)^{-\frac{2\alpha-1}{2\alpha}} = M^{-\frac{(2\alpha-1)}{4\alpha}}Q^{-\frac{2\alpha-1}{2\alpha}},$$

hence

$$L(N) \;\eqsim\; M^{-\frac{6\alpha-1}{4\alpha}} \left(\gamma_0 I(N)\right)^{-\frac{2\alpha-1}{2\alpha}}, \qquad I(N) := \int_0^N \frac{du}{\sqrt{L(u)}}. \tag{62}$$

Raise both sides of (62) to the power $-\frac{2\alpha}{2\alpha-1}$ so that the integral becomes linear:

$$L(N)^{-\frac{2\alpha}{2\alpha-1}} \;=\; M^{\frac{6\alpha-1}{4\alpha-2}} \gamma_0 \, I(N) \;\eqsim\; M^{\frac{6\alpha-1}{4\alpha-2}} \gamma_0 \int_0^N \frac{du}{\sqrt{L(u)}}. \tag{63}$$

Differentiating (63) with respect to $t$ yields

$$-\frac{2\alpha}{2\alpha-1} \, L(t)^{-\frac{2\alpha}{2\alpha-1}-1} L'(t) = M^{\frac{6\alpha-1}{4\alpha-2}} \gamma_0 \frac{1}{\sqrt{L(t)}}.$$

Rearranging gives a separable ODE of the usual power form

$$L'(t) \;=\; -\kappa_2 \, L(t)^{\beta_2}, \qquad \beta_2 \;=\; \frac{2\alpha}{2\alpha-1} + \frac{1}{2} \;=\; \frac{6\alpha-1}{4\alpha-2} \;>\; 1, \tag{64}$$

with

$$\kappa_2 \;=\; \frac{2\alpha-1}{2\alpha} \, \gamma_0 \, M^{\frac{6\alpha-1}{4\alpha-2}} \;>\; 0. \tag{65}$$

Since $\beta_2 > 1$, solving (64) gives

$$L(t)^{-(\beta_2-1)} \;=\; (\beta_2-1)\,\kappa_2 \, t + \text{const.}$$

Absorbing harmless absolute constants into $\eqsim$ and setting $t = N$,

$$L_2(N) \;\eqsim\; \left(\gamma_0 \, M^{\frac{6\alpha-1}{4\alpha-2}} \, N\right)^{-p_2}, \qquad p_2 \;=\; \frac{1}{\beta_2-1} \;=\; \boxed{\frac{2(2\alpha-1)}{2\alpha+1}}. \tag{66}$$

**Crossover scale.** Equating (61) and (66) gives

$$N_1 \;\eqsim\; \gamma_0^{-1} \, M^\eta, \qquad \eta = \frac{2\alpha+1-4\beta}{4\beta},$$

so $R_1$ dominates for $N \lesssim N_1$ and $L_2$ for $N \gtrsim N_1$ (when $\alpha > 0.5$ and $0.5 < \beta < \alpha + 0.5$).

### E.4.2  LIMIT STAGE (APPROX AND NOISE FLOORS)

As in the case $\alpha < 0.5$ or $\beta < 0.5$, the stationary analysis with $f \equiv 1$ yields

$$L_\infty \;\eqsim\; \max\{\gamma_0^2 \, \text{Tr}(\text{diag}(\boldsymbol{K})^{1/2})^2, \|\boldsymbol{H}^{1/2}\boldsymbol{w}_\perp\|^2\}.$$

Under our standing model $\text{Tr}(\text{diag}(\boldsymbol{K})^{1/2}) \eqsim M^{0.5}$ and by the results from Paquette et al. (2024); Lin et al. (2024), and note in Appendix K.3, $\|\boldsymbol{H}^{1/2}\boldsymbol{w}_\perp\|^2 \eqsim M^{-2\alpha}$, hence the floor

$$C \;:=\; \gamma_0^2 \, M \;+\; M^{-2\alpha}.$$

### E.4.3  COMBINED PROXY

$$\begin{aligned}
L_{\text{px}}(N) &:= L_1(N) + L_2(N) + C \\
&= \left(\gamma_0 \, M^{0.5} N\right)^{-p_1} \;+\; \left(\gamma_0 \, M^{\frac{6\alpha-1}{4\alpha-2}} N\right)^{-p_2} \;+\; C,
\end{aligned} \tag{67}$$

where

$$p_1 = \frac{2(2\alpha+2\beta-1)}{2\alpha+1-2\beta}, \qquad p_2 = \frac{4\alpha-2}{2\alpha+1}.$$

### E.4.4  VERIFICATION OF THE PROXY

We show that $L_{\text{px}}$ satisfies (59) up to absolute constants.

**Lower bound.** We claim

$$\underbrace{\left(M^{0.5}Q_{L_{\text{px}}}(N)\right)^{-\frac{2\alpha+2\beta-1}{2\alpha}}}_{\text{drift}_1} + \underbrace{M^{-1}\left(M^{0.5}Q_{L_{\text{px}}}(N)\right)^{-1+\frac{1}{2\alpha}}}_{\text{drift}_2} + \underbrace{M^{-2\alpha}}_{\text{approx}} \tag{68}$$

$$+ \underbrace{\frac{2\gamma_0^2}{\pi}\sum_{i=1}^{M} V_i \int_0^N \exp\left(-\frac{4\gamma_0}{\pi}\lambda_i(\overline{K})\int_z^N \frac{du}{\sqrt{L_{\text{px}}(u)}}\right)dz}_{\text{noise}} \gtrsim L_{\text{px}}(N). \tag{69}$$

*Drift part.* Using $L_{\text{px}} \geq R_1$ inside $Q$,

$$\left(M^{0.5}Q_{L_{\text{px}}}(N)\right)^{-\frac{2\alpha+2\beta-1}{2\alpha}} \gtrsim \left(M^{0.5}\gamma_0\int_0^N \frac{du}{\sqrt{L_1(u)}}\right)^{-\frac{2\alpha+2\beta-1}{2\alpha}} \asymp \left(\gamma_0 M^{0.5}N\right)^{-p_1} \asymp L_1(N).$$

Similarly, using $L_{\text{px}} \geq L_2$ inside $Q$,

$$M^{-1}\left(M^{0.5}Q_{L_{\text{px}}}(N)\right)^{-1+\frac{1}{2\alpha}} \gtrsim M^{-1}\left(M^{0.5}\gamma_0\int_0^N \frac{du}{\sqrt{L_2(u)}}\right)^{-1+\frac{1}{2\alpha}} \asymp \left(\gamma_0 M^{\frac{6\alpha-1}{4\alpha-2}}N\right)^{-p_2} \asymp L_2(N).$$

Therefore,

$$\text{drift}_1 + \text{drift}_2 \gtrsim L_1(N) + L_2(N). \tag{70}$$

*Noise + approx.* Since $L_{\text{px}} \geq C$,

$$\int_z^N \frac{du}{\sqrt{L_{\text{px}}(u)}} \leq \frac{N-z}{\sqrt{C}}.$$

As in the Equation 53,

$$\text{noise} \gtrsim \gamma_0\sqrt{C}\sum_{i=1}^{M} \frac{V_i}{\lambda_i(\overline{K})} = \frac{\gamma_0}{2}\,\text{Tr}\big(\text{diag}(K)^{1/2}\big)\sqrt{C} \asymp \gamma_0 M^{0.5}\sqrt{C} \gtrsim \gamma_0^2 M.$$

Thus noise + approx $\gtrsim C$. Together with (70), this proves (69).

**Upper bound.** We will prove

$$\underbrace{\left(M^{0.5}Q_{L_{\text{px}}}(N)\right)^{-\frac{2\alpha+2\beta-1}{2\alpha}}}_{\text{drift}_1} + \underbrace{M^{-1}\left(M^{0.5}Q_{L_{\text{px}}}(N)\right)^{-1+\frac{1}{2\alpha}}}_{\text{drift}_2} + \underbrace{M^{-2\alpha}}_{\text{approx}} \tag{71}$$

$$+ \underbrace{\frac{2\gamma_0^2}{\pi}\sum_{i=1}^{M} V_i \int_0^N \exp\left(-\frac{4\gamma_0}{\pi}\lambda_i(\overline{K})\int_z^N \frac{du}{\sqrt{L_{\text{px}}(u)}}\right)dz}_{\text{noise}} \lesssim L_{\text{px}}(N). \tag{72}$$

Let

$$A(N) = \begin{cases} (\gamma_0 M^{0.5}N)^{-p_1}, & N \leq N_1, \\ \left(\gamma_0 M^{\frac{6\alpha-1}{4\alpha-2}}N\right)^{-p_2}, & N_1 \leq N \leq N_2, \\ C, & N > N_2, \end{cases}$$

where $N_1$ and $N_2$ are the crossover points between the three terms. There exists a constant $B \geq 1$ such that

$$L_{\text{px}}(N) \leq B\,A(N) \qquad (\forall N \geq 0). \tag{73}$$

It suffices to show

$$\underbrace{\left(M^{0.5}Q_{B\cdot A}(N)\right)^{-\frac{2\alpha+2\beta-1}{2\alpha}}}_{\text{drift}_1} + \underbrace{M^{-1}\left(M^{0.5}Q_{B\cdot A}(N)\right)^{-1+\frac{1}{2\alpha}}}_{\text{drift}_2} + \underbrace{M^{-2\alpha}}_{\text{approx}} \tag{74}$$

$$+ \underbrace{\frac{2\gamma_0^2}{\pi}\sum_{i=1}^{M} V_i \int_0^N \exp\left(-\frac{4\gamma_0}{\pi}\lambda_i(\overline{K})\int_z^N \frac{du}{\sqrt{B\cdot A(u)}}\right)dz}_{\text{noise}} \lesssim L_{\text{px}}(N). \tag{75}$$

**Case** $N \leq N_1$. It is enough to prove

$$\underbrace{\left(M^{0.5}Q_{B \cdot A}(N)\right)^{-\frac{2\alpha+2\beta-1}{2\alpha}}}_{\text{drift}_1} + \underbrace{M^{-1}\left(M^{0.5}Q_{B \cdot A}(N)\right)^{-1+\frac{1}{2\alpha}}}_{\text{drift}_2} + \underbrace{M^{-2\alpha}}_{\text{approx}} \tag{76}$$

$$+ \underbrace{\frac{2\gamma_0^2}{\pi}\sum_{i=1}^{M} V_i \int_0^N \exp\left(-\frac{4\gamma_0}{\pi}\lambda_i(\overline{K})\int_z^N \frac{du}{\sqrt{B \cdot A(u)}}\right) dz}_{\text{noise}} \lesssim (\gamma_0\, M^{0.5} N)^{-p_1}. \tag{77}$$

We have $M^{-2\alpha} \lesssim (\gamma_0\, M^{0.5}N)^{-p_1}$ directly. Also, the following holds with straightforward integration.

$$\left(M^{0.5}Q_{B \cdot A}(N)\right)^{-\frac{2\alpha+2\beta-1}{2\alpha}} \approx (\gamma_0\, M^{0.5}N)^{-p_1}.$$

Since $N \leq N_1 \approx \gamma_0^{-1} M^\eta$ with $\eta = \frac{2\alpha+1-4\beta}{4\beta}$, following holds by integration and calculation.

$$M^{-1}\left(M^{0.5}Q_{B \cdot A}(N)\right)^{-1+\frac{1}{2\alpha}} \lesssim (\gamma_0\, M^{0.5}N)^{-p_1}.$$

Finally, arguing as in the $N \leq N_0$ case of Section E.3.4,

$$\frac{2\gamma_0^2}{\pi}\sum_{i=1}^{M} V_i \int_0^N \exp\left(-\frac{4\gamma_0}{\pi}\lambda_i(\overline{K})\int_z^N \frac{du}{\sqrt{B \cdot A(u)}}\right) dz \lesssim (\gamma_0\, M^{0.5}N)^{-p_1}.$$

Hence, the claim holds for $N \leq N_1$.

**Case** $N_1 \leq N \leq N_2$. We will show

$$\underbrace{\left(M^{0.5}Q_{BA}(N)\right)^{-\frac{2\alpha+2\beta-1}{2\alpha}}}_{\text{drift}_1} + \underbrace{M^{-1}\left(M^{0.5}Q_{BA}(N)\right)^{-1+\frac{1}{2\alpha}}}_{\text{drift}_2} + \underbrace{M^{-2\alpha}}_{\text{approx}} \tag{78}$$

$$+ \underbrace{\frac{2\gamma_0^2}{\pi}\sum_{i=1}^{M} V_i \int_0^N \exp\left(-\frac{4\gamma_0}{\pi}\lambda_i(\overline{K})\int_z^N \frac{du}{\sqrt{BA(u)}}\right) dz}_{\text{noise}} \lesssim \left(\gamma_0\, M^{\frac{6\alpha-1}{4\alpha-2}}N\right)^{-p_2}, \tag{79}$$

where

$$p_1 = \frac{2(2\alpha+2\beta-1)}{2\alpha+1-2\beta}, \qquad p_2 = \frac{2(2\alpha-1)}{2\alpha+1}, \qquad A(u) = \begin{cases} (\gamma_0\, M^{0.5}u)^{-p_1}, & u \leq N_1, \\ (\gamma_0\, M^{\frac{6\alpha-1}{4\alpha-2}}u)^{-p_2}, & N_1 < u \leq N, \end{cases}$$

and $Q_{BA}(N) = \frac{4\gamma_0}{\pi}\int_0^N \frac{du}{\sqrt{BA(u)}}$.

**Approx term.** Since $N \leq N_2$,

$$M^{-2\alpha} \lesssim \left(\gamma_0\, M^{\frac{6\alpha-1}{4\alpha-2}}N\right)^{-p_2}.$$

**Drift term.** If $N_1 \leq N \leq 2N_1$, using the case $N \leq N_1$, we get an inequality for two drift terms.

$$\underbrace{\left(M^{0.5}Q_{BA}(N)\right)^{-\frac{2\alpha+2\beta-1}{2\alpha}}}_{\text{drift}_1} + \underbrace{M^{-1}\left(M^{0.5}Q_{BA}(N)\right)^{-1+\frac{1}{2\alpha}}}_{\text{drift}_2} \tag{80}$$

$$\leq \underbrace{\left(M^{0.5}Q_{BA}(N_1)\right)^{-\frac{2\alpha+2\beta-1}{2\alpha}}}_{\text{drift}_1} + \underbrace{M^{-1}\left(M^{0.5}Q_{BA}(N_1)\right)^{-1+\frac{1}{2\alpha}}}_{\text{drift}_2} \tag{81}$$

$$\lesssim (\gamma_0\, M^{0.5}N_1)^{-p_1} \lesssim \left(\gamma_0\, M^{\frac{6\alpha-1}{4\alpha-2}}N\right)^{-p_2}. \tag{82}$$

So while covering the drift term, we will temporarily assume $2N_1 \leq N$.

**Lower bound on** $Q_{BA}(N)$. Split the integral at $N_1$:

$$Q_{BA}(N) \approx \gamma_0 \int_0^{N_1} \frac{du}{\sqrt{A(u)}} + \gamma_0 \int_{N_1}^N \frac{du}{\sqrt{A(u)}} =: \gamma_0\,(I_1 + I_2). \tag{83}$$

For the first part, using $A(u) = (\gamma_0 M^{0.5} u)^{-p_1}$ on $[0, N_1]$,

$$I_1 = (\gamma_0 M^{0.5})^{p_1/2} \int_0^{N_1} u^{p_1/2} \, du = \frac{(\gamma_0 M^{0.5})^{p_1/2}}{1 + p_1/2} \, N_1^{1+p_1/2}. \tag{84}$$

For the second part, using $A(u) = (\gamma_0 M^{\frac{6\alpha-1}{4\alpha-2}} u)^{-p_2}$ on $[N_1, N]$,

$$I_2 = (\gamma_0 M^{\frac{6\alpha-1}{4\alpha-2}})^{p_2/2} \int_{N_1}^N u^{p_2/2} \, du = \frac{(\gamma_0 M^{\frac{6\alpha-1}{4\alpha-2}})^{p_2/2}}{1 + p_2/2} \left( N^{1+p_2/2} - N_1^{1+p_2/2} \right). \tag{85}$$

Since we temporarily assumed $N \geq 2N_1$, we have

$$I_2 \gtrsim (\gamma_0 M^{\frac{6\alpha-1}{4\alpha-2}})^{p_2/2} N^{1+p_2/2}.$$

Hence, from (83),

$$Q_{BA}(N) \gtrsim \gamma_0 \, (\gamma_0 M^{\frac{6\alpha-1}{4\alpha-2}})^{p_2/2} \, N^{1+p_2/2}. \tag{86}$$

**drift$_1$ vs. drift$_2$.** From $N \geq N_1$ and (86), we have $Q_{BA}(N) \geq Q_{BA}(N_1)$. It follows that

$$\text{drift}_1 = \left( M^{0.5} Q_{BA}(N) \right)^{-\frac{2\alpha+2\beta-1}{2\alpha}} \leq M^{-1} \left( M^{0.5} Q_{BA}(N) \right)^{-1+\frac{1}{2\alpha}} = \text{drift}_2,$$

so it suffices to control drift$_2$.

**drift$_2$ bound.** Using (86),

$$\text{drift}_2 = M^{-1} \left( M^{0.5} \, Q_{BA}(N) \right)^{-1+\frac{1}{2\alpha}}$$

$$\lesssim M^{-1} \left( M^{0.5} \cdot \gamma_0^{1+p_2/2} M^{\frac{6\alpha-1}{4\alpha-2} \cdot \frac{p_2}{2}} N^{1+p_2/2} \right)^{-1+\frac{1}{2\alpha}}. \tag{87}$$

Now compute the exponents of $N$, $\gamma_0$, and $M$ separately.

*(i) $N$–exponent:*

$$\left( 1 + \tfrac{p_2}{2} \right) \left( -1 + \tfrac{1}{2\alpha} \right) = \left( 1 + \frac{2\alpha-1}{2\alpha+1} \right) \left( \frac{1}{2\alpha} - 1 \right) = \frac{4\alpha}{2\alpha+1} \cdot \left( -\frac{2\alpha-1}{2\alpha} \right) = -\frac{2(2\alpha-1)}{2\alpha+1} = -p_2.$$

*(ii) $\gamma_0$–exponent:* the same calculation as in (i) gives $-p_2$.

*(iii) $M$–exponent:* the total exponent equals

$$-1 + \left( -1 + \tfrac{1}{2\alpha} \right) \left( 0.5 + \frac{6\alpha-1}{4\alpha-2} \cdot \frac{p_2}{2} \right).$$

A direct simplification shows this equals $-\frac{6\alpha-1}{4\alpha-2} p_2$. Therefore, from (87),

$$\text{drift}_2 \lesssim \left( \gamma_0 M^{\frac{6\alpha-1}{4\alpha-2}} N \right)^{-p_2}. \tag{88}$$

Since drift$_1 \leq$ drift$_2$, we also have drift$_1 \lesssim (\gamma_0 M^{\frac{6\alpha-1}{4\alpha-2}} N)^{-p_2}$.

**Noise bound.** It suffices to show

$$\frac{2\gamma_0^2}{\pi} \sum_{i=1}^M V_i \int_0^{N_1} \exp\left( -\frac{4\gamma_0}{\pi} \lambda_i(\overline{K}) \int_z^N \frac{du}{\sqrt{B \cdot A(u)}} \right) dz \tag{89}$$

$$+ \frac{2\gamma_0^2}{\pi} \sum_{i=1}^M V_i \int_{N_1}^N \exp\left( -\frac{4\gamma_0}{\pi} \lambda_i(\overline{K}) \int_z^N \frac{du}{\sqrt{B \cdot A(u)}} \right) dz \lesssim \left( \gamma_0 M^{\frac{6\alpha-1}{4\alpha-2}} N \right)^{-p_2}. \tag{90}$$

*Integral over $[N_1, N]$.* As in the case $N \leq N_0$ of Section E.3.4, with $A(u) = (\gamma_0 M^{\frac{6\alpha-1}{4\alpha-2}} u)^{-p_2}$ on $[N_1, N]$,

$$\frac{2\gamma_0^2}{\pi} \sum_{i=1}^M V_i \int_{N_1}^N \exp\left( -\frac{4\gamma_0}{\pi} \lambda_i(\overline{K}) \int_z^N \frac{du}{\sqrt{B \cdot A(u)}} \right) dz \lesssim \left( \gamma_0 M^{\frac{6\alpha-1}{4\alpha-2}} N \right)^{-p_2}.$$

*Integral over* $[0, N_1]$. First,

$$\frac{2\gamma_0^2}{\pi} \sum_{i=1}^{M} V_i \int_0^{N_1} \exp\left(-\frac{4\gamma_0}{\pi}\lambda_i(\overline{K})\int_z^N \frac{du}{\sqrt{B \cdot A(u)}}\right) dz \leq \frac{2\gamma_0^2}{\pi} \sum_{i=1}^{M} V_i \int_0^{N_1} \exp\left(-\frac{4\gamma_0}{\pi}\lambda_i(\overline{K})\int_z^{N_1} \frac{du}{\sqrt{B \cdot A(u)}}\right) dz.$$

As in the case $N \leq N_0$ of Section E.3.4,

$$\frac{2\gamma_0^2}{\pi} \sum_{i=1}^{M} V_i \int_0^{N_1} \exp\left(-\frac{4\gamma_0}{\pi}\lambda_i(\overline{K})\int_z^{N_1} \frac{du}{\sqrt{B \cdot A(u)}}\right) dz \lesssim \sqrt{C}\,\big(\gamma_0 M^{0.5} N\big)^{-p_1/2} \lesssim \big(\gamma_0\, M^{\frac{6\alpha-1}{4\alpha-2}}\, N_1\big)^{-p_2}.$$

If $N \leq 2N_1$, this already implies

$$\frac{2\gamma_0^2}{\pi} \sum_{i=1}^{M} V_i \int_0^{N_1} \exp\left(-\frac{4\gamma_0}{\pi}\lambda_i(\overline{K})\int_z^N \frac{du}{\sqrt{B \cdot A(u)}}\right) dz \lesssim \big(\gamma_0\, M^{\frac{6\alpha-1}{4\alpha-2}}\, N\big)^{-p_2}.$$

If $N > 2N_1$, then

$$\int_0^{N_1} \exp\left(-\frac{4\gamma_0}{\pi}\lambda_i(\overline{K})\int_z^N \frac{du}{\sqrt{B \cdot A(u)}}\right) dz \leq N_1\,\exp\left(-\frac{4\gamma_0}{\pi}\lambda_i(\overline{K})\int_{N_1}^N \frac{du}{\sqrt{B \cdot A(u)}}\right),$$

and, using $e^{-x} \leq 1/x$ together with the lower bound $\int_{N_1}^N \frac{du}{\sqrt{B \cdot A(u)}} \gtrsim N^{1+p_2/2}\,(\gamma_0 M^{\frac{6\alpha-1}{4\alpha-2}})^{p_2/2}$, we get

$$\frac{2\gamma_0^2}{\pi} \sum_{i=1}^{M} V_i \int_0^{N_1} \cdots dz \lesssim \frac{2\gamma_0^2}{\pi} \sum_{i=1}^{M} V_i\, \frac{N_1}{\frac{4\gamma_0}{\pi}\lambda_i(\overline{K})\int_{N_1}^N \frac{du}{\sqrt{B \cdot A(u)}}}$$

$$\lesssim \gamma_0 \sum_{i=1}^{M} \frac{V_i}{\lambda_i(\overline{K})}\,\big(\gamma_0 M^{\frac{6\alpha-1}{4\alpha-2}} N\big)^{-p_2/2}$$

$$\asymp \gamma_0\, M^{0.5}\,\big(\gamma_0 M^{\frac{6\alpha-1}{4\alpha-2}} N\big)^{-p_2/2} \lesssim \big(\gamma_0\, M^{\frac{6\alpha-1}{4\alpha-2}}\, N\big)^{-p_2},$$

where the last step uses $\gamma_0 M^{0.5} \leq (\gamma_0 M^{\frac{6\alpha-1}{4\alpha-2}} N)^{-p_2/2}$ which holds from $N \leq N_2$.

Combining the $[N_1, N]$ and $[0, N_1]$ bounds yields

$$\text{noise} \lesssim \big(\gamma_0\, M^{\frac{6\alpha-1}{4\alpha-2}}\, N\big)^{-p_2},$$

as required for the case $N_1 \leq N \leq N_2$.

**Case $N \geq N_2$.** We have $M^{-2\alpha} \lesssim C$ directly. As in the above case,

$$\big(M^{0.5}Q_{B \cdot A}(N)\big)^{-\frac{2\alpha+2\beta-1}{2\alpha}} \leq M^{-1}\big(M^{0.5}Q_{B \cdot A}(N)\big)^{-1+\frac{1}{2\alpha}}.$$

Using the estimate from the previous case,

$$M^{-1}\big(M^{0.5}Q_{B \cdot A}(N)\big)^{-1+\frac{1}{2\alpha}} \lesssim M^{-1}\big(M^{0.5}Q_{B \cdot A}(N_2)\big)^{-1+\frac{1}{2\alpha}} \lesssim \big(\gamma_0\, M^{\frac{6\alpha-1}{4\alpha-2}} N_2\big)^{-p_2} \lesssim C.$$

Finally, as in the $N > N_0$ case of Section E.3.4,

$$\frac{2\gamma_0^2}{\pi} \sum_{i=1}^{M} V_i \int_0^N \exp\left(-\frac{4\gamma_0}{\pi}\lambda_i(\overline{K})\int_z^N \frac{du}{\sqrt{B \cdot A(u)}}\right) dz \lesssim C.$$

Therefore, the bound holds for $N \geq N_2$ as well.

## E.5 NOTE ON THE REGIME $\beta > \alpha + 0.5$

When $\beta > \alpha + 0.5$, the assumption $\zeta > 1$ used in step 42 no longer holds. In this case, the first drift term takes a different form:

$$L_{\text{drift}_1}(N) \asymp \left(1 - \kappa\gamma_0 M^{\min(\alpha, 0.5)} N\right)^{\frac{2(2\alpha+2\beta-1)}{2\beta-2\alpha-1}},$$

for a finite horizon and some constant $\kappa$. Inserting the max function, we can represent it as a global function.

$$L_{\text{drift}_1}(N) \ \eqsim \ \left(\max\left(1 - \kappa\,\gamma_0\,M^{\min(\alpha,0.5)}\,N, 0\right)\right)^{\frac{2(2\alpha+2\beta-1)}{2\beta-2\alpha-1}}.$$

Now we explain the behavior of the term. When $N$ is asymptotically smaller than $(\gamma_0\,M^{\min(\alpha,0.5)})^{-1}$, the term is asymptotically constant. On $N \ \eqsim \ (\gamma_0\,M^{\min(\alpha,0.5)})^{-1}$, the term suddenly drops from a constant scale to 0.

For the case $\alpha < 0.5$ or $\beta < 0.5$ the valid proxy is

$$L_{\text{px}}(N) := \left(\max\left(1 - \kappa\,\gamma_0\,M^{\min(\alpha,0.5)}\,N, 0\right)\right)^{\frac{2(2\alpha+2\beta-1)}{2\beta-2\alpha-1}} + \gamma_0^2\,M^{2-2\min(\alpha,\,0.5)} + M^{-2\alpha+\max(0,\,1-2\beta)},$$

and for the case $\alpha > 0.5$ and $\beta > 0.5$ the valid proxy is

$$L_{\text{px}}(N) := \left(\max\left(1 - \kappa\,\gamma_0\,M^{0.5}\,N, 0\right)\right)^{\frac{2(2\alpha+2\beta-1)}{2\beta-2\alpha-1}} + M^{-\frac{6\alpha-1}{2\alpha+1}}\,(N\gamma_0)^{-\frac{2(2\alpha-1)}{2\alpha+1}} + \gamma_0^2\,M + M^{-2\alpha}.$$

These satisfy the implicit integral equation, same as Sections E.3.4 and E.4.4.

Therefore, for the case $\alpha < 0.5$, $\beta > \alpha + 0.5$,

$$R(M, N, \gamma_0) = \left(\max\left(1 - \kappa\,\gamma_0\,M^{\alpha}\,N, 0\right)\right)^{\frac{2(2\alpha+2\beta-1)}{2\beta-2\alpha-1}} + \gamma_0^2\,M^{2-2\alpha} + M^{-2\alpha}, \qquad (91)$$

and for the case $\alpha > 0.5$ and $\beta > 0.5$,

$$R(M, N, \gamma_0) = \left(\max\left(1 - \kappa\,\gamma_0\,M^{0.5}\,N, 0\right)\right)^{\frac{2(2\alpha+2\beta-1)}{2\beta-2\alpha-1}} + M^{-\frac{6\alpha-1}{2\alpha+1}}\,(N\gamma_0)^{-\frac{2(2\alpha-1)}{2\alpha+1}} + \gamma_0^2\,M + M^{-2\alpha}.$$
$$(92)$$

## F   DERIVATION OF THE COMPUTE-OPTIMAL RESULT

**Goal.**   The main goal of this section is to derive compute-optimal scaling laws of signSGD in the following form:

$$M^\star \asymp \mathfrak{f}^\xi, \qquad R\left(M^\star, \tfrac{\mathfrak{f}}{M^\star}, \gamma_0^\star\right) \asymp \mathfrak{f}^{-\eta}.$$

Here $R(M, N, \gamma_0)$ denote the $L(\boldsymbol{\theta}_N)$ under learning rate $\gamma_0$ and fixed model size $M$. We define the computational budget in terms of FLOPS as $\mathfrak{f} = MN$, and consider the optimal model size $M^\star$ under fixed $\mathfrak{f}$, and optimal scaling of learning rate in the form $\gamma_0^\star = M^{-e^*}$.

**Proof Overview.**   Substituting the learning rate $\gamma_0 = M^{-e}$ into our loss formula

$$R(M, N, \gamma_0) \asymp M^{-2\alpha+\max(0,\,1-2\beta)} + \left(M^{\min(\alpha,0.5)} N\gamma_0\right)^{-\frac{2(2\alpha+2\beta-1)}{2\alpha-2\beta+1}}$$
$$+ M^{-\frac{6\alpha-1}{2\alpha+1}}\left(N\gamma_0\right)^{-\frac{2(2\alpha-1)}{2\alpha+1}} + \gamma_0^2\, M^{2-\min(1,2\alpha)},$$

we can represent the risk as a function of three variables $M$, $N$, $e$, and two parameters $\alpha$, $\beta$.

Then for fixed compute $\mathfrak{f} = MN$, we substitute $M = \mathfrak{f}^x$ and $N = \mathfrak{f}^{1-x}$ to express the risk as the function of three variables $\mathfrak{f}$, $x$, $e$ and two parameters $\alpha$, $\beta$. Four terms in the loss formula convert to four terms with exponential of FLOPS $\mathfrak{f}$ with exponent functions $\ell_1$ to $\ell_4$.

$$R(\mathfrak{f}, x, e, \alpha, \beta) \asymp \mathfrak{f}^{-\ell_1(x,e,\alpha,\beta)} + \mathfrak{f}^{-\ell_2(x,e,\alpha,\beta)} + \mathfrak{f}^{-\ell_3(x,e,\alpha,\beta)} + \mathfrak{f}^{-\ell_4(x,e,\alpha,\beta)}.$$

Since each term is a power of $\mathfrak{f}$, and assuming $\mathfrak{f} \geq 1$, the loss simplifies to

$$R(\mathfrak{f}, x, e) \asymp \mathfrak{f}^{-h(x,e,\alpha,\beta)}, \quad \text{where } h(x,e,\alpha,\beta) = \min(\ell_1, \ell_2, \ell_3, \ell_4).$$

We find the optimal learning rate exponent $e^*$ and the optimal model size exponent by

$$x^*, e^* = \arg\max_{x,e} h(x, e, \alpha, \beta).$$

As we optimize over two variables $x$ and $e$, three terms among $\ell_1$ to $\ell_4$ balance on the optimal values $x^*$ and $e^*$.

Then the optimal learning rate is $\gamma_0^* = M^{-e^*}$, and the optimal model size is $M^\star = \mathfrak{f}^{x^*}$. Finally, the compute-optimal scaling law is

$$R(M^\star, \mathfrak{f}/M^\star, \gamma_0^*) = \mathfrak{f}^{-h(x^*,e^*,\alpha,\beta)},$$

and $h(x^*, e^*, \alpha, \beta)$ will be the compute-optimal slope in absolute value.

### F.1   COMPUTE-OPTIMAL RESULT FOR MAXIMAL LEARNING RATE

We now discuss the maximal learning rate case deferred from the main text. Note that Paquette et al. (2024) showed that the maximal learning rate for SGD is $\gamma_0 \asymp 1$ when $\alpha > \frac{1}{2}$, and $\gamma_0 \asymp M^{-(1-2\alpha)}$ when $\alpha < \frac{1}{2}$.

Now, we discuss the maximal learning rate for signSGD. Because the noise term is $\gamma_0^2\, M^{2-\min(1,2\alpha)}$, stability requires

$$\gamma_0^2\, M^{2-\min(1,2\alpha)} \lesssim 1.$$

Otherwise, the signSGD noise term explodes as $M$ grows. This condition is satisfied by choosing

$$\gamma_0 = M^{-1+\min(\alpha,0.5)},$$

which ensures $\gamma_0^2\, M^{2-\min(1,2\alpha)} \asymp 1$ while the other terms still decay appropriately.

For $\alpha < 0.5$, the term

$$\left(M^{\min(\alpha,0.5)} N\,\gamma_0\right)^{-\frac{2(2\alpha+2\beta-1)}{2\alpha-2\beta+1}} = \left(M^{-(1-2\alpha)} N\right)^{-\frac{2(2\alpha+2\beta-1)}{2\alpha-2\beta+1}}$$

decreases with $N$ but increases with $M$. However, under a fixed compute budget $\mathfrak{f} = MN$, one can allocate resources so that this term does not cause an exploding loss; hence we do not classify it as unstable.

Thus, the maximal learning rate for signSGD is

$$\gamma_0 = M^{-1+\min(\alpha, 0.5)}.$$

In this case, however, we obtain $R(M, N, \gamma_0) \eqsim 1$, so the slope of the compute-optimal curve is always zero.

## F.2 DERIVATION OF COMPUTE-OPTIMAL RESULT FOR OPTIMAL LEARNING RATE

We assume $\alpha + \beta > 0.5$ throughout, even for the case where it is not specified.

### F.2.1 $\alpha > 0.5$, $\beta < 0.5$ (PHASE A$a$)

We start from

$$R(M, N, \gamma_0) \eqsim \left(M^{1/2} N \gamma_0\right)^{-\frac{2(2\alpha+2\beta-1)}{2\alpha+1-2\beta}} + M^{-(2\alpha+2\beta-1)} + \gamma_0^2 M.$$

Substitute

$$\gamma_0 = M^{-e}, \qquad N = \frac{\mathfrak{f}}{M}, \qquad M = \mathfrak{f}^x,$$

so that, up to constant factors,

$$R \eqsim \mathfrak{f}^{\max\{\ell_1(x), \ell_2(x), \ell_3(x)\}},$$

where

$$\ell_1(x) = -(2\alpha + 2\beta - 1)\,x,$$
$$\ell_2(x) = \frac{2(2\alpha + 2\beta - 1)}{2\alpha + 1 - 2\beta}\left(e + \tfrac{1}{2}\right)x \;-\; \frac{2(2\alpha + 2\beta - 1)}{2\alpha + 1 - 2\beta},$$
$$\ell_3(x) = (1 - 2e)\,x.$$

We minimize the convex, piecewise–linear function $f(x, e) = \max_i \ell_i(x, e)$ over $x \in (0, 1)$ and $e \in \mathbb{R}$. By convexity, any interior minimizer must occur at a kink where at least two lines are active. In our regime $\alpha + \beta > 0.5$ and $\beta < \alpha + 0.5$, the only admissible triple intersection is $\{\ell_1, \ell_2, \ell_3\}$. Solving $\ell_1 = \ell_3$ and $\ell_2 = \ell_3$ yields

$$e^* = \alpha + \beta, \qquad x^* = \frac{1}{2\alpha + 1}, \qquad h^* = \ell_1(x^*) = \ell_2(x^*) = \ell_3(x^*) = -\frac{2\alpha + 2\beta - 1}{2\alpha + 1}.$$

To verify that this kink is the global minimizer, note first that $x^* \in (0, 1)$ when $\alpha > 0.5$, hence it is interior. Next, the subgradient optimality condition for convex max-of-lines problems requires $(0, 0) \in \partial f(x^*, e^*)$. At $(x^*, e^*)$ the active lines have slopes that straddle zero in both coordinates:

$$\partial_x \ell_1 = -(2\alpha+2\beta-1) < 0, \quad \partial_x \ell_2 = \frac{2(2\alpha + 2\beta - 1)}{2\alpha + 1 - 2\beta}\left(e^* + \tfrac{1}{2}\right) > 0, \quad \partial_x \ell_3 = 1 - 2e^* = 1 - 2(\alpha+\beta) < 0,$$

and

$$\partial_e \ell_1 = 0, \qquad \partial_e \ell_2 = \frac{2(2\alpha + 2\beta - 1)}{2\alpha + 1 - 2\beta}\,x^* > 0, \qquad \partial_e \ell_3 = -2x^* < 0.$$

Since 0 lies in the convex hull of the active slopes in both $x$ and $e$, we have $(0, 0) \in \partial f(x^*, e^*)$, so the interior triple intersection is the global minimizer; no boundary check is needed.

$$\boxed{\gamma_0 = M^{-(\alpha+\beta)}, \quad M^\star \eqsim \mathfrak{f}^{1/(2\alpha+1)}, \quad R\left(M^\star, \tfrac{\mathfrak{f}}{M^\star}\right) \eqsim \mathfrak{f}^{-\frac{2\alpha+2\beta-1}{2\alpha+1}}.}$$

### F.2.2 $\alpha < 0.5$, $\beta < 0.5$ (PHASE A$b$)

We start from

$$R(M, N, \gamma_0) = \left(M^\alpha N \gamma_0\right)^{-\frac{2(2\alpha + 2\beta - 1)}{2\alpha + 1 - 2\beta}} + M^{-(2\alpha + 2\beta - 1)} + \gamma_0^2 M^{2 - 2\alpha}.$$

Substitute

$$\gamma_0 = M^{-e}, \qquad N = \frac{\mathfrak{f}}{M}, \qquad M = \mathfrak{f}^x,$$

so that, up to constant factors,

$$R \asymp \mathfrak{f}^{\max\{\ell_1(x), \ell_2(x), \ell_3(x)\}},$$

where

$$\ell_1(x) = -(2\alpha + 2\beta - 1)\, x,$$
$$\ell_2(x) = -\frac{2(2\alpha + 2\beta - 1)}{2\alpha + 1 - 2\beta} \left(\alpha - e - 1\right) x \; - \; \frac{2(2\alpha + 2\beta - 1)}{2\alpha + 1 - 2\beta},$$
$$\ell_3(x) = \left(2 - 2\alpha - 2e\right) x.$$

We minimize the convex, piecewise–linear function $f(x, e) = \max_i \ell_i(x, e)$ over $x \in (0, 1)$ and $e \in \mathbb{R}$. Under our standing assumptions $\alpha + \beta > 0.5$ and $\beta < \alpha + 0.5$, the only admissible triple intersection is $\{\ell_1, \ell_2, \ell_3\}$. Solving $\ell_1 = \ell_3$ and $\ell_2 = \ell_3$ gives

$$e^* = \beta + \tfrac{1}{2}, \qquad x^* = \tfrac{1}{2}, \qquad h^* = \ell_1(x^*) = \ell_2(x^*) = \ell_3(x^*) = -\frac{2\alpha + 2\beta - 1}{2}.$$

To certify optimality, note that $x^* \in (0, 1)$ (since $x^* = \tfrac{1}{2}$) and check the subgradient condition $(0, 0) \in \partial f(x^*, e^*)$. At $(x^*, e^*)$ the active lines have slopes straddling zero in both coordinates:

$$\partial_x \ell_1 = -(2\alpha + 2\beta - 1) < 0, \quad \partial_x \ell_2 = \frac{2(2\alpha + 2\beta - 1)}{2\alpha + 1 - 2\beta}(e^* + 1 - \alpha) > 0, \quad \partial_x \ell_3 = 2 - 2\alpha - 2e^* = 1 - 2(\alpha + \beta) < 0,$$

and

$$\partial_e \ell_1 = 0, \qquad \partial_e \ell_2 = \frac{2(2\alpha + 2\beta - 1)}{2\alpha + 1 - 2\beta}\, x^* > 0, \qquad \partial_e \ell_3 = -2x^* < 0.$$

Hence $0$ lies in the convex hull of the active slopes in both variables, so the interior kink $(x^*, e^*)$ is the global minimizer; no boundary check is required.

$$\boxed{\gamma_0 = M^{-(\beta + 0.5)}, \quad M^\star \asymp \mathfrak{f}^{1/2}, \quad R\left(M^\star, \tfrac{\mathfrak{f}}{M^\star}\right) \asymp \mathfrak{f}^{-\frac{2\alpha + 2\beta - 1}{2}}.}$$

### F.2.3 $\alpha < 0.5$, $0.5 < \beta < \alpha + 0.5$ (PHASE A$c$)

We start from

$$R(M, N, \gamma_0) = \left(M^\alpha N \gamma_0\right)^{-\frac{2(2\alpha + 2\beta - 1)}{2\alpha + 1 - 2\beta}} + M^{-2\alpha} + \gamma_0^2 M^{2 - 2\alpha}.$$

Substitute

$$\gamma_0 = M^{-e}, \qquad N = \frac{\mathfrak{f}}{M}, \qquad M = \mathfrak{f}^x,$$

so that, up to constant factors,

$$R \asymp \mathfrak{f}^{\max\{\ell_1(x), \ell_2(x), \ell_3(x)\}},$$

where

$$\ell_1(x) = -2\alpha\, x,$$
$$\ell_2(x) = -\frac{2(2\alpha + 2\beta - 1)}{2\alpha + 1 - 2\beta} \left(\alpha - e - 1\right) x \; - \; \frac{2(2\alpha + 2\beta - 1)}{2\alpha + 1 - 2\beta},$$
$$\ell_3(x) = \left(2 - 2\alpha - 2e\right) x.$$

We minimize the convex, piecewise–linear objective $f(x, e) = \max_i \ell_i(x, e)$ over $x \in (0, 1)$ and $e \in \mathbb{R}$. In the regime $\alpha + \beta > 0.5$ and $\beta < \alpha + 0.5$ (with $\alpha < 0.5 < \beta$), the only admissible triple intersection is $\{\ell_1, \ell_2, \ell_3\}$. Solving $\ell_1 = \ell_3$ and $\ell_2 = \ell_3$ yields

$$e^* = 1, \qquad x^* = \frac{2\alpha + 2\beta - 1}{-4\alpha\beta + 6\alpha + 4\beta - 2}, \qquad h^* = \ell_1(x^*) = \ell_2(x^*) = \ell_3(x^*) = -\frac{2\alpha\,(2\alpha + 2\beta - 1)}{-4\alpha\beta + 6\alpha + 4\beta - 2}.$$

One checks that the denominator is positive in this regime and exceeds the positive numerator $2\alpha + 2\beta - 1$, hence $x^* \in (0, 1)$.

*Interior optimality.* At $(x^*, e^*)$ the active lines' slopes straddle zero in both coordinates:

$$\partial_x \ell_1 = -2\alpha < 0, \quad \partial_x \ell_2 = \frac{2(2\alpha + 2\beta - 1)}{2\alpha + 1 - 2\beta}(e^* + 1 - \alpha) > 0, \quad \partial_x \ell_3 = 2 - 2\alpha - 2e^* = -2\alpha < 0,$$

and

$$\partial_e \ell_1 = 0, \qquad \partial_e \ell_2 = \frac{2(2\alpha + 2\beta - 1)}{2\alpha + 1 - 2\beta} x^* > 0, \qquad \partial_e \ell_3 = -2x^* < 0.$$

Thus $(0, 0) \in \partial f(x^*, e^*)$ and, with $x^* \in (0, 1)$, the interior kink is the global minimizer; no boundary check is required.

$$\boxed{\gamma_0 = M^{-1}, \quad M^\star \approx \mathfrak{f}^{\frac{2\alpha + 2\beta - 1}{-4\alpha\beta + 6\alpha + 4\beta - 2}}, \quad R(M^\star, \tfrac{\mathfrak{f}}{M^\star}) \approx \mathfrak{f}^{-\frac{2\alpha\,(2\alpha + 2\beta - 1)}{-4\alpha\beta + 6\alpha + 4\beta - 2}}.}$$

### F.2.4 $\alpha > 0.5,\ 0.5 < \beta < \alpha + 0.5$ (PHASE B$a$)

We start from

$$R(M, N, \gamma_0) = \left(M^{1/2} N \gamma_0\right)^{-\frac{2(2\alpha + 2\beta - 1)}{2\alpha + 1 - 2\beta}} + \left(M^{\frac{6\alpha - 1}{4\alpha - 2}} N \gamma_0\right)^{-\frac{2(2\alpha - 1)}{2\alpha + 1}} + M^{-2\alpha} + \gamma_0^2 M.$$

Substitute $\gamma_0 = M^{-e}$, $N = \mathfrak{f}/M$, $M = \mathfrak{f}^x$. Then, up to $\mathfrak{f}$-independent factors,

$$R \approx \mathfrak{f}^{\max_{i=1,\ldots,4} \ell_i(x, e)},$$

where

$$\ell_1(x) = -2\alpha\, x,$$
$$\ell_2(x) = \frac{2(2\alpha + 2\beta - 1)}{2\alpha + 1 - 2\beta}\left(e + \tfrac{1}{2}\right)x - \frac{2(2\alpha + 2\beta - 1)}{2\alpha + 1 - 2\beta},$$
$$\ell_3(x) = \left(\frac{2(2\alpha - 1)}{2\alpha + 1} e - 1\right)x - \frac{2(2\alpha - 1)}{2\alpha + 1},$$
$$\ell_4(x) = (1 - 2e)\, x.$$

We minimize the convex, piecewise–linear function $f(x, e) = \max_i \ell_i(x, e)$ over $x \in (0, 1)$, $e \in \mathbb{R}$. In the regime $\alpha > 0.5$, $\beta > 0.5$, the only admissible interior kink with three active lines is $\{\ell_2, \ell_3, \ell_4\}$. Solving $\ell_2 = \ell_4$ and $\ell_3 = \ell_4$ yields

$$e^* = \frac{2\alpha + 4\beta - 1}{4\beta}, \qquad x^* = \frac{\beta}{\alpha + \beta}, \qquad h^* = \ell_2(x^*, e^*) = \ell_3(x^*, e^*) = \ell_4(x^*, e^*) = -\frac{2\alpha + 2\beta - 1}{2\alpha + 2\beta}.$$

*Interior optimality.* First, $x^* \in (0, 1)$ since $\alpha, \beta > 0.5$. Second, the subgradient condition $(0, 0) \in \partial f(x^*, e^*)$ holds because the active slopes straddle zero in both variables:

$$\partial_x \ell_2 = \tfrac{2(2\alpha + 2\beta - 1)}{2\alpha + 1 - 2\beta}\left(e^* + \tfrac{1}{2}\right) > 0, \quad \partial_x \ell_3 = \tfrac{2(2\alpha - 1)}{2\alpha + 1} e^* - 1 < 0, \quad \partial_x \ell_4 = 1 - 2e^* = \tfrac{1 - 2\alpha - 2\beta}{2\beta} < 0,$$

and

$$\partial_e \ell_2 = \tfrac{2(2\alpha + 2\beta - 1)}{2\alpha + 1 - 2\beta} x^* > 0, \qquad \partial_e \ell_3 = \tfrac{2(2\alpha - 1)}{2\alpha + 1} x^* > 0, \qquad \partial_e \ell_4 = -2x^* < 0.$$

Hence $(x^*, e^*)$ is the global minimizer among interior points. It remains to exclude $\ell_1$ at $(x^*, e^*)$:

$$\ell_1(x^*) = -2\alpha \frac{\beta}{\alpha + \beta} \leq -\frac{2\alpha + 2\beta - 1}{2(\alpha + \beta)} = h^*,$$

since $4\alpha\beta - 2\alpha - 2\beta + 1 = 4(\alpha - \tfrac{1}{2})(\beta - \tfrac{1}{2}) \geq 0$ for $\alpha, \beta > 0.5$. Therefore the triple intersection $\{\ell_2, \ell_3, \ell_4\}$ is the global optimum.

$$\boxed{\gamma_0 = M^{-\frac{2\alpha + 4\beta - 1}{4\beta}}, \quad M^\star \approx \mathfrak{f}^{\frac{\beta}{\alpha + \beta}}, \quad R\left(M^\star, \tfrac{\mathfrak{f}}{M^\star}\right) \approx \mathfrak{f}^{-\frac{2\alpha + 2\beta - 1}{2\alpha + 2\beta}}.}$$

### F.2.5 $\alpha < 0.5,\ \beta > \alpha + 0.5$ (PHASE A$d$)

Recall the loss formula (91)

$$R(M, N, \gamma_0) = \Big(\max\big(1 - \kappa\,\gamma_0\,M^\alpha\,N, 0\big)\Big)^{\frac{2(2\alpha+2\beta-1)}{2\beta-2\alpha-1}} + \gamma_0^2\,M^{2-2\alpha} + M^{-2\alpha}.$$

Note that the drift term vanishes at $N \asymp (\gamma_0\,M^\alpha)^{-1}$.

Let $\gamma_0 = M^{-e}$. Note that because of the approximation error $M^{-2\alpha}$, there is no gain from setting $e$ bigger than 1. So we will only consider the case $e \le 1$. In that case, loss is a constant scale before $N \asymp M^{e-\alpha}$, and it drops to the scale of $M^{-2e-2\alpha+2}$.

Since a constant scale loss cannot be compute-optimal, the loss $M^{-2e-2\alpha+2}$ at $N \asymp M^{e-\alpha}$ will be a candidate for the compute-optimal point. In that case $\mathfrak{f} = MN = M^{1+e-\alpha}$ holds and it leads to $M = \mathfrak{f}^{\frac{1}{1+e-\alpha}}$. So the loss $M^{-2e-2\alpha+2}$ has the size $\mathfrak{f}^{\frac{-2e-2\alpha+2}{1+e-\alpha}}$.

Since $e = 1$ minimizes $\frac{-2e-2\alpha+2}{1+e-\alpha}$, $\gamma_0 = M^{-1}$ is the optimal learning rate. This leads to the following result.

$$\boxed{\gamma_0 = M^{-1}, \quad M^\star \asymp \mathfrak{f}^{\frac{1}{2-\alpha}}, \quad R\Big(M^\star, \tfrac{\mathfrak{f}}{M^\star}\Big) \asymp \mathfrak{f}^{-\frac{2\alpha}{2-\alpha}}.}$$

### F.2.6 $\alpha > 0.5,\ \beta > \alpha + 0.5$ (PHASE B$b$)

Recall the loss formula (92)

$$R(M, N, \gamma_0) = \Big(\max\big(1 - \kappa\,\gamma_0\,M^{0.5}\,N, 0\big)\Big)^{\frac{2(2\alpha+2\beta-1)}{2\beta-2\alpha-1}} + M^{-\frac{6\alpha-1}{2\alpha+1}}(N\gamma_0)^{-\frac{2(2\alpha-1)}{2\alpha+1}} + \gamma_0^2\,M + M^{-2\alpha}.$$

Note that the first term vanishes at $N \asymp (\gamma_0\,M^\alpha)^{-1}$. At that point second term becomes $M^{-\frac{6\alpha-1}{2\alpha+1}}(N\gamma_0)^{-\frac{2(2\alpha-1)}{2\alpha+1}} \asymp M^{-\frac{4\alpha}{2\alpha+1}}$.

As we optimize over three parameters $N, M, \gamma_0$, and one constraint $\mathfrak{f} = MN$, we have two degrees of freedom. So this means three terms may balance together at the compute-optimal point.

The first possible case is the balance of the first three terms, and in this case, $\gamma_0^2 M = M^{-\frac{4\alpha}{2\alpha+1}}$ and $N \asymp (\gamma_0\,M^\alpha)^{-1}$ must hold. Here, the loss is $M^{-\frac{4\alpha}{2\alpha+1}}$ and $\mathfrak{f} = MN = M^{\frac{2\alpha+1}{4\alpha+1}}$ holds, so the loss is $\mathfrak{f}^{-\frac{4\alpha}{4\alpha+1}}$.

The second possible case is the balance of the last three terms, and after solving the equations, the loss is $\mathfrak{f}^{-\frac{2\alpha}{2\alpha+1}}$.

The first case has a steeper decay, so it is the compute-optimal. This leads to the following result.

$$\boxed{\gamma_0 = M^{-\frac{6\alpha+1}{4\alpha+2}}, \quad M^\star \asymp \mathfrak{f}^{\frac{2\alpha+1}{4\alpha+1}}, \quad R\Big(M^\star, \tfrac{\mathfrak{f}}{M^\star}\Big) \asymp \mathfrak{f}^{-\frac{4\alpha}{4\alpha+1}}.}$$

### F.3 DISCUSSION FOR THE SUBOPTIMAL LEARNING RATE

In this section, we calculate the compute-optimal exponent for a general size of learning rate in the form of $\gamma_0 = M^{-e}$. We will focus on Phase Aa. In that phase, the maximal learning rate was $\gamma_0 = M^{-1/2}$ and optimal learning rate was $\gamma_0^\star = M^{-(\alpha+\beta)}$.

In this section, we will calculate the compute-optimal exponent for general $e \ge 1/2$.

Recall that we have the following loss formula for Phase Aa.

$$R(M, N, \gamma_0) \asymp \big(M^{1/2}N\gamma_0\big)^{-\frac{2(2\alpha+2\beta-1)}{2\alpha+1-2\beta}} + M^{-(2\alpha+2\beta-1)} + \gamma_0^2\,M.$$

For the case $1/2 \le e \le (\alpha+\beta)$, $\big(M^{1/2}N\gamma_0\big)^{-\frac{2(2\alpha+2\beta-1)}{2\alpha+1-2\beta}}$ and $\gamma_0^2 M$ are dominant terms. Substituting $\gamma_0 = M^{-e}$ and balancing them, we get $N = M^{\left(\frac{4\alpha}{2\alpha+2\beta-1}\right)(e-1/2)}$. As $\mathfrak{f} = MN$ holds, it leads

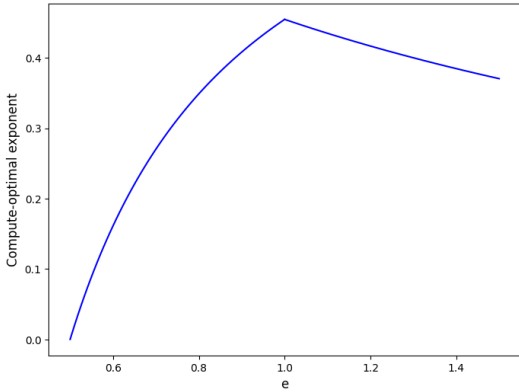

Figure 21: **Compute-optimal exponent with respect to** $e$ **of** $\gamma_0 = M^{-e}$ **for** $(\alpha, \beta) = (0.6, 0.4)$**.** Colored line shows the compute-optimal exponent $x$ in the formula $R\left(M^\star, \frac{\mathfrak{f}}{M^\star}, \gamma_0\right) \eqsim \mathfrak{f}^{-x}$.

to

$$M^\star \eqsim \mathfrak{f}^{1/((\frac{4\alpha}{2\alpha+2\beta-1})(e-1/2)+1)}, \quad R\left(M^\star, \frac{\mathfrak{f}}{M^\star}, \gamma_0\right) \eqsim \mathfrak{f}^{-\frac{(2e-1)(2\alpha+2\beta-1)}{2\alpha(2e-1)+(2\alpha+2\beta-1)}}.$$

For the case $e \geq (\alpha + \beta)$, $\left(M^{1/2}N\gamma_0\right)^{-\frac{2(2\alpha+2\beta-1)}{2\alpha+1-2\beta}}$ and $M^{-(2\alpha+2\beta-1)}$ are dominant terms. Substituting $\gamma_0 = M^{-e}$ and balancing them, we get $N = M^{\alpha-\beta+e}$. As $\mathfrak{f} = MN$ holds, it leads to

$$M^\star \eqsim \mathfrak{f}^{1/(\alpha-\beta+e+1)}, \quad R\left(M^\star, \frac{\mathfrak{f}}{M^\star}, \gamma_0\right) \eqsim \mathfrak{f}^{-\frac{2\alpha+2\beta-1}{\alpha-\beta+e+1}}.$$

In Figure 21, we provide a graph of the compute-optimal exponent with respect to $e$ of $\gamma_0 = M^{-e}$ for $(\alpha, \beta) = (0.6, 0.4)$. As the graph is continuous, the absolute value of the compute-optimal exponent gradually decreases as we move away from the optimal choice. Also, we can observe that the degradation is smaller for the learning rates with larger $e$ in $\gamma_0 = M^{-e}$ than that of the optimal learning rate. So in terms of tuning the learning rate, we may aggressively set a high $e$ in $\gamma_0 = M^{-e}$ for the initial attempt, and gradually decrease the $e$ for later attempts.

## G  DERIVATION FOR THE STABLE-DECAY AND WARMUP-STABLE-DECAY SCHEDULING

We first derive a scaling-law bound for the stable-decay schedule, a simplified variant of the warmup–stable-decay (WSD) schedule, and then extend it to the full warmup–stable-decay schedule in Section G.4.

We set the learning rate as $\gamma_k = \gamma_0 f(k)$. Previously, we considered the constant–learning–rate case ($f \equiv 1$). In this section, we start with a general decaying learning rate by taking $f$ to be a decreasing function, and then substitute the stable-decay scheduling. Throughout, for simplicity, we assume $\alpha > 0.5$ and $\beta < 0.5$ (Phase Aa).

Recall the implicit integral equation (26):

$$L(N) = \|\boldsymbol{H}^{1/2}\boldsymbol{w}_\perp\|^2 + \sum_{i=1}^{M} r_i(0)\,\exp\!\left(-\frac{4\lambda_i\gamma_0}{\pi}\int_0^N \frac{f(u)}{\sqrt{L(u)}}\,du\right) \tag{93}$$

$$+\frac{2\gamma_0^2}{\pi}\sum_{i=1}^{M}V_i\int_0^N \exp\!\left(-\frac{4\lambda_i\gamma_0}{\pi}\int_z^N \frac{f(u)}{\sqrt{L(u)}}\,du\right)f(z)^2\,dz. \tag{94}$$

Also recall Equations 27 and 28.

$$L^{\mathrm{drift}}(N) = \sum_{i=1}^{M} r_i(0)\,e^{-\frac{4\lambda_i\gamma_0}{\pi}\int_0^N \frac{f(u)}{\sqrt{L(u)}}\,du}, \qquad L^{\mathrm{noise}}(N) = \frac{2\gamma_0^2}{\pi}\sum_{i=1}^{M}V_i\int_0^N e^{-\frac{4\lambda_i\gamma_0}{\pi}\int_z^N \frac{f(u)}{\sqrt{L(u)}}\,du}f(z)^2\,dz. \tag{95}$$

$$L(N) = \|\boldsymbol{H}^{1/2}\boldsymbol{w}_\perp\|^2 + L^{\mathrm{drift}}(N) + L^{\mathrm{noise}}(N). \tag{96}$$

Recall also the drift/approximation transformation (35):

$$L^{\mathrm{drift}}(N) + \|\boldsymbol{H}^{1/2}\boldsymbol{w}_\perp\|^2 \;\eqsim\; M^{-(2\alpha+2\beta-1)} \;+\; \left(M^{0.5}Q_L(N)\right)^{-\frac{2\alpha+2\beta-1}{2\alpha}},$$

$$Q_L(z) := \frac{4\gamma_0}{\pi}\int_0^z \frac{f(u)}{\sqrt{L(u)}}\,du.$$

Hence,

$$L(N) \;\eqsim\; M^{-(2\alpha+2\beta-1)} \;+\; \left(M^{0.5}Q_L(N)\right)^{-\frac{2\alpha+2\beta-1}{2\alpha}} \tag{97}$$

$$+\;\frac{2\gamma_0^2}{\pi}\sum_{i=1}^{M}V_i\int_0^N \exp\!\left(-\frac{4\lambda_i\gamma_0}{\pi}\int_z^N \frac{f(u)}{\sqrt{L(u)}}\,du\right)f(z)^2\,dz. \tag{98}$$

*Remark* 2 (Early-iteration proxy).  In early iterations the drift term $\left(M^{0.5}Q_L(N)\right)^{-\frac{2\alpha+2\beta-1}{2\alpha}}$ dominates. Solving $L(N) \eqsim \left(M^{0.5}Q_L(N)\right)^{-\frac{2\alpha+2\beta-1}{2\alpha}}$ yields

$$L(N) \;\eqsim\; \left(M^{0.5}\gamma_0 F(N)\right)^{-\frac{2(2\alpha+2\beta-1)}{2\alpha+1-2\beta}}, \qquad F(N) := \int_0^N f(u)\,du.$$

Now we move on to stable-decay scheduling.

**Stable-decay Schedule.**  For the stable-decay schedule, we set the learning rate to $\gamma_k = \gamma_0 f(k)$ with

$$f(k) = \begin{cases} 1, & k \le pN, \\ \left(1+\tau(k-pN)\right)^{-c}, & k > pN, \end{cases} \tag{99}$$

where $p, c \in (0,1)$ and $\tau > 0$. In other words, the learning rate remains constant for the first $pN$ steps, and then decays polynomially with exponent $c$ for the remaining $(1-p)N$ steps.

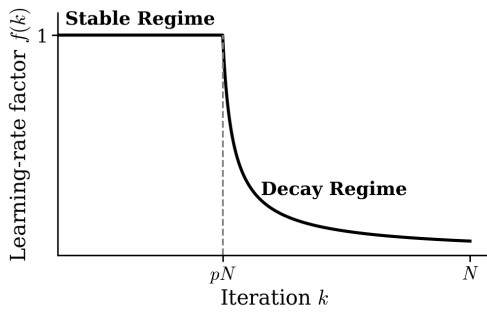

Figure 22: **Visualization of Stable-decay Scheduling.**

*Remark* 3. Note that $f$ depends on the total training steps $N$. To be precise, we have to represent it as $f_N$, but for simplicity, we write it as $f$ throughout the analysis.

First, we will make an upper bound on the noise term under stable-decay scheduling.

### G.1 UPPER BOUND OF THE NOISE TERM

Fix $p < q < 1$ close to 1 and split $L^{\text{noise}}(N)$ as

$$
L^{\text{noise}}(N) = \frac{2\gamma_0^2}{\pi} \sum_{i=1}^{M} V_i \int_0^{qN} \exp\left(-\frac{4\lambda_i\gamma_0}{\pi} \int_z^N \frac{f(u)}{\sqrt{L(u)}} \, du\right) f(z)^2 \, dz
$$
$$
+ \frac{2\gamma_0^2}{\pi} \sum_{i=1}^{M} V_i \int_{qN}^N \exp\left(-\frac{4\lambda_i\gamma_0}{\pi} \int_z^N \frac{f(u)}{\sqrt{L(u)}} \, du\right) f(z)^2 \, dz =: T_{\leq qN} + T_{>qN}.
$$

**Bounding $T_{>qN}$.** Note that $f(N) \asymp f(z)$ holds for $qN < z < N$. So

$$
\int_{qN}^N \exp\left(-\frac{4\lambda_i\gamma_0}{\pi} \int_z^N \frac{f(u)}{\sqrt{L(u)}} \, du\right) f(z)^2 \, dz \asymp f(N)^2 \int_{qN}^N \exp\left(-\frac{4\lambda_i\gamma_0}{\pi} \int_z^N \frac{f(u)}{\sqrt{L(u)}} \, du\right) dz.
$$

For $q$ sufficiently close to 1, there exist constants $c_0, c_1 > 0$ such that for $qN < z < N$

$$
c_0 \frac{(N-z)f(N)}{\sqrt{L(N)}} \leq \int_z^N \frac{f(u)}{\sqrt{L(u)}} \, du \leq c_1 \frac{(N-z)f(N)}{\sqrt{L(N)}}.
$$

Therefore,

$$
T_{>qN} \leq \frac{2\gamma_0^2}{\pi} f(N)^2 \sum_{i=1}^{M} V_i \int_{qN}^N \exp\left(-\frac{4\lambda_i\gamma_0}{\pi} c_0 \frac{(N-z)f(N)}{\sqrt{L(N)}}\right) dz
$$
$$
\asymp \frac{2\gamma_0^2}{\pi} f(N)^2 \sum_{i=1}^{M} V_i \frac{\pi \sqrt{L(N)}}{4\lambda_i\gamma_0 c_0 f(N)} \asymp \gamma_0 f(N) \sqrt{L(N)} \sum_{i=1}^{M} \frac{V_i}{\lambda_i}
$$
$$
\asymp \gamma_0 f(N) \sqrt{L(N)} \operatorname{Tr}\left(\operatorname{diag}(K)^{1/2}\right) \asymp \gamma_0 f(N) \sqrt{L(N)} M^{0.5}.
$$

To summarize, we have

$$
T_{>qN} \lesssim \gamma_0 f(N) \sqrt{L(N)} M^{0.5} \asymp \gamma_0 M^{1/2} N^{-c} \sqrt{L(N)}.
$$

**Bounding $T_{\leq qN}$.** Let $Q(z, N) = \frac{4\gamma_0}{\pi} \int_z^N \frac{f(u)}{\sqrt{L(u)}} \, du$. Then

$$
T_{\leq qN} = \frac{2\gamma_0^2}{\pi} \sum_{i=1}^{M} (w_i^{\mathsf{T}} \boldsymbol{K}_\sigma K u_i) \int_0^{qN} e^{-\frac{4\gamma_0}{\pi} \lambda_i(\overline{\boldsymbol{K}}) \int_z^N \frac{f(u)}{\sqrt{L(u)}} \, du} f(z)^2 dz
$$
$$
= \frac{2\gamma_0^2}{\pi} \sum_{i=1}^{M} \operatorname{Tr}(\boldsymbol{K}_\sigma K u_i w_i^{\mathsf{T}}) \int_0^{qN} e^{-\lambda_i(\overline{\boldsymbol{K}})Q(z,N)} f(z)^2 dz
$$
$$
= \frac{2\gamma_0^2}{\pi} \int_0^{qN} \sum_{i=1}^{M} \operatorname{Tr}(\boldsymbol{K}_\sigma K u_i w_i^{\mathsf{T}}) e^{-\lambda_i(\overline{\boldsymbol{K}})Q(z,N)} f(z)^2 dz
$$
$$
= \frac{2\gamma_0^2}{\pi} \int_0^{qN} \operatorname{Tr}\left(\boldsymbol{K}_\sigma K \sum_{i=1}^{M} e^{-\lambda_i(\overline{\boldsymbol{K}})Q(z,N)} u_i w_i^{\mathsf{T}}\right) f(z)^2 dz
$$
$$
= \frac{2\gamma_0^2}{\pi} \int_0^{qN} \operatorname{Tr}(\boldsymbol{K}_\sigma K e^{-\overline{\boldsymbol{K}}Q(z,N)}) f(z)^2 dz.
$$

Using $\arcsin x \approx x$ approximation on $\boldsymbol{K}_\sigma = \arcsin(\mathrm{diag}(K)^{-1/2} \cdot K \cdot \mathrm{diag}(K)^{-1/2})$, we get

$$\mathrm{Tr}(\boldsymbol{K}_\sigma K e^{-\overline{K}Q(z,N)}) = \mathrm{Tr}(\boldsymbol{K}_\sigma S H^{1/2} e^{-\overline{K}_1 Q(z,N)} H^{1/2} S^\mathsf{T})$$
$$= \mathrm{Tr}(H^{1/2} S^\mathsf{T} \boldsymbol{K}_\sigma S H^{1/2} e^{-\overline{K}_1 Q(z,N)}) \approx \mathrm{Tr}(\overline{K}_1^2 e^{-\overline{K}_1 Q(z,N)}).$$

Using same contour representation method and deterministic approximation with Section E.2.2 we get

$$T_{\leq qN} \asymp \frac{2\gamma_0^2}{\pi} \int_0^{qN} \mathrm{Tr}(\overline{K}_1^2 e^{-\overline{K}_1 Q(z,N)}) \, f(z)^2 dz$$
$$= \frac{2\gamma_0^2}{\pi} \int_0^{qN} \mathrm{Tr}\left(\frac{-1}{2\pi i} \oint_{\Gamma_2} z_1^2 e^{-Q(z,N)z_1} (\overline{K}_1 - z_1 \boldsymbol{I})^{-1} \, dz_1\right) f(z)^2 dz$$
$$\approx \frac{2\gamma_0^2}{\pi} \int_0^{qN} \mathrm{Tr}\left(\frac{-1}{2\pi i} \oint_{\Gamma} p_d^2 z_1^2 e^{-p_d Q(z,N)z_1} \mathcal{R}(z_1) \, dz_1\right) f(z)^2 dz$$
$$\asymp \frac{2\gamma_0^2}{\pi} M \int_0^{qN} \mathrm{Tr}\left(\frac{-1}{2\pi i} \oint_{\Gamma} z_1^2 e^{-p_d Q(z,N)z_1} \mathcal{R}(z_1) \, dz_1\right) f(z)^2 dz$$

Adopting the method in Paquette et al. (2024) same as Section K.1, we get

$$\mathrm{Tr}\left(\frac{-1}{2\pi i} \oint_{\Gamma} z_1^2 e^{-p_d Q(z,N)z_1} \mathcal{R}(z_1) \, dz_1\right) \asymp (p_d Q(z,N))^{-2+1/(2\alpha)} \asymp (M^{1/2} Q(z,N))^{-2+1/(2\alpha)}.$$

It leads to

$$T_{\leq qN} \asymp \frac{2\gamma_0^2}{\pi} M \int_0^{qN} (M^{1/2} Q(z,N))^{-2+1/(2\alpha)} f(z)^2 dz$$
$$\asymp \gamma_0^2 M^{1/(4\alpha)} \int_0^{qN} (Q(z,N))^{-2+1/(2\alpha)} f(z)^2 \, dz$$

Finally,

$$\gamma_0^2 M^{1/(4\alpha)} \int_0^{qN} (Q(z,N))^{-2+1/(2\alpha)} f(z)^2 \, dz$$
$$\asymp \gamma_0^{1/(2\alpha)} M^{1/(4\alpha)} \int_0^{pN} \frac{f(z)^2}{(\int_z^N \frac{f(u)}{\sqrt{L(u)}} du)^{2-1/(2\alpha)}} \, dz + \gamma_0^{1/(2\alpha)} M^{1/(4\alpha)} \int_{pN}^{qN} \frac{f(z)^2}{(\int_z^N \frac{f(u)}{\sqrt{L(u)}} du)^{2-1/(2\alpha)}} \, dz$$
$$\lesssim \gamma_0^{1/(2\alpha)} M^{1/(4\alpha)} \int_0^{pN} \frac{1}{(\frac{pN-z}{\sqrt{L(0)}} + \frac{1}{\sqrt{L(pN)}} \int_{pN}^N f(u)du)^{2-1/(2\alpha)}} \, dz$$
$$+ \gamma_0^{1/(2\alpha)} M^{1/(4\alpha)} \int_{pN}^{qN} \frac{f(z)^2}{(\frac{1}{\sqrt{L(pN)}} \int_{qN}^N f(u)du)^{2-1/(2\alpha)}} \, dz$$
$$\lesssim \gamma_0^{1/(2\alpha)} M^{1/(4\alpha)} \int_0^{pN} \frac{1}{(\frac{pN-z}{\sqrt{L(0)}} + \frac{N^{1-c}}{\sqrt{L(pN)}})^{2-1/(2\alpha)}} \, dz + \gamma_0^{1/(2\alpha)} M^{1/(4\alpha)} \int_{pN}^{qN} \frac{f(z)^2}{(\frac{N^{1-c}}{\sqrt{L(pN)}})^{2-1/(2\alpha)}} \, dz$$
$$\asymp \gamma_0^{1/(2\alpha)} M^{1/(4\alpha)} \sqrt{L(0)}((\frac{N^{1-c}}{\sqrt{L(pN)}})^{1/(2\alpha)-1} - (pN + \frac{N^{1-c}}{\sqrt{L(pN)}})^{1/(2\alpha)-1})$$
$$+ \gamma_0^{1/(2\alpha)} M^{1/(4\alpha)} N^{\max(1-2c,0)}(\frac{N^{1-c}}{\sqrt{L(pN)}})^{1/(2\alpha)-2}$$
$$\lesssim \gamma_0^{1/(2\alpha)} M^{1/(4\alpha)} N^{-(1-c)(1-1/(2\alpha))} L(pN)^{(1/2-1/(4\alpha))} \lesssim \gamma_0^{1/(2\alpha)} M^{1/(4\alpha)} N^{-(1-c)(1-1/(2\alpha))}$$

$$\tag{100}$$

So we have

$$T_{\leq qN} \lesssim \gamma_0^{1/(2\alpha)} M^{1/(4\alpha)} N^{-(1-c)(1-1/(2\alpha))}.$$

## G.2 COMBINING TERMS

Combining the bounds,

$$L(N) \lesssim M^{-(2\alpha+2\beta-1)} + \left(M^{0.5}\gamma_0 N\right)^{-\frac{2(2\alpha+2\beta-1)}{2\alpha+1-2\beta}} + \gamma_0 M^{0.5} N^{-c}\sqrt{L(N)} + \gamma_0^{\frac{1}{2\alpha}} M^{\frac{1}{4\alpha}} N^{-(1-c)\left(1-\frac{1}{2\alpha}\right)}.$$

We replaced the drift part with $\left(M^{0.5}\gamma_0 N\right)^{-\frac{2(2\alpha+2\beta-1)}{2\alpha+1-2\beta}}$ temporarily based on Remark 2, and justify this on our selected parameters in Remark 4. Solving the inequality asymptotically yields

$$L(N) \lesssim M^{-(2\alpha+2\beta-1)} + \left(M^{0.5}\gamma_0 N\right)^{-\frac{2(2\alpha+2\beta-1)}{2\alpha+1-2\beta}} + \gamma_0^2 M N^{-2c} + \gamma_0^{\frac{1}{2\alpha}} M^{\frac{1}{4\alpha}} N^{-(1-c)\left(1-\frac{1}{2\alpha}\right)}.$$

Finally, substituting $\gamma_0 = M^{-e}$ and $N = \mathfrak{f}/M$ yields

$$R(M,\mathfrak{f}) \lesssim M^{-(2\alpha+2\beta-1)} + \left(M^{-e-0.5}\mathfrak{f}\right)^{-\frac{2(2\alpha+2\beta-1)}{2\alpha+1-2\beta}} + M^{1+2c-2e}\mathfrak{f}^{-2c}$$
$$+ M^{\frac{1}{4\alpha} - \frac{e}{2\alpha} + (1-c)\left(1-\frac{1}{2\alpha}\right)}\mathfrak{f}^{-(1-c)\left(1-\frac{1}{2\alpha}\right)}.$$

Optimizing over $M$ gives a bound of the form $R(M^\star,\mathfrak{f}) \leq \mathfrak{f}^{-h(\alpha,\beta,c,e)}$, and we then optimize over $c, e$ to maximize $h(\alpha,\beta,c,e)$.

## G.3 OPTIMIZING OVER $c, e$ TO MAXIMIZE $h(\alpha,\beta,c,e)$

Assume throughout $\alpha > 0.5$, $\beta < 0.5$, and $2\alpha + 2\beta > 1$. Consider the upper bound

$$R_U(M,\mathfrak{f}) = M^{-(2\alpha+2\beta-1)} + \left(M^{-e-0.5}\mathfrak{f}\right)^{-\frac{2(2\alpha+2\beta-1)}{2\alpha+1-2\beta}} + M^{1+2c-2e}\mathfrak{f}^{-2c} + M^{\frac{1}{4\alpha} - \frac{e}{2\alpha} + (1-c)\left(1-\frac{1}{2\alpha}\right)}\mathfrak{f}^{-(1-c)\left(1-\frac{1}{2\alpha}\right)}.$$

For large $\mathfrak{f}$, define

$$R_{\min}(\mathfrak{f}) := \min_{M>0} R_U(M,\mathfrak{f}).$$

We show $R_{\min}(\mathfrak{f}) \asymp \mathfrak{f}^{h^\star(\alpha,\beta)}$ with $h^\star(\alpha,\beta) < 0$, and identify $c^\star(\alpha,\beta)$, $e^\star(\alpha,\beta)$, and $M = \mathfrak{f}^{m^\star(\alpha,\beta)}$.

**Logarithmic reduction to exponent balancing**

Let $M = \mathfrak{f}^m$ with $m \in \mathbb{R}$. Writing each term as $\mathfrak{f}^{L_i}$ gives

$$L_1(m) = -(2\alpha + 2\beta - 1)\,m, \tag{101}$$

$$L_2(m,e) = -\frac{2(2\alpha + 2\beta - 1)}{2\alpha + 1 - 2\beta} + \frac{2(2\alpha + 2\beta - 1)}{2\alpha + 1 - 2\beta}\,m\left(e + 0.5\right), \tag{102}$$

$$L_3(m,c,e) = m(1 + 2c - 2e) - 2c, \tag{103}$$

$$L_4(m,c,e) = m\left(\frac{1}{4\alpha} - \frac{e}{2\alpha} + (1-c)\left(1 - \frac{1}{2\alpha}\right)\right) - (1-c)\left(1 - \frac{1}{2\alpha}\right). \tag{104}$$

Thus minimizing $R_U$ is equivalent to

$$\min_{m,e\in\mathbb{R},\,0<c<1} \max\{L_1, L_2, L_3, L_4\}. \tag{105}$$

Introduce $h \in \mathbb{R}$ and rewrite as

$$\min_{m,c,e,h} h \quad \text{s.t.} \quad L_i(m,c,e) \leq h \ (i = 1,2,3,4), \ 0 < c < 1. \tag{106}$$

At an interior optimum ($0 < c < 1$), constraints equalize:

$$L_1 = L_2 = L_3 = L_4 = h. \tag{107}$$

Solving the equality yields

$$c^\star = \frac{-8\alpha\beta + 2\alpha + 2\beta - 1}{16\alpha^2 + 8\alpha\beta - 6\alpha - 2\beta + 1}, \tag{108}$$

$$e^\star = \frac{8\alpha^2 + 16\alpha\beta - 4\alpha - 4\beta + 1}{2(4\alpha - 1)}, \tag{109}$$

$$m^\star = \frac{2(4\alpha - 1)}{16\alpha^2 + 8\alpha\beta + 2\alpha - 2\beta - 1}, \tag{110}$$

$$h^\star = -\frac{2(4\alpha - 1)(2\alpha + 2\beta - 1)}{16\alpha^2 + 8\alpha\beta + 2\alpha - 2\beta - 1}. \tag{111}$$

*Feasibility.* Since $\alpha > 0.5$, denominators are positive. The condition $c^\star > 0$ is equivalent to

$$-8\alpha\beta + 2\alpha + 2\beta - 1 > 0 \quad \Longleftrightarrow \quad \beta < \frac{2\alpha - 1}{2(4\alpha - 1)} := B^\star(\alpha),$$

which is stricter than $\beta < 0.5$. Moreover, $c^\star < 1$ holds automatically for $\beta > 0$. Hence, the interior solution is feasible whenever

$$\boxed{0.5 - \alpha < \beta < B^\star(\alpha)} \qquad \text{with} \quad B^\star(\alpha) = \frac{2\alpha - 1}{2(4\alpha - 1)}. \tag{112}$$

In this band,

$$\boxed{M = \mathfrak{f}^{m^*}, \qquad R_{\min}(\mathfrak{f}) \asymp \mathfrak{f}^{h^\star}}$$

with $m^\star, h^\star$ as in (110)–(111). Note $m^\star > 0$ and $h^\star < 0$.

**Result** As $\mathfrak{f} \to \infty$, the choice $M = \mathfrak{f}^{m^*}$ with

$$m^\star = \frac{2(4\alpha - 1)}{16\alpha^2 + 8\alpha\beta + 2\alpha - 2\beta - 1},$$

$$c^\star = \frac{-8\alpha\beta + 2\alpha + 2\beta - 1}{16\alpha^2 + 8\alpha\beta - 6\alpha - 2\beta + 1},$$

$$e^\star = \frac{8\alpha^2 + 16\alpha\beta - 4\alpha - 4\beta + 1}{2(4\alpha - 1)},$$

$$h^\star = -\frac{2(4\alpha - 1)(2\alpha + 2\beta - 1)}{16\alpha^2 + 8\alpha\beta + 2\alpha - 2\beta - 1}$$

is optimal for $\alpha > 0.5$, $0.5 - \alpha < \beta < B^\star(\alpha)$, where

$$B^\star(\alpha) = \frac{2\alpha - 1}{2(4\alpha - 1)}.$$

This choice minimizes $\max\{L_1, L_2, L_3, L_4\}$ in (105). Consequently,

$$R_{\min}(\mathfrak{f}) \asymp \mathfrak{f}^{h^\star(\alpha, \beta)} \qquad \text{with } h^\star(\alpha, \beta) < 0.$$

*Remark* 4 (Justification on drift term conversion). Note that $M = \mathfrak{f}^{M^\star}$ and $N = \mathfrak{f}^{1-M^\star}$ holds for the selected parameters.

For $pN$ iterations the stable-decay scheduling behaves same as the constant learning rate. Let $N_0$ be the crossover point in constant learning rate. Note that $N \gtrsim N_0$ holds, and $N$ is asymptotically strictly bigger than $N_0$. So $L(u) \lesssim \gamma_0^2 M + M^{-2\alpha - 2\beta + 1}$ holds for $u \geq N_0$.

Also for selected $\gamma_0 = M^{-e^*}$, $\gamma_0^2 M \gtrsim M^{-2\alpha - 2\beta + 1}$ holds.

So we have $L(u) \lesssim \gamma_0^2 M$ for $u \geq N_0$.

$$\left(M^{0.5}Q_L(N)\right)^{-\frac{2\alpha+2\beta-1}{2\alpha}} = \left(M^{0.5}\frac{4\gamma_0}{\pi}\int_0^N \frac{f(u)}{\sqrt{L(u)}}\,du\right)^{-\frac{2\alpha+2\beta-1}{2\alpha}}$$

$$\lesssim \left(M^{0.5}\frac{4\gamma_0}{\pi}\int_{N_0}^{pN} \frac{f(u)}{\sqrt{L(u)}}\,du\right)^{-\frac{2\alpha+2\beta-1}{2\alpha}}$$

$$\approx \left(\gamma_0 M^{0.5}\int_{N_0}^{pN} \frac{1}{\sqrt{L(u)}}\,du\right)^{-\frac{2\alpha+2\beta-1}{2\alpha}}$$

$$\approx \left(\gamma_0 M^{0.5}\int_{N_0}^{pN} \frac{1}{\sqrt{\gamma_0^2 M}}\,du\right)^{-\frac{2\alpha+2\beta-1}{2\alpha}}$$

$$\approx (pN-N_0)^{-\frac{2\alpha+2\beta-1}{2\alpha}} \approx N^{-\frac{2\alpha+2\beta-1}{2\alpha}}$$

For selected parameters $M = \mathfrak{f}^{M^\star}$, $N = \mathfrak{f}^{1-M^\star}$, $c^*$, and $\gamma_0 = M^{-e^*}$ following holds.

$$N^{-\frac{2\alpha+2\beta-1}{2\alpha}} \lesssim \gamma_0^2 M N^{-2c^*}.$$

As $\left(M^{0.5}Q_L(N)\right)^{-\frac{2\alpha+2\beta-1}{2\alpha}} \lesssim \gamma_0^2 M N^{-2c^*}$, replacing the drift term with a proxy does not alter the argument.

### G.4 ANALYSIS FOR WARMUP-STABLE-DECAY

The analysis for warmup-stable-decay is almost identical to that for stable-decay. The only difference occurs in the step leading to (100), but the final bound is the same. Thus, the loss bound for warmup-stable-decay matches that for stable-decay. We provide the corresponding analysis to the procedure of (100) at the end of this subsection.

Finally, the bound

$$R_f(M^\star, \mathfrak{f}/M^\star, (M^\star)^{-e^*}) \lesssim \mathfrak{f}^{-\frac{2(4\alpha-1)(2\alpha+2\beta-1)}{16\alpha^2+8\alpha\beta+2\alpha-2\beta-1}}. \tag{113}$$

introduced in (15) also holds for warmup-stable-decay.

For the warmup-stable-decay schedule, we set the learning rate to $\gamma_k = \gamma_0 f(k)$ with

$$f(k) = \begin{cases} k/(wN), & k \le wN, \\ 1, & wN \le k \le pN, \\ \left(1+\tau(k-pN)\right)^{-c}, & k > pN, \end{cases} \tag{114}$$

where $w, p, c \in (0,1)$ and $\tau > 0$. $w$ is the ratio for the warmup stage, and we assume that $w$ is smaller than $p/2$.

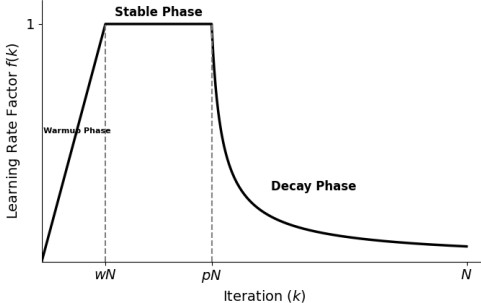

Figure 23: **Visualization of Warmup-stable-decay Scheduling.**

Following is the corresponding analysis to the procedure of (100).

$$\gamma_0^2 M^{1/(4\alpha)} \int_0^{qN} (Q(z,N))^{-2+1/(2\alpha)} f(z)^2 \, dz$$

$$\approx \gamma_0^{1/(2\alpha)} M^{1/(4\alpha)} \int_0^{pN} \frac{f(z)^2}{(\int_z^N \frac{f(u)}{\sqrt{L(u)}} du)^{2-1/(2\alpha)}} \, dz + \gamma_0^{1/(2\alpha)} M^{1/(4\alpha)} \int_{pN}^{qN} \frac{f(z)^2}{(\int_z^N \frac{f(u)}{\sqrt{L(u)}} du)^{2-1/(2\alpha)}} \, dz$$

$$\lesssim \gamma_0^{1/(2\alpha)} M^{1/(4\alpha)} \int_0^{wN} \frac{1}{(\frac{pN/2}{\sqrt{L(0)}} + \frac{1}{\sqrt{L(pN)}} \int_{pN}^N f(u)du)^{2-1/(2\alpha)}} \, dz$$

$$+ \gamma_0^{1/(2\alpha)} M^{1/(4\alpha)} \int_{wN}^{pN} \frac{1}{(\frac{pN-z}{\sqrt{L(0)}} + \frac{1}{\sqrt{L(pN)}} \int_{pN}^N f(u)du)^{2-1/(2\alpha)}} \, dz$$

$$+ \gamma_0^{1/(2\alpha)} M^{1/(4\alpha)} \int_{pN}^{qN} \frac{f(z)^2}{(\frac{1}{\sqrt{L(pN)}} \int_{qN}^N f(u)du)^{2-1/(2\alpha)}} \, dz$$

$$\lesssim \gamma_0^{1/(2\alpha)} M^{1/(4\alpha)} \int_0^{wN} \frac{1}{(\frac{pN/2}{\sqrt{L(0)}} + \frac{N^{1-c}}{\sqrt{L(pN)}})^{2-1/(2\alpha)}} \, dz + \gamma_0^{1/(2\alpha)} M^{1/(4\alpha)} \int_{wN}^{pN} \frac{1}{(\frac{pN-z}{\sqrt{L(0)}} + \frac{N^{1-c}}{\sqrt{L(pN)}})^{2-1/(2\alpha)}} \, dz$$

$$+ \gamma_0^{1/(2\alpha)} M^{1/(4\alpha)} \int_{pN}^{qN} \frac{f(z)^2}{(\frac{N^{1-c}}{\sqrt{L(pN)}})^{2-1/(2\alpha)}} \, dz$$

$$\approx \gamma_0^{1/(2\alpha)} M^{1/(4\alpha)} wN (\frac{pN/2}{\sqrt{L(0)}})^{-2+1/(2\alpha)}$$

$$+ \gamma_0^{1/(2\alpha)} M^{1/(4\alpha)} \sqrt{L(0)}((\frac{N^{1-c}}{\sqrt{L(pN)}})^{1/(2\alpha)-1} - (pN - wN + \frac{N^{1-c}}{\sqrt{L(pN)}})^{1/(2\alpha)-1})$$

$$+ \gamma_0^{1/(2\alpha)} M^{1/(4\alpha)} N^{\max(1-2c,0)} (\frac{N^{1-c}}{\sqrt{L(pN)}})^{1/(2\alpha)-2}$$

$$\lesssim \gamma_0^{1/(2\alpha)} M^{1/(4\alpha)} N^{-(1-1/(2\alpha))} + \gamma_0^{1/(2\alpha)} M^{1/(4\alpha)} N^{-(1-c)(1-1/(2\alpha))} L(pN)^{(1/2-1/(4\alpha))}$$

$$\lesssim \gamma_0^{1/(2\alpha)} M^{1/(4\alpha)} N^{-(1-c)(1-1/(2\alpha))}$$

## G.5 Scheduling on SGD

In this subsection, we explain that the scheduling does not lift the compute-optimal exponent of SGD in the Phase I and Phase II. Assume a bounded scheduling function $f$, and define $F(k) = \int_0^k f(z) \, dz$.

Ferbach et al. (2025) proved

$$R_f(M,N,\gamma_0) \gtrsim M^{-2\alpha+\max(0,1-2\beta)} + (\gamma_0 F(N))^{-(2\alpha+2\beta-1)/(2\alpha)} + M^{-1}(\gamma_0 F(N))^{-1+1/(2\alpha)}$$

for the risk $R_f(M,N,\gamma_0)$ with general bounded scheduling function $f$.

Since $f$ is bounded, we have $F(N) \lesssim N$. Therefore,

$$R_f(M,N,\gamma_0) \gtrsim M^{-2\alpha+\max(0,1-2\beta)} + (\gamma_0 F(N))^{-(2\alpha+2\beta-1)/(2\alpha)} + M^{-1}(\gamma_0 F(N))^{-1+1/(2\alpha)}$$
$$\gtrsim M^{-2\alpha+\max(0,1-2\beta)} + (\gamma_0 N)^{-(2\alpha+2\beta-1)/(2\alpha)} + M^{-1}(\gamma_0 N)^{-1+1/(2\alpha)}$$
$$\gtrsim R_1(M,N,\gamma_0),$$

where $R_1(M,N,\gamma_0)$ is the loss under a constant schedule $f \equiv 1$.

Thus, scheduling does not improve the compute-optimal exponent of SGD in Phase I and Phase II.

# H ANALYSIS FOR LINEAR DECAYING SCHEDULING AND COSINE SCHEDULING

## H.1 ANALYSIS FOR LINEAR DECAYING SCHEDULING

In this section, we analyze the following linear decaying scheduling.

$$f(t) = 1 - \left(1 - \frac{1}{\sqrt{N}}\right)\frac{t}{N} \tag{115}$$

It decays from 1 to $\frac{1}{\sqrt{N}}$ linearly.

We will focus on Phase Aa, and follow a similar procedure to stable-decay scheduling.

Note that we have to handle the following equation, where $Q_L(z) := \frac{4\gamma_0}{\pi}\int_0^z \frac{f(u)}{\sqrt{L(u)}}\,du$.

$$L(N) \asymp M^{-(2\alpha+2\beta-1)} + \left(M^{0.5}Q_L(N)\right)^{-\frac{2\alpha+2\beta-1}{2\alpha}}$$
$$+ \frac{2\gamma_0^2}{\pi}\sum_{i=1}^M V_i \int_0^N \exp\left(-\frac{4\lambda_i\gamma_0}{\pi}\int_z^N \frac{f(u)}{\sqrt{L(u)}}\,du\right)f(z)^2\,dz.$$

In early iterations the drift term $\left(M^{0.5}Q_L(N)\right)^{-\frac{2\alpha+2\beta-1}{2\alpha}}$ dominates. Solving $L(N) \asymp \left(M^{0.5}Q_L(N)\right)^{-\frac{2\alpha+2\beta-1}{2\alpha}}$ yields

$$L(N) \asymp \left(M^{0.5}\gamma_0 F(N)\right)^{-\frac{2(2\alpha+2\beta-1)}{2\alpha+1-2\beta}}, \qquad F(N) := \int_0^N f(u)\,du.$$

For linear decaying scheduling $F(N) \asymp N$ holds, so the drift term becomes $\left(M^{0.5}\gamma_0 N\right)^{-\frac{2(2\alpha+2\beta-1)}{2\alpha+1-2\beta}}$.

Now we move to the noise term. We split the noise term $L^{\text{noise}}(N)$ as

$$L^{\text{noise}}(N) = \frac{2\gamma_0^2}{\pi}\sum_{i=1}^M V_i \int_0^{N-\sqrt{N}} \exp\left(-\frac{4\lambda_i\gamma_0}{\pi}\int_z^N \frac{f(u)}{\sqrt{L(u)}}\,du\right)f(z)^2\,dz$$
$$+ \frac{2\gamma_0^2}{\pi}\sum_{i=1}^M V_i \int_{N-\sqrt{N}}^N \exp\left(-\frac{4\lambda_i\gamma_0}{\pi}\int_z^N \frac{f(u)}{\sqrt{L(u)}}\,du\right)f(z)^2\,dz =: T_{\leq(N-\sqrt{N})} + T_{>(N-\sqrt{N})}.$$

**Bounding** $T_{>(N-\sqrt{N})}$**.** Note that $f(N) \asymp f(z)$ holds for $(N-\sqrt{N}) < z < N$. So

$$\int_{(N-\sqrt{N})}^N \exp\left(-\frac{4\lambda_i\gamma_0}{\pi}\int_z^N \frac{f(u)}{\sqrt{L(u)}}\,du\right)f(z)^2\,dz \asymp f(N)^2 \int_{(N-\sqrt{N})}^N \exp\left(-\frac{4\lambda_i\gamma_0}{\pi}\int_z^N \frac{f(u)}{\sqrt{L(u)}}\,du\right)dz.$$

There exist constants $c_0, c_1 > 0$ such that for $(N-\sqrt{N}) < z < N$

$$c_0\,\frac{(N-z)f(N)}{\sqrt{L(N)}} \leq \int_z^N \frac{f(u)}{\sqrt{L(u)}}\,du \leq c_1\,\frac{(N-z)f(N)}{\sqrt{L(N)}}.$$

Therefore,

$$T_{>qN} \leq \frac{2\gamma_0^2}{\pi}f(N)^2\sum_{i=1}^M V_i \int_{(N-\sqrt{N})}^N \exp\left(-\frac{4\lambda_i\gamma_0}{\pi}c_0\,\frac{(N-z)f(N)}{\sqrt{L(N)}}\right)dz$$
$$\asymp \frac{2\gamma_0^2}{\pi}f(N)^2\sum_{i=1}^M V_i\,\frac{\pi\sqrt{L(N)}}{4\lambda_i\gamma_0 c_0 f(N)} \asymp \gamma_0 f(N)\sqrt{L(N)}\sum_{i=1}^M \frac{V_i}{\lambda_i}$$
$$\asymp \gamma_0 f(N)\sqrt{L(N)}\,\text{Tr}\left(\text{diag}(K)^{1/2}\right) \asymp \gamma_0 f(N)\sqrt{L(N)}\,M^{0.5}.$$

To summarize, we have

$$T_{>(N-\sqrt{N})} \lesssim \gamma_0 f(N)\sqrt{L(N)}\,M^{0.5} \asymp \gamma_0 M^{1/2}N^{-1/2}\sqrt{L(N)}.$$

**Bounding** $T_{\leq(N-\sqrt{N})}$. Let $Q(z,N) = \frac{4\gamma_0}{\pi}\int_z^N \frac{f(u)}{\sqrt{L(u)}}\,du$.

By the same procedure as the stable-decaying case, we can get

$$
T_{\leq(N-\sqrt{N})} \approx \frac{2\gamma_0^2}{\pi} M \int_0^{(N-\sqrt{N})} (M^{1/2}Q(z,N))^{-2+1/(2\alpha)} f(z)^2 dz
$$

$$
\approx \gamma_0^2 M^{1/(4\alpha)} \int_0^{(N-\sqrt{N})} (Q(z,N))^{-2+1/(2\alpha)} f(z)^2\, dz.
$$

We have

$$
\gamma_0^2 M^{1/(4\alpha)} \int_0^{(N-\sqrt{N})} (Q(z,N))^{-2+1/(2\alpha)} f(z)^2\, dz
$$

$$
\approx \gamma_0^{1/(2\alpha)} M^{1/(4\alpha)} \int_0^{(N-\sqrt{N})} \frac{f(z)^2}{(\int_z^N \frac{f(u)}{\sqrt{L(u)}} du)^{2-1/(2\alpha)}}\, dz
$$

$$
\lesssim \gamma_0^{1/(2\alpha)} M^{1/(4\alpha)} \int_0^{(N-\sqrt{N})} \frac{f(z)^2}{(\int_z^N \frac{f(u)}{\sqrt{L(0)}} du)^{2-1/(2\alpha)}}\, dz \tag{116}
$$

$$
\approx \gamma_0^{1/(2\alpha)} M^{1/(4\alpha)} \int_0^{(N-\sqrt{N})} \frac{f(z)^2}{(\int_z^N f(u)du)^{2-1/(2\alpha)}}\, dz
$$

Let the integral term be $\mathcal{I}$. First, we use the change of variables $z = N - u$, which transforms the integration interval $[0, N - \sqrt{N}]$ into $[\sqrt{N}, N]$. In the regime of large $N$, the linear schedule $f(N-u)$ can be approximated as

$$
f(N-u) = 1 - \left(1 - \frac{1}{\sqrt{N}}\right)\frac{N-u}{N} \approx \frac{1}{\sqrt{N}}\left(1 + \frac{u}{\sqrt{N}}\right). \tag{117}
$$

Using this approximation, we evaluate the inner integral in the denominator:

$$
\int_{N-u}^N f(s)\, ds \approx \int_0^u \frac{1}{\sqrt{N}}\left(1 + \frac{v}{\sqrt{N}}\right) dv = \frac{u}{\sqrt{N}}\left(1 + \frac{u}{2\sqrt{N}}\right). \tag{118}
$$

Substituting these terms back into $\mathcal{I}$, we obtain

$$
\mathcal{I} \approx \int_{\sqrt{N}}^N \frac{\left[\frac{1}{\sqrt{N}}\left(1 + \frac{u}{\sqrt{N}}\right)\right]^2}{\left[\frac{u}{\sqrt{N}}\left(1 + \frac{u}{2\sqrt{N}}\right)\right]^{2-\frac{1}{2\alpha}}}\, du. \tag{119}
$$

To decouple the dependency on $N$, we apply the scaling $u = \sqrt{N}y$, which implies $du = \sqrt{N}dy$. The integration limits change from $[\sqrt{N}, N]$ to $[1, \sqrt{N}]$. The integral is then reformulated as

$$
\mathcal{I} \approx \int_1^{\sqrt{N}} \frac{\frac{1}{N}(1+y)^2}{(y(1+y/2))^{2-\frac{1}{2\alpha}}} \sqrt{N}\, dy
$$

$$
= \frac{1}{\sqrt{N}} \int_1^{\sqrt{N}} \frac{(1+y)^2}{y^{2-\frac{1}{2\alpha}}(1+y/2)^{2-\frac{1}{2\alpha}}}\, dy. \tag{120}
$$

The asymptotic behavior is determined by the convergence of the remaining integral. As $y \to \infty$, the integrand behaves as

$$
\frac{y^2}{y^{2-\frac{1}{2\alpha}}(y/2)^{2-\frac{1}{2\alpha}}} \propto y^{2-2(2-\frac{1}{2\alpha})} = y^{\frac{1}{\alpha}-2}. \tag{121}
$$

Integrating this term from 1 to $\sqrt{N}$ leads to the following cases depending on the exponent $\frac{1}{\alpha} - 2$:

$$
\mathcal{I} \sim \frac{1}{\sqrt{N}} \times \begin{cases} (\sqrt{N})^{\frac{1}{\alpha}-1} = N^{\frac{1}{2\alpha}-\frac{1}{2}} & \text{if } \frac{1}{\alpha} - 2 > -1 \implies \alpha < 1, \\ \ln(\sqrt{N}) \sim \ln N & \text{if } \frac{1}{\alpha} - 2 = -1 \implies \alpha = 1, \\ \text{const} & \text{if } \frac{1}{\alpha} - 2 < -1 \implies \alpha > 1. \end{cases} \tag{122}
$$

Simplifying the final exponents, we get the asymptotic order:

$$\mathcal{I} \sim \begin{cases} \mathcal{O}\left(N^{\frac{1}{2\alpha}-1}\right) & \text{if } 0.5 < \alpha < 1, \\ \mathcal{O}\left(N^{-\frac{1}{2}}\ln N\right) & \text{if } \alpha = 1, \\ \mathcal{O}\left(N^{-\frac{1}{2}}\right) & \text{if } \alpha > 1. \end{cases} \tag{123}$$

For $0.5 < \alpha < 1$, we have

$$T_{\leq(N-\sqrt{N})} \lesssim \gamma_0^{1/(2\alpha)} M^{1/(4\alpha)} N^{-(1-1/(2\alpha))}.$$

For $0.5 < \alpha < 1$, combining the bounds for the drift term and noise term, we have

$$L(N) \lesssim M^{-(2\alpha+2\beta-1)} + \left(M^{0.5}\gamma_0 N\right)^{-\frac{2(2\alpha+2\beta-1)}{2\alpha+1-2\beta}} + \gamma_0 M^{0.5} N^{-1/2}\sqrt{L(N)} + \gamma_0^{\frac{1}{2\alpha}} M^{\frac{1}{4\alpha}} N^{-(1-\frac{1}{2\alpha})}.$$

In intersection of Area Aa$^\star$ and $0.5 < \alpha < 1$, with choice of $e^*$ in $\gamma_0 = M^{-e^*}$ and $c^*$ we used for stable-decaying scheduling, we have

$$L(N) \lesssim M^{-(2\alpha+2\beta-1)} + \left(M^{0.5}\gamma_0 N\right)^{-\frac{2(2\alpha+2\beta-1)}{2\alpha+1-2\beta}} + \gamma_0 M^{0.5} N^{-c^*}\sqrt{L(N)} + \gamma_0^{\frac{1}{2\alpha}} M^{\frac{1}{4\alpha}} N^{-(1-c^*)(1-\frac{1}{2\alpha})}.$$

So in intersection of Area Aa$^\star$ and $0.5 < \alpha < 1$, we have

$$R_f(M^\star, \mathfrak{f}/M^\star, (M^\star)^{-e^*}) \lesssim \mathfrak{f}^{-\frac{2(4\alpha-1)(2\alpha+2\beta-1)}{16\alpha^2+8\alpha\beta+2\alpha-2\beta-1}}. \tag{124}$$

Therefore, linear decaying scheduling has an advantage compared to constant learning rate in the intersection of Area Aa$^\star$ and $0.5 < \alpha < 1$.

## H.2 Analysis for Cosine Scheduling

In this section, we analyze the following cosine scheduling.

$$f(t) = \frac{1+1/N}{2} + \frac{1-1/N}{2}\cos\left(\frac{\pi}{N}t\right) \tag{125}$$

It decays from 1 to $\frac{1}{N}$.

We will focus on Phase Aa, and follow a similar procedure to stable-decay scheduling.

Note that we have to handle the following equation, where $Q_L(z) := \frac{4\gamma_0}{\pi}\int_0^z \frac{f(u)}{\sqrt{L(u)}}\,du$.

$$L(N) \approx M^{-(2\alpha+2\beta-1)} + \left(M^{0.5}Q_L(N)\right)^{-\frac{2\alpha+2\beta-1}{2\alpha}}$$
$$+ \frac{2\gamma_0^2}{\pi}\sum_{i=1}^M V_i \int_0^N \exp\left(-\frac{4\lambda_i\gamma_0}{\pi}\int_z^N \frac{f(u)}{\sqrt{L(u)}}\,du\right)f(z)^2\,dz.$$

In early iterations the drift term $\left(M^{0.5}Q_L(N)\right)^{-\frac{2\alpha+2\beta-1}{2\alpha}}$ dominates. Solving $L(N) \approx \left(M^{0.5}Q_L(N)\right)^{-\frac{2\alpha+2\beta-1}{2\alpha}}$ yields

$$L(N) \approx \left(M^{0.5}\gamma_0 F(N)\right)^{-\frac{2(2\alpha+2\beta-1)}{2\alpha+1-2\beta}}, \qquad F(N) := \int_0^N f(u)\,du.$$

For cosine scheduling $F(N) \approx N$ holds, so the drift term becomes $\left(M^{0.5}\gamma_0 N\right)^{-\frac{2(2\alpha+2\beta-1)}{2\alpha+1-2\beta}}$.

Now we move to the noise term. We split the noise term $L^{\text{noise}}(N)$ as

$$L^{\text{noise}}(N) = \frac{2\gamma_0^2}{\pi} \sum_{i=1}^{M} V_i \int_0^{N-\sqrt{N}} \exp\left(-\frac{4\lambda_i\gamma_0}{\pi} \int_z^N \frac{f(u)}{\sqrt{L(u)}} du\right) f(z)^2 \, dz$$

$$+ \frac{2\gamma_0^2}{\pi} \sum_{i=1}^{M} V_i \int_{N-\sqrt{N}}^{N} \exp\left(-\frac{4\lambda_i\gamma_0}{\pi} \int_z^N \frac{f(u)}{\sqrt{L(u)}} du\right) f(z)^2 \, dz =: T_{\leq(N-\sqrt{N})} + T_{>(N-\sqrt{N})}.$$

**Bounding $T_{>(N-\sqrt{N})}$.** Note that $f(N) \approx f(z)$ holds for $(N - \sqrt{N}) < z < N$. So

$$\int_{(N-\sqrt{N})}^{N} \exp\left(-\frac{4\lambda_i\gamma_0}{\pi} \int_z^N \frac{f(u)}{\sqrt{L(u)}} du\right) f(z)^2 \, dz \approx f(N)^2 \int_{(N-\sqrt{N})}^{N} \exp\left(-\frac{4\lambda_i\gamma_0}{\pi} \int_z^N \frac{f(u)}{\sqrt{L(u)}} du\right) dz.$$

There exist constants $c_0, c_1 > 0$ such that for $(N - \sqrt{N}) < z < N$

$$c_0 \frac{(N-z)f(N)}{\sqrt{L(N)}} \leq \int_z^N \frac{f(u)}{\sqrt{L(u)}} du \leq c_1 \frac{(N-z)f(N)}{\sqrt{L(N)}}.$$

Therefore,

$$T_{>qN} \leq \frac{2\gamma_0^2}{\pi} f(N)^2 \sum_{i=1}^{M} V_i \int_{(N-\sqrt{N})}^{N} \exp\left(-\frac{4\lambda_i\gamma_0}{\pi} c_0 \frac{(N-z)f(N)}{\sqrt{L(N)}}\right) dz$$

$$\approx \frac{2\gamma_0^2}{\pi} f(N)^2 \sum_{i=1}^{M} V_i \frac{\pi\sqrt{L(N)}}{4\lambda_i\gamma_0 c_0 f(N)} \approx \gamma_0 f(N) \sqrt{L(N)} \sum_{i=1}^{M} \frac{V_i}{\lambda_i}$$

$$\approx \gamma_0 f(N) \sqrt{L(N)} \, \text{Tr}\left(\text{diag}(K)^{1/2}\right) \approx \gamma_0 f(N) \sqrt{L(N)} M^{0.5}.$$

To summarize, we have

$$T_{>(N-\sqrt{N})} \lesssim \gamma_0 f(N) \sqrt{L(N)} M^{0.5} \approx \gamma_0 M^{1/2} N^{-1} \sqrt{L(N)}.$$

**Bounding $T_{\leq(N-\sqrt{N})}$.** Let $Q(z, N) = \frac{4\gamma_0}{\pi} \int_z^N \frac{f(u)}{\sqrt{L(u)}} du$.

By the same procedure as the stable-decaying case, we can get

$$T_{\leq(N-\sqrt{N})} \approx \frac{2\gamma_0^2}{\pi} M \int_0^{(N-\sqrt{N})} (M^{1/2} Q(z,N))^{-2+1/(2\alpha)} f(z)^2 dz$$

$$\approx \gamma_0^2 M^{1/(4\alpha)} \int_0^{(N-\sqrt{N})} (Q(z,N))^{-2+1/(2\alpha)} f(z)^2 \, dz.$$

We have

$$\gamma_0^2 M^{1/(4\alpha)} \int_0^{(N-\sqrt{N})} (Q(z,N))^{-2+1/(2\alpha)} f(z)^2 \, dz$$

$$\approx \gamma_0^{1/(2\alpha)} M^{1/(4\alpha)} \int_0^{(N-\sqrt{N})} \frac{f(z)^2}{(\int_z^N \frac{f(u)}{\sqrt{L(u)}} du)^{2-1/(2\alpha)}} \, dz$$

$$\lesssim \gamma_0^{1/(2\alpha)} M^{1/(4\alpha)} \int_0^{(N-\sqrt{N})} \frac{f(z)^2}{(\int_z^N \frac{f(u)}{\sqrt{L(0)}} du)^{2-1/(2\alpha)}} \, dz \tag{126}$$

$$\approx \gamma_0^{1/(2\alpha)} M^{1/(4\alpha)} \int_0^{(N-\sqrt{N})} \frac{f(z)^2}{(\int_z^N f(u) du)^{2-1/(2\alpha)}} \, dz$$

Let the integral term be $\mathcal{I}$. First, we use the change of variables $z = N - u$, which transforms the integration interval $[0, N - \sqrt{N}]$ into $[\sqrt{N}, N]$. The integral can be written as:

$$\mathcal{I} = \int_{\sqrt{N}}^{N} \frac{f(N-x)^2}{\left(\int_0^x f(N-v)\, dv\right)^{2-\frac{1}{2\alpha}}} \, dx. \tag{127}$$

We evaluate the asymptotic magnitude of $\mathcal{I}$ by analyzing the dominant contributions from the lower limit ($x \approx \sqrt{N}$) and the upper limit ($x \approx N$).

Contribution near the lower limit ($x \approx \sqrt{N}$): In the region where $x$ is small, the learning rate approaches its minimum, $f(N-x) \approx \frac{1}{N}$. Consequently, the cumulative sum scales linearly with the inverse of $N$, i.e., $\int_0^x f(N-v)\, dv \approx \frac{x}{N}$. Substituting these approximations, the integrand becomes:

$$\frac{(1/N)^2}{(x/N)^{2-\frac{1}{2\alpha}}} = N^{-2} \cdot N^{2-\frac{1}{2\alpha}} \cdot x^{-2+\frac{1}{2\alpha}} = N^{-\frac{1}{2\alpha}} x^{-2+\frac{1}{2\alpha}}. \tag{128}$$

Integrating this term with respect to $x$ near the lower limit $\sqrt{N}$:

$$N^{-\frac{1}{2\alpha}} \left[ x^{-1+\frac{1}{2\alpha}} \right]_{x=\sqrt{N}} \sim N^{-\frac{1}{2\alpha}} (\sqrt{N})^{-1+\frac{1}{2\alpha}} = N^{-\frac{1}{2\alpha}} N^{-\frac{1}{2}+\frac{1}{4\alpha}}. \tag{129}$$

Simplifying the exponents yields the scaling $N^{-\frac{1}{2}-\frac{1}{4\alpha}}$.

Contribution near the upper limit ($x \approx N$): In the region where $x$ is large, $f(N-x) \sim \mathcal{O}(1)$ and the cumulative sum scales as $\mathcal{O}(x)$. The integrand is dominated by $x^{-\left(2-\frac{1}{2\alpha}\right)}$. Integrating this term near the upper limit $N$:

$$\left[ x^{-1+\frac{1}{2\alpha}} \right]^{x=N} \sim N^{-1+\frac{1}{2\alpha}}. \tag{130}$$

The asymptotic behavior of $\mathcal{I}$ is determined by the maximum of these two contributions. The contribution from the lower limit dominates when $-\frac{1}{2} - \frac{1}{4\alpha} > -1 + \frac{1}{2\alpha}$, which corresponds to $\alpha > 1.5$. Otherwise, the contribution from the upper limit dominates. Thus,

$$\mathcal{I} \approx \begin{cases} N^{-\frac{1}{2}-\frac{1}{4\alpha}} & \text{if } \alpha > 1.5, \\ N^{-1+\frac{1}{2\alpha}} & \text{if } 0.5 < \alpha < 1.5. \end{cases} \tag{131}$$

For $0.5 < \alpha < 1.5$, we have

$$T_{\leq(N-\sqrt{N})} \lesssim \gamma_0^{1/(2\alpha)} M^{1/(4\alpha)} N^{-(1-1/(2\alpha))}.$$

For $0.5 < \alpha < 1.5$, combining the bounds for the drift term and noise term, we have

$$L(N) \lesssim M^{-(2\alpha+2\beta-1)} + \left(M^{0.5}\gamma_0 N\right)^{-\frac{2(2\alpha+2\beta-1)}{2\alpha+1-2\beta}} + \gamma_0 M^{0.5} N^{-1}\sqrt{L(N)} + \gamma_0^{\frac{1}{2\alpha}} M^{\frac{1}{4\alpha}} N^{-(1-\frac{1}{2\alpha})}.$$

In intersection of Area Aa$^\star$ and $0.5 < \alpha < 1.5$, with choice of $e^*$ in $\gamma_0 = M^{-e^*}$ and $c^*$ we used for stable-decaying scheduling, we have

$$L(N) \lesssim M^{-(2\alpha+2\beta-1)} + \left(M^{0.5}\gamma_0 N\right)^{-\frac{2(2\alpha+2\beta-1)}{2\alpha+1-2\beta}} + \gamma_0 M^{0.5} N^{-c^*}\sqrt{L(N)} + \gamma_0^{\frac{1}{2\alpha}} M^{\frac{1}{4\alpha}} N^{-(1-c^*)(1-\frac{1}{2\alpha})}.$$

So in intersection of Area Aa$^\star$ and $0.5 < \alpha < 1.5$, we have

$$R_f(M^\star, f/M^\star, (M^\star)^{-e^*}) \lesssim f^{-\frac{2(4\alpha-1)(2\alpha+2\beta-1)}{16\alpha^2+8\alpha\beta+2\alpha-2\beta-1}}. \tag{132}$$

Therefore, linear decaying scheduling has an advantage compared to constant learning rate in the intersection of Area Aa$^\star$ and $0.5 < \alpha < 1.5$.

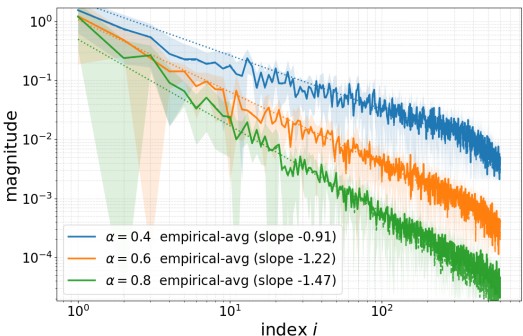

Figure 24: **Decay of gradient under the basis of $U$.** Colored solid lines show the average of gradients under the basis of $U$ for the parameter $(\alpha, \beta) = (0.4, 0.5), (0.6, 0.5), (0.8, 0.5)$. On the legend, we only noted the $\alpha$. The dotted line is fitted for the average of gradients, and we noted its slope in the legend. Slope is similar to $2\alpha$ within error $0.13$.

# I   ANALYSIS ABOUT HYPOTHESIS FOR THE POSITION OF THE BENEFICIAL AREA

In this section, we cover the analysis of stochastic gradient decay, which was deferred from Section 5.1.

We examine the decaying structure of the stochastic gradient. Assume a feature vector $\boldsymbol{x}$ is drawn from the distribution $\mathcal{N}(0, \boldsymbol{H})$, and its label is $y = \langle \boldsymbol{x}, \boldsymbol{w}^* \rangle$. Then the stochastic gradient for that feature vector is

$$\boldsymbol{g} = 2\big(\langle \boldsymbol{S}\boldsymbol{x}_t, \boldsymbol{\theta}_{t-1} \rangle - y\big)\,\boldsymbol{S}\boldsymbol{x}_t$$
$$= 2\boldsymbol{S}\boldsymbol{x}\boldsymbol{x}^\mathsf{T}\boldsymbol{S}^\mathsf{T}(\boldsymbol{\theta} - \boldsymbol{\theta}^*) - 2\boldsymbol{S}\boldsymbol{x}\boldsymbol{x}^\mathsf{T}\boldsymbol{w}_\perp.$$

Taking the expectation of the stochastic gradient and using $\boldsymbol{S}\boldsymbol{H}\boldsymbol{w}_\perp = 0$, we obtain

$$\mathbb{E}[\boldsymbol{g}] = 2\boldsymbol{S}\boldsymbol{H}\boldsymbol{S}^\mathsf{T}(\boldsymbol{\theta} - \boldsymbol{\theta}^*) - 2\boldsymbol{S}\boldsymbol{H}\boldsymbol{w}_\perp$$
$$= 2\boldsymbol{S}\boldsymbol{H}\boldsymbol{S}^\mathsf{T}(\boldsymbol{\theta} - \boldsymbol{\theta}^*).$$

Lin et al. (2024) proved that the eigenvalues $\lambda_i$ of $\boldsymbol{S}\boldsymbol{H}\boldsymbol{S}^\mathsf{T}$ satisfy $\lambda_i \asymp i^{-2\alpha}$. Let the eigenvalue decomposition of $\boldsymbol{S}\boldsymbol{H}\boldsymbol{S}^\mathsf{T}$ be $\boldsymbol{S}\boldsymbol{H}\boldsymbol{S}^\mathsf{T} = \boldsymbol{U}\,\mathrm{diag}(\lambda_i)\boldsymbol{U}^\mathsf{T}$. Then

$$\boldsymbol{U}^\mathsf{T}\mathbb{E}[\boldsymbol{g}] = 2\,\mathrm{diag}(\lambda_i)\,\boldsymbol{U}^\mathsf{T}(\boldsymbol{\theta} - \boldsymbol{\theta}^*),$$

which provides the intuition that $\mathbb{E}[\boldsymbol{g}]$, expressed in the basis of the columns of $\boldsymbol{U}$, decays as $i^{-2\alpha}$. Figure 24 shows that the expected gradient decays similarly to $i^{-2\alpha}$. Also, note that a larger $\alpha$ leads to a steeper gradient decay.

## J    SCALING LAW OF ADAM WITH HEURISTIC

First, we recall the Adam (Kingma & Ba, 2015) update and notation. For the stochastic gradient

$$\boldsymbol{g}_k = 2\big(\langle \boldsymbol{Sx}_k, \boldsymbol{\theta}_k \rangle - y_k\big)\, \boldsymbol{Sx}_k.$$

Adam maintains first and second moment estimates

$$\boldsymbol{m}_k = \beta_1\, \boldsymbol{m}_{k-1} + (1 - \beta_1)\, \boldsymbol{g}_k,$$
$$\boldsymbol{v}_k = \beta_2\, \boldsymbol{v}_{k-1} + (1 - \beta_2)\, \boldsymbol{g}_k^{\odot 2},$$

with bias corrections $\hat{\boldsymbol{m}}_k = \boldsymbol{m}_k/(1 - \beta_1^k)$, $\hat{\boldsymbol{v}}_k = \boldsymbol{v}_k/(1 - \beta_2^k)$. The update is

$$\boldsymbol{\theta}_{k+1} = \boldsymbol{\theta}_k - \gamma_k\, \hat{\boldsymbol{m}}_k \odot \big(\epsilon \boldsymbol{1} + \hat{\boldsymbol{v}}_k\big)^{-1/2},$$

where $\odot$ denotes elementwise multiplication and the $(-1/2)$ power is taken elementwise; $\epsilon > 0$ is the usual damping (we will set $\epsilon = 0$ in the asymptotic analysis).

Xiao et al. (2025) proposed a heuristic for Adam: take $\beta_2$ sufficiently close to 1 so that the second moment can be treated as an expectation.

We present results under a same heuristic. In addition, Ferbach et al. (2025) prove that SGD with momentum obeys the same scaling law as SGD; motivated by this, we set $\beta_1 = 0$ and omit the first-moment term for simplicity.

**Second-moment proxy and normalized update.**    Under the heuristic of Xiao et al. (2025),

$$\hat{\boldsymbol{v}}_k \approx 4\mathbb{E}\Big[(\boldsymbol{Sx}_k)^{\odot 2} \big(\langle \boldsymbol{Sx}_k, \boldsymbol{\theta}_k \rangle - y_k\big)^2 \,\Big|\, \mathcal{F}_k\Big]$$

Let $\boldsymbol{\xi} := \boldsymbol{Sx}_k$ be the sketched feature vector. Under the Gaussian assumption on $\boldsymbol{x}_k$, $\boldsymbol{\xi}$ follows a multivariate Gaussian distribution with covariance $\boldsymbol{K} = \boldsymbol{SHS}^\mathsf{T}$.

First, we analyze the residual term. Using the decomposition $\boldsymbol{w}^* = \boldsymbol{S}^\mathsf{T}\boldsymbol{\theta}^* + \boldsymbol{w}_\perp$, the residual at step $k$ is:

$$\begin{aligned}
\langle \boldsymbol{Sx}_k, \boldsymbol{\theta}_k \rangle - y_k &= \boldsymbol{x}_k^\mathsf{T}\boldsymbol{S}^\mathsf{T}\boldsymbol{\theta}_k - \boldsymbol{x}_k^\mathsf{T}\boldsymbol{w}^* \\
&= \boldsymbol{x}_k^\mathsf{T}\boldsymbol{S}^\mathsf{T}\boldsymbol{\theta}_k - \boldsymbol{x}_k^\mathsf{T}(\boldsymbol{S}^\mathsf{T}\boldsymbol{\theta}^* + \boldsymbol{w}_\perp) \\
&= (\boldsymbol{Sx}_k)^\mathsf{T}(\boldsymbol{\theta}_k - \boldsymbol{\theta}^*) - \boldsymbol{x}_k^\mathsf{T}\boldsymbol{w}_\perp \\
&= \boldsymbol{\xi}^\mathsf{T}\delta_k - \zeta_k,
\end{aligned}$$

where $\delta_k := \boldsymbol{\theta}_k - \boldsymbol{\theta}^*$ is the parameter error, and $\zeta_k := \boldsymbol{x}_k^\mathsf{T}\boldsymbol{w}_\perp$ is the irreducible residual term induced by approximation error.

We focus on the $i$-th coordinate of the second moment vector. Let $X := (\boldsymbol{\xi})_i$ and $Y := \boldsymbol{\xi}^\mathsf{T}\delta_k - \zeta_k$ (the residual). Since both are linear combinations of the Gaussian vector $\boldsymbol{x}_k$, they are jointly Gaussian. We apply Isserlis' theorem:

$$\mathbb{E}[X^2 Y^2] = \mathbb{E}[X^2]\mathbb{E}[Y^2] + 2\big(\mathbb{E}[XY]\big)^2.$$

We evaluate each term:

1. **Variance of the feature ($\mathbb{E}[X^2]$):**

$$\mathbb{E}[(\boldsymbol{\xi})_i^2] = (\boldsymbol{SHS}^\mathsf{T})_{ii} = (\mathrm{diag}(\boldsymbol{K}))_i.$$

2. **Variance of the residual ($\mathbb{E}[Y^2]$):** By definition, the expected squared residual is the population risk:

$$\mathbb{E}[Y^2] = \mathbb{E}\big[(\langle \boldsymbol{Sx}_k, \boldsymbol{\theta}_k \rangle - y_k)^2\big] = L(\boldsymbol{\theta}_k).$$

3. **Covariance term ($\mathbb{E}[XY]$):** This term involves the correlation between the feature and the residual.

$$\begin{aligned}
\mathbb{E}[XY] &= \mathbb{E}\Big[(\boldsymbol{\xi})_i\big(\boldsymbol{\xi}^\mathsf{T}\delta_k - \zeta_k\big)\Big] \\
&= \mathbb{E}\Big[(\boldsymbol{Sx}_k)_i(\boldsymbol{x}_k^\mathsf{T}\boldsymbol{S}^\mathsf{T}\delta_k)\Big] - \mathbb{E}\Big[(\boldsymbol{Sx}_k)_i(\boldsymbol{x}_k^\mathsf{T}\boldsymbol{w}_\perp)\Big].
\end{aligned}$$

The first part is the standard covariance calculation:

$$\mathbb{E}\Big[(\boldsymbol{S}\boldsymbol{x}_k)_i(\boldsymbol{x}_k^\mathsf{T}\boldsymbol{S}^\mathsf{T}\delta_k)\Big] = \big(\boldsymbol{S}\mathbb{E}[\boldsymbol{x}_k\boldsymbol{x}_k^\mathsf{T}]\boldsymbol{S}^\mathsf{T}\delta_k\big)_i = (\boldsymbol{S}\boldsymbol{H}\boldsymbol{S}^\mathsf{T}\delta_k)_i = (\boldsymbol{K}\delta_k)_i.$$

The second part vanishes due to the orthogonality property of the projected solution $(\boldsymbol{S}\boldsymbol{H}\boldsymbol{w}_\perp = 0)$:

$$\mathbb{E}\Big[(\boldsymbol{S}\boldsymbol{x}_k)_i(\boldsymbol{x}_k^\mathsf{T}\boldsymbol{w}_\perp)\Big] = \big(\boldsymbol{S}\mathbb{E}[\boldsymbol{x}_k\boldsymbol{x}_k^\mathsf{T}]\boldsymbol{w}_\perp\big)_i = (\boldsymbol{S}\boldsymbol{H}\boldsymbol{w}_\perp)_i = 0.$$

Thus, $\mathbb{E}[XY] = (\boldsymbol{K}\delta_k)_i$.

Substituting these back into Isserlis' formula yields:

$$\mathbb{E}[X^2Y^2] = (\mathrm{diag}(\boldsymbol{K}))_i\, L(\boldsymbol{\theta}_k) \;+\; 2\big((\boldsymbol{K}(\boldsymbol{\theta}_k - \boldsymbol{\theta}^*))_i\big)^2.$$

Stacking the coordinates gives the following equation.

$$\mathbb{E}\Big[(\boldsymbol{S}\boldsymbol{x}_k)^{\odot 2}\,\big(\langle\boldsymbol{S}\boldsymbol{x}_k,\boldsymbol{\theta}_k\rangle - y_k\big)^2 \,\Big|\, \mathcal{F}_k\Big] = \mathrm{diag}(\boldsymbol{K})\cdot L(\boldsymbol{\theta}_k) \;+\; 2\,(\boldsymbol{K}(\boldsymbol{\theta}_k - \boldsymbol{\theta}^*))^{\odot 2}. \tag{133}$$

Moreover, by Cauchy–Schwarz,

$$(\boldsymbol{K}(\boldsymbol{\theta}_k - \boldsymbol{\theta}^*))_j^2 \le \boldsymbol{K}_{jj}\,(\boldsymbol{\theta}_k - \boldsymbol{\theta}^*)^\mathsf{T}\boldsymbol{K}(\boldsymbol{\theta}_k - \boldsymbol{\theta}^*) \le \boldsymbol{K}_{jj}\,L(\boldsymbol{\theta}_k),$$

and hence the exact second moment admits the coordinate-wise bounds

$$\mathrm{diag}(\boldsymbol{K})\cdot L(\boldsymbol{\theta}_k) \;\preceq\; \mathbb{E}\Big[(\boldsymbol{S}\boldsymbol{x}_k)^{\odot 2}\,\big(\langle\boldsymbol{S}\boldsymbol{x}_k,\boldsymbol{\theta}_k\rangle - y_k\big)^2 \,\Big|\, \mathcal{F}_k\Big] \;\preceq\; 3\,\mathrm{diag}(\boldsymbol{K})\cdot L(\boldsymbol{\theta}_k), \tag{134}$$

where $\preceq$ denotes elementwise inequality. In particular, replacing the exact second moment by $\mathrm{diag}(\boldsymbol{K})\cdot L(\boldsymbol{\theta}_k) = \mathrm{diag}(\boldsymbol{S}\boldsymbol{H}\boldsymbol{S}^\mathsf{T})\cdot L(\boldsymbol{\theta}_k)$ changes the normalization by at most a universal constant factor.

Hence, the (elementwise) normalized update satisfies

$$\boldsymbol{\theta}_{k+1} - \boldsymbol{\theta}_k \approx -\gamma_k\frac{\big(\langle\boldsymbol{S}\boldsymbol{x}_k,\boldsymbol{\theta}_k\rangle - y_k\big)\,\boldsymbol{S}\boldsymbol{x}_k}{\sqrt{\mathrm{diag}(\boldsymbol{S}\boldsymbol{H}\boldsymbol{S}^\mathsf{T})\cdot L(\boldsymbol{\theta}_k)}}.$$

**One-step update formula.** Recalling the Taylor expansion used for signSGD,

$$\mathbb{E}\big[q(\boldsymbol{\theta}_{k+1}) - q(\boldsymbol{\theta}_k)\,\big|\,\mathcal{F}_k\big] = \mathbb{E}\Big[\langle\nabla q(\boldsymbol{\theta}_k),\,\boldsymbol{\theta}_{k+1} - \boldsymbol{\theta}_k\rangle\,\Big|\,\mathcal{F}_k\Big] + \frac{1}{2}\,\mathbb{E}\Big[\langle\nabla^2 q,\,(\boldsymbol{\theta}_{k+1} - \boldsymbol{\theta}_k)^{\otimes 2}\rangle\,\Big|\,\mathcal{F}_k\Big].$$

*Gradient term:*

$$\mathbb{E}\Big[\langle\nabla q(\boldsymbol{\theta}_k),\,\boldsymbol{\theta}_{k+1} - \boldsymbol{\theta}_k\rangle\,\Big|\,\mathcal{F}_k\Big] \approx -\gamma_k\left\langle\nabla q(\boldsymbol{\theta}_k),\,\frac{\boldsymbol{S}\boldsymbol{H}\boldsymbol{S}^\mathsf{T}\boldsymbol{\theta}_k - \boldsymbol{S}\boldsymbol{H}\boldsymbol{w}^*}{\sqrt{\mathrm{diag}(\boldsymbol{S}\boldsymbol{H}\boldsymbol{S}^\mathsf{T})\cdot L(\boldsymbol{\theta}_k)}}\right\rangle.$$

*Quadratic term:*

$$\mathbb{E}\Big[\langle\nabla^2 q,\,(\boldsymbol{\theta}_{k+1} - \boldsymbol{\theta}_k)^{\otimes 2}\rangle\,\Big|\,\mathcal{F}_k\Big]$$

$$\approx \gamma_k^2\mathbb{E}\bigg[\Big\langle\nabla^2 q,\,\mathrm{diag}(\boldsymbol{S}\boldsymbol{H}\boldsymbol{S}^\mathsf{T})^{-1/2}\,\boldsymbol{S}\boldsymbol{x}_k\boldsymbol{x}_k^\mathsf{T}\boldsymbol{S}^\mathsf{T}\,\mathrm{diag}(\boldsymbol{S}\boldsymbol{H}\boldsymbol{S}^\mathsf{T})^{-1/2}\Big\rangle\frac{\big(\langle\boldsymbol{S}\boldsymbol{x}_k,\boldsymbol{\theta}_k\rangle - y_k\big)^2}{L(\boldsymbol{\theta}_k)}\,\bigg|\,\mathcal{F}_k\bigg]$$

$$= \frac{\gamma_k^2}{L(\boldsymbol{\theta}_k)}\Big(\big\langle\mathrm{diag}(\boldsymbol{S}\boldsymbol{H}\boldsymbol{S}^\mathsf{T})^{-1/2}\nabla^2 q\,\mathrm{diag}(\boldsymbol{S}\boldsymbol{H}\boldsymbol{S}^\mathsf{T})^{-1/2},\,\boldsymbol{S}\boldsymbol{H}\boldsymbol{S}^\mathsf{T}\big\rangle\,L(\boldsymbol{\theta}_k)$$

$$+ 2\big\langle\boldsymbol{S}\boldsymbol{H}\boldsymbol{S}^\mathsf{T}\,\mathrm{diag}(\boldsymbol{S}\boldsymbol{H}\boldsymbol{S}^\mathsf{T})^{-1/2}\nabla^2 q\,\mathrm{diag}(\boldsymbol{S}\boldsymbol{H}\boldsymbol{S}^\mathsf{T})^{-1/2}\,\boldsymbol{S}\boldsymbol{H}\boldsymbol{S}^\mathsf{T},\,(\boldsymbol{\theta}_k - \boldsymbol{\theta}^*)^{\otimes 2}\big\rangle\Big).$$

Combining the two contributions,

$$\mathbb{E}\big[q(\boldsymbol{\theta}_{k+1}) - q(\boldsymbol{\theta}_k)\,\big|\,\mathcal{F}_k\big] \approx -\frac{\gamma_k}{\sqrt{L(\boldsymbol{\theta}_k)}}\,\big\langle\nabla q(\boldsymbol{\theta}_k),\,\overline{\boldsymbol{K}}(\boldsymbol{\theta}_k - \boldsymbol{\theta}^*)\big\rangle + \frac{\gamma_k^2}{2}\,\big\langle\nabla^2 q,\,\boldsymbol{K}_\tau\big\rangle$$

$$+ \frac{\gamma_k^2}{L(\boldsymbol{\theta}_k)}\,\big\langle\boldsymbol{S}\boldsymbol{H}\boldsymbol{S}^\mathsf{T}\,\mathrm{diag}(\boldsymbol{S}\boldsymbol{H}\boldsymbol{S}^\mathsf{T})^{-1/2}\nabla^2 q\,\mathrm{diag}(\boldsymbol{S}\boldsymbol{H}\boldsymbol{S}^\mathsf{T})^{-1/2}\,\boldsymbol{S}\boldsymbol{H}\boldsymbol{S}^\mathsf{T},\,(\boldsymbol{\theta}_k - \boldsymbol{\theta}^*)^{\otimes 2}\big\rangle,$$

where $\boldsymbol{K}_\tau := \mathrm{diag}(\boldsymbol{S}\boldsymbol{H}\boldsymbol{S}^\mathsf{T})^{-1/2}\,\boldsymbol{S}\boldsymbol{H}\boldsymbol{S}^\mathsf{T}\,\mathrm{diag}(\boldsymbol{S}\boldsymbol{H}\boldsymbol{S}^\mathsf{T})^{-1/2}$.

**Mode-wise recursion.** For $r_i(k) := (\boldsymbol{\theta}_k - \boldsymbol{\theta}^*)^\mathsf{T}(\boldsymbol{K}\boldsymbol{u}_i \otimes \boldsymbol{w}_i)(\boldsymbol{\theta}_k - \boldsymbol{\theta}^*)$ (cf. Appendix E.1),

$$\mathbb{E}\big[r_i(k+1) - r_i(k) \,\big|\, \mathcal{F}_k\big] \approx -\frac{2\gamma_k}{\sqrt{L(\boldsymbol{\theta}_k)}}\,\lambda_i(\overline{\boldsymbol{K}})\,r_i(k) + \gamma_k^2\,(\boldsymbol{w}_i^\mathsf{T}\boldsymbol{K}_\tau\boldsymbol{K}\boldsymbol{u}_i) + \frac{2\gamma_k^2}{L(\boldsymbol{\theta}_k)}\,\lambda_i(\overline{\boldsymbol{K}})\,r_i(k)$$

$$= -\Big(\frac{2\gamma_k}{\sqrt{L(\boldsymbol{\theta}_k)}} - \frac{2\gamma_k^2}{L(\boldsymbol{\theta}_k)}\Big)\lambda_i(\overline{\boldsymbol{K}})\,r_i(k) + \gamma_k^2\,(\boldsymbol{w}_i^\mathsf{T}\boldsymbol{K}_\tau\boldsymbol{K}\boldsymbol{u}_i).$$

We now assume $f \equiv 1$, and $\gamma_k = \gamma_0$ for simplicity. Passing to the ODE limit as in Section E.2 we get following ODE for $P(t) = L(t/\gamma_0)$ and $p_i(t) = r_i(t/\gamma_0)$.

$$\frac{dp_i}{dt} \approx -2\left(\frac{1}{\sqrt{P(t)}} - \frac{\gamma_0}{P(t)}\right)\lambda_i(\overline{\boldsymbol{K}})\,p_i(t) + \gamma_0\,V_i', \tag{135}$$

where $V_i' := \boldsymbol{w}_i^\mathsf{T}\boldsymbol{K}_\tau\boldsymbol{K}\boldsymbol{u}_i$.

Interpreting the solution of the ODE as an implicit integral equation and summing over $i$, similar to Section E.2, and writing

$$Q_2(N) = 2\gamma_0\int_0^N\Big(\frac{1}{\sqrt{L(u)}} - \gamma_0\frac{1}{L(u)}\Big)\,du,$$

we obtain the implicit integral inequality for some $c_1, c_2 > 0$.

$$\|\boldsymbol{H}^{1/2}\boldsymbol{w}_\perp\|^2 + \sum_{i=1}^M r_i(0)\,\exp\Big(-c_1\lambda_i Q_2(N)\Big)$$

$$+ \gamma_0^2\sum_{i=1}^M V_i'\int_0^N \exp\Big(-2c_1\lambda_i\gamma_0\int_z^N\Big(\frac{1}{\sqrt{L(u)}} - \gamma_0\frac{1}{L(u)}\Big)\,du\Big)\,dz$$

$$\leq L(N) \leq \|\boldsymbol{H}^{1/2}\boldsymbol{w}_\perp\|^2 + \sum_{i=1}^M r_i(0)\,\exp\Big(-c_2\lambda_i Q_2(N)\Big)$$

$$+ \gamma_0^2\sum_{i=1}^M V_i'\int_0^N \exp\Big(-2c_2\lambda_i\gamma_0\int_z^N\Big(\frac{1}{\sqrt{L(u)}} - \gamma_0\frac{1}{L(u)}\Big)\,du\Big)\,dz.$$

**Drift transformation and limit phase.** By the same drift/approximation transformation as in equation 35,

$$M^{-2\alpha+\max(0,\,1-2\beta)} + \big(M^{\min(\alpha,0.5)}\,Q_2(N)\big)^{-\frac{2\alpha+2\beta-1}{2\alpha}}$$

$$+ \mathbf{1}_{\{\alpha>0.5,\,\beta>0.5\}}\,M^{-1}\big(M^{\min(\alpha,0.5)}Q_2(N)\big)^{-1+\frac{1}{2\alpha}} + \gamma_0^2\sum_{i=1}^M V_i'\int_0^N e^{-2c_1\lambda_i\gamma_0\int_z^N(\frac{f(u)}{\sqrt{L(u)}}-\gamma_0\frac{f(u)^2}{L(u)})\,du}\,f(z)^2\,dz$$

$$\lesssim L(N) \lesssim M^{-2\alpha+\max(0,\,1-2\beta)} + \big(M^{\min(\alpha,0.5)}\,Q_2(N)\big)^{-\frac{2\alpha+2\beta-1}{2\alpha}}$$

$$+ \mathbf{1}_{\{\alpha>0.5,\,\beta>0.5\}}\,M^{-1}\big(M^{\min(\alpha,0.5)}Q_2(N)\big)^{-1+\frac{1}{2\alpha}} + \gamma_0^2\sum_{i=1}^M V_i'\int_0^N e^{-2c_2\lambda_i\gamma_0\int_z^N(\frac{f(u)}{\sqrt{L(u)}}-\gamma_0\frac{f(u)^2}{L(u)})\,du}\,f(z)^2\,dz.$$

We will first handle the limit phase, similar to Section E.3.2. At stationarity, let $p_i(t) \to s_i$ and $P(t) \to L_\infty$, we must have

$$-2\left(\frac{1}{\sqrt{L_\infty}} - \frac{\gamma_0}{L_\infty}\right)\lambda_i(\overline{\boldsymbol{K}})\,s_i + \gamma_0\,V_i \approx 0 \implies s_i \approx \frac{\gamma_0\sqrt{L_\infty}}{2\,\lambda_i(\overline{\boldsymbol{K}})}\,V_i'\frac{1}{1-\frac{\gamma_0}{\sqrt{L_\infty}}} = \frac{\gamma_0\sqrt{L_\infty}}{2\,\lambda_i(\overline{\boldsymbol{K}})}\,(\boldsymbol{w}_i^\mathsf{T}\boldsymbol{K}_\tau\boldsymbol{K}\boldsymbol{u}_i)\frac{1}{1-\frac{\gamma_0}{\sqrt{L_\infty}}}.$$

Using the loss decomposition $P(t) = \sum_{i=1}^{M} p_i(t) + \|\boldsymbol{H}^{1/2}\boldsymbol{w}_\perp\|^2$, we obtain

$$
\begin{aligned}
L_\infty &= \sum_{i=1}^{M} s_i + \|\boldsymbol{H}^{1/2}\boldsymbol{w}_\perp\|^2 \asymp \frac{\gamma_0}{2}\Big(\sum_{i=1}^{M} \frac{w_i^\mathsf{T}\boldsymbol{K}_\tau \boldsymbol{K} u_i}{\lambda_i(\overline{K})}\Big)\sqrt{L_\infty}\frac{1}{1 - \frac{\gamma_0}{\sqrt{L_\infty}}} + \|\boldsymbol{H}^{1/2}\boldsymbol{w}_\perp\|^2 \\
&= \frac{\gamma_0}{2}\operatorname{Tr}\!\big(\operatorname{diag}(K)^{1/2}\boldsymbol{K}_\tau\big)\sqrt{L_\infty}\frac{1}{1 - \frac{\gamma_0}{\sqrt{L_\infty}}} + \|\boldsymbol{H}^{1/2}\boldsymbol{w}_\perp\|^2 \\
&= \frac{\gamma_0}{2}\operatorname{Tr}\!\big(\operatorname{diag}(K)^{1/2}\big)\sqrt{L_\infty}\frac{1}{1 - \frac{\gamma_0}{\sqrt{L_\infty}}} + \|\boldsymbol{H}^{1/2}\boldsymbol{w}_\perp\|^2.
\end{aligned}
$$

Then we get

$$
L_\infty \asymp \max\Big\{\gamma_0^2 \operatorname{Tr}\!\big(\operatorname{diag}(K)^{1/2}\big)^2, \|\boldsymbol{H}^{1/2}\boldsymbol{w}_\perp\|^2\Big\} \asymp \max\Big\{\gamma_0^2 M^{\,2-2\min(\alpha,0.5)}, M^{-2\alpha+\max(0,\,1-2\beta)}\Big\}.
$$

So we have the same floor as for signSGD.

Since $f$ is bounded and $L(N) \geq \gamma_0^2 M^{\,2-2\min(\alpha,0.5)}$,

$$
\frac{\frac{f(u)}{\sqrt{L(u)}}}{\gamma_0 \frac{f(u)^2}{L(u)}} = \frac{\sqrt{L(u)}}{\gamma_0 f(u)} \gtrsim M^{1-\min(\alpha,0.5)},
$$

so the subtraction inside $Q_2$ is asymptotically negligible and $Q_2(N) \asymp Q(N)$. Hence, the drift contribution coincides with that of signSGD.

**Scaling law (constant learning rate).** For $f \equiv 1$, Adam (under this heuristic) follows the same scaling law as signSGD:

$$
\begin{aligned}
R(M, N, \gamma_0) \asymp\ & M^{-2\alpha+\max(0,\,1-2\beta)} + \big(M^{\min(\alpha,0.5)}\,N\,\gamma_0\big)^{-\frac{2(2\alpha+2\beta-1)}{2\alpha+1-2\beta}} \\
& + \big(M^{\frac{6\alpha-1}{4\alpha-2}}\,N\,\gamma_0\big)^{-\frac{2(2\alpha-1)}{2\alpha+1}} + \gamma_0^2\,M^{\,2-2\min(\alpha,0.5)}.
\end{aligned}
$$

Since the loss formula $R(M, N, \gamma_0)$ is the same as signSGD, the compute-optimal scaling law will also be the same as signSGD. So we expect that Adam has the compute-optimal scaling law in Table 1. Figure 25 shows that exponents in the Table 1 and measured compute-optimal loss slope and optimal model size slope (in log-log plot) for Adam match well.

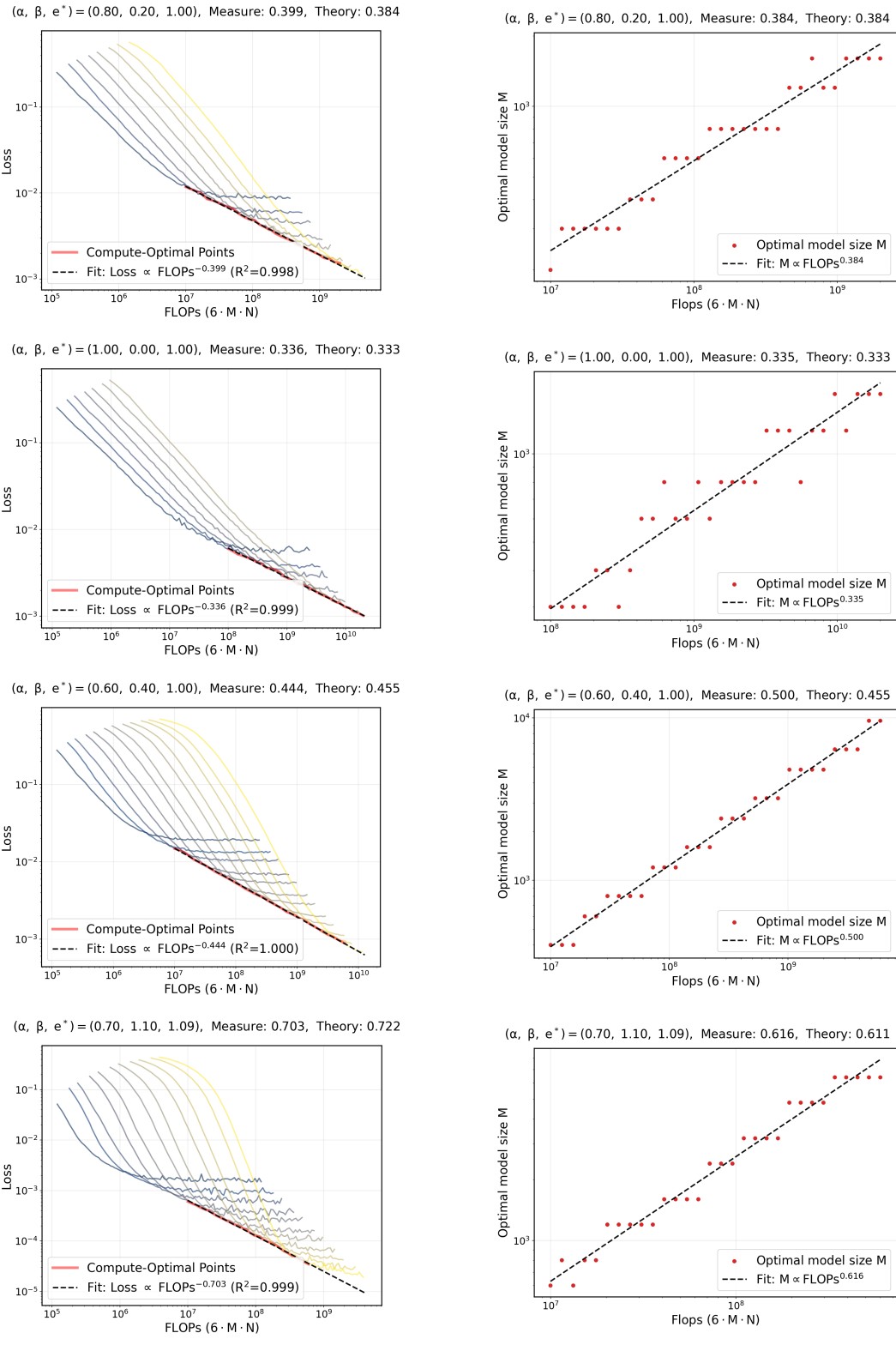

Figure 25: **Measure of compute-optimal loss slope and optimal model size slope for Adam.** We validate the exponent of $R\left(M^\star, \frac{\mathfrak{f}}{M^\star}, \gamma_0^\star\right)$ and $M^\star$ with respect to $\mathfrak{f}$ in the Table 1. The left plot shows the compute-optimal loss with respect to FLOPS $6MN$. The right plot shows the optimal model size with respect to FLOPS $6MN$. Each plot includes the measured slope and the theoretical slope from the Table 1. Parameters : $\beta_1 = 0.9$, $\beta_2 = 0.999$, $\epsilon = 10^{-8}$, $\gamma_0 = 0.002$.

# K    OMITTED ANALYSIS FROM SECTION E

## K.1    OMITTED PROOF OF (31) AND (33)

In this section, we cover omitted proof of (31) and (33). Note that the proof is almost similar to Paquette et al. (2024), but we cover it briefly for completeness. Refer to Appendix F, G, H of Paquette et al. (2024) for more details.

It is enough to prove

$$-\frac{1}{2\pi\mathrm{i}} \oint_\Gamma e^{-p_d\,Q(N)z}\, \langle \mathcal{L}(z), v^{\otimes 2}\rangle\, dz \;\eqsim\; M^{-2\alpha+\max(0,\,1-2\beta)}$$
$$+ \left(M^{\min(\alpha,\,0.5)}\,Q(N)\right)^{-\frac{2\alpha+2\beta-1}{2\alpha}}$$
$$+ 1_{\{\alpha>0.5,\beta>0.5\}}M^{-1}\left(M^{\min(\alpha,\,0.5)}\,Q(N)\right)^{-1+\frac{1}{2\alpha}}.$$

From now on, we will use similar notation to Paquette et al. (2024), except in the inevitable case, to facilitate easy comparison for the reader. Note that we use $M$ and $d$ for model size and initial dimension before projection, while Paquette et al. (2024) uses $d$ and $v$.

We use $\Gamma$ for the contour containing the spectrum of $\boldsymbol{K}$, while Paquette et al. (2024) used $\Gamma \cup \Gamma_0$ for that, where $\Gamma_0$ is a small circle containing the origin.

Let

$$\mathcal{F}(N) \;:=\; -\frac{1}{2\pi i} \oint_\Gamma \langle \mathcal{R}(z), (H^{1/2}w^*)^{\otimes 2}\rangle\, e^{-p_d Q(N)\,z}\, dz. \tag{136}$$

The exponential kernel $e^{-p_d Q(N)z}$ replaces all polynomial weights in the analysis of Paquette et al. (2024). The resulting leading orders remain the same while constants and exponents are altered in a transparent way; precise statements follow.

We can split the $\mathcal{F}(N)$ by splitting the keyhole contour $\Gamma$. We let

$$\mathcal{F}(N) = \mathcal{F}_0(N) + \mathcal{F}_{\mathrm{caps}}(N) + \mathcal{F}_C(N) + (\text{lower-order}), \tag{137}$$

where $\mathcal{F}_0$ collects the small circle around the origin, $\mathcal{F}_{\mathrm{caps}}$ collects the right/left caps adjacent to the positive real axis, and $\mathcal{F}_C$ collects the central arc close to $[0,1]$. Refer to Appendix F of Paquette et al. (2024) for more details about the picture of contour and decomposition of contour.

In the following proposition the function $(x)_+ := \max(x, 0)$ is used.

*Proposition* K.1.  $\mathcal{F}_0(N)$ is independent of $N$ and obeys

$$\left| \mathcal{F}_0(0) - \sum_{j=1}^d \frac{j^{-2\alpha-2\beta}}{1 + j^{-2\alpha}M^{2\alpha}\,\kappa(d/M)} \right| \;\le\; C\,M^{-2\alpha+(2\beta-1)_++-1}.$$

*Sketch.*  Putting $z = 0$ to the exponential leads to 1, so we can reduce to the analysis of Paquette et al. (2024). So the error bound is identical.  □

After this $\mathcal{F}_0(N) \eqsim M^{-2\alpha+\max(0,\,1-2\beta)}$ holds by identical procedure calculating $\sum_{j=1}^d \frac{j^{-2\alpha-2\beta}}{1+j^{-2\alpha}M^{2\alpha}\kappa(d/M)}$.

*Proposition* K.2.  There exist functions $f, g \geq 0$ with

$$f(N) \leq C\exp\!\left(-p_d Q(N)\,M^{-2\alpha}\right), \qquad g(N) \leq C\exp\!\left(-p_d Q(N)\right),$$

so that

$$\left| \mathcal{F}_{\mathrm{caps}}(N)\right| \;\leq\; C\,f(N)\,M^{-2\alpha+(1-2\beta)_+} + C\,g(N).$$

*Sketch.*  Use $|m(z) - 1| \lesssim M^{-\min\{2\alpha,1\}}$ (as in Paquette et al. (2024)) on a cap pushed $\mathcal{O}(1)$-close to $[0,1]$ to replace $\langle \mathcal{R}(z), (H^{1/2}w^*)^{\otimes 2}\rangle$ by a simple partial fraction, and control the remainder by the real part of $z$.  □

The main contribution arises from the arc parameterized by $z(u) = u + i\eta(u)$ with $u \in [M^{-2\alpha}, 1]$ and $|\eta(u)| \ll u$. Along this arc we have the uniform approximation

$$\left| m\big(z(u)\big) - \left(1 - \tfrac{\pi}{2\alpha}(c(u) + i)\, u^{-1/(2\alpha)} M^{-1}\right) \right| \leq \varepsilon\, u^{-1/(2\alpha)} M^{-1} \tag{138}$$

for some bounded real $c(u)$. Inserting (138) in $\mathcal{R}(z) = (-zI + m(z)H)^{-1}$ and extracting the imaginary part produces two canonical integrals,

$$\mathcal{F}_{pp}(N) := \frac{1}{2\alpha} \int_0^1 u^{(2\beta-1)/(2\alpha)} e^{-p_d Q(N)u}\, du, \qquad \mathcal{F}_{ac}(N) := \frac{c_\beta}{2\alpha} \int_{M^{-2\alpha}}^1 u^{-1/(2\alpha)} M^{-1} e^{-p_d Q(N)u}\, du, \tag{139}$$

with $c_\beta = \sum_{j\geq 1} j^{-2\beta}$ if $2\beta > 1$ and $c_\beta = 0$ otherwise.

*Proposition* K.3. There exists $C > 0$ such that for all $N \geq 0$, $|\mathcal{F}_C(N)| \leq C\big(\mathcal{F}_{pp}(N) + \mathcal{F}_{ac}(N)\big)$. Moreover, there are $A > 0$ and a bounded function $C(N) > 0$ with $C(N) \leq 1 + \varepsilon$ whenever $p_d Q(N) \in [A, M^{2\alpha}/A]$, and

$$\frac{1}{C(N)}\big(\mathcal{F}_{pp}(N) + \mathcal{F}_{ac}(N)\big) \leq \mathcal{F}_C(N) \leq C(N)\big(\mathcal{F}_{pp}(N) + \mathcal{F}_{ac}(N)\big).$$

*Sketch.* Parameterize $\Gamma_C$ by $u$ and use (138) to separate real/imaginary parts. The imaginary terms integrate exactly to (139), while the real part is smaller by a factor $\mathcal{O}(\varepsilon)$ since $|\eta(u)| \ll u$. $\qquad\square$

*Proposition* K.4 (Asymptotics of $\mathcal{F}_{pp}$). Assume $2\alpha + 2\beta > 1$ and set $X := p_d Q(N)$. For any $\varepsilon > 0$ there exists $A > 0$ such that for $X \geq A$,

$$\big|\mathcal{F}_{pp}(N) - g_{pp}(N)\big| \leq \varepsilon\, g_{pp}(N),$$

where

$$g_{pp}(N) := (2\alpha)^{-1} X^{-(1+\beta/\alpha)+1/(2\alpha)}\, \Gamma\big(\tfrac{\beta}{\alpha} - \tfrac{1}{2\alpha} + 1\big).$$

Moreover, if $X \leq \widetilde{A}$ then $c \leq \mathcal{F}_{pp}(N) \leq C$ for constants $c, C > 0$, and if $X \geq \widetilde{A} M^{2\alpha}$ then $\mathcal{F}_{pp}(N) \leq \widetilde{C}\, \mathcal{F}_0(N)$ for some $\widetilde{C} > 0$ independent of $M$.

*Sketch.* With the change of variables $w = Xu$, we get

$$\mathcal{F}_{pp}(N) = (2\alpha)^{-1} X^{-(1+\beta/\alpha)+1/(2\alpha)} \int_0^X w^{(2\beta-1)/(2\alpha)} e^{-w}\, dw.$$

Comparing to the complete gamma integral yields the relative error bound in terms of the upper incomplete gamma tail, which can be made $\leq \varepsilon$ by choosing $A$ large. The remaining bounds follow by monotonicity and elementary estimates. $\qquad\square$

*Proposition* K.5 (Asymptotics of $\mathcal{F}_{ac}$). Let $X := p_d Q(N)$. There exists $C(\alpha, \beta) > 0$ such that

$$\mathcal{F}_{ac}(N) \leq \begin{cases} C\, \mathcal{F}_0(N), & 2\beta > 1,\ 2\alpha < 1, \\ 0, & 2\beta < 1. \end{cases}$$

If in addition $2\alpha > 1$ and $2\beta > 1$, then for any $\varepsilon > 0$ there is $A > 0$ such that whenever $X \in [A, M^{2\alpha}/A]$,

$$\big|\mathcal{F}_{ac}(N) - g_{ac}(N)\big| \leq \varepsilon\, g_{ac}(N), \qquad g_{ac}(N) := \Big(\sum_{j=1}^\nu j^{-2\beta}\Big) (2\alpha)^{-1} \Gamma\big(1 - \tfrac{1}{2\alpha}\big) X^{-1+1/(2\alpha)} M^{-1}.$$

Furthermore, for any $\widetilde{A} > 0$ there exist constants $C, c > 0$ (independent of $M$) such that

$$\mathcal{F}_{ac}(N) \leq \begin{cases} C\, M^{-1}, & X \leq \widetilde{A}, \\ c\, \mathcal{F}_0(N), & X \geq \widetilde{A} M^{2\alpha}. \end{cases}$$

*Sketch.* Compare the truncated integral in (139) with its extension to $[0, \infty)$ and control the two tails $[0, M^{-2\alpha}]$ and $[1, \infty)$ separately. The first is at most $\widetilde{c} M^{-2\alpha}$; the second is bounded by $M^{-1} X^{-1} e^{-X}$. Normalizing by $g_{ac}(N)$ shows both are relatively small for $X \in [A, M^{2\alpha}/A]$ with $A$ large. The endpoint bounds follow from dropping the exponential and from a crude $\int e^{-Xu} du \leq X^{-1} e^{-X M^{-2\alpha}}$ estimate when $X \gtrsim M^{2\alpha}$. $\qquad\square$

Finally we get

$$-\frac{1}{2\pi i}\oint_{\Gamma}e^{-p_d\,Q(N)z}\,\langle\mathcal{L}(z),\,v^{\otimes 2}\rangle\,dz \;\asymp\; \mathcal{F}_0(N)+\mathcal{F}_{\mathrm{caps}}(N)+\mathcal{F}_C(N)$$

$$\asymp M^{-2\alpha+\max(0,\,1-2\beta)}+\left(p_d\,Q(N)\right)^{-\frac{2\alpha+2\beta-1}{2\alpha}}$$

$$+\,\mathbf{1}_{\{\alpha>0.5,\beta>0.5\}}M^{-1}\left(p_d\,Q(N)\right)^{-1+\frac{1}{2\alpha}}M^{-2\alpha+\max(0,\,1-2\beta)}$$

$$\asymp M^{-2\alpha+\max(0,\,1-2\beta)}+\left(M^{\min(\alpha,\,0.5)}\,Q(N)\right)^{-\frac{2\alpha+2\beta-1}{2\alpha}}$$

$$+\,\mathbf{1}_{\{\alpha>0.5,\beta>0.5\}}M^{-1}\left(M^{\min(\alpha,\,0.5)}\,Q(N)\right)^{-1+\frac{1}{2\alpha}}.$$

## K.2 NOTE ON THE $\arcsin x\approx x$ APPROXIMATION

We explain that it is possible to replace the linear approximation $\arcsin x\approx x$ by an inequality, and the main results of our paper remain unchanged.

**Replacing the $\arcsin$–linearization by a uniform sandwich.** Fix $0<\rho\le 1$ and define

$$c_1(\rho)\;:=\;\inf_{|t|\le\rho}\frac{\arcsin t}{t}=1,\qquad c_2(\rho)\;:=\;\sup_{|t|\le\rho}\frac{\arcsin t}{t}=\frac{\arcsin\rho}{\rho}\;\le\;\frac{\pi}{2}.$$

For $x\in\mathbb{R}^d$ with $\|x\|_\infty\le\rho$, the entrywise odd and monotone map $t\mapsto\arcsin t$ satisfies the componentwise bounds

$$c_1(\rho)\,x\;\le\;\arcsin(x)\;\le\;c_2(\rho)\,x.$$

In our update, put $v_k:=\boldsymbol{\theta}_k-\boldsymbol{\theta}^*$ and

$$x_k\;:=\;\frac{\overline{K}\,v_k}{\sqrt{L(\boldsymbol{\theta}_k)}},\qquad\text{so that}\quad\arcsin(x_k)=D_k\,x_k,$$

for some diagonal $D_k=\mathrm{diag}(\kappa_{k,1},\dots,\kappa_{k,d})$ with $c_1(\rho)\le\kappa_{k,j}\le c_2(\rho)$. Using $K^\mathsf{T}=K$ and $K^\mathsf{T}\overline{K}=\overline{K}^\mathsf{T}K^\mathsf{T}$, the one–step drift can be written as

$$\mathbb{E}[r_i(k{+}1)-r_i(k)\mid\mathcal{F}_k]=-\frac{2\gamma_k}{\pi\sqrt{L(\boldsymbol{\theta}_k)}}\,v_k^\mathsf{T}\left(K\boldsymbol{u}_i\boldsymbol{w}_i^\mathsf{T}+\boldsymbol{w}_i\boldsymbol{u}_i^\mathsf{T}K\right)D_k\,\overline{K}\,v_k\;+\;\frac{2\gamma_k^2}{\pi}\left(\boldsymbol{w}_i^\mathsf{T}K_\sigma K\boldsymbol{u}_i\right).$$

Since $D_k$ is diagonal with $c_1(\rho)I\preceq D_k\preceq c_2(\rho)I$, the quadratic form is sandwiched between the same expression with $D_k$ replaced by $c_1(\rho)I$ and $c_2(\rho)I$. Recalling the identity used earlier,

$$v_k^\mathsf{T}\left(K\boldsymbol{u}_i\boldsymbol{w}_i^\mathsf{T}+\boldsymbol{w}_i\boldsymbol{u}_i^\mathsf{T}K\right)\overline{K}\,v_k=2\,\lambda_i(\overline{K})\,r_i(k),$$

we obtain the two–sided one–step bound

$$-\frac{4\,c_2(\rho)\,\gamma_k}{\pi\sqrt{L(\boldsymbol{\theta}_k)}}\,\lambda_i(\overline{K})\,r_i(k)\;+\;\frac{2\gamma_k^2}{\pi}\left(\boldsymbol{w}_i^\mathsf{T}K_\sigma K\boldsymbol{u}_i\right)\;\le\;\mathbb{E}[r_i(k{+}1)-r_i(k)\mid\mathcal{F}_k]$$

$$\le\;-\frac{4\,c_1(\rho)\,\gamma_k}{\pi\sqrt{L(\boldsymbol{\theta}_k)}}\,\lambda_i(\overline{K})\,r_i(k)\;+\;\frac{2\gamma_k^2}{\pi}\left(\boldsymbol{w}_i^\mathsf{T}K_\sigma K\boldsymbol{u}_i\right).$$

**Consequences for the ODE limit and the implicit integral equation.** Let $\gamma_k=\gamma_0 f(k)$, $t=k\gamma_0$, $p_i(t):=r_i(k)$, and $P(t):=L(\boldsymbol{\theta}_k)$, as in Appendix E.2. Then we obtain the differential inequalities

$$-\frac{4\,c_2(\rho)}{\pi\sqrt{P(t)}}\,\lambda_i(\overline{K})\,f(t/\gamma_0)\,p_i(t)+\frac{2\gamma_0}{\pi}f(t/\gamma_0)^2 V_i\;\le\;\dot{p}_i(t)\;\le\;-\frac{4\,c_1(\rho)}{\pi\sqrt{P(t)}}\,\lambda_i(\overline{K})\,f(t/\gamma_0)\,p_i(t)+\frac{2\gamma_0}{\pi}f(t/\gamma_0)^2 V_i,$$

with $V_i:=\boldsymbol{w}_i^\mathsf{T}K_\sigma K\boldsymbol{u}_i$. Solving these linear comparison inequalities yields the bounds

$$p_i^{(c_2)}(t)\;\le\;p_i(t)\;\le\;p_i^{(c_1)}(t),\qquad P^{(c_2)}(t)\;\le\;P(t)\;\le\;P^{(c_1)}(t),$$

where $p_i^{(c)}(\cdot)$ and $P^{(c)}(\cdot)$ denote the solutions of the ODE/integral equations from Appendix E.2 with the factor $\frac{4}{\pi}$ replaced by $\frac{4c}{\pi}$. Equivalently, defining

$$Q_c(N) \;:=\; \frac{4c\,\gamma_0}{\pi} \int_0^N \frac{f(u)}{\sqrt{P(u)}}\,du,$$

the drift/noise expressions remain valid with $Q(N)$ replaced by $Q_c(N)$, and all proofs carry through verbatim.

**Only multiplicative constants change; scaling exponents and phases do not.** Every appearance of $Q(N)$ in the final formulas enters either through an exponential $e^{-\lambda Q(N)}$ or through a polynomial factor $(M^\mu Q(N))^{-\nu}$. Replacing $Q$ by $Q_c = c\,Q$ only multiplies these terms by constants: $e^{-\lambda cQ}$ converts to $(M^\mu cQ)^{-\nu} = c^{-\nu}(M^\mu Q)^{-\nu}$. Hence the *rates, exponents, and phase boundaries* of the scaling laws are unchanged; only the prefactors are rescaled by fixed constants depending on $c_1(\rho), c_2(\rho) \in [1, \pi/2]$. In particular, all "$\eqsim$" statements (equalities up to absolute constants) remain valid with the same exponents.

### K.3  NOTE ON APPROXIMATION ERROR

Though proof of Paquette et al. (2024) implicitly implies

$$\left\| \boldsymbol{H}^{1/2}\boldsymbol{w}_\perp \right\|^2 \eqsim M^{-2\alpha+\max(0,\,1-2\beta)}.$$

It was not explicitly specified. So we clarify it here.

First,

$$-\frac{1}{2\pi i} \oint_{|z|=\varepsilon} \left\langle (\widehat{\boldsymbol{K}} - z\boldsymbol{I})^{-1},\, (\boldsymbol{H}^{1/2}\boldsymbol{w}^*)^{\otimes 2} \right\rangle dz \eqsim M^{-2\alpha+\max(0,\,1-2\beta)},$$

is directly implied from Proposition H.3 of Paquette et al. (2024). So it is enough to prove the following claim.

**Claim.** Let

$$\widehat{\boldsymbol{K}} = \boldsymbol{H}^{1/2}\boldsymbol{S}^\mathsf{T}\boldsymbol{S}\,\boldsymbol{H}^{1/2}, \qquad \boldsymbol{w}^* = \boldsymbol{S}^\mathsf{T}\boldsymbol{\theta}^* + \boldsymbol{w}_\perp, \qquad \boldsymbol{S}\,\boldsymbol{H}\,\boldsymbol{w}_\perp = \boldsymbol{0}.$$

For a sufficiently small circle $|z| = \varepsilon$ enclosing only the eigenvalue $0$ of $\widehat{\boldsymbol{K}}$,

$$-\frac{1}{2\pi i} \oint_{|z|=\varepsilon} \left\langle (\widehat{\boldsymbol{K}} - z\boldsymbol{I})^{-1},\, (\boldsymbol{H}^{1/2}\boldsymbol{w}^*)^{\otimes 2} \right\rangle dz \;=\; \left\| \boldsymbol{H}^{1/2}\boldsymbol{w}_\perp \right\|^2.$$

**Proof.** By the Riesz projection theorem (Dunford–Riesz functional calculus), for a small circle $|z| = \varepsilon$ enclosing only the eigenvalue $0$ of $\widehat{\boldsymbol{K}}$,

$$\Pi_0 \;:=\; -\frac{1}{2\pi i} \oint_{|z|=\varepsilon} (\widehat{\boldsymbol{K}} - z\boldsymbol{I})^{-1}\,dz$$

is the spectral Riesz projector onto the $0$-eigenspace; since $\widehat{\boldsymbol{K}}$ is Hermitian, $\Pi_0$ is the *orthogonal* projector onto $\ker(\widehat{\boldsymbol{K}})$.

Then we have

$$-\frac{1}{2\pi i} \oint_{|z|=\varepsilon} \left\langle (\widehat{\boldsymbol{K}} - z\boldsymbol{I})^{-1},\, (\boldsymbol{H}^{1/2}\boldsymbol{w}^*)^{\otimes 2} \right\rangle dz = \left\langle \Pi_0,\, (\boldsymbol{H}^{1/2}\boldsymbol{w}^*)^{\otimes 2} \right\rangle = \left\| \Pi_0\,\boldsymbol{H}^{1/2}\boldsymbol{w}^* \right\|_2^2.$$

Since $\ker(\widehat{\boldsymbol{K}}) = \{\,\boldsymbol{x} : \boldsymbol{S}\,\boldsymbol{H}^{1/2}\boldsymbol{x} = \boldsymbol{0}\,\} = \big(\mathrm{Im}(\boldsymbol{H}^{1/2}\boldsymbol{S}^\mathsf{T})\big)^\perp$, We have the orthogonal decomposition

$$\boldsymbol{H}^{1/2}\boldsymbol{w}^* = \underbrace{\boldsymbol{H}^{1/2}\boldsymbol{S}^\mathsf{T}\boldsymbol{\theta}^*}_{\in\,\mathrm{Im}(\boldsymbol{H}^{1/2}\boldsymbol{S}^\mathsf{T})} + \underbrace{\boldsymbol{H}^{1/2}\boldsymbol{w}_\perp}_{\in\,(\mathrm{Im}(\boldsymbol{H}^{1/2}\boldsymbol{S}^\mathsf{T}))^\perp},$$

where the second membership uses $\boldsymbol{S}\,\boldsymbol{H}\,\boldsymbol{w}_\perp = \boldsymbol{0}$. Hence $\Pi_0\,\boldsymbol{H}^{1/2}\boldsymbol{w}^* = \boldsymbol{H}^{1/2}\boldsymbol{w}_\perp$, and therefore

$$\left\langle \Pi_0,\, (\boldsymbol{H}^{1/2}\boldsymbol{w}^*)^{\otimes 2} \right\rangle = \left\| \boldsymbol{H}^{1/2}\boldsymbol{w}_\perp \right\|^2. \qquad \square$$

## K.4 PROOF OF MATRIX INEQUALITY FOR $\operatorname{diag}(\boldsymbol{SHS}^\top)^{-1/2}$

We will prove the inequality in the following form in this section.

$$\boxed{c_1\, M^{\min(0.5,\alpha)}\, I \;\preceq\; \operatorname{diag}(\boldsymbol{SHS}^\top)^{-1/2} \;\preceq\; c_2\, M^{\min(0.5,\alpha)}\, I}$$

**Setup.** Let $\boldsymbol{S} \in \mathbb{R}^{M \times d}$ have i.i.d. entries $S_{ij} \sim \mathcal{N}(0, 1/M)$, and let

$$\boldsymbol{H} = \operatorname{diag}\big(1^{-2\alpha},\, 2^{-2\alpha},\, \ldots,\, d^{-2\alpha}\big), \qquad \alpha > 0.$$

Then, for each $i \in \{1, \ldots, M\}$,

$$\big[\operatorname{diag}(\boldsymbol{SHS}^\top)\big]_{ii} = \sum_{j=1}^{d} H_{jj} S_{ij}^2 = \frac{1}{M}\sum_{j=1}^{d} j^{-2\alpha}\chi_j^2,$$

where $\chi_1^2, \ldots, \chi_d^2$ are i.i.d. $\chi^2(1)$.

*Remark* 5 (Rough intuition for what we will prove).

$$\big[\operatorname{diag}(\boldsymbol{SHS}^\top)\big]_{ii} = \frac{1}{M}\sum_{j=1}^{d} j^{-2\alpha}\chi_j^2 \;\approx\; \begin{cases} M^{-1}, & \alpha > \frac{1}{2}, \\ M^{-1}d^{1-2\alpha} \approx M^{-2\alpha}, & \alpha \leq \frac{1}{2} \text{ with } d \approx M, \end{cases}$$

So, we want to obtain $\operatorname{diag}(\boldsymbol{SHS}^\top)^{-1/2} \approx M^{\min(0.5,\alpha)}I$.

Define

$$S_d(\alpha) := \sum_{j=1}^{d} j^{-2\alpha}\chi_j^2 \quad \Longrightarrow \quad \big[\operatorname{diag}(\boldsymbol{SHS}^\top)\big]_{ii} = \frac{1}{M}S_d(\alpha).$$

Hence, any high–probability upper/lower bounds on $S_d(\alpha)$ translate into corresponding bounds on $\operatorname{diag}(\boldsymbol{SHS}^\top)^{-1/2}$ via

$$\frac{1}{M}S_d(\alpha) \leq U \quad \Longrightarrow \quad \big[\operatorname{diag}(\boldsymbol{SHS}^\top)\big]^{-1/2} \succeq \sqrt{\frac{M}{U}}\,I,$$

$$\frac{1}{M}S_d(\alpha) \geq L \quad \Longrightarrow \quad \big[\operatorname{diag}(\boldsymbol{SHS}^\top)\big]^{-1/2} \preceq \sqrt{\frac{M}{L}}\,I.$$

We consider two regimes and then unify them through $M^{\min(0.5,\alpha)}$.

REGIME I: $\alpha > \frac{1}{2}$ (SUMMABLE WEIGHTS)

In this regime, $\sum_{j=1}^{\infty} j^{-2\alpha} = \zeta(2\alpha) < \infty$. Write $X_j := j^{-2\alpha}(\chi_j^2 - 1)$, so that

$$S_d(\alpha) = \mathbb{E}[S_d(\alpha)] + \sum_{j=1}^{d} X_j, \qquad \mathbb{E}[S_d(\alpha)] = \sum_{j=1}^{d} j^{-2\alpha} \leq \zeta(2\alpha).$$

Moreover, $\operatorname{Var}(S_d(\alpha)) = 2\sum_{j=1}^{d} j^{-4\alpha} \leq 2\zeta(4\alpha)$.

**Upper tail (to lower–bound $\operatorname{diag}^{-1/2}$).** For $\lambda = \frac{1}{2}$,

$$\mathbb{E}\big[e^{\lambda X_j}\big] = e^{-\lambda j^{-2\alpha}}(1 - 2\lambda j^{-2\alpha})^{-1/2} \;\leq\; \exp\!\Big(\tfrac{1}{2}j^{-4\alpha}\Big),$$

hence

$$\mathbb{E}\Big[e^{\frac{1}{2}(S_d(\alpha) - \mathbb{E}S_d(\alpha))}\Big] \leq \exp\!\Big(\tfrac{1}{2}\sum_{j=1}^{d} j^{-4\alpha}\Big) \leq \exp\!\Big(\tfrac{1}{2}\zeta(4\alpha)\Big).$$

By Markov and a union bound over the $M$ diagonal entries, setting the per–entry failure probability to $\delta_0 := \delta_{\text{total}}/M$,

$$\Pr\!\Big(S_d(\alpha) \leq \zeta(2\alpha) + \zeta(4\alpha) + 2\log\frac{M}{\delta_{\text{total}}}\Big) \;\geq\; 1 - \delta_{\text{total}}.$$

Therefore, with probability at least $1 - \delta_{\text{total}}$,

$$\text{diag}(\boldsymbol{SHS}^{\mathsf{T}})^{-1/2} \;\succeq\; \frac{\sqrt{M}}{\left(\zeta(2\alpha) + \zeta(4\alpha) + 2\log\frac{M}{\delta_{\text{total}}}\right)^{1/2}}\, I.$$

**Lower tail (to upper–bound $\text{diag}^{-1/2}$).** A Chernoff bound on the lower tail of $S_d(\alpha)$ (via the mgf of $e^{-t\,j^{-2\alpha}\chi^2}$) gives, for any $\delta \in (0,1)$, the existence of a constant

$$c_{\downarrow}(\alpha) \;=\; \left(\tfrac{2\alpha-1}{2}\right)^{2\alpha-1} \big/\, 2^{\,2\alpha-1}$$

such that

$$\Pr\!\Big(S_d(\alpha) \;\geq\; c_{\downarrow}(\alpha)\,\big(\log(1/\delta)\big)^{-(2\alpha-1)}\Big) \;\geq\; 1 - \delta.$$

With $\delta = \delta_0 = \delta_{\text{total}}/M$ and a union bound over the $M$ rows, with probability at least $1 - \delta_{\text{total}}$,

$$\text{diag}(\boldsymbol{SHS}^{\mathsf{T}})^{-1/2} \;\preceq\; \frac{\sqrt{M}}{\left(c_{\downarrow}(\alpha)\right)^{1/2}} \left(\log\frac{M}{\delta_{\text{total}}}\right)^{\frac{2\alpha-1}{2}} I.$$

**Conclusion for $\alpha > \tfrac{1}{2}$.** Combining the two displays,

$$\boxed{\;\frac{\sqrt{M}}{\left(\zeta(2\alpha) + \zeta(4\alpha) + 2\log\frac{M}{\delta_{\text{total}}}\right)^{1/2}}\, I \;\preceq\; \text{diag}(\boldsymbol{SHS}^{\mathsf{T}})^{-1/2} \;\preceq\; \frac{\sqrt{M}}{\left(c_{\downarrow}(\alpha)\right)^{1/2}} \left(\log\frac{M}{\delta_{\text{total}}}\right)^{\frac{2\alpha-1}{2}} I\;} \quad (\alpha > \tfrac{1}{2}).$$

REGIME II: $\alpha \leq \tfrac{1}{2}$ (DIVERGING WEIGHTS)

Assume $d \geq r\,M$ for some fixed $r > 1$ (as in our setup). Then

$$\mathbb{E}[S_d(\alpha)] = \sum_{j=1}^{d} j^{-2\alpha} \quad \text{satisfies} \quad \frac{(d+1)^{1-2\alpha} - 1}{1 - 2\alpha} \;\leq\; \mathbb{E}[S_d(\alpha)] \;\leq\; 1 + \frac{d^{1-2\alpha} - 1}{1 - 2\alpha}.$$

Hence $\mathbb{E}[S_d(\alpha)] \asymp d^{1-2\alpha}$. Moreover,

$$\text{Var}\big(S_d(\alpha)\big) = 2\sum_{j=1}^{d} j^{-4\alpha} \begin{cases} = O(1), & \alpha > \tfrac{1}{4}, \\ = \Theta\big(d^{1-4\alpha}\big), & \alpha < \tfrac{1}{4}, \end{cases}$$

so in all cases $\sqrt{\text{Var}(S_d(\alpha))} = o\big(\mathbb{E}[S_d(\alpha)]\big)$ as $d \to \infty$. Thus, by Bernstein and a union bound over the $M$ rows, for all sufficiently large $M$ we get, with probability at least $1 - \delta_{\text{total}}$,

$$\frac{1}{2}\,\mathbb{E}[S_d(\alpha)] \;\leq\; S_d(\alpha) \;\leq\; \frac{3}{2}\,\mathbb{E}[S_d(\alpha)].$$

Using $d \geq rM$ and the integral bounds for $\mathbb{E}[S_d(\alpha)]$,

$$\frac{(rM)^{1-2\alpha} - 1}{2(1 - 2\alpha)} \;\leq\; S_d(\alpha) \;\leq\; \frac{3}{2}\Big(1 + \frac{(rM)^{1-2\alpha} - 1}{1 - 2\alpha}\Big).$$

Dividing by $M$ and inverting the square–root yields constants

$$C_L(\alpha, r) \;:=\; \left(\frac{3}{1 - 2\alpha}\, r^{1-2\alpha}\right)^{-1/2}, \qquad C_U(\alpha, r) \;:=\; \left(\frac{1}{2(1 - 2\alpha)}\, r^{1-2\alpha}\right)^{-1/2},$$

such that, with probability at least $1 - \delta_{\text{total}}$,

$$\boxed{\;C_L(\alpha, r)\, M^{\alpha}\, I \;\preceq\; \text{diag}(\boldsymbol{SHS}^{\mathsf{T}})^{-1/2} \;\preceq\; C_U(\alpha, r)\, M^{\alpha}\, I\;} \quad (\alpha \leq \tfrac{1}{2}).$$

UNIFIED STATEMENT

Combining Regimes I and II, there exist positive constants $c_1(\alpha, r, \delta_{\text{total}})$ and $c_2(\alpha, r, \delta_{\text{total}})$ such that, with probability at least $1 - \delta_{\text{total}}$,

$$\boxed{c_1(\alpha, r, \delta_{\text{total}}) \, M^{\min(0.5, \alpha)} \, I \; \preceq \; \operatorname{diag}(\boldsymbol{SHS}^\mathsf{T})^{-1/2} \; \preceq \; c_2(\alpha, r, \delta_{\text{total}}) \, M^{\min(0.5, \alpha)} \, I}$$

with the following explicit choices:

- If $\alpha > \frac{1}{2}$:

$$c_1(\alpha, \cdot, \delta_{\text{total}}) = \left(\zeta(2\alpha) + \zeta(4\alpha) + 2\log \frac{M}{\delta_{\text{total}}}\right)^{-1/2}, \qquad c_2(\alpha, \cdot, \delta_{\text{total}}) = \left(c_\downarrow(\alpha)\right)^{-1/2} \left(\log \frac{M}{\delta_{\text{total}}}\right)^{\frac{2\alpha-1}{2}},$$

  where one admissible choice is $c_\downarrow(\alpha) = \left(\frac{2\alpha-1}{2}\right)^{2\alpha-1} / 2^{\,2\alpha-1}$.

- If $\alpha \le \frac{1}{2}$ and $d \ge rM$:

$$c_1(\alpha, r, \cdot) = C_L(\alpha, r), \qquad c_2(\alpha, r, \cdot) = C_U(\alpha, r),$$

  with $C_L, C_U$ as defined above.

## L    ANALYSIS FOR THE CASE WITH LABEL NOISE

For the case with label noise, only Phase Ia is solved for SGD by Lin et al. (2024). So we will focus on the Phase Ia where $\alpha > 0.5$ and $\beta < 0.5$ holds.

Now we set an assumption for label noise. For selected data $x$, we assume that label $y$ satisfies

$$y = \langle \boldsymbol{x}, \boldsymbol{w}^* \rangle + \epsilon$$

where $\epsilon$ is a label noise with mean 0 and variance $\sigma^2$ satisfying $\epsilon \perp\!\!\!\perp \boldsymbol{x}$.

Note that for the case with label noise $L(\boldsymbol{\theta}) = \mathbb{E}_{\boldsymbol{x}}\left[(\langle \boldsymbol{Sx}, \boldsymbol{\theta}\rangle - y)^2\right]$ and $L(\boldsymbol{\theta}) = \|\boldsymbol{H}^{1/2}(\boldsymbol{S}^\mathsf{T}\boldsymbol{\theta} - \boldsymbol{w}^*)\|^2$ are not equivalent.

So in this section, we will use a notation $L_{\text{true}}(\boldsymbol{\theta}) = \mathbb{E}_{\boldsymbol{x}}\left[(\langle \boldsymbol{Sx}, \boldsymbol{\theta}\rangle - y)^2\right]$.

Then $L_{\text{true}}(\boldsymbol{\theta}) = \|\boldsymbol{H}^{1/2}(\boldsymbol{S}^\mathsf{T}\boldsymbol{\theta} - \boldsymbol{w}^*)\|^2 + \sigma^2 = L(\boldsymbol{\theta}) + \sigma^2$

Here $\sigma^2$ is the irreducible risk. Lin et al. (2024) discussed compute-optimal scaling for $L(\boldsymbol{\theta}) = L_{\text{true}}(\boldsymbol{\theta}) - \sigma^2$. Therefore, we will also discuss compute-optimal scaling for $L(\boldsymbol{\theta}) = L_{\text{true}}(\boldsymbol{\theta}) - \sigma^2$.

Also in this section, we let $R(M, N, \gamma_0)$ as the $L_{\text{true}}(\boldsymbol{\theta}_N)$ under learning rate $\gamma_0$ and fixed model size $M$. We will discuss the scaling law of $R(M, N, \gamma_0) - \sigma^2$.

### L.1    DERIVING ODE AND INTEGRAL EQUATION

For a quadratic function $q$, by Taylor's theorem, we have

$$\mathbb{E}[q(\boldsymbol{\theta}_{k+1}) - q(\boldsymbol{\theta}_k) \,|\, \mathcal{F}_k] = \mathbb{E}[\langle \nabla q(\boldsymbol{\theta}_k), \boldsymbol{\theta}_{k+1} - \boldsymbol{\theta}_k\rangle \,|\, \mathcal{F}_k] + \tfrac{1}{2}\,\mathbb{E}\left[\langle \nabla^2 q, (\boldsymbol{\theta}_{k+1} - \boldsymbol{\theta}_k)^{\otimes 2}\rangle \,\big|\, \mathcal{F}_k\right],$$

where $\mathcal{F}_k = \sigma(\boldsymbol{S}, \boldsymbol{\theta}_0, \dots, \boldsymbol{\theta}_k)$. Since

$$\boldsymbol{\theta}_{k+1} - \boldsymbol{\theta}_k = -\gamma_k \, \text{sign}(\langle \boldsymbol{Sx}_k, \boldsymbol{\theta}_k\rangle - y_k) \, \text{sign}(\boldsymbol{Sx}_k),$$

We can expand the two terms using sign-Gaussian identities. We let label noise for the same $(\boldsymbol{x}_k, y_k)$ as $\epsilon_k$ and $y_k = \langle \boldsymbol{x}_k, w^*\rangle + \epsilon_k$ holds.

**Gradient term.**

$$\begin{aligned}
&\mathbb{E}[\langle \nabla q(\boldsymbol{\theta}_k), \boldsymbol{\theta}_{k+1} - \boldsymbol{\theta}_k\rangle \,|\, \mathcal{F}_k] \\
&= -\gamma_k \left\langle \nabla q(\boldsymbol{\theta}_k), \mathbb{E}\left[\text{sign}(\boldsymbol{Sx}_k)\,\text{sign}\left(\langle \boldsymbol{x}_k, \boldsymbol{S}^\mathsf{T}\boldsymbol{\theta}_k - \boldsymbol{w}^*\rangle - \epsilon_k\right)\,|\, \mathcal{F}_k\right] \right\rangle \\
&= -\gamma_k \left\langle \nabla q(\boldsymbol{\theta}_k), \frac{2}{\pi} \arcsin\left(\frac{\text{diag}(\boldsymbol{SHS}^\mathsf{T})^{-1/2}\,\boldsymbol{SH}(\boldsymbol{S}^\mathsf{T}\boldsymbol{\theta}_k - \boldsymbol{w}^*)}{\sqrt{(\boldsymbol{S}^\mathsf{T}\boldsymbol{\theta}_k - \boldsymbol{w}^*)^\mathsf{T}\boldsymbol{H}(\boldsymbol{S}^\mathsf{T}\boldsymbol{\theta}_k - \boldsymbol{w}^*) + \sigma^2}}\right) \right\rangle \\
&= -\gamma_k \left\langle \nabla q(\boldsymbol{\theta}_k), \frac{2}{\pi} \arcsin\left(\frac{\text{diag}(\boldsymbol{K})^{-1/2}\,\boldsymbol{K}(\boldsymbol{\theta}_k - \boldsymbol{\theta}^*)}{\sqrt{\|\boldsymbol{H}^{1/2}(\boldsymbol{S}^\mathsf{T}\boldsymbol{\theta}_k - \boldsymbol{w}^*)\|^2 + \sigma^2}}\right) \right\rangle,
\end{aligned}$$

where $\boldsymbol{K} = \boldsymbol{SHS}^\mathsf{T}$.

**Quadratic term.**

$$\begin{aligned}
&\mathbb{E}\left[\langle \nabla^2 q, (\boldsymbol{\theta}_{k+1} - \boldsymbol{\theta}_k)^{\otimes 2}\rangle \,\big|\, \mathcal{F}_k\right] \\
&= \gamma_k^2 \left\langle \nabla^2 q, \mathbb{E}\left[\left(\text{sign}(\boldsymbol{Sx}_k)\,\text{sign}(\langle \boldsymbol{x}_k, \boldsymbol{S}^\mathsf{T}\boldsymbol{\theta}_k - \boldsymbol{w}^*\rangle - \epsilon_k)\right)^{\otimes 2} \,\big|\, \mathcal{F}_k\right] \right\rangle \\
&= \gamma_k^2 \left\langle \nabla^2 q, \mathbb{E}\left[(\text{sign}(\boldsymbol{Sx}_k))^{\otimes 2} \,\big|\, \mathcal{F}_k\right] \right\rangle \\
&= \gamma_k^2 \left\langle \nabla^2 q, \frac{2}{\pi} \arcsin\left(\text{diag}(\boldsymbol{SHS}^\mathsf{T})^{-1/2}\,\boldsymbol{SHS}^\mathsf{T}\,\text{diag}(\boldsymbol{SHS}^\mathsf{T})^{-1/2}\right) \right\rangle \\
&= \gamma_k^2 \left\langle \nabla^2 q, \frac{2}{\pi} \arcsin\left(\text{diag}(\boldsymbol{K})^{-1/2}\,\boldsymbol{K}\,\text{diag}(\boldsymbol{K})^{-1/2}\right) \right\rangle.
\end{aligned}$$

**One-step update formula.** Substituting the gradient and quadratic terms yields the desired one-step update formula for signSGD.

$$\mathbb{E}[q(\boldsymbol{\theta}_{k+1}) - q(\boldsymbol{\theta}_k) \,|\, \mathcal{F}_k] = -\frac{2\gamma_k}{\pi} \left\langle \nabla q(\boldsymbol{\theta}_k)\,, \arcsin\left(\frac{\overline{\boldsymbol{K}}\,(\boldsymbol{\theta}_k - \boldsymbol{\theta}^*)}{\sqrt{L(k) + \sigma^2}}\right) \right\rangle + \frac{\gamma_k^2}{\pi} \left\langle \nabla^2 q\,, \boldsymbol{K}_\sigma \right\rangle.$$

By the same procedure as the noiseless case, while $\sqrt{L(k)}$ in the denominator is replaced by $\sqrt{L(k) + \sigma^2}$, we get the following ODE, where $P(t) = L(t/\gamma_0)$ and $p_i(t) = r_i(t/\gamma_0)$.

$$\frac{dp_i}{dt} = -\frac{4}{\pi\sqrt{P(t) + \sigma^2}} \lambda_i(\overline{\boldsymbol{K}})\, f(t/\gamma_0)\, p_i(t) + \frac{2f(t/\gamma_0)^2\gamma_0}{\pi} V_i. \tag{140}$$

**Integral equation.** Also, by the same procedure as the noiseless case, while $\sqrt{L(u)}$ in the denominator is replaced by $\sqrt{L(u) + \sigma^2}$, we get the following integral equation.

$$L(N) = \|\boldsymbol{H}^{1/2}\boldsymbol{w}_\perp\|^2 + \sum_{i=1}^{M} r_i(0)\, e^{-\frac{4\lambda_i\gamma_0}{\pi} \int_0^N \frac{f(u)}{\sqrt{L(u)+\sigma^2}}\, du} + \frac{2\gamma_0^2}{\pi} \sum_{i=1}^{M} V_i \int_0^N e^{-\frac{4\lambda_i\gamma_0}{\pi} \int_z^N \frac{f(u)}{\sqrt{L(u)+\sigma^2}}\, du} f(z)^2\, dz. \tag{141}$$

By using the same drift/approximation-term transformation as the noiseless case, we get

$$L(N) \,\asymp\, \underbrace{M^{-2\alpha-2\beta+1}}_{\textbf{approx}} + \underbrace{\left(M^{0.5}\, Q(N)\right)^{-\frac{2\alpha+2\beta-1}{2\alpha}}}_{\textbf{drift}} \tag{142}$$

$$+ \underbrace{\frac{2\gamma_0^2}{\pi} \sum_{i=1}^{M} V_i \int_0^N \exp\left(-\frac{4\gamma_0}{\pi}\lambda_i(\overline{K})\int_z^N \frac{du}{\sqrt{L(u)+\sigma^2}}\right) dz}_{\textbf{noise}}. \tag{143}$$

where $f(z) \equiv 1$ (which means constant learning rate) and

$$Q(N) = \frac{4\gamma_0}{\pi} \int_0^N \frac{du}{\sqrt{L(u)+\sigma^2}}.$$

## L.2 Early Stage for a Noisy Label

Similar to the noiseless case, we first solve for the early stage. Here we have to solve the following equation

$$L(N) \,\asymp\, \left(M^{0.5}\, Q(N)\right)^{-\frac{2\alpha+2\beta-1}{2\alpha}}$$

It can be converted to

$$L(N)^{-\frac{2\alpha}{2\alpha+2\beta-1}} \,\asymp\, M^{0.5} \gamma_0 \int_0^N \frac{du}{\sqrt{L(u)+\sigma^2}}. \tag{144}$$

Replacing $\asymp$ by equality in (144) and differentiating with respect to $N$ (viewed as a continuous time variable $t$) yields

$$-\frac{2\alpha}{2\alpha+2\beta-1} L(t)^{-\frac{2\alpha}{2\alpha+2\beta-1}-1} L'(t) \,\asymp\, M^{0.5} \gamma_0 \frac{1}{\sqrt{L(t)+\sigma^2}}. \tag{145}$$

Equivalently,

$$L(t)^{A-1} L'(t) \,\asymp\, M^{0.5} \gamma_0 \frac{1}{\sqrt{L(t)+\sigma^2}}, \qquad A := -\frac{2\alpha}{2\alpha+2\beta-1}. \tag{146}$$

For any $\sigma > 0$ and $x \geq 0$ we have the elementary bounds

$$\frac{1}{\sqrt{2}} \min\!\left(x^{-1/2},\, \sigma^{-1}\right) \leq \frac{1}{\sqrt{x+\sigma^2}} \leq \min\!\left(x^{-1/2},\, \sigma^{-1}\right). \tag{147}$$

Indeed, if $x \geq \sigma^2$ then $x \leq x + \sigma^2 \leq 2x$, so

$$\frac{1}{\sqrt{2}} x^{-1/2} \leq \frac{1}{\sqrt{x + \sigma^2}} \leq x^{-1/2},$$

whereas if $0 \leq x \leq \sigma^2$ then $\sigma^2 \leq x + \sigma^2 \leq 2\sigma^2$, so

$$\frac{1}{\sqrt{2}} \sigma^{-1} \leq \frac{1}{\sqrt{x + \sigma^2}} \leq \sigma^{-1}.$$

Combining the two cases yields (147). Applying (147) with $x = L(t)$ in (146), we obtain

$$L(t)^{A-1} L'(t) \; \eqsim \; M^{0.5} \gamma_0 \begin{cases} L(t)^{-1/2}, & L(t) \geq \sigma^2, \\ \sigma^{-1}, & L(t) \leq \sigma^2. \end{cases} \tag{148}$$

This naturally splits the dynamics into a *large-L* regime $L \geq \sigma^2$ and a *small-L* regime $L \leq \sigma^2$.

Suppose $L(t) \geq \sigma^2$. Then from (148) we have

$$L(t)^{A-1} L'(t) \; \eqsim \; M^{0.5} \gamma_0 L(t)^{-1/2},$$

or equivalently

$$L'(t) \; \eqsim \; M^{0.5} \gamma_0 L(t)^{1-A-\frac{1}{2}}. \tag{149}$$

Define

$$\zeta := 1 - A - \frac{1}{2} = -A + \frac{1}{2} = \frac{2\alpha}{2\alpha + 2\beta - 1} + \frac{1}{2}.$$

The assumptions $\alpha > 0.5$, $\beta < 0.5$, and $\alpha + \beta > 0.5$ imply $\zeta > 1$. Then (149) takes the canonical form

$$\frac{dL}{dt} \; \eqsim \; M^{0.5} \gamma_0 L^{\zeta}.$$

Separating variables and integrating gives

$$\int L^{-\zeta} \, dL \; \eqsim \; M^{0.5} \gamma_0 \int dt \quad \implies \quad L(t)^{-(\zeta-1)} \; \eqsim \; M^{0.5} \gamma_0 t,$$

where we have absorbed additive constants into the implicit comparison. Thus, in the large-$L$ regime,

$$L(t) \; \eqsim \; \left( M^{0.5} \gamma_0 t \right)^{-1/(\zeta-1)}. \tag{150}$$

Writing

$$p := \frac{1}{\zeta - 1} = \frac{2(2\alpha + 2\beta - 1)}{2\alpha + 1 - 2\beta},$$

we recover exactly the original early-phase exponent:

$$L(t) \; \eqsim \; \left( M^{0.5} \gamma_0 t \right)^{-p}, \qquad L(t) \geq \sigma^2. \tag{151}$$

In particular, the presence of $\sqrt{L + \sigma^2}$ in the denominator does not change the scaling exponent $p$ in the regime where $L$ is larger than the noise floor $\sigma^2$; it only affects the constant factors hidden in $\eqsim$.

Now suppose $L(t) \leq \sigma^2$ and $t$ is sufficiently large so that the small-$L$ regime dominates. From (148) we obtain

$$L(t)^{A-1} L'(t) \; \eqsim \; M^{0.5} \gamma_0 \sigma^{-1}.$$

Observing that $\frac{d}{dt} L(t)^A = A L(t)^{A-1} L'(t)$, we can rewrite this as

$$\frac{d}{dt} L(t)^A \; \eqsim \; M^{0.5} \gamma_0 \sigma^{-1}.$$

Integrating in $t$ and absorbing additive constants into $\eqsim$ yields

$$L(t)^A \; \eqsim \; M^{0.5} \gamma_0 \sigma^{-1} t.$$

Since $A < 0$, we invert this relation to obtain

$$L(t) \approx \left(M^{0.5}\,\gamma_0\,t/\sigma\right)^{1/A} = \left(M^{0.5}\,\gamma_0\,t/\sigma\right)^{-p'}, \qquad p' := -\frac{1}{A} = \frac{2\alpha + 2\beta - 1}{2\alpha}. \tag{152}$$

Thus, in the small-$L$ (noise-dominated) regime,

$$L(t) \approx \left(M^{0.5}\,\gamma_0\,t/\sigma\right)^{-p'}, \qquad L(t) \leq \sigma^2. \tag{153}$$

Combining (151) and (153), we obtain the following formula for the early-stage.

$$L(t) \approx \left(M^{0.5}\,\gamma_0\,t\right)^{-p} + \left(M^{0.5}\,\gamma_0\,t/\sigma\right)^{-p'}, \qquad p = \frac{2(2\alpha + 2\beta - 1)}{2\alpha + 1 - 2\beta}, \quad p' = \frac{2\alpha + 2\beta - 1}{2\alpha}. \tag{154}$$

### L.3 LIMIT STAGE FOR A NOISY LABEL

By the same procedure as Appendix E.3.2, we get an equation

$$L_\infty = \frac{\gamma_0 \pi}{4}\,\mathrm{Tr}\big(\mathrm{diag}(\boldsymbol{K})^{1/2}\big)\sqrt{L_\infty + \sigma^2} + \|\boldsymbol{H}^{1/2}\boldsymbol{w}_\perp\|^2.$$

Solving the quadratic equation, we get

$$L_\infty \approx \gamma_0^2\,\mathrm{Tr}\big(\mathrm{diag}(\boldsymbol{K})^{1/2}\big)^2 + \sigma\gamma_0\,\mathrm{Tr}\big(\mathrm{diag}(\boldsymbol{K})^{1/2}\big) + \|\boldsymbol{H}^{1/2}\boldsymbol{w}_\perp\|^2$$

Under our setup,

$$\mathrm{Tr}\big(\mathrm{diag}(\boldsymbol{K})^{1/2}\big) = \sum_{i=1}^{M}\sqrt{(\boldsymbol{SHS}^\mathsf{T})_{ii}} \approx M \cdot \sqrt{\frac{1}{M}M^{\max(1-2\alpha,0)}} \approx M^{1-\min(\alpha,\,0.5)}.$$

By the results from Paquette et al. (2024); Lin et al. (2024), and note in Appendix K.3,

$$\|\boldsymbol{H}^{1/2}\boldsymbol{w}_\perp\|^2 \approx M^{-2\alpha+\max(0,\,1-2\beta)}.$$

Hence

$$L_\infty \approx \gamma_0^2 M + \sigma\gamma_0\sqrt{M} + M^{-(2\alpha+2\beta-1)}$$

### L.4 EVALUATING COMPUTE-OPTIMAL SCALING

Combining the early stage and the limit stage, we get

$$R(M, N, \gamma_0) - \sigma^2 \approx \left(M^{1/2}N\gamma_0\right)^{-\frac{2(2\alpha+2\beta-1)}{2\alpha+1-2\beta}} + \left(M^{1/2}N\gamma_0/\sigma\right)^{-\frac{2\alpha+2\beta-1}{2\alpha}} + \gamma_0^2 M + \sigma\gamma_0\sqrt{M} + M^{-(2\alpha+2\beta-1)}.$$

Note that we use $R$ instead of $L$ when we are writing the loss as a three-variable function.

We let $\gamma_0 = M^{-e}$. Also assume $\sigma \approx 1$ (this covers values such as $\sigma = 1, 0.2, 0.01$, etc.).

Compute-optimal occurs when the three terms balance. For the loss formula in this section, compute-optimal occurs when $\left(M^{1/2}N\gamma_0/\sigma\right)^{-\frac{2\alpha+2\beta-1}{2\alpha}}$ and $\sigma\gamma_0\sqrt{M}$ and $M^{-(2\alpha+2\beta-1)}$ balances. Solving $\sigma\gamma_0\sqrt{M} = M^{-(2\alpha+2\beta-1)}$, we get $\gamma_0^\star = M^{-(2\alpha+2\beta-0.5)}$. Solving $\left(M^{1/2}N\gamma_0/\sigma\right)^{-\frac{2\alpha+2\beta-1}{2\alpha}} = M^{-(2\alpha+2\beta-1)}$, we get $N = M^{4\alpha+2\beta-1}$ and it leads to $\mathfrak{f} = MN = M^{4\alpha+2\beta}$.

So finally we get

$$M^\star = \mathfrak{f}^{1/(4\alpha+2\beta)}, \qquad R(M^\star, \mathfrak{f}/M^\star, \gamma_0^\star) - \sigma^2 \approx \mathfrak{f}^{-(2\alpha+2\beta-1)/(4\alpha+2\beta)}. \tag{155}$$

Figure 26 shows that exponents in the (155) and measured compute-optimal loss slope and optimal model size slope (in log-log plot) for the case with the label noise match well. In the experiments, we used $\sigma = 0.1$.

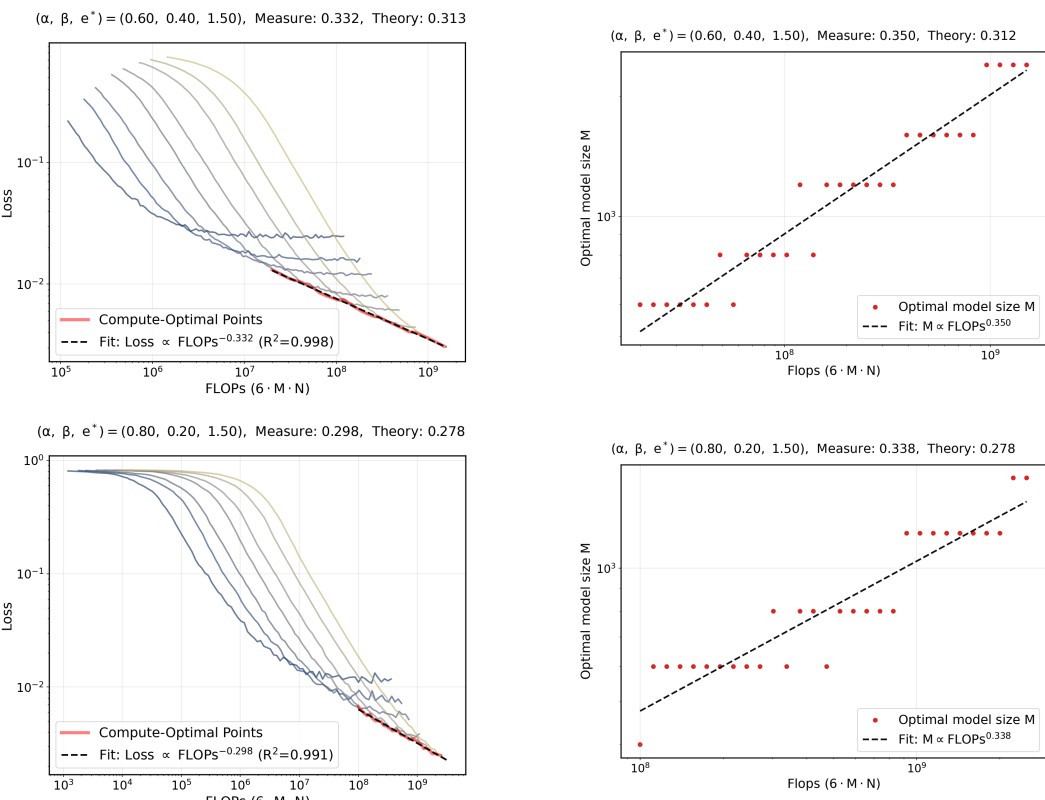

Figure 26: **Measure of compute-optimal loss slope and optimal model size slope for the case with label noise.** We validate the exponent of $R\left(M^{\star}, \frac{\mathfrak{f}}{M^{\star}}, \gamma_0^{\star}\right)$ and $M^{\star}$ with respect to $\mathfrak{f}$ for the case with label noise. The left plot shows the compute-optimal loss with respect to FLOPS $6MN$. The right plot shows the optimal model size with respect to FLOPS $6MN$. Note that we evaluate the region with big FLOPS, as we aim to evaluate asymptotic behavior.

