# OpenReview forum: "Scaling Laws of SignSGD in Linear Regression: When Does It Outperform SGD?"
_ICLR.cc/2026/Conference — ICLR 2026 Poster_

### Official Review · Reviewer_5nJx · 2025-10-23

**Soundness:** 3
**Presentation:** 3
**Contribution:** 3
**Rating:** 4
**Confidence:** 4

**Summary:**

This paper studies the scaling law of SignSGD in linear regression. It shows that the compute-optimal scaling of SignSGD is better than SGD in certain regimes, and the stable-decay schedule can further benefit the scaling.

**Strengths:**

1. A clear benefit of SignSGD compared with SGD is shown.
2. The authors made an important attempt towards understanding the learning rate schedule by studying the stable-decay schedule.
3. Presentation is clear overall.

**Weaknesses:**

1. The assumption of $H$ being diagonal and $(w_*)\_i$ being $i^{-\beta}$ can be problematic for SignSGD. We omit the projection $S$ for convenience. In existing works about SGD, assumptions are usually about the eigenvalue spectrum of $H$ as well as the inner product of $w_*$ with the eigenvectors of $H$. Meanwhile, we can assume that $H$ is diagonal to simplify analysis because in SGD, we can apply rotations to $x$ and $w$: If $H$ is not diagonal, then assume that $H=UDU^\top$ where $U$ is orthogonal and $D$ is diagonal. We can then apply the mapping $x'=U^\top x$ and $w'=U^\top x$, then the SGD update is
$$
w'\_{t+1}=U^\top w_{t+1}=U^\top(w_t-\gamma_tg_t)=w_t'-\gamma_tU^\top g_t.
$$
Since $g_t=x_t(x_t^\top w_t-y_t)$, we have $U^\top g_t=x_t'((x_t')^\top w_t'-y_t)\coloneqq g'_t$, indicating that the SGD update does not change under rotation. However, this property does not hold for SignSGD because $U^\top\mathrm{sign}(g_t)\neq\mathrm{sign}(g'_t)$. Hence, we cannot apply rotations to $x$ and $w$ to make the covariance matrix of $x$ diagonal. This is my major concern because intuitively SignSGD can perform better if the model has some "coordinate-wise" structure, which is the case when $H$ is diagonal. I will raise my score if a good explanation is made.
2. In terms of presentation, it can be beneficial to compare the drift and noise terms of signSGD and SGD in a table (with a comparison in each area of Figure 2). It can also be beneficial if the stable-decay structure can be introduced in greater detail, possibly in a separate section.
3. The stable-decay schedule can be restrictive. It captures the case where the learning rate decays polynomially in the iteration number, but does not include the linear decay schedule (and the similar cosine schedule, both decaying to 0 after a finite number of iterations).
4. (Minor) Given the existing results in Xiao et al. (2024), the technical contribution is minor. However, the insights of this paper is fruitful.

Xiao et al. Exact Risk Curves of signSGD in High-Dimensions: Quantifying Preconditioning and Noise-Compression Effects. 2024.

**Questions:**

1. Does SignSGD improve the exponent in area Ic? Ic is adjacent to IV, and it intersects with Ac and Ad for SignSGD. It seems that SGD and SignSGD performs differently at least in part of Ic.
2. Does SGD benefit from stable-decay scheduling? If so, it can be unfair to compare singSGD **with** stable-decay scheduling against SGD **without** such scheduling.

---

> ### Author Response · Authors · 2025-11-23
> **Response to Reviewer 5nJx (1/2)**
>
> We thank the reviewer for the thoughtful comments and helpful feedback.
> ### **W1. Extendability to the General Covariance Matrix $H$**
> Thank you for bringing this point up. We should have been clearer on this point, but assuming that $H$ is diagonal is in fact without loss of generality. Here, the Gaussian sketch matrix $S$ plays an important role, thanks to its rotational invariance.
>
> Allow us to elaborate: starting from the general covariance matrix $H$, we will convert it to a diagonal covariance matrix case. Assume $H$ has eigenvalues $1^{-2\alpha},2^{-2\alpha},\dots,d^{-2\alpha}$, and assume $ \langle v_i, w^\ast \rangle = i^{-\beta}$ where $v_i$ is the eigenvector of $H$ corresponding to eigenvalue $i^{-\alpha}$ for $i=1,\dots,d$. Let $H = UDU^T$ be an eigenvalue decomposition of $H$.
> Then $w^\ast=Uw_0^\ast$ holds, where $ w_0^\ast = [1^{-\beta},2^{-\beta},\dots,d^{-\beta}]^T $.
>
> The signSGD update rule in our setting is
> $$
> \theta_{k+1}
> = \theta_{k} - \gamma_k  \text{sign} \big(\langle S x_k  , \theta_{k} \rangle -   \langle x_k, w^\ast \rangle \big) \text{sign}(S x_k).
> $$
> Letting $x_k’ = U^T x_k$, and substituting $x_k = Ux_k'$, it converts to
> $$
> \theta_{k+1}
> = \theta_{k} - \gamma_k  \text{sign} \big(\langle SU x_k'  , \theta_{k} \rangle -   \langle Ux_k', Uw_0^\ast \rangle \big) \text{sign}(S Ux_k') =  \theta_{k} - \gamma_k  \text{sign} \big(\langle (SU) x_k'  , \theta_{k} \rangle -   \langle x_k', w_0^\ast \rangle \big) \text{sign}((SU) x_k').
> $$
> On the other hand, the loss formula
> $$
> L(\theta) =  (S^T \theta - w^\ast)^T H (S^T \theta - w^\ast)
> $$
> converts to
> $$
> L(\theta) = (S^T \theta - U w_0^\ast)^T U D U^T (S^T \theta - U w_0^\ast) = ((S U)^T \theta - w_0^\ast)^T  D ((S U)^T \theta - w_0^\ast).
> $$
>
> Note that $x_k'$ has a diagonal covariance matrix $D$. Also note that $S$ and $SU$ have the same distribution due to the rotational invariance of standard Gaussian vectors. So the general covariance matrix problem can be converted to a diagonal covariance matrix problem. Therefore, our analysis for the diagonal covariance matrix can also cover the case of a general covariance matrix with the eigenvalue spectrum assumption. We added this explanation as Section D of the Appendix.
>
>
> ### **W2. Weaknesses in Presentation**
>
> We added two tables (Table 2 and Table 3 in lines 877-909 of Appendix B.4), each for signSGD and SGD, which represent the approximation term, the drift term, and the noise term for different phases. Also, we added a visualization of the stable-decay scheduling (Figure 3 in lines 403-414) to help the readers understand better.
>
> ### **W3. About the Linear Decay Schedule and the Cosine Schedule**
>
> Thank you for the suggestion. We have added an analysis for linear decaying and cosine scheduling in Section H of the Appendix.
>
> For the linear decay schedule, we let the learning rate decay from $1$ to $1/ \sqrt{N}$ over $N$ iterations. We have gone through a similar analysis with the stable-decay scheduling, and we find that linear decay has an advantage compared to a constant learning rate in the intersection of Area Aa* and $0.5 < \alpha < 1$, as it satisfies the same inequality
> $$R_f(M^{\star}, \mathfrak{f} / M^\star, (M^\star)^{-e^\ast}) \lesssim \mathfrak{f}^{-\tfrac{2(4\alpha-1)(2\alpha+2\beta-1)}{16\alpha^2 + 8\alpha\beta + 2\alpha - 2\beta - 1}}$$
> as the stable-decay schedule in that area.
>
> For the cosine schedule, we let the learning rate decay from $1$ to $1/N$, and it has an advantage compared to a constant learning rate in the intersection of Area Aa* and $0.5 < \alpha < 1.5$, as it satisfies the same inequality with the stable-decay schedule in that area.
>
> While we managed to show an advantage of the linear decay schedule and the cosine schedule relative to a constant learning rate, it is hard to compare different schedules and determine which one is better, because we currently have only upper bounds for these scheduling schemes and leave lower bounds open. We leave deriving exact asymptotic forms and comparing the relative performance between different scheduling schemes for future work. We also note that learning-rate-schedules that decay all the way to 0 exactly are technically harder to analyze, but we conjecture their scaling laws will coincide with those for schedules truncated at $1/ \sqrt{N}$ or $1/N$.

---

> > ### Comment · Reviewer_5nJx · 2025-11-23
> >
> > Many thanks for the response! The explanation of assuming $H$ being diagonal and the additional texts in the manuscript are very nice. I have increased my rating.
> >
> > Just a minor follow-up question: In Appendix H.1, the learning rate of the linear decay schedule is assumed to decay to $1/\sqrt N$, while in Appendix H.2, the final learning rate of the cosine schedule is $1/N$. What are the major concerns of choosing the final learning rate, and why are the final learning rates of the linear schedule and the cosine schedule different?

---

> > > ### Author Response · Authors · 2025-11-24
> > >
> > > We are glad to know that our response and revision were helpful, and thank you for raising the score! We address your follow-up question below:
> > >
> > > ### **Answer for the Follow-up Question about Scheduling**
> > >
> > > In short, for cosine scheduling, both final values $1/\sqrt{N}$ and $1/N$ lead to the same scaling behavior in our analysis, but for the linear decaying schedule, selecting a final value of $1/N$  raises a technical issue, so we instead use $1/\sqrt{N}$.
> > >
> > > Let the learning rate for $k$-th iteration be $\gamma_k = f(k) \gamma_0$. The function $f$ could be thought of as a learning rate scheduling function. The main technical concern is the behavior of the schedule near the end of training. In our noise analysis, we need a tail interval $[h(N), N]$ such that (1) $N - h(N) \to \infty$, and (2) $f(z)$ is asymptotically of the same order as $f(N)$ for all $z \in [h(N), N]$. This allows us to treat the tail of the noise integral as if the learning rate were essentially constant at scale $f(N)$.
> > >
> > > When analyzing the noise term, we split the integral
> > > $$\int_0^{N} \exp\Bigl(
> > > -\frac{4\lambda_i \gamma_0}{\pi} \int_z^N \frac{f(u)}{\sqrt{L(u)}} du \Bigr) f(z)^2 dz$$
> > > into the integral on $[0, h(N)]$ and the integral on $[h(N), N]$ for some $h(N)$.
> > > Inside the formula, $L(u)$ is the loss function, and $\lambda_i$ is the $i$-th eigenvalue of the matrix $\text{diag}(SHS^T)^{-1/2} SHS^T$.
> > >
> > > Here, if the conditions $f(N) \eqsim f(z)$ for all $z \in [h(N), N]$, and $N - h(N) \gg 1$ are satisfied, the tail integral on $[h(N), N]$ becomes technically easier to analyze, because the learning-rate schedule is effectively flat on that interval at the scale of $f(N)$.
> > >
> > >
> > > For the stable-decay scheduling, choosing $h(N) = qN$ for some constant $q<1$ close to $1$ was enough.
> > > However, for linear-decaying and cosine schedules with a final learning rate $0 \le f(N) \le 1$, taking $h(N) = qN$ is not a good choice.
> > >
> > > For a linear decaying function decaying from $1$ to $f(N)$, we can take $f(t) = 1- (1-f(N)) \frac{t}{N}$. Also, for a cosine scheduling function decaying from $1$ to $f(N)$, we can take $f(t) = (1+\cos(\frac{\pi t}{N})) \frac{1}{2} + (1 - \cos(\frac{\pi t}{N})) \frac{f(N)}{2} $.
> > >
> > > If we select $h(N) = qN$, then for the linear decaying schedule, $f(h(N)) = f(qN) = 1 - q(1-f(N)) \eqsim 1$ and for the cosine schedule, $f(h(N)) = f(qN) =(1+\cos(q \pi)) \frac{1}{2} + (1 - \cos(q \pi)) \frac{f(N)}{2} \eqsim 1$ as long as $0 \le f(N) \le 1$.
> > >
> > > Then, in order to satisfy the desired condition $f(N) \eqsim f(h(N))$, we would be forced to choose $f(N) \eqsim 1$, which does not correspond to a decaying learning rate. In other words, this makes $\gamma_k$ of the same scale with the $\gamma_0$ for all $k \in [1, N]$, and thus, $h(N) = qN$ cannot give both a decaying schedule and an asymptotically flat tail at the scale of $f(N)$.
> > > Therefore, we select $h(N) = N - \sqrt{N}$ instead of $h(N) = qN$ for linear decaying and cosine scheduling, so that the last $\sqrt{N}$ steps form the tail region.
> > >
> > > * For the linear decaying schedule, we let $f(N) = 1/\sqrt{N}$, which leads to $f(N) \eqsim f(z) \eqsim 1/\sqrt{N} $ for $N - \sqrt{N} \le z \le N$. However, if we select $f(N) = 1/N$, then the two terms, $f(N) \eqsim 1/N$ and $f(N - \sqrt{N}) \eqsim 1/\sqrt{N} $, are no longer asymptotically of the same order. This breaks the “flat tail” condition and makes the tail noise integral technically harder to analyze. For this reason, we select $1/\sqrt{N}$ for the linear decay schedule.
> > >
> > > * For the cosine scheduling, both $1/N$ and $1/\sqrt{N}$ satisfy the “flat-tail” condition on $[N - \sqrt{N}, N]$, and are possible choices. Near $t = N$, the derivative of the cosine schedule vanishes, so $f(z)$ changes only by $O(1/N)$ over the last $\sqrt{N}$​ steps, and $f(z)$ remains of the same asymptotic order as $f(N)$ in both cases. Thus, both choices lead to the same scaling exponent in our analysis.
> > > For cosine scheduling, we select $1/N$ over $1/\sqrt{N}$ just because of notational simplicity.

---

> ### Author Response · Authors · 2025-11-23
> **Response to Reviewer 5nJx (2/2)**
>
> ### **W4. Explanation for the technical contribution**
>
> We are glad that the reviewer finds our insights fruitful. We would like to gently clarify that although the ODE (7) we derive has the same form as in Xiao et al. (2024), our derivation takes a different path under a more relaxed set of assumptions: in particular, we do not require the spectrum of the covariance matrix to be uniformly lower-bounded away from 0, whereas their analysis assumes a positive lower bound on all eigenvalues. Moreover, the procedure we develop to derive the four-term loss formula (12) from this ODE, and to interpret each term as a distinct contribution (drift, noise, initialization, approximation), is new in our paper; note that Xiao et al. (2024) do not investigate that procedure. Since our original submission, we have also included a detailed side-by-side comparison with Xiao et al. (2024) in Section B.2 of the Appendix.
>
> ### **Q1. Does SignSGD improve the exponent in Area Ic?**
>
> In short, SignSGD does not strictly improve the compute-optimal exponent anywhere in Area Ic.
>
> In the region where Phase Ic and Ac overlap, SignSGD and SGD have the same exponents. Note that the loss formulas of SignSGD and SGD are different in that region, so SignSGD and SGD behave differently, but after optimizing the learning rate, the compute-optimal exponents coincide.
>
> In the region where Phase Ic and Ad overlap, SignSGD has worse exponents compared to SGD. The drift-normalization effect sharpens the loss curve of the drift term, but in Phase Ad, the loss already drops sharply at the boundary $\beta = \alpha+0.5$, so in the interior of Phase Ad the additional benefit from the drift-normalization effect is limited.
>
>
> ### **Q2. Does SGD benefit from stable-decay scheduling?**
>
> In our original submission, we stated in the last paragraph of Section 4.3 that SGD does not benefit from scheduling in Phases I and II (which includes Area Aa*). So we believe that the comparison made between (scheduled) signSGD and SGD in Phase Aa is fair. In the revised version, the aforementioned remark on the effect of scheduling for SGD can be found in lines 427-430 and Section G.5 of the Appendix.

---

### Official Review · Reviewer_AVVQ · 2025-10-31

**Soundness:** 3
**Presentation:** 2
**Contribution:** 3
**Rating:** 4
**Confidence:** 3

**Summary:**

This work analyzes the one-pass SignSGD algorithm on a random sketch model $f_{\theta}(x)=\langle Sx,\theta\rangle$ with $S\in\mathbb{R}^{M\times d}$ a Gaussian matrix with $\mathcal{N}(0,1/M)$ entries and data generated by a linear model $y_{k}=\langle w_{\star},x_{k}\rangle$, under a power-law assumption: $x_{k}\sim \mathcal{N}(0,H)$ with $H_{ij}=i^{-2\alpha}\delta_{ij}$ and $w_{\star,j}=j^{-\beta}$ with $\alpha,\beta\geq 0$.

Leveraging an ODE description from Xiao et al. 2024, the main contribution is to characterize the scaling behavior of the risk $R \asymp \mathfrak{f}^{-\eta}$ as a function of the number of flops $\mathfrak{f}=MN$ (model size $\times$ running time), which is summarized in the diagram in Fig. 2 (left)

The authors then compare it to one-pass SGD, concluding as long as the learning rate is properly chosen, the SignSGD rates are always equal or better than SGD, depending on the interplay between $\alpha$ and $\beta$. This can be understood by comparing the ODE description of these two algorithms, which differ by a noise-reshaping and drift-normalization terms that lead to improved scaling.

**Strengths:**

With the surge of interest in scaling laws, the results are timely and of interest to the ICLR community interested in this topic. The fact that the authors can derive different scaling behavior in the $(\alpha,\beta)$ plane and characterize the cross-over between them is interesting. Although the paper is technically loaded, the authors make a good job in giving intuitive explanations to the technical steps.

**Weaknesses:**

The primary weakness - shared by much of the literature on “theory of neural scaling laws” for linear models - is the gap between the stated motivation and the actual model studied. The setting considered here is highly idealized in the context of this motivation (linear model, linear feature map, noiseless observations: no feature learning), and it remains unclear to what extent the interesting theoretical results obtained in such a simplified regime meaningfully translate to the more complex models they aim to explain.

Closely related settings have been extensively investigated in the context of kernel methods and their finite-width approximations (e.g., random features, Nyström methods), where the motivation for studying scaling behavior in linear models is arguably more natural. The power-law decay of covariates and target coefficients corresponds to the classical source and capacity conditions in this literature, and has been analyzed in kernel ridge regression (Caponetto and De Vito, 2007; Cui et al., 2021), random-features ridge regression (Rudi and Rosasco, 2017; Bach, 2017; Defilippis et al., 2024), as well as in work more directly related to the present paper on one-pass SGD (Yao et al., 2007; Ying et al., 2008; Carratino et al., 2018) and multi-pass SGD (Pillaud-Vivien et al., 2018).

While I recognize that there are substantive differences in terms of motivation, specific parameter regimes ($2\alpha<1$), and the quantities of interest (compute versus running time), there remains a meaningful overlap in the models and regimes studied. Therefore, the absence of discussion of this prior work — some of which precides Paquette et al. (2024) and Lin et al. (2024) by decades - is an important weakness. I encourage the authors not only to acknowledge this in the introduction and related work but also to compare with the overlapping regime.

- (Caponnetto and De Vito 2007) Optimal rates for the regularized least-squares algorithm. Foundations of Computational Mathematics 7 (3), 331-368.
- (Cui et al. 2021) Generalization Error Rates in Kernel Regression: The Crossover from the Noiseless to Noisy Regime. NeurIPS 2021
- (Rudi and Rosasco 2017). Generalization properties of learning with random features.  NeurIPS 2017.
- (Bach et al. 2017) On the Equivalence between Kernel Quadrature Rules and Random Feature Expansions. JMLR 2017.
- (Defilippis et al. 2024) Dimension-free deterministic equivalents and scaling laws for random feature regression. NeurIPS 2024
- (Yao et al. 2007) On Early Stopping in Gradient Descent Learning. Constr Approx 26, 289–315 (2007).
- (Ying et al. 2008) Online gradient descent learning algorithms. Foundations of Computational Mathematics, 8, 561-596.
- (Carratino et al. 2018) Learning with SGD and Random Features.  NeurIPS 2018.
- (Pillaud-Vivien et al. 2018) Statistical optimality of stochastic gradient descent on hard learning problems. NeurIPS 2018.
- (Berthier et al. 2020) Tight Nonparametric Convergence Rates for Stochastic Gradient Descent under the Noiseless Linear Model. NeurIPS 2020

**Questions:**

1. The observation that the maximal learning rate leads to a plateau regime is intriguing. Do you have any intuition for why this is the case? How does the maximal and optimal learning rate compare in terms of scaling?

2. In L305:
> *We therefore focus on the optimal learning rate $\gamma_0^\star$, which maximizes the decay exponent $\eta$*

Is the optimal learning rate $\gamma_0^\star$ maximizing the exponent $\eta$ also maximizing the risk?

3. This work focus on a noiseless target. Why? Previous work on SGD (Berthier et al. 2020) shows that the noiseless rates can be faster, and in the context of ridge that there can be cross-overs in the rates depending on the interplay between the noise and the regularization (Cui et al. 2021; Defilippis et al. 2024). Do you expect your results to be independent of the noise of to observe something similar here?

---

> ### Author Response · Authors · 2025-11-23
> **Response to Reviewer AVVQ (1/2)**
>
> We thank the reviewer for the detailed feedback and useful perspective.
>
> ### **W1. Weaknesses of the Setup (linear model, linear feature map, noiseless observations, no feature learning)**
>
> We agree with the reviewer that the PLRF setting—and more broadly, the linear-model analyses commonly used in the theory of neural scaling laws—remains far from practical scenarios such as training modern LLMs with AdamW. Building a rigorous theoretical understanding for such large-scale nonlinear models is extremely challenging, and it is not something that can be achieved in a single step.
>
> However, we believe that progress toward this long-term goal necessarily begins with simpler models where clean analysis is possible. In this sense, the PLRF framework has served as an important testbed for developing intuition about the scaling laws. Prior works have analyzed gradient flow, GD, one-pass SGD, multi-pass SGD, and momentum methods in this setting. Our work extends this line of research by providing a new analysis for signSGD and comparing it with SGD.
>
> These incremental advances form the foundation for extending scaling-law analysis to complex models. We expect that such results will eventually generalize to feature learning and more realistic architectures. For example, the loss formulas for the PLRF and for certain feature-learning models trained with SGD are closely related (Paquette et al., 2024; Bordelon et al., 2025), so we expect the signSGD loss formula derived in our work can help build a proxy of loss formula for feature learning with signSGD. Exploring these directions is an important avenue for future research.
>
> Finally, regarding noisy observations, our analysis can be extended to the noisy-label setting. We outline this extension in Section L of the Appendix and in our response to Question 3 below.
>
>
> ### **W2. Comparison with Work in the Context of Kernel Methods**
> Thank you for the constructive comment. We agree that it is important to compare with the papers in the context of kernel methods and their finite-width approximations, so we have added a detailed comparison in Section B.3 of the Appendix, and now explicitly acknowledge this line of work in the related work part (lines 122-130) of the main text.
>
> In the revised text we now discuss classical kernel ridge regression results as well as random-feature and SGD-based methods. We note in particular that Berthier et al. (2020) studied a setting closest to ours, namely, linear regression with SGD and noiseless labels. Converting their source and capacity parameters into $(\alpha, \beta)$, their upper bound on the loss behaves as $n^{-\min((2\alpha+2\beta-1)/(2\alpha), 1-1/(2\alpha))}$ where $n$ is the number of samples. Later work, e.g., by Paquette et al. (2024), obtains the same exponents for the drift terms, as they also analyze SGD under a noiseless-label assumption. The difference between exponents in Berthier et al. (2020) and the exponents of the drift term in our work stems from the drift-normalization effect of signSGD.
>
> Please find more detailed comparisons against other existing papers in Section B.3 of the Appendix. There are many other interesting prior works in the context of kernel methods, but since they do not consider signSGD, it is difficult to directly match and compare the four terms in our decomposition (12) term by term.
>
>
> - Bordelon, B., Atanasov, A., & Pehlevan, C. (2025). How feature learning can improve neural scaling laws. Journal of Statistical Mechanics: Theory and Experiment, 2025(8), 084002.

---

> ### Author Response · Authors · 2025-11-23
> **Response to Reviewer AVVQ (2/2)**
>
> ### **Q1. Why does the maximal learning rate lead to a plateau regime?**
>
> When we are close to the optimal point, the size of the gradient gets smaller. So for SGD, this naturally reduces the size of the update when it gets closer to the optimal point, and it converges well.
>
> In contrast, for signSGD, as we normalize the gradient, the size of the update does not shrink as we approach the optimal point, and this introduces a bottleneck term of order $o(\gamma^2)$ in (12), where $\gamma$ is the learning rate. Because of this bottleneck term, using the maximal learning rate causes the loss to saturate at a plateau, and the compute-optimal learning rate  is strictly smaller than the maximal learning rate.
>
> For example, in Phase Aa, the maximal learning rate scales as $M^{-1/2}$, while the optimal learning rate scales as $M^{-(\alpha+\beta)}$, where $M$ is the model size. In this phase, $\alpha + \beta > 1/2$, so the ratio between the maximal and optimal learning rate grows as we scale up $M$. In terms of loss, the maximal learning rate suffers from the plateau caused by the bottleneck term, whereas the optimal learning rate avoids this plateau and therefore yields a better scaling exponent.
>
> ### **Q2. Does the optimal learning rate maximize the risk?**
> No, the optimal learning rate $\gamma_0^*$ that maximizes $\eta$ actually *minimizes* the risk. In our notation, the loss takes the form $\mathfrak{f}^{- \eta(\gamma_0)}$, so a larger exponent $\eta$ leads to a smaller loss because of the minus sign on the exponent.
>
> ### **Q3. About the Noisy Target**
> The primary motivation for this paper was to analyze signSGD scaling laws and to compare them against the SGD scaling laws presented in Paquette et al. (2024), so we focused on the noiseless target setting, following Paquette et al. (2024). However, we agree that the rate for noisy labels is also an important aspect to study, so we have added analysis and experiment for the noisy target in Section L of the Appendix.
>
> We denote by $\sigma^2$ the variance of the label noise; this contributes an irreducible term $\sigma^2$ to the risk. We analyze the rate of convergence of the excess risk (Risk – $\sigma^2$) in Phase Aa, and find that, in the compute-optimal case, the rate is $\mathfrak{f}^{-(2\alpha+2\beta-1)/(4\alpha+2\beta)}$. For comparison, the rate in the noiseless setting is $\mathfrak{f}^{-(2\alpha+2\beta-1)/(2\alpha+1)}$. Since $4\alpha+2\beta>2\alpha+1$, the exponent with noise is smaller, and thus, the noisy rate is slower than the noiseless rate, in line with the behavior reported in the prior work (Berthier et al., 2020) the reviewer mentioned.
>
> Thus, the scaling behavior is not independent of the noise: the presence of label noise degrades the compute-optimal exponent. More generally, as in Cui et al. (2021), we expect that varying the relative scales of noise and compute would lead to cross-overs between noise-dominated and bias-dominated regimes. Our current analysis in Phase Aa illustrates one such noisy regime, and a complete classification of noisy phases is an interesting direction for future work. Since the compute-optimal exponent of SGD with noisy labels for a constant learning rate is not known yet, we also leave the comparison with SGD for future work.

---

> > ### Comment · Reviewer_AVVQ · 2025-11-25
> >
> > I thank the authors for their detailed rebutal and the revised manuscript. I am updating my score accordingly.
> >
> > > *W1. Weaknesses of the Setup (linear model, linear feature map, noiseless observations, no feature learning)*
> >
> > Completely agree witg the authors reply.
> >
> > > *W2. Comparison with Work in the Context of Kernel Methods*
> >
> > I appreciate the detailed comparison of the exponents with the kernel literature in the new Appendix B3.
> >
> > Despite the different origins and vocabulary, these two literatures share a common endeavor, and the quantities computed are the same. Putting forward this comparison actually strengths the novelty of the analysis (differences between SignSGD, KRR and SGD), not the opposite.
> >
> > > *Q2. Does the optimal learning rate maximize the risk?*
> >
> > Yes, my bad. I meant minimize. But your reply actually answered my question.
> >
> > > *Q3. About the Noisy Target*
> >
> > These preliminary results are very interesting. I look forward for the follow up work characterizing the cross-overs with the noise-dominated regions.

---

> > > ### Author Response · Authors · 2025-11-25
> > >
> > > Thank you very much for your reassessment and for updating your score. Your comments and suggestions were very helpful in improving the paper, and we sincerely appreciate the time and effort you invested in our work.

---

### Official Review · Reviewer_Hokd · 2025-10-31

**Soundness:** 3
**Presentation:** 3
**Contribution:** 2
**Rating:** 4
**Confidence:** 4

**Summary:**

This paper develops a theoretical analysis of the scaling laws for signSGD under the power-law random features (PLRF) model, extending the recent theoretical framework of Paquette et al. (2024) for SGD. The authors derive asymptotic risk formulas and compute-optimal tradeoffs for signSGD, identifying two novel mechanisms—drift-normalization and noise-reshaping—that differentiate its scaling behavior from SGD. They also propose a simplified stable-decay learning-rate schedule (related to warmup–stable–decay) that further improves compute-optimal exponents in certain parameter regimes. Theoretical predictions are validated with numerical simulations.

Overall, this work makes a theoretically interesting extension of the PLRF scaling-law framework to signSGD, introducing two interpretable mechanisms that may explain optimizer-dependent scaling differences. However, it requires improved exposition, stronger intuition, and at least limited empirical grounding on neural networks to reach full impact.

**Strengths:**

1. Extending scaling-law theory to optimizers beyond SGD is a natural and important step, especially given that sign-based methods underlie Adam and its variants, which dominate large-scale model training. The derivation of scaling exponents for signSGD is nontrivial, and the introduction of drift-normalization and noise-reshaping as analytic effects is both insightful and original.
2. The analysis connects optimizer dynamics (sign normalization, step-size scheduling) to compute-optimal scaling exponents, potentially offering insight into why adaptive methods empirically outperform SGD. The paper demonstrates strong technical depth, following a structured progression from stochastic dynamics → ODE approximation → scaling law formulation → compute-optimal tradeoffs. The analysis is self-contained and references relevant recent work (Paquette et al., Lin et al., Xiao et al., etc.).
3. Experiments (though small-scale) consistently confirm the predicted exponents and qualitative trends from the theory.

**Weaknesses:**

1.The paper’s presentation is algebra-heavy and difficult to parse. While mathematically correct, it often lacks guiding intuition or concrete interpretation of the results in the context of real neural network training. The drift-normalization and noise-reshaping effects, while named, are only partially explained mechanistically.
2. The experiments are limited to synthetic settings (Gaussian-sketch features) and are primarily confirmatory. No evidence is given that the derived scaling laws hold qualitatively in deep networks, or that the “beneficial areas” (e.g., Area III–IV_sub) correspond to practically relevant regimes.
3. Although signSGD is an important extension, much of the framework—including PLRF setup, compute-optimal derivation, and phase-plane analysis—is directly inherited from prior work. The primary novelty is analytical substitution of the sign-based update rule into the existing framework.
4. The paper’s claim that Adam follows the same scaling law “heuristically” (Appendix H) is speculative and unsupported by rigorous derivation. This is a key potential impact point, but it’s treated informally.
5. The text is dense, with overuse of symbols and nested definitions (e.g., equations (6)–(12)). Several steps (especially in Appendices D–E) are highly opaque. Without significant simplification or visualization, the theoretical results remain inaccessible to a broad audience.
6. The empirical mapping of where signSGD outperforms SGD in the (α, β)-plane feels ad hoc and under-motivated. The heuristic in Section 5.1 (decay mismatch between target and gradient) is suggestive but not tested quantitatively.

**Questions:**

1. How sensitive are the scaling-law exponents to the Gaussian-sketch assumption? Would structured or learned feature maps change the analysis?
2. Can the “drift-normalization” and “noise-reshaping” effects be observed empirically in loss/gradient trajectories of real models trained with Adam?
3. The “stable-decay” schedule is claimed to approximate warmup stable-decay; can this be verified experimentally on a practical model?
4. How does the asymptotic noise term ( N_{\text{sign}}(M, \eta_0) ) compare quantitatively with SGD noise in finite-N settings?

---

> ### Author Response · Authors · 2025-11-23
> **Response to Reviewer Hokd (1/3)**
>
> We thank the reviewer for the helpful comments and time devoted to our work.
>
> ### **W1-1. Intuition for Real Neural Network Training**
>
> In modern LLM training, Adam combined with warmup–stable–decay (WSD) schedules is widely used in practice. In this work, we analyze signSGD—viewed as a simple surrogate for Adam—under stable–decay learning rates and identify power-law regimes in which it can outperform SGD within our PLRF model. This suggests, at a theoretical level, that using Adam-like signSGD updates together with WSD-style schedules can be advantageous, at least in certain regimes, when scaling up models, in line with current practice. Although practitioners already use such schemes, we believe that providing a rigorous explanation for their benefits is also important. While our setting is still far from the full complexity of real neural networks, we believe that our results shed light on the benefit of practical schemes such as Adam and WSD and help account for part of the theory–practice gap observed in large-scale neural network training.
>
> Zipf’s law for token frequencies ($p_r \propto r^{-1}$ for the rank-$r$ and frequency $p_r$) suggests that simple language-modeling data have covariance spectra $\lambda_k \propto k^{-1}$ (where $\lambda_k$ denotes the $k$-th eigenvalue), corresponding to $\alpha \approx 0.5$ in our parametrization. For regimes with $\alpha \approx 0.5$ and $\beta > 0.5$, our analysis shows that signSGD attains a better scaling behavior than SGD, and this parameter regime could be related to scenarios where the loss is more influenced by high-frequency components or common tokens in language modeling. While our setting abstracts away many complexities of real neural networks, we believe that these observations connect our “beneficial areas” in the $(\alpha,\beta)$-plane to practically relevant regimes.
>
> ### **W1-2. About Drift-normalization and Noise-reshaping Effects**
>
> Mechanistically, the drift-normalization effect arises because SignSGD normalizes the gradient, which amplifies the contribution of small-eigenvalue directions relative to SGD and increases the exponent of the drift term in our four-term loss formula. The noise-reshaping effect describes how this normalization changes the dependence of the noise plateau on the learning rate, making the noise contribution grow with $\gamma_0$ for SignSGD in contrast to the SGD case. We also added more explanation for the drift-normalization and noise-reshaping effects (lines 297-313), expanding on how the difference between signSGD and SGD at the ODE level translates to the effects in the four-term loss decomposition formulas.
>
> ### **W2. Experiments for Non-synthetic Settings**
>
> We added an experiment comparing AdamW and SGD on a Transformer in Section C.5.1 in the Appendix.
>
> We evaluated Transformers of depth up to 8. In terms of compute-optimal scaling, we observed exponents of approximately -0.021 for AdamW and -0.005 for SGD, indicating that AdamW achieves a better compute-optimal exponent than SGD in this setting.
> Although our analysis is about signSGD (which is a surrogate for Adam-type methods), in a simple linear PLRF model, this experiment provides preliminary qualitative evidence that an advantage in the compute-optimal scaling aspect may also appear for a practical optimizer, AdamW, on Transformers.
>
> As in prior work such as Ferbach et al. (2025), the exact absolute exponents for practical deep networks are smaller than in the linear models. Therefore, we focus on the relative difference between different optimizers rather than on matching exponents exactly. A more direct comparison between SignSGD, AdamW, and SGD in larger-scale architectures would be a valuable direction for future empirical work.
>
> - Ferbach, D., Everett, K., Gidel, G., Paquette, E., & Paquette, C. (2025). Dimension-adapted Momentum Outscales SGD. arXiv preprint arXiv:2505.16098.

---

> ### Author Response · Authors · 2025-11-23
> **Response to Reviewer Hokd (2/3)**
>
> ### **W3. About the Similarity of Framework to Prior Work**
>
> The primary motivation for this paper was to analyze signSGD scaling laws and to compare them against the SGD scaling laws presented in Paquette et al. (2024). We thus deliberately inherited the setting of Paquette et al. (2024), to make sure that fair comparisons are made. As the reviewer correctly points out, our novelty lies in the shift from SGD to signSGD; however, we emphasize that this is much more than a simple "substitution" of update rules.
>
> In Paquette et al. (2024), after evaluating the contour integral for the drift term, they obtain an explicit loss formula expressed in terms of the model size, number of steps, and learning rate. However, in our analysis, due to the normalization in the SignSGD update, the analogous calculation leads instead to an expression involving $\int_{0}^{N}\frac{du}{\sqrt{L(u)}}$ term, where $N$ is the number of steps and $L(u)$ is the loss function. So after a contour-integral argument similar in spirit to Paquette et al. (2024), we obtain a self-consistent equation for $L$ rather than a closed-form loss formula. Solving this self-consistent equation is a new technical ingredient in our paper.
>
> Also, for the noise term in (10), which take the form $\int_{0}^{N} h(z) dz$, the contour-integral technique cannot be applied uniformly up to the part where $z$ is near $N$. Therefore, we treat the region near $N$ separately, by using the limit loss value to control that contribution. This additional analysis is required specifically because of the SignSGD normalization.
>
>
>
> ### **W4. About the Heuristic Analysis for Adam**
>
> The heuristic analysis for Adam is not a main contribution of our paper, and therefore, we deliberately did not mention Adam in the title or abstract. We included the heuristic analysis for Adam and discussion in the Appendix to share our conjecture for Adam with the community.
>
> Having said that, following the suggestions by Reviewer 1qNE, we have included a brief discussion on the scaling laws of Adam along with a pointer to Appendix J in Section 5.2 of our revised manuscript.
>
> ### **W5.  Accessibility to a broad audience**
>
> Thank you for the comment. At the beginning of Appendices E and F (which were Appendices D and E in the initial version), we have added a proof overview and specified the goal of each section to help readers navigate the technical arguments. We hope that these modifications will make the paper more accessible while preserving the full mathematical detail.
>
>
> ### **W6. About the Heuristic in Section 5.1**
>
> We expanded our explanation of why a larger $\beta$ can lead to a faster plateau of the target decay in the revised version (lines 489-506). In the initial version, we only provided empirical evidence, but in the revision, we add theoretical intuition by analyzing the decay of $U^T \theta^\ast$ (the target expressed in the eigen-basis $U$). In particular, we split $U^T \theta^\ast$ into an expectation term and a fluctuation term, and show that a larger $\beta$ could cause the error term to dominate earlier, thereby leading to a faster plateau. This connects the heuristic “decay mismatch” picture to a more precise decomposition.

---

> ### Author Response · Authors · 2025-11-23
> **Response to Reviewer Hokd (3/3)**
>
> ### **Q1. How sensitive are the scaling-law exponents to the Gaussian-sketch assumption?**
>
> If Gaussian-sketch is replaced by structured or learned feature maps, we expect that the absolute value of scaling-law exponents may increase for some $(\alpha, \beta)$ regions.
>
> For learned feature maps, Bordelon et.al.(2025) analyzed that for SGD, feature learning leads to a decrease of aligned feature loss in Phase Aa (the factor $(N\gamma_0)^{-x}$ changes to $(N\gamma_0)^{-(2x)/(1+x)}$ where $x = (2\alpha+2\beta-1)/(2\alpha)$).
>
> By analogy, we conjecture that for the signSGD scaling law under feature learning, the absolute value of the exponent in the aligned drift term in the signSGD loss formula will also increase. We think analyzing this precisely will be an interesting future direction.
>
> ### **Q2. Can the drift-normalization and noise-reshaping effects be observed empirically in loss/gradient trajectories of real models trained with Adam?**
>
> We investigated the loss curve of AdamW and SGD on a Transformer. The experimental setup is similar to the experiment used in our answer to W2, and the details are given in Section C.5.2 of the Appendix.
>
> To observe the drift-normalization effect, we experimented with a batch size of 16 and gradient accumulation steps of 32 to reduce the noise term. Since the loss curve can be viewed as the sum of the drift, noise, and approximation terms, suppressing the noise term allows the drift contribution to dominate, letting us observe the drift-normalization effect more clearly. We measured the slope of the loss curve in a log-log plot over the linear-decay interval, where the drift term is dominant, and observed that the slope for AdamW is larger than SGD. This qualitative behavior is consistent with the drift-normalization effect in our PLRF analysis, where the exponent of the drift term increases for signSGD compared to SGD.
>
> To observe the noise-reshaping effect, we focus on the plateau regime in a batch-size-1 experiment. To see how the loss value of the plateau regime depends on the learning rate, we experiment with two learning rate values: 0.00266 and 0.00133 for both AdamW and SGD. We observed that the plateau loss value, which is dominated by the noise term, increases for AdamW when we use the larger learning rate, but does not increase for SGD. This is qualitatively consistent with the noise-reshaping effect in PLRF, where the size of the noise term in signSGD increases with the learning rate, in contrast to SGD.
>
> ### **Q3. Is it okay to approximate the warmup-stable-decay with the stable-decay schedule?**
>
> Thank you for bringing this point up. The reason why we used a stable-decay schedule in the main analysis is to emphasize that the stable and decay part of a warmup-stable-decay schedule is what improves the compute-optimal exponent of signSGD, not because we cannot analyze the warmup-stable-decay itself. Actually, our claims in Section 4.3 for the stable-decay schedule also hold for warmup-stable-decay; we have added the analysis of warmup-stable-decay as Section G.4 of the Appendix in the revision.
>
> ### **Q4. How does the asymptotic noise term ($\mathcal{N}^{\text{sign}}(M, \eta_0)$) compare quantitatively with SGD noise in finite-$N$ settings?**
>
> It highly depends on the learning rate $\gamma_0$. For the case $\gamma_0 \eqsim 1$, the signSGD noise term is larger than the SGD noise term, because SGD reduces the size of its updates as the gradient norm shrinks near an optimal point, whereas signSGD removes this “self-regularizing” property through  normalization.
>
> When $\gamma_0$ becomes smaller (equivalently, when $e$ increases in the scaling $\gamma_0 \eqsim M^{-e}$), the signSGD noise decreases, since it is proportional to $\gamma_0^2$ in our PLRF formulas once the self-regularizing effect has been removed. In contrast, for $2\alpha>1$, the SGD noise increases as $\gamma_0$ decreases, because it depends on a power of the effective number of steps $N\gamma_0$ (intuitively, SGD updates also effectively reduce the impact of noise from previous timesteps).
>
> Therefore, when $N$ and $M$ are fixed and we decrease $\gamma_0$ (increase $e$, where $\gamma_0 \eqsim M^{-e}$), SGD noise is initially smaller, but after a crossover point, signSGD noise becomes smaller.
> Regarding $N$ and $M$, increasing $N$ will decrease the SGD noise but does not affect the signSGD noise in our asymptotic regime. Increasing $M$ decreases the signSGD noise if $e>1-\min(0.5, \alpha)$ (where $\gamma_0 \eqsim M^{-e}$), and increases it otherwise; in contrast, the asymptotic SGD noise does not depend on $M$ in this regime.

---

### Official Review · Reviewer_1qNE · 2025-11-01

**Soundness:** 3
**Presentation:** 2
**Contribution:** 3
**Rating:** 6
**Confidence:** 2

**Summary:**

This paper explores the scaling laws of one-pass signSGD under the power-law random features (PLRF) model with two parameters (feature decay and target decay). It derives a four-term population risk decomposition and an explicit scaling law that depends on model size M, training steps N, and learning rate. By comparing this to SGD, the authors uncover two phenomena: drift-normalization, which accelerates training under conditions,  and noise-reshaping, which yields an N-independent noise floor and additional M dependence. Based on the scaling law, this paper theoretically shows that signSGD achieves a steeper compute-optimal slope than SGD in certain regimes, and optimal learning rate scheduling (a simplified warmup–stable–decay) further improves the slope. There are empirical experiments that confirm the theory.

**Strengths:**

* The theoretical argument is rigorous and sound to me. The four-term risk formula decomposition mirrors prior SGD analyses and uncovers signSGD-specific properties in a clear way. This gives immediate intuition for when signSGD can help.
* The paper provides further insights, including the comprehensive phase transitions of scaling laws, analysis for compute-optimal slope, and learning rate scheduling that reflects the practice. These analyses serve as a step towards understanding this optimizer in theory.
* The paper checks predicted slopes and compute-optimal scaling across representative regimes (Appendix C) via synthetic experiments. These empirical results validate and give us confidence in the theory.

**Weaknesses:**

* **Limitations in Setting:** There are certain assumptions in this paper, for example, batch size = 1, diagonal covariance, one-pass training, and Gaussian sketching in the PLRF model, that limit direct generalization to more practical pipelines. The paper acknowledges some limitations such as mini-batching and momentum/Adam invariants, but these still limit the novelty to some extent, and results would be stronger with at least partial extensions (or tests) beyond this PLRF setting.

* **Discussion of Theory:** The structure of the paper could be improved. In particular, the context of the problem and proof ideas occupy too much space, while some useful contents are pushed to the appendices, and this makes the important Section 5 seem rushed. The two main effects, drift normalization and noise reshaping, should definitely be discussed in more detail. For instance, from Lines 413-424 ,the mechanism of signSGD superiority is attributed to an intricate interplay between these two effects that “creates room for a balance”. It is helpful if this could be quantified further.

* **Adam Optimizer:** In the introduction, one claimed relevance of this work is that the state-of-the-art LLMs are trained with the Adam optimizer (rather than SGD), which can be indirectly studied through the more tractable signSGD. Since this is why signSGD scaling law helps connect theory with empirical optimizer choice in practice, there should also be more discussion on Adam in the main text. For now, Adam is not mentioned until in the conclusion, which claims that they also analyze Adam in a heuristic manner in the Appendix (while I do not think we should mention new contents in the conclusion). Given the paper’s motivation, these parts will be selectively included earlier to make the organization less detached, and it would strengthen the case to include at least one non-PLRF synthetic setting for Adam, too.

Overall, I think this paper is sound. My main concerns are in possible preliminary extension from this setting and issues in presentation.

**Questions:**

Based on the phase transitions, can you provide some summary for practitioners to decide (say from coarse empirical estimates of “feature/target decay”) when to try signSGD + stable-decay vs. SGD?

Please refer to the Weakness section for other questions.

---

> ### Author Response · Authors · 2025-11-23
> **Response to Reviewer 1qNE (1/2)**
>
> We thank the reviewer for the detailed feedback and useful perspective.
>
> ### **W1. Limitations in Setting**
> Thank you for the detailed comment. We will explain each of the four limitations one-by-one.
>
> - Diagonal covariance : We would like to first point out that a general covariance matrix with eigenvalues $1^{-2\alpha}, …, d^{-2\alpha}$ can be reduced to our diagonal case; hence, assuming diagonal $H$ is without loss of generality. We added a detailed explanation for that as Section D of the Appendix.
>
> - One-pass training: Modern LLM training is typically done in a single (if not just a few) pass through the dataset, due to the abundant amount of available data. As our study of the scaling laws in the PLRF setting is mainly motivated by LLM, we believe that one-pass training is the setting closer to the practice of language model training.
>
> - Batch size 1:  Our experiments show that with the choices of batch size of 10 and 128, signSGD has compute-optimal exponents similar to the batch size 1 case; we added these experiments in Section C.4 of the Appendix. Therefore, we conjecture that mini-batching with constant-order batch size has the same compute-optimal exponents as the batch size 1 case; this is plausible because constant factors in the loss formula are ignored in the exponent analysis. However, mathematically analyzing the mini-batch case is highly nontrivial, as the sign of the gradient does not directly relate to the sign of the input data vector whenever the batch size is greater than 1. We believe this is an important future research direction. We would also like to point out that prior work (Xiao et al., 2024) on ODE and SDE representations of signSGD also focuses on the batch size 1 case.
>
> - Gaussian sketching: Replacing Gaussian sketching with learnable parameters (i.e., "feature learning") is an interesting future direction. Bordelon et al. (2025) analyzed feature learning with SGD and found that it leads to a decrease of aligned feature loss in Phase Aa (the factor $(N\gamma_0)^{-x}$ changes to $(N\gamma_0)^{-(2x)/(1+x)}$, where $x = (2\alpha+2\beta-1)/(2\alpha)$). We conjecture that for feature learning with signSGD as well, the absolute value of the exponent in the aligned drift term will increase compared to the sketching-based signSGD loss formula in our paper.
>
> ### **W2. Discussion of Theory**
> Thank you very much for the great suggestions. We added an explanation for the reason why the target decay could be faster for a bigger $\beta$ in Section 5.1 (lines 489-506). In the added explanation, we split the vector $U^T \theta^\ast$ (showing the target decay in the basis of $U$) into an expectation term and a fluctuation term. Here, we find that a larger choice of $\beta$ could lead to an earlier transition to the regime where the error term dominates, which then results in a faster plateau.
>
> We also expanded an explanation for the drift-normalization and noise-reshaping effects (lines 297-313). We discussed more on how the difference in the ODE stage introduces the two effects in the loss formula.
>
> Further, we elaborated more about balancing between the drift term and the noise term in the Area III–IV_sub of signSGD in the revised version (lines 450-460). We discussed the actual exponents, which we believe makes the explanation less abstract.
>
>
> ### **W3. Adam Optimizer**
>
> Thank you for the helpful suggestions. We added Section 5.2, which explains our conjecture for Adam based on heuristic analysis in the main text. In Section C.6 of the Appendix, we have also added an experiment with Adam on a feature learning setting of Bordelon et al. (2025). This experiment introduces a synthetic non-PLRF setting in which the sketch matrix is learnable, unlike the fixed Gaussian sketch matrix in PLRF. We experimented on the $(\alpha, \beta)$ parameter inside Area III–IV_sub, and observed that Adam and signSGD have similar compute-optimal exponents, while SGD has a smaller exponent compared to them: an observation consistent with the PLRF setting.
>
> - Xiao, K. L., Marshall, N., Agarwala, A., & Paquette, E. (2024). Exact risk curves of signsgd in high-dimensions: Quantifying preconditioning and noise-compression effects. arXiv preprint arXiv:2411.12135.
> - Bordelon, B., Atanasov, A., & Pehlevan, C. (2025). How feature learning can improve neural scaling laws. Journal of Statistical Mechanics: Theory and Experiment, 2025(8), 084002.

---

> ### Author Response · Authors · 2025-11-23
> **Response to Reviewer 1qNE (2/2)**
>
> ### **Q1. Based on the phase transitions, can you provide some summary for practitioners to decide (say from coarse empirical estimates of “feature/target decay”) when to try signSGD + stable-decay vs. SGD?**
>
> Based on our analysis, we could propose two things. First, we recommend using stable-decay by default, because it does not harm the scaling law while giving benefits for some regions.
>
> In terms of signSGD vs SGD, we recommend using signSGD for most cases, except for cases with very small feature decay and large target decay (the lower right part of Phase Ad), because signSGD scales similar to or better than SGD in terms of the compute-optimal scaling law. Intuitively, this indicates that it is a good idea to try signSGD (or Adam) in most language tasks, because this “small feature decay, large target decay” regime corresponds to uncommon and unlikely settings where the text distribution is nearly uniform (diverse words and styles appearing with similar probabilities) but the loss is dominated by a few simple, high-frequency patterns such as basic grammar, common phrases, or an easy downstream label.

---

### Official Review · Reviewer_dH5K · 2025-11-04

**Soundness:** 3
**Presentation:** 2
**Contribution:** 3
**Rating:** 6
**Confidence:** 3

**Summary:**

This manuscript builds on the asymptotic characterization of SignSGD for quadratic problems in Xiao et al. (2024) to study its scaling properties, in a manner reminiscent of Paquette et al. (2024). It shows that, in phases where the scaling exponent of SGD is determined by gradient noise, SignSGD can improve the scaling. The benefit is attributed to SignSGD’s ability to balance drift-normalization and noise-reshaping via learning-rate tuning.

**Strengths:**

The paper derives the scaling exponent for SignSGD, a simplified variant of Adam, and empirically demonstrates that the theoretical predictions also hold for Adam.

**Weaknesses:**

- The technical part of the paper closely follows the ideas in Xiao et al. (2024) and Paquette et al. (2024).
- As far as I understand, the results in the paper depend on choosing a good learning rate, where the optimal rate depends on both the target and feature exponents. However, the authors do not discuss how to tune the learning rate. Discussing this point—and showing how sensitive the results are to suboptimal choices—would strengthen the paper.

**Questions:**

- As fast as I can follow, there is no label noise in the setting studied. Can you explain why?

---

> ### Author Response · Authors · 2025-11-23
> **Response to Reviewer dH5K (1/2)**
>
> We thank the reviewer for the careful reading and valuable remarks.
>
> ### **W1. Explanation of the Technical Contribution**
>
> We would like to gently clarify that although the ODE (7) we derive has the same form as in Xiao et al. (2024), our derivation takes a different path under a more relaxed set of assumptions: in particular, we do not require the spectrum of the covariance matrix to be uniformly lower-bounded away from 0, whereas their analysis assumes a positive lower bound on all eigenvalues. Moreover, the procedure we develop to derive the four-term loss formula (12) from this ODE, and to interpret each term as a distinct contribution (drift, noise, initialization, approximation), is new in our paper; note that Xiao et al. (2024) do not investigate that procedure. Since our original submission, we have also included a detailed side-by-side comparison with Xiao et al. (2024) in Section B.2 of the Appendix.
>
> Now, we will explain technical differences compared to Paquette et al. (2024). In Paquette et al. (2024), after evaluating the contour integral for the drift term, they obtain an explicit loss formula expressed in terms of the model size, number of steps, and learning rate. However, in our analysis, due to the normalization in the SignSGD update, the analogous calculation leads instead to an expression involving $\int_{0}^{N}\frac{du}{\sqrt{L(u)}}$ term, where $N$ is the number of steps and $L(u)$ is the loss function. So, after a contour-integral argument similar in spirit to Paquette et al. (2024), we obtain a self-consistent equation for $L$ rather than a closed-form loss formula. Solving this self-consistent equation is a new technical ingredient in our paper.
>
> Also, for the noise term in (10), which take the form $\int_{0}^{N} h(z) dz$, the contour-integral technique cannot be applied uniformly up to the part where $z$ is near $N$. Therefore, we treat the region near $N$ separately, by using the limit loss value to control that contribution. This additional analysis is required specifically because of the SignSGD normalization.
>
>
>
> ### **W2. Effect of Suboptimal Choice of Learning Rate and Way to Tune the Learning Rate**
>
> Thank you for the constructive comment. We added a calculation of the compute-optimal exponent for general learning rates of the form $\gamma_0 = M^{-e}$ (this includes suboptimal choices) in Section F.3 of the Appendix. We focused on Phase Aa and expressed the final exponent in terms of $e$.
>
> In the analysis, we find that the absolute value of the compute-optimal exponent as a function of $e$ is continuous, and the absolute value gradually decreases as we move away from the optimal choice. Also, we observed that the degradation of performance is milder for the learning rates with larger-than-optimal $e$ than that of smaller-than-optimal values of $e$. Therefore, in terms of learning rate tuning, we would suggest initially trying a high $e$ and then gradually decreasing $e$ for later attempts.

---

> ### Author Response · Authors · 2025-11-23
> **Response to Reviewer dH5K (2/2)**
>
> ### **Q1. About the Label Noise**
>
> Our primary motivation was to compare our signSGD scaling laws against the SGD counterparts studied in Paquette et al. (2024). As Paquette et al. (2024) use a framework with no label noise, we carried out our analysis in the noise-less setting. However, we agree that label noise is also an important and nonnegligible component, so we added analysis and experiment for the case with label noise in Section L of the Appendix.
>
> We denote by $\sigma^2$ the variance of the label noise; this contributes an irreducible term $\sigma^2$ to the risk. We analyze the rate of convergence of the excess risk (Risk – $\sigma^2$) in Phase Aa, and find that, in the compute-optimal case, the rate is $\mathfrak{f}^{-(2\alpha+2\beta-1)/(4\alpha+2\beta)}$. For comparison, the rate in the noiseless setting is $\mathfrak{f}^{-(2\alpha+2\beta-1)/(2\alpha+1)}$. Since $4\alpha+2\beta>2\alpha+1$, the exponent with noise is smaller, and thus, the noisy rate is slower than the noiseless rate, in line with the behavior reported in the prior work (Berthier et al., 2020).
>
> Thus, the scaling behavior is not independent of the noise: the presence of label noise degrades the compute-optimal exponent. More generally, as in Cui et al. (2021), we expect that varying the relative scales of noise and compute would lead to cross-overs between noise-dominated and bias-dominated regimes. Our current analysis in Phase Aa illustrates one such noisy regime, and a complete classification of noisy phases is an interesting direction for future work. Since the compute-optimal exponent of SGD with noisy labels for a constant learning rate is not known yet, we also leave the comparison with SGD for future work.
>
> - Berthier, R., Bach, F., & Gaillard, P. (2020). Tight nonparametric convergence rates for stochastic gradient descent under the noiseless linear model. Advances in Neural Information Processing Systems, 33, 2576-2586.
> - Cui, H., Loureiro, B., Krzakala, F., & Zdeborová, L. (2021). Generalization error rates in kernel regression: The crossover from the noiseless to noisy regime. Advances in Neural Information Processing Systems, 34, 10131-10143.

---

> > ### Comment · Reviewer_dH5K · 2025-11-26
> >
> > Thank you very much for the detailed response! I’d like to ask one follow-up question.
> >
> > In Xiao et al. (2024), it is suggested that a lower bound on the spectrum of the covariance matrix is **necessary** for analyzing how the sign function affects the stochastic gradient (see the paragraph following Assumption 3 in the arXiv version). However, although you suggest that you can go beyond this requirement, Section B.2 does not make clear why the earlier study considered this assumption necessary or how you remove this assumption in your analysis. Could you provide a more detailed comparison on this point (ideally, with some pointers to the relevant parts of your proof)?

---

> > > ### Author Response · Authors · 2025-11-30
> > >
> > > Thanks for the question. We believe that the difference stems from an early step in the analysis, namely, how the term $\mathbb{E}[\mathrm{sign}(X)\mathrm{sign}(Y)]$ is treated; from that point on, the two proofs diverge.
> > >
> > > In our proof, we handle the sign function using the sign-Gaussian identity
> > > $$
> > > \mathbb{E}[\text{sign}(X)\text{sign}(Y)] = \frac{2}{\pi} \arcsin\left(\frac{\mathbb{E}[XY]}{\sqrt{\mathbb{E}[X^2]\mathbb{E}[Y^2]}}\right),
> > > $$
> > > in lines 1797–1800 and lines 1812–1815, both on page 34 of the revised version of our paper.
> > > In our analysis, the terms that play the role of $X$ and $Y$ are linear functionals of a Gaussian vector, so the identity applies directly, and the sign function does not appear again in the proof.
> > >
> > > In contrast, Xiao et al. (2024) handle a similar term using the tower property
> > > $$
> > > \mathbb{E}[\text{sign}(X)\text{sign}(Y)] = \mathbb{E}[\text{sign}(X) \mathbb{E}[\text{sign}(Y) | X]],
> > > $$
> > > as shown in equations (38) and (39) on page 17 of the arXiv version of Xiao et al. (2024).
> > > Even after evaluating the inner term $\mathbb{E}[\text{sign}(Y) | X]$, the sign function in the outer expectation remains, so they use bounds such as equation (171) in Lemma 13 (page 33) of the arXiv version to handle the remaining term.
> > >
> > > Since the term $\mathbb{E}[\text{sign}(X)\text{sign}(Y)]$ arises very early in both analyses, the choice between the conditional-expectation route (Xiao et al.) and the sign-Gaussian identity (ours) determines the rest of the proof, making them diverge significantly. Once we rewrite $\mathbb{E}[\text{sign}(X)\text{sign}(Y)]$, we never introduce an analogue of the bounds used in Lemma 13 of Xiao et al. (2024).  Consequently, the lower-bound assumption on the covariance spectrum is not needed in our derivation of the ODE.
> > >
> > > We expect that these methodological differences stem from the differing goals of the two works. The primary goal of Xiao et al. (2024) is to derive both an SDE and an ODE representation of signSGD; they first derive the SDE and then obtain the ODE from it.
> > > In contrast, the main goal of our paper is to express the loss of signSGD in terms of model size $M$, number of steps $N$, learning rate $\gamma_0$, and power-law parameters $\alpha$, $\beta$. Since an ODE representation is sufficient for this purpose, we derive the ODE directly without involving an SDE, and this allows us to handle the term $\mathbb{E}[\text{sign}(X)\text{sign}(Y)]$ differently and to avoid the spectral lower-bound assumption.
> > >
> > > Xiao et al. (2024) also make the following comment in their discussion section (second line on page 12 of the arXiv version). Note that Theorem 2 in the comment is their result about ODE:
> > > > We believe that a version of Theorem 2 is true in much greater generality than we have proven it
> > >
> > > We interpret this as indicating that the lower-bound assumption is primarily a limitation of their current proof technique rather than a fundamental requirement for the validity of the ODE itself. Our analysis provides one such extension: by working directly with the sign-Gaussian identity, we obtain an ODE for SignSGD without assuming a uniform lower bound on the covariance spectrum.

---

### Author Response · Authors · 2025-11-23
**Revision of the Paper**

We thank the reviewers and the area chair for their time, careful reading, and thoughtful feedback. Your comments have been very helpful for improving our paper. For the reviewers' convenience, we highlighted our modifications in blue.

Below, we provide a brief summary of the major changes in the revised manuscript.

- We added a proof of equivalence to the general covariance matrix case with eigenvalues $1^{-2\alpha},2^{-2\alpha},\dots,d^{-2\alpha}$ in Section D of the Appendix. This proof indicates that the choice of $H$ as a diagonal matrix is without loss of generality.

- We added an analysis for the linear decaying scheduling and the cosine scheduling in Section H of the Appendix.

- We added an analysis and experiments for the case with noisy labels in Section L of the Appendix.

- We added a comparison with the existing literature on kernel methods in Section B.3 of the Appendix, and also acknowledged them in the lines 122-130 of the main text.

- We expanded our explanation of why a larger $\beta$ can lead to a faster plateau of the target decay in the revised version (lines 489-506).

- We added an analysis of the warmup-stable-decay schedule in Section G.4 of the Appendix.

- We added an experiment for AdamW and SGD optimizer on the Transformer architecture in the Section C.5 of Appendix, demonstrating the drift-normalization and noise-reshaping effects.

---

### Author Response · Authors · 2025-12-03
**Summary of Contributions, Discussion, and Revisions**

Dear (New) AC,

Thank you for taking over our submission. To assist your meta-review, we briefly summarize our main contributions, reviewers’ key concerns, our responses, and resulting changes in evaluations.

## **Summary of the Contribution**
In this paper, we study the scaling laws of signSGD under the Power-Law Random Features (PLRF) framework, previously analyzed by Paquette et al. (2024) for SGD.

First, we derive a loss formula for signSGD with constant learning rate $\gamma_0$ in terms of model size $M$, training steps $N$, and PLRF model parameters (feature decay $\alpha$, target decay $\beta$).
Compared to the SGD loss formula of Paquette et al. (2024), we observe two characteristic effects of signSGD: drift-normalization and noise-reshaping.

Second, under a fixed compute budget $\mathfrak{f}=MN$, we balance $M$ and $N$, and optimize over $\gamma_0$, expressing the compute-optimal loss as a power of $\mathfrak{f}$.
We compare the resulting exponent of $\mathfrak{f}$ with that of SGD in Paquette et al. (2024), identify regions where signSGD scales better, and empirically validate them.

Third, we show that a stable-decay schedule yields better compute-optimal scaling than a constant learning rate in the region $\alpha>0.5$, $0.5-\alpha<\beta<\tfrac{2\alpha-1}{2(4\alpha-1)}$.

## **Summary of the Discussion**

### **Reviewer dH5K**
Reviewer dH5K mainly questioned the technical novelty relative to Xiao et al. (2024) and Paquette et al. (2024).
We clarified that we provide a new loss formula and compute-optimal scaling characterization for signSGD under PLRF, with new technical ingredients, rather than a mechanical extension of prior SGD analyses.

The reviewer then asked why our analysis does not require the lower bound on the spectrum of the covariance matrix, unlike Xiao et al. (2024). We explained that our technique handling the sign nonlinearity obviates the need for such a bound.


### **Reviewer 1qNE**
Reviewer 1qNE was concerned about the restrictive setting (diagonal covariance, one-pass training, batch size 1, Gaussian sketching).

We
* explained how a general covariance matrix can be reduced to the diagonal case,
* argued that one-pass training is closer to the practice in large-scale language-model training,
* and discussed how our results may extend beyond batch size 1 and Gaussian sketching.

Following the reviewer’s suggestion, we added a non-PLRF synthetic experiment comparing Adam, signSGD, and SGD in a feature learning setting with a learnable sketch matrix $S$ (Appendix C.6).


### **Reviewer Hokd**
Reviewer Hokd requested stronger connections to practical neural network training and more experiments on neural networks.
We added experiments on the Transformer architecture using AdamW and SGD (Appendix C.5).
We observe a steeper compute-optimal slope for AdamW compared to SGD, and observe drift-normalization and noise-reshaping effects by comparing two optimizers.

The reviewer also raised a concern about approximating the warmup-stable-decay with the stable-decay schedule. We clarified that the reason why we used a stable-decay schedule is to emphasize that the stable and decay phases of a warmup-stable-decay schedule are what improve the compute-optimal exponent of signSGD, not because we cannot analyze the warmup-stable-decay itself.


### **Reviewer AVVQ**
Reviewer AVVQ was concerned about the simplified setup (linear model and feature map, noiseless observations, no feature learning).
We extended the analysis to noisy targets (Appendix L); the reviewer called these preliminary results “very interesting” and encouraged follow-up work on noise-dominated regimes.
We also argued that our work, together with related scaling-law analyses, is a stylized yet important step toward understanding large-scale nonlinear models, with which the reviewer “completely agree[d].”

Finally, we added a detailed comparison with kernel-method scaling results (Appendix B.3), which the reviewer “appreciate[d].”


### **Reviewer 5nJx**
Reviewer 5nJx questioned whether the diagonal-covariance analysis extends to a general covariance matrix in the signSGD setting due to its coordinate dependence.
We proved that a general covariance with eigenvalues $1^{-2\alpha},…,d^{-2\alpha}$ can be reduced to our diagonal case, using the rotational invariance of the Gaussian sketch matrix $S$. We added this proof as Appendix D. The reviewer described the explanation as “very nice,” thereby resolving the concern and increasing the rating.

Following the reviewer’s suggestion, we added an analysis of linear decay and cosine schedule (Appendix H).

### **Overall Outcome**
We strengthened both the theoretical and empirical aspects of the paper in response to all reviews. During the discussion period, two reviewers (AVVQ and 5nJx) raised their scores from 4 to 6.
Because the discussion phase ended early, we did not receive responses from 1qNE and Hokd, and our exchanges with 5nJx and dH5K halted before a conclusion.

---

### Meta-Review · Area_Chair_HvLx · 2025-12-21

**Summary:**

This paper extends the results of Paquette et al. (2024) from one-pass SGD to signSGD, and then study the scaling laws. Compared to SGD, there are two effects by signSGD, one is drift-normalization and another one is noise-reshaping. Under the fixed compute budget, the authors can similarly identify the regimes which one is better than SGD or not. Two reviewers complain about the motivation, setting, or difficult to follow. The AC also checked this paper and recommend to accept this paper.

**Reviewer Concerns:**

Reviewer dH5K concerned about the technical novelty when compared to Xiao et al. (2024) and Paquette et al. (2024). The authors have clarified this.

Reviewer 1qNE concerned about the restrictive setting (diagonal covariance, one-pass training, batch size 1, Gaussian sketching). The authors have clarified this.

Reviewer Hokd requested stronger connections to practical neural network training and more experiments on neural networks. The authors added these experiments.

Reviewer AVVQ concerned the simplified setup (linear model and feature map, noiseless observations, no feature learning). The authors explained this.

Reviewer 5nJx questioned about the diagonal-covariance analysis. The authors added an analysis in Appendix H.

**Reviewer Scores:**

Reviewer dH5K and Reviewer 1qNE would keep their positive evaluation.

Reviewer Hokd may increase the score.

Reviewer AVVQ and Reviewer 5nJx decided to increase the score.

---

### Decision · Program_Chairs · 2026-01-26

Accept (Poster)